# Enhanced subglacial discharge amplifies Petermann Ice Shelf melting when ocean thermal forcing saturates

Abhay Prakash [1,2] ✉, Qin Zhou [3], Tore Hattermann [4,5] & Nina Kirchner[1,2]

Increased basal melting of the Petermann Ice Shelf is typically attributed to rising ocean temperatures. While subglacial discharge is known to intensify basal melt, the underlying mechanisms and their evolution in a warming climate remain unresolved. Using a 3-D numerical regional ice shelf–ocean model centered on the Petermann Fjord, we identify a regime shift in heat flux efficiency within the ice shelf cavity when discharge exceeds the current peak summer value. In this regime, thermal driving saturates, and discharge-intensified currents increase melt by enhancing shear-driven turbulent mixing across the ice shelf-ocean boundary layer. Increases in melt are most profound at the crests of basal channels, where vigorous meltwater confluence amplifies friction velocity. Challenging conventional attributions of increased ice shelf basal melting to ocean warming alone, our results demonstrate how atmospheric warming exacerbates ocean-driven melt processes and is likely to play a dominant role in amplifying future basal melt.

The Greenland Ice Sheet is now the dominant single source of barystatic sea-level rise[1,2]. However, observational estimates of contemporary mass loss likely represent conservative baselines[3], as process-level uncertainties in ice-ocean interactions and surface melt amplification suggest future rates will exceed current projections[4,5]. The northern sector of the Greenland Ice Sheet hosts several of the last remaining marine-terminating outlet glaciers featuring floating ice tongues (also called ice shelves) that exert buttressing forces to the inland ice. During 2001 – 2021, the dominating factors of mass loss from these ice shelves were (in decreasing order) basal melt, calving, and negative surface mass balance. Between 2003 and 2010, collapses were observed at Hagen Bræ, C.H. Ostenfeld glacier, and Zachariæ Isstrøm[6]. Dynamic thinning, retreat and even collapse combined with perturbations of the outlet glacier's grounding line can lead to a loss of backstress and accelerated mass loss from the northern Greenland Ice Sheet sectors[7] holding an ice volume of ca. 4 m global mean sea level equivalent[8,9].

Along the Nares Strait coast of the northern Greenland Ice Sheet, Petermann Ice Shelf terminates into the Petermann Fjord. Formerly the longest ice shelf in the northern hemisphere at ca. 80 km in length, two recent massive episodic calving events in 2010 and 2012 reduced its length by ca. 30 km[10]. The ice shelf draft thins from ca. 400–600 m near the grounding line to ca. 50–100 m near the calving front. Evenly spaced across its 15–20 km width are 1–2 km wide basal melt channels that incise several-100 m deep into the draft[11]. Warming of Atlantic Water in Petermann Fjord by ca. 0.3 °C has been observed since 1960, with much (ca. 0.2 °C) of the warming occurring in the last two decades[12,13]. Average basal melt rates at the Petermann Ice Shelf grounding line, at ca. 600 m below sea level, are currently reported to exceed 17 m/yr, leading studies[6,13] to suggest that increasing ice shelf basal melt is following increasing ocean temperatures in Greenland fjords. Numerical modelling study and satellite observations have revealed that the highest basal melt rates occur in the kilometre-size grounding zone of the glacier, driven by tidally modulated intrusions of seawater[14,15]. Seasonal and climate-warming driven long-term changes in regional sea ice cover are also known to enhance basal melting. According to model results[16,17], elevated temperatures and an energetic ocean circulation under the ice shelf, driven by variations in Nares

[1]Department of Physical Geography, Stockholm University, Stockholm, Sweden. [2]Bolin Centre for Climate Research, Stockholm University, Stockholm, Sweden. [3]Akvaplan-niva, Tromsø, Norway. [4]Norwegian Polar Institute, Tromsø, Norway. [5]Complex Systems Group, Department of Mathematics and Statistics, UiT - The Arctic University of Norway, Tromsø, Norway. ✉e-mail: abhay.prakash@natgeo.su.se

Strait sea ice mobility and thickness, increased basal melting by a factor ranging between 1.2 and 1.7. Also, melting can be enhanced by routing of glacier surface melt to the glacier bed, and further to the grounding line, where it enters the fjord as subglacial discharge. Modelling study suggests that a higher subglacial discharge during summer (June-August) triples basal melt at Petermann Ice Shelf compared to winter[18]. Furthermore, within the central basal channel, for the region extending from the grounding line to 16 km seaward of it, between August and December 2015, an estimated area-averaged melt rate maximum of 170 m/yr has been reported[19], consistent with episodes of enhanced discharge-driven sub-ice shelf currents. This suggests that subglacial discharge may play a more important role in modulating Petermann Ice Shelf basal melt than sea ice-driven changes in ocean circulation. However, with atmospheric and oceanic warming expected to amplify in the Arctic by the end of the century[20,21], a salient question remains unanswered: How would Petermann Ice Shelf basal melt respond to projected increases in discharge under a warmer future climate, and what would be the mechanisms driving this response?

In this study, we extend a high-resolution unstructured grid 3-D ocean–sea ice–ice shelf regional model setup[17,22] centred over the Petermann Ice Shelf and Petermann Fjord to include subglacial discharge at the grounding line (Fig. 1). The fjord bathymetry and the ice shelf draft are derived from BedMachine v3[8]. Modifications have been applied to the topographical datasets (discussed further below) to amend an inaccurately represented sub-ice shelf water column thickness, and furthermore, to implement a 540–610 m deep inner sill situated ca. 25 km from the grounding line, retrieved using aerogravity data from Operation IceBridge[23] (Supplementary Fig. E1). Open boundaries in the Lincoln Sea and Baffin Bay are used to provide lateral ocean boundary conditions. These include temperature, salinity, and seawater velocities derived from A4 ROMS[24] (4 km pan-Arctic Regional Ocean Modelling System grid), and merged sea surface elevation derived using A4 ROMS and 5 km AOTIM[25] (Arctic Ocean Tidal Inverse Model). Data from the 5.5 km polar (p) version of RACMO2.3p2[26] (Regional Atmospheric Climate Model) is used to compute the momentum, heat, and freshwater fluxes. Sea ice concentration and thickness, bulk ice salinity, and sea ice velocities are obtained from A4 ROMS - CICE[27] (Community Ice CodE). Using this setup, we perform a control run with zero subglacial discharge, and three test runs designed to simulate the response of the cavity circulation and ice shelf basal melt rates to present day subglacial discharge, increased discharge under high-emissions Representative Concentration Pathway8.5 (RCP8.5) climate warming scenario, and an intermediate discharge scenario between the two (Table 1). We show that discharge increases basal melting by (a) accelerating a quasi-estuarine circulation[28] which increases the advective heat transport into the cavity, and by (b) enhancing the shear-driven mixing across the ice shelf-ocean boundary layer. The latter exerts a dominant control over the increase in basal melting seen for higher discharge under future warming scenarios while the former (advective heat transport) saturates, suggesting a shift towards a regime where melt increases are primarily driven by increases in friction velocity (i.e., velocity scale representing shear stress in the boundary layer). Our findings challenge the prevailing focus on increased oceanic heat forcing as the primary driver of basal melting at the Petermann Ice Shelf, revealing that heightened subglacial discharge significantly enhances (channelised) basal melt. This suggests that atmospheric warming-induced excitation of oceanic melt processes will critically influence the stability of the ice shelf.

## Results

### Response of cavity circulation and basal melt to contemporary discharge

In the control run, Petermann Ice Shelf basal melt rates largely range from a few m/yr near the calving front to ca. 40 – 60 m/yr under the

deeper (400– 600 m) regions of the ice shelf base near the grounding line (Fig. 2a). Melt rates decreasing with distance from the grounding line is consistent with results obtained from steady state ice discharge divergence approach[11] and satellite remote sensing[29]. For the period from 2011–2015, remotely sensed estimates reveal melt rates of up to ca. 80 m/yr near the grounding line, although they largely range between 40 – 50 m/yr decreasing to ca. 10 m/yr under the shallower (< 200 m) portions of the ice shelf[29], and are comparable to our modelled estimates. With the introduction of contemporary summer mean subglacial discharge ($Q_{sg}$-present) into the ice shelf-fjord system, the modelled mean melt strengthens nearly throughout the domain and appears to be concentrated along the basal channels that run along the length of the ice shelf (Figs. 1d, 2a, b). Spatial variability of our modelled melt on scales similar to the basal channel configuration is consistent with observational findings[11,29]. Strongest increases of up to ca. 100 m/yr are seen within these channels under the deeper regions, which gradually decreases towards the calving front. This increase lies within the in situ conductivity-temperature and radar-derived estimates of discharge-enhanced seasonal melt rate maxima of 40–170 m/yr[19]. The summer mean basal melt rate ($m_b$), friction velocity ($u^*$), and thermal driving ($\Delta T$) are averaged over (a) the entire ice shelf base, and (b) over parts of the ice shelf where draft is deeper than 300 m, and are indicated, respectively, by subscripts $pgis$ and 300 m appended to the variable in question. Compared to the summer mean modelled melt rates in the control experiment, the contemporary discharge-driven increase in $m_{bpgis}$ and $m_{b300m}$, respectively, is 41.2% and 68.1%, driven by similar increases in $u^*_{pgis}$ (47%) and $u^*_{300m}$ (67.4%) and modest increases in $\Delta T_{pgis}$ (1.1%) and $\Delta T_{300m}$ (4%) (Fig. 1d).

The presence of subglacial discharge at the grounding line enhances the fjord-scale overturning circulation (Fig. 3i). We observe a more robust Atlantic Water inflow at depth, extending into the inner fjord basin and up to the grounding line (Fig. 3c, f). Hereafter, we refer to this process as the thermal pump, which promotes the entrainment of heat and salt deep into the ice shelf cavity (columns 1 and 2 in Fig. 3). We note an increase in bottom temperature of 0.03 °C at a location ca. 10 km from the grounding line, and a temperature excess (thermal driving) of 0.1 °C under the ice shelf base at 300 m depth (Figs. 2j, 3g). A higher thermal driving near the grounding line (Fig. 2i, j) drives more melting at depth. A more energetic outflow/strengthened meltwater plume dynamics, driven by the introduction of discharge, exacerbates the shear-driven turbulent mixing and heat transfer across the ice shelf-ocean interface (Fig. 2e, f), thus driving more melt. The spatial variability of friction velocity anomalies is consistent with the corresponding melt rate anomalies (cf. panels b and f in Fig. 2). Regions exhibiting a strong increase in friction velocity are concentrated along the longitudinal basal channels near the grounding line, and gradually decrease downstream of it. However, a higher meltwater production (Fig. 2b) and stronger outflow (Fig. 3f, i) deposits larger volumes of meltwater from the deeper to the shallower regions of the ice shelf base, considerably lowering the thermal driving locally (Figs. 2j, 3d, e). Thus, while the increase in $u^*_{pgis}$ and $u^*_{300m}$ is significant, the contribution of thermal driving is largely restricted to the deeper regions of the ice shelf draft ($\Delta T_{300m}$) (Fig. 1d). This implies that for subglacial discharge equivalent to present-day summer mean magnitudes, increase in basal melt under the deeper (> 300 m) drafts is driven by increases in both thermal driving (increased thermal pumping) and friction velocity (increased shear-driven mixing).

### Heightened discharge and saturation of the thermal pump

When transitioning from the control experiment to heightened discharge in the RCP 8.5 scenario ($Q_{sg}$-RCP 8.5), and an intermediate discharge scenario between the two ($Q_{sg}$-median) (Table 1), a higher modelled discharge, as expected in a future warmer atmosphere[30,31], results in considerably higher increases in basal melt as compared to $Q_{sg}$-present (Figs. 1d, 2a–d). In particular, an increase in discharge from

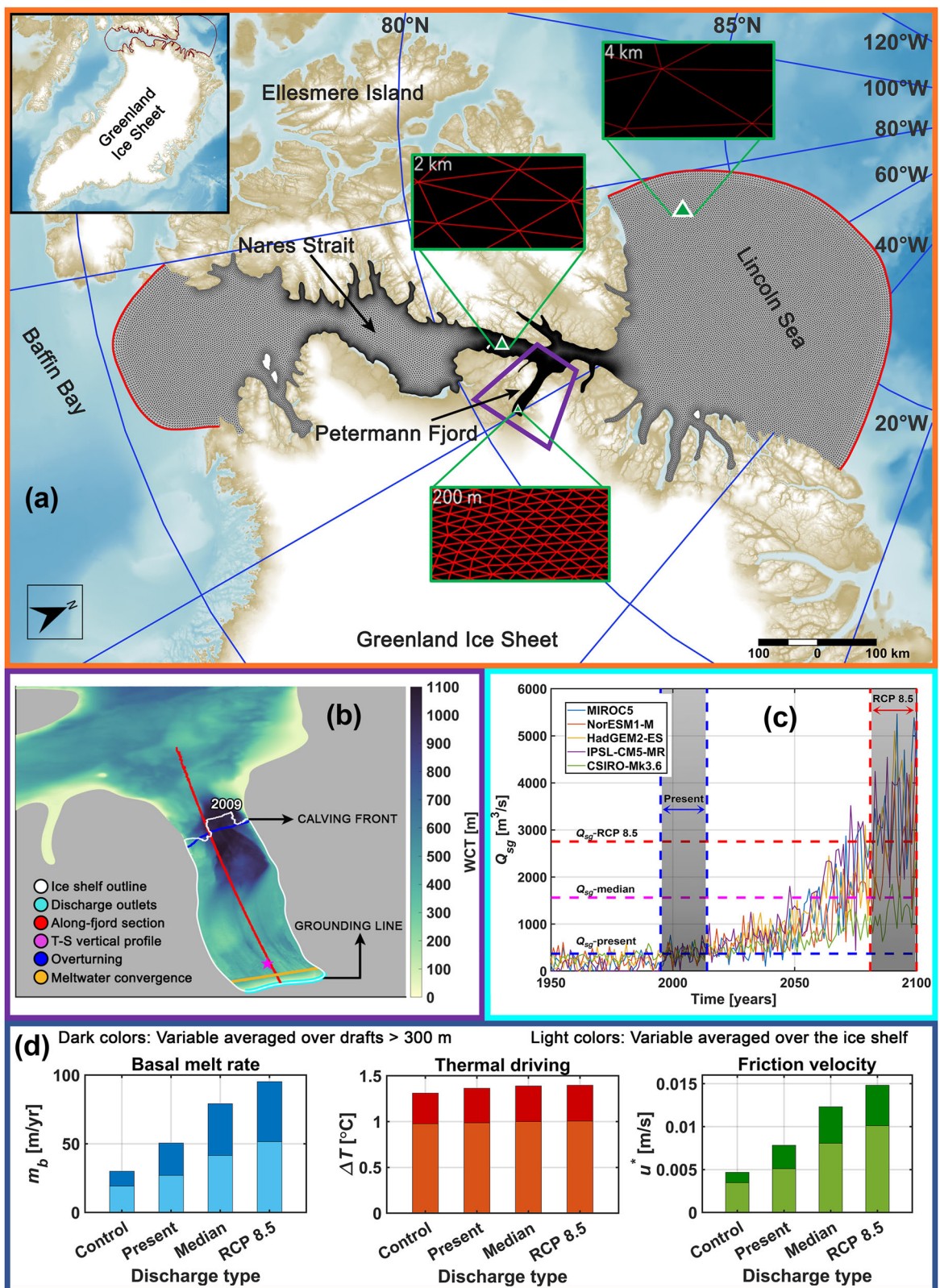

control to RCP 8.5 results in more than twofold (168.4%) and threefold (217.2%) increase in $m_{bpgis}$ and $m_{b300m}$, respectively. These increases are primarily driven by increases in $u^*_{pgis}$ (191.8%) and $u^*_{300m}$ (216.9%), whereas $\Delta T_{pgis}$ and $\Delta T_{300m}$ show relatively nominal increases of 2.9% and 6.5%, respectively. For the median experiment, 2.3% and 5.9% increase in $\Delta T_{pgis}$ and $\Delta T_{300m}$, and 132.5% and 163.4% increase in $u^*_{pgis}$ and $u^*_{300m}$ generate 116.3% and 163.6% increase in $m_{bpgis}$ and $m_{b300m}$,

respectively. As noted for $Q_{sg}$-present, the strongest increases occur under the deeper regions of the ice shelf base, which gradually decreases downstream (Fig. 2c, d), however, channelised melting is seen to strengthen substantially (discussed further below).

Combining insights from all the experiments provides a novel perspective on how discharge dictates the interplay between Petermann Ice Shelf basal melt and its drivers: Increase in the heat entrained

**Fig. 1 | Petermann Ice Shelf-fjord system: Model domain, fjord hydrography, and discharge-driven mean response of melt and its drivers.** A map of our regional Petermann Ice Shelf and fjord model domain (**a**) (in a wider context shown in the inset in the upper left corner) overlaid on a shaded relief representation of the bathymetric grid from the International Bathymetric Chart of the Arctic Ocean v4.0[69]. The mesh size in the model domain varies regionally, as exemplified by the three green-framed boxes. Open ocean boundaries in the Lincoln Sea and Baffin Bay are marked with red lines. Petermann Fjord is surrounded by a purple polygon, detailed in panel (**b**). **b** Petermann Ice Shelf extent (white outline), from CE 2009, overlaid on water column thickness (WCT) in Petermann Fjord and adjacent parts of Nares Strait. The cyan line marks the grounding line, where discharge (detailed in panel (**c**)) is injected into the model. The red line marks the location of the transect over which the temperature, salinity, and flow along the fjord are investigated. The

magenta star denotes the node ca. 10 km from the grounding line at which the vertical profile of temperature and salinity are shown. Blue and yellow lines, respectively, represent the transects across which overturning circulation and diagnostics for meltwater convergence are shown. **c** Discharge ($Q_{sg}$) scenarios, 1950–2100 CE, obtained by forcing the Modéle Atmosphérique Régional with five general circulation models[43,65]. Shaded vertical stripes with blue and red stippled outlines denote the periods over which $Q_{sg}$-present (CE 1995–2014) and $Q_{sg}$-RCP 8.5 (CE 2081–2100), respectively, are averaged over (Table 1). Blue, magenta, and red horizontal stippled lines correspond to $Q_{sg}$-present (372 m³/s), $Q_{sg}$-median (1563 m³/s), and $Q_{sg}$-RCP 8.5 (2754 m³/s) discharge magnitudes, respectively. **d** A graphical summary of the results showing basal melt rate ($m_b$), and its drivers, namely, thermal driving ($\Delta T$) and friction velocity ($u^*$). The discharge type categories correspond to the experiment design (Table 1).

in the ice shelf cavity as a result of the thermal pump is strongest at the inclusion of present-day summer mean estimate of subglacial discharge, with much of the heat delivered by the thermal pump concentrated under the deeper ice shelf base in the inner fjord basin (Figs. 1d, 3d, g). At a summer mean discharge value which is approximately fourfold higher than the present (i.e., $Q_{sg}$-median vs. $Q_{sg}$-present), thermal driving increases at a lower rate (Figs. 1d, 3g; cf. Fig. 3d and Supplementary Fig. C1a). We note an increase of 1.9% in $Q_{sg}$-median $\Delta T_{300m}$ as compared to $Q_{sg}$-present (Fig. 1d). Moreover, a larger meltwater deposition further lowers the thermal driving under the shallower drafts (Figs. 2b, c, j, k, 3i and Supplementary Fig. C1a–c), which results in a relatively weaker increase in $\Delta T_{pgis}$ (Fig. 1d). Lastly, in the RCP 8.5 scenario, discharge is nearly doubled as compared to the $Q_{sg}$-median experiment, however, the efficiency of the thermal pump in increasing basal melting saturates (Fig. 3g and Supplementary Fig. C1). Here, increases in $\Delta T_{pgis}$ and $\Delta T_{300m}$ are indiscernible as compared to the $Q_{sg}$-median experiment (Fig. 1d). Instead, we find that discharge under the high-emissions RCP 8.5 scenario enhances melting by further accelerating the cavity circulation, wherein, 24.1% and 20.4% increase in $m_{bpgis}$ and $m_{b300m}$ are seen, driven predominantly by 25.5% and 20.3% increase in $u^*_{pgis}$ and $u^*_{300m}$, respectively.

In sum, our experiments reveal a regime change in heat flux efficiency within the ice shelf cavity. In the control experiment, the ice shelf-ocean boundary exchange consumes a considerable amount of heat from the ocean that is available for melting the ice. As such, the latent heat loss and meltwater release cools the ambient ocean inside the cavity. In such a scenario, an increased fjord-scale overturning circulation due to increased discharge (e.g., ≤ $Q_{sg}$-median experiment) entrains more heat in the cavity (thermal pump), which increases melting through enhanced thermal driving, in addition to the enhanced shear-driven turbulent heat exchange. This saturates at some point, when more discharge (e.g., > $Q_{sg}$-median experiment, as seen for the RCP 8.5 scenario) accelerates the overturning circulation such that there is always enough heat coming into the cavity to maintain a high thermal driving (i.e., heat supply at an elevated level is maintained). In this case, melt rates become entirely a function of the friction velocity.

### Convergence of meltwater and channel stability

We use surface layer salinity as a proxy for tracking the spatial distribution of meltwater under the deeper regions of the ice shelf (Fig. 4 and Supplementary Fig. D1). Freshening occurs nearly throughout the domain in the discharge-driven experiments as compared to the control run. Furthermore, regions of substantial freshening align with four prominent ice shelf basal channels (Figs. 4a, b, 5a, b and Supplementary Fig. B1) along which the draft thins from 300–450 m ca. 5 km from the grounding line to 150 m ca. 25–30 km seaward of it, suggesting a topographically steered horizontal pathway for the meltwater outflow that is determined by the ice shelf basal morphology and which in turn influences it (discussed further below). Changes appear to be most profound within these

channels, particularly at the crests, where salinity decreases by up to 1.06 psu in the RCP 8.5 experiment with respect to the control run (Figs. 4, 5a, b). These locations of meltwater convergence are concordant with locations of the largest melt rate increases (discussed further below) of up to 181.5 m/yr; a sevenfold increase compared to the control run. Thermal driving is not higher at the crests where meltwater converges, but under the deeper drafts, which is consistent with the pressure-dependent melting point of seawater (Fig. 5c). Although thermal driving may not increase the most at the crests, we observe increases ranging from 0.02 to 0.12 °C in the RCP 8.5 experiment compared to the control run across the transverse cross-section near the grounding line (Fig. 5c). We find that the friction velocity profile across this section is (nearly) congruent with the basal melt rate profile, and exhibits highest increases at the crests of up to 0.03 m/s (Fig. 5d).

Three ice shelf basal channel sections are outlined (Fig. 6a), whose lengths are demarcated based on the apex of the 150 m ice shelf draft contour (e.g., Fig. 4a and Table 2). The channel averaged annual mean basal melt rates for the western ($S_w$), central ($S_c$), and eastern ($S_e$) sections for the RCP 8.5 experiment are calculated as 45 m/yr, 51 m/yr, and 47 m/yr, respectively. Enhanced melting at the crests will promote channel opening, and we can calculate the changes in channel averaged annual mean ice volume above each channel over time to estimate the incision/erosion timescale for that channel under the respective basal melt rates (Eqn. (5), (6)). To bracket the role of ice advection from upstream of the grounding line, three different rates ($AR_c$) are considered for each channel which correspond to the contemporary mean[10] (1 km/yr), and 50% lower (0.5 km/yr) and higher (1.5 km/yr) than contemporary mean values. While $S_w$ exhibits the lowest mean basal melt rate, it also features the shallowest mean draft value (Table 2). Notably, its incision timescales are comparable to those of $S_c$, which has the highest mean melt rate but a relatively deeper mean draft compared to $S_w$. The number of years ($N$) required for the mean ice volume above the channels to become zero is estimated to be 5.88, 6.52, and 7.31 years for $S_w$, in contrast to 5.95, 6.75, and 7.8 years for $S_c$, at $AR_c$ of 0.5 km/yr, 1 km/yr, and 1.5 km/yr, respectively (Fig. 6b). It has been shown that the viscous ice response to a basal channel is insufficient to significantly delay the breakthrough of basal channels in ice shelves that are thinner than 400 m[32]. With a lower mean melt rate than $S_c$ and the deepest mean draft value, $S_e$ demonstrates the greatest resilience. We estimate $N$ to be 8.06, 9.41, and 11.31 years for $AR_c$ of 0.5 km/yr, 1 km/yr, and 1.5 km/yr, respectively (Fig. 6b). These estimates suggest that ice advection from upstream of the grounding line (alone) may not be sufficient to compensate for the thinning of the ice shelf due to intense channelised basal melting that may be expected under the RCP 8.5 scenario (discussed further below).

## Discussion
Our model is forced with a contemporary summer mean discharge estimate and expected increases of it under future warming. Present-

**Table 1 | The ensemble averaged June-July-August (JJA) mean discharge ($Q_{sg}$) magnitude for the Petermann catchment [in m³/s] that is distributed uniformly across the grounding line for present conditions (1995–2014), future conditions (2081–2100, under the RCP 8.5 scenario), a control scenario, and a median scenario**

| Experiment | JJA-mean $Q_{sg}$ [m³/s] | JJA-mean time period |
|---|---|---|
| Control | N/A | N/A |
| $Q_{sg}$-present | 372 | 1995–2014 |
| $Q_{sg}$-median | 1563 | N/A |
| $Q_{sg}$-RCP 8.5 | 2754 | 2081–2100 |

day maximum discharge estimates[18,33] at Petermann are two to four times higher than the present-day summer mean discharge used in this study, and comparable to the $Q_{sg}$-median experiment. Therefore, while $Q_{sg}$-median represents an intermediate state between present and extreme future mean scenarios, it is plausible that such increases in discharge, and thus basal melt, currently occur briefly during summer with inter-annual variability. We posit that if exceptionally high discharge is sustained for (say) several weeks, it may temporarily nudge the ice shelf-fjord system into a regime where basal melting is largely driven by friction velocity. We note that, unlike Gaussian-like patterns[33], our applied discharge forcing follows a boxcar function. Thus, studies investigating the sensitivity of the system to seasonal and inter-annual variability in subglacial discharge, derived, for e.g., from a subglacial hydrological model, are needed to quantify potential transitions in basal melting mechanisms. Modelling work[7] has demonstrated that the loss of thicker and rheologically stiffer sections of the Petermann Ice Shelf within 12 km from the grounding line (draft ≥ 300 m) could sufficiently disrupt grounding line stresses, accelerating interior ice flow and increasing dynamic ice discharge, significantly contributing to sea level rise. We thus suggest that discharge-driven intensified basal melting of the deeper regions near the grounding line (Figs. 1d, 2), which are dynamically resilient, will have a profound impact on the future stability of the Petermann Glacier, as their loss could trigger an unopposed acceleration of the inland ice.

Thus far, basal melting of the Petermann Ice Shelf has been largely investigated from the perspective of (increased) oceanic heat delivery to Petermann Fjord. However, both observational and modelled estimates concur on the abundance of heat supply into the fjord, i.e., heat availability exceeds what is needed to generate contemporary estimates of basal melt[12,17,34]. Our findings show that increased basal melting of the Petermann Ice Shelf in a warming climate is not limited to an increase in oceanic heat forcing and underscore the importance of atmospheric warming in controlling the basal melting of Northern Greenland Ice Sheet glaciers[35]. Uncovering how this unfolds from an ice shelf-ocean interaction perspective, we find that the effect of the thermal pump on melt is strongest when discharge is introduced into the system (as $Q_{sg}$-present; Table 1) and saturates for discharge greater than $Q_{sg}$-median (Figs. 1d, 3 and Supplementary Fig. C1), wherein $Q_{sg}$-median is equivalent to present-day estimates of maximum discharge[18]. We see an increase in bottom temperature near the grounding line (increased heat advection towards the interior of the ice shelf cavity via the thermal pump) as discharge increases from control to present to median, but less so for substantially higher discharge under the RCP 8.5 scenario (Fig. 3g). Therefore, we posit that a regime shift in the mechanisms driving basal melting may occur when catchment-integrated meltwater runoff under future warming scenarios exceeds the present-day maximum discharge threshold. In such a scenario, the significance of the thermal pump in driving increased melting wanes, and the higher discharge being released into the fjord accelerates the sub-ice shelf currents, nudging the system towards a regime where basal melt increase is predominantly dictated by

substantially higher shear-driven turbulence, which increases heat fluxes at the ice shelf-ocean interface (Fig. 2). Observations show that Atlantic Water reaching the fjord in 2020 was 0.3 °C warmer compared to 1970, with 0.2 °C of this warming occurring between 2000 and 2020[13]. Further modelling work is required to determine the characteristics of Atlantic Water reaching the fjord by the end of this century under different climate warming scenarios. While we do not consider the impact of rising ocean temperatures in our study, we acknowledge that, realistically, in a future warmer climate, it would likely act in concert with increased discharge to further amplify basal melting. However, with a much lower heat capacity compared to the ocean, increases in surface melt over grounded ice in response to atmospheric warming (and consequently, subglacial discharge) are highly likely to outpace the warming of deep-ocean water masses, which alone would be required to bring about such substantial increases in basal melting in the upcoming centuries (e.g., increases of up to ca. 200 m/yr in the RCP 8.5 scenario) as modelled here. Thus, while the combined effect of increased ocean heat forcing and increased discharge on basal melting in a future warming climate needs to be examined, discharge-driven basal melting is likely to pose a more immediate and pressing threat to the glacier's long-term stability.

Seasonal changes, including, but not limited to an increase in fjord temperature of ca. 0.1 °C during summer as compared to winter[17] drives up to ca. 47% increase in mean channelised (sections $S_w$, $S_c$, and $S_e$ in Fig. 6a and Table 2) basal melting. Without taking into consideration the compensating viscous ice response (discussed further below), and for contemporary mean ice advection rate from upstream of the grounding line of 1 km/yr[10], this increase results in up to ca. three-fold increase in erosion rates. The strongest seasonal influence on melt is imposed by the addition of subglacial discharge into the fjord. Compared to winter (no discharge), we calculate that contemporary summer mean discharge increases mean melting in the channels by up to ca. three-fold, yielding up to an order of magnitude increase in erosion rates. We note that although $S_w$ and $S_c$ may exhibit higher rates of erosion as compared to $S_e$, the highest increases in erosion rates are modelled for $S_e$.

Across all discharge experiments (Table 1), we consistently find channelised strengthening of basal melt (Fig. 2a–d), driven by the complex ice shelf basal morphology[11]. Within these channels (e.g., Fig. 5a), compared to the control experiment, larger volumes of buoyant meltwater rise along the steepest slopes of the ice base towards the channel's crest (Fig. 5). From there, the stronger buoyancy-driven overturning circulation in the fjord transports it further downstream along the longitudinal channel axis (Figs. 3i, 4 and Supplementary Fig. B1, D1). The steep ice shelf basal slopes characterising these channels support stronger entrainment of ambient Atlantic Water into the buoyant meltwater plume[29]. Furthermore, the vigorous confluence of meltwater towards the crest and its downstream advection increases the friction velocity, which predominantly drives the intensified melting (Figs. 2, 5b, d). Profound melting at the crests may accelerate the vertical growth of the channels. To that end, our estimates of mean ice volume changes above the channels indicate that sustained high mean melt rates due to enhanced summer discharge, as seen in the RCP 8.5 experiment, could undermine the ice shelf's structural integrity (Fig. 6). If such intensified melting persists for about a decade, it may fully incise the channels, likely leading to ice shelf disintegration. While the accuracy of this estimate depends on uncertainties in the applied melt rate parameterisation (Eqns. (1)–(4)), e.g., not accounting for enhanced stratification effects due to meltwater discharge[36,37], the simulated acceleration of cavity circulation remains a robust feature of our simulations. Together with observational evidence[19], this suggests significant impacts on the basal mass loss of the Petermann Ice Shelf. We also note that our static ice shelf geometry does not account for ice dynamical processes, which could

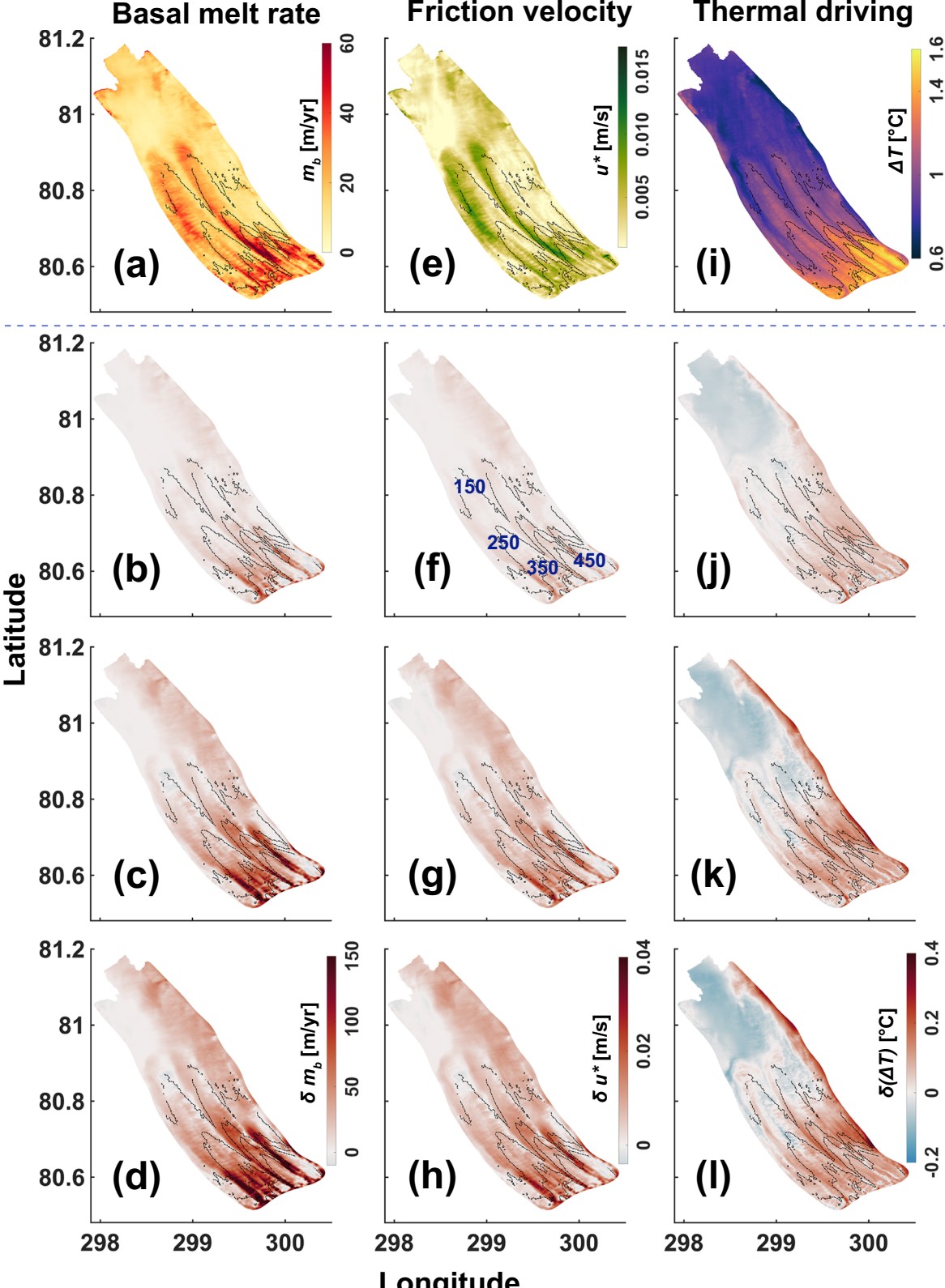

**Fig. 2 | Spatial distribution of summer mean melt and its drivers under varying discharge scenarios.** Response of ice shelf basal melt rate and its drivers to discharge ($Q_{sg}$) distributed uniformly across the grounding line (Fig. 1b), for the experiments defined in Table 1. Summer mean basal melt rate ($m_b$) (**a**–**d**), friction velocity ($u^*$) (**e**–**h**), and thermal driving ($\Delta T$) (**i**–**l**) for the control (first row), $Q_{sg}$-present relative to control (change: $\delta$; second row), $Q_{sg}$-median relative to control ($\delta$; third row), and $Q_{sg}$-RCP 8.5 relative to control ($\delta$; last row) experiments. Contours of the ice shelf draft are plotted at 100 m intervals (450–150 m; panel(f)) and overlaid as dotted black lines on each panel.

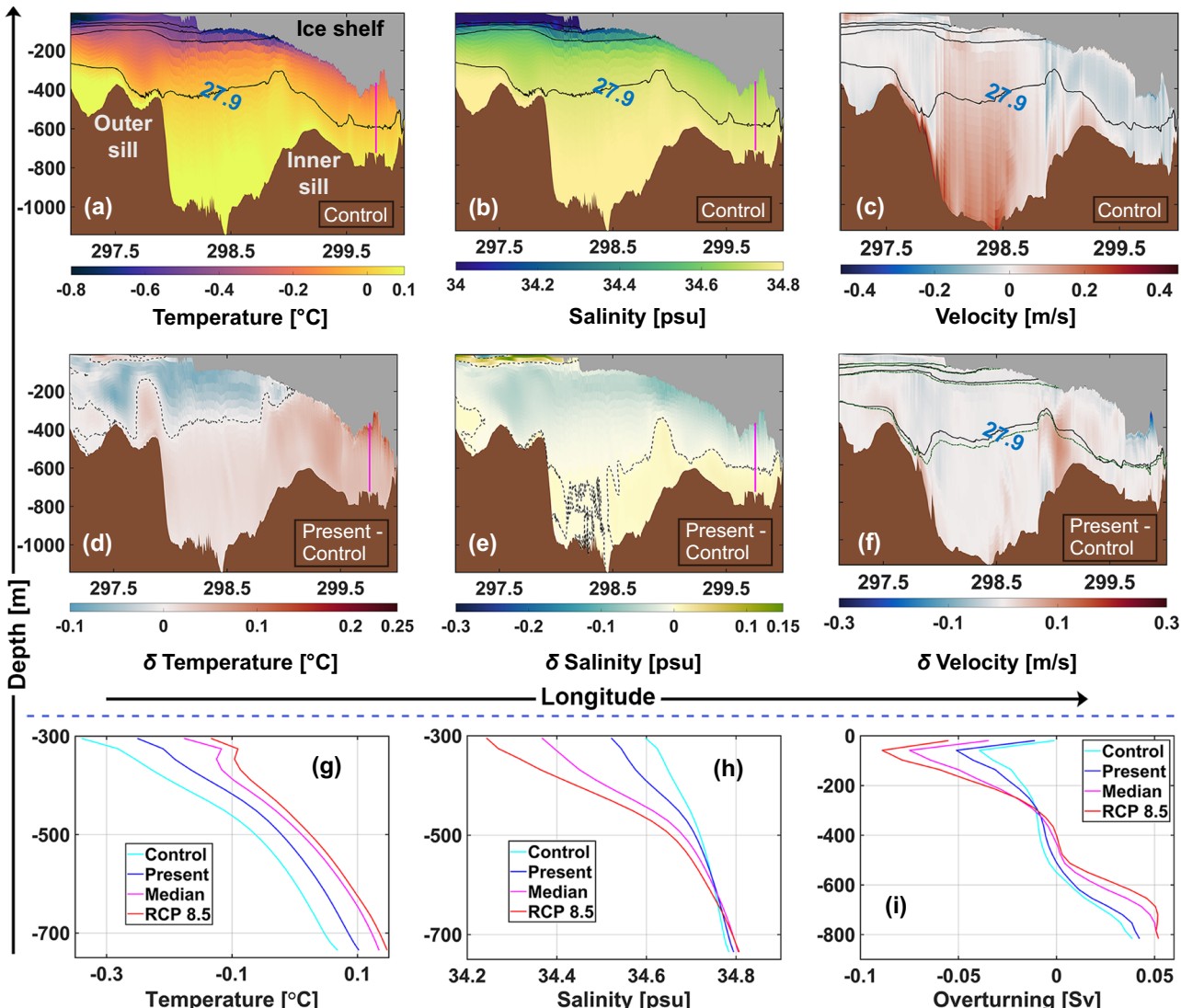

**Fig. 3 | Indicators of a regime shift in heat flux efficiency beneath the Petermann Ice Shelf.** Along-fjord summer mean temperature (**a**), salinity (**b**), and flow (**c**) for the control experiment for the section shown in Fig. 1b. Mean isopycnals (solid black lines) are overlaid at equal intervals of 0.2 kg/m³. The 27.9 kg/m³ isopycnal corresponds to the dense Atlantic Water that contacts the grounding line. For the same section, changes ($\delta$) in summer mean temperature (**d**), salinity (**e**), and flow (**f**) for the discharge ($Q_{sg}$) experiment $Q_{sg}$-present relative to control are shown. Black stippled lines in panels d and e indicate the zero temperature and salinity difference contour, respectively. In panel f, solid black lines and stippled green lines depict the mean isopycnals for the control and $Q_{sg}$-present experiments, respectively, plotted at equal intervals of 0.2 kg/m³. In each panel, the Petermann Ice Shelf grounding line (at ca. 600 m depth) is on the right margin, and the open ocean is to the left. Vertical magenta line in panels (**a**, **b**, **d**, and **e**) shows the transect location closest to the model node ca. 10 km from the grounding line (Fig. 1b) at which the vertical profile of summer mean temperature (**g**) and salinity (**h**) are shown for the control, $Q_{sg}$-present, $Q_{sg}$-median, and $Q_{sg}$-RCP 8.5 experiments (Table 1). **i** Laterally integrated summer mean vertical overturning profile[17] [in Sverdrup (Sv)] for all the experiments (Table 1) for the section shown in Fig. 1b. Positive [Sv] values correspond to (laterally integrated) inflow, and negative [Sv] values correspond to (laterally integrated) outflow across the section.

partially offset melting. While it has been shown that secondary ice flow is generally not strong enough to significantly hinder basal channel development in ice shelves thinner than 400 m³[32], certain sectors of the Petermann Ice Shelf, such as the southeastern sector near the grounding line, exceed this threshold. Further work is required to ascertain how localised (sub-km) channel closures or openings would impact broader ice dynamics and, consequently, the glacier's long-term stability. We suggest that future work using coupled ice-flow–ocean models, frameworks of which have been recently developed[38], or offline coupling of an ice-flow model with high-resolution ice shelf basal melt rates from sophisticated regional ocean models is needed to improve our understanding of the nuanced dynamical variability within these channels.

Our results may be applicable to other warm cavity environments where pronounced increases in discharge are expected and the ice shelf features a complex basal morphology. Specifically, we propose that these conditions may be met in the Ryder and Nioghalvfjerdsbræ (79N) glaciers. The ice shelves of both glaciers exhibit channelised basal geometry, leading to notable spatial melt rate variability and a dependency of basal melt on slope, similar to that of the Petermann Ice Shelf[17,29]. The lower ice advection rates from upstream of the grounding line make Ryder Glacier more prone to developing deep channels. At 79N, studies suggest that the recent expansion of its central basal channel results from increased discharge, attributed to a significantly expanded summer surface melt area, brought about by atmospheric warming[39,40]. Moreover, the presence of moulins along the Greenland Ice Sheet margins in summer[33,41,42], together with projected increases in surface melt[20,30], suggests that end-of-century discharge estimates for these sites[43] may be comparable to those of Petermann.

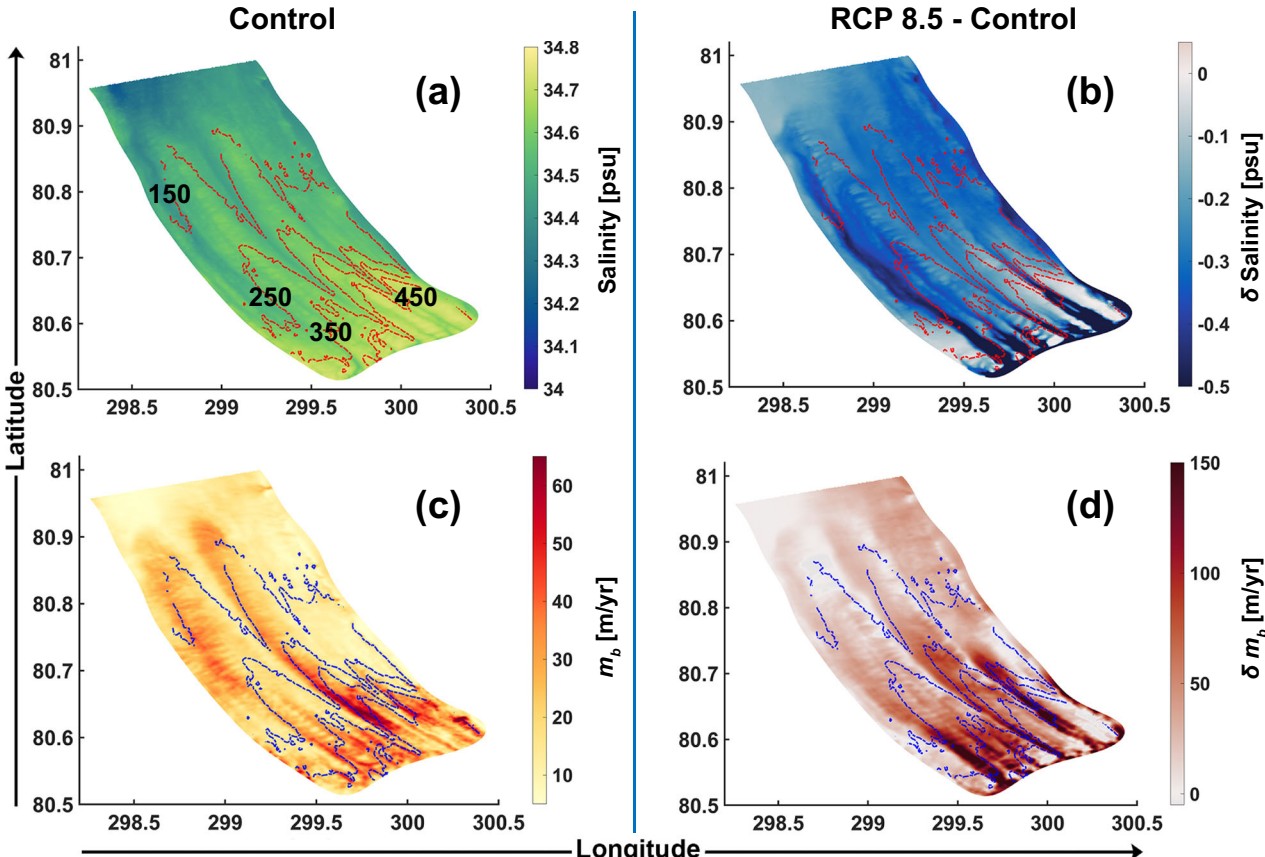

**Fig. 4 | Congruent patterns of increased freshening and melting.** Maps of the summer mean surface layer salinity (**a**, **b**) and basal melt rate ($m_b$) (**c**, **d**) for the control experiment (column 1) and the discharge ($Q_{sg}$) experiment $Q_{sg}$-RCP 8.5 relative to control (change: $\delta$; column 2). Contours of the ice shelf draft are plotted at 100 m intervals (450–150 m; panel(**a**)) and overlaid as dotted red and blue lines over the surface layer salinity and melt rate maps, respectively.

Analogous to Greenland, the intricate basal morphology of Antarctic ice shelves, sculpted by their interactions with the bed and/or melting, exhibits a diverse range of features (e.g., channels, keels, crevasses, and terraces) that strongly influence their melting and structural integrity, which depend on basal slope[44,45]. The role of a basally injected additional buoyancy source into the fjord in enhancing basal melting and retreat has been extensively investigated in Greenlandic settings[18,28,46,47] and has garnered recent attention in Antarctica[48,49]. Studies indicate that subglacial discharge played a critical role in influencing the mass balance of the Antarctic Ice Sheet during the last deglaciation period and is expected to remain a key factor in a warming climate[50,51]. In addition, ice shelf collapses linked to hydrofracturing and flexurally-induced fractures caused by supraglacial lake drainage events have been documented[52,53]. However, unlike Greenland[54,55], there is no evidence of surface meltwater infiltrating to the bed. Greenland's interconnected surface and subglacial hydrological systems provide a blueprint for understanding how Antarctica may evolve in a warmer world, as its surface hydrology increasingly resembles present-day conditions on the Greenland Ice Sheet[56]. Projected increases in Antarctic surface melt rates[57] suggest a shift towards a more expansive and dynamic surface hydrology, one that could eventually link the currently isolated bed via seasonal influxes of surface meltwater. We note that such scenarios depend on future emission pathways and whether the stringent targets set by the Paris Climate Agreement are met[58]. However, if these targets are exceeded, as anticipated under the RCP 8.5 projection, we posit that a fundamental shift in Antarctic ice dynamics and the processes controlling it cannot been precluded. In particular, energetic currents driven by intensified discharge in a cavity characterised by complex channelised

basal morphology, as modelled here, are likely to accelerate the heat flow towards the ice shelf base. The substantial shear-driven turbulence could independently enhance melting, and may even be sufficient to abrade the stable boundary layer stratification observed in critical regions where basal melting remains hitherto suppressed due to latent ocean environments[37].

Ice shelf basal topography is often represented in numerical models using an idealised tapered form, typically either completely flat or smoothly curved as a function of the along-fjord slope. Moreover, sub-ice shelf bathymetry in these models is derived from simplified interpolation techniques, which estimate depths between known points near the fjord mouth and the grounding line. These critical shortcomings are addressed in our model setup[22]. However, efforts to mitigate these topographic caveats in numerical models that investigate real-world locations, despite their well-acknowledged impetus on basal melting, are still not being widely undertaken. Indeed, an unknown sub-ice shelf bathymetry for most ice shelf-fjord systems is a limiting factor. As such, more targeted airborne and ground-based geophysical campaigns are needed to better characterise these environments. While our findings underscore the importance of implementing realistic ice shelf geometries, we also acknowledge the limitations of current datasets. Available resolutions are typically on the order of hundred to several hundred metres[8,59], which are insufficient to capture small-scale basal features, on the order of several (tens of) metres, as revealed by in situ under-ice shelf observations[37,45]. These features play a crucial role in modulating spatial melt rate variability. To fully understand the intricate interplay between subglacial discharge, basal morphology, and melt processes, we emphasise that a more extensive dataset of high-resolution observations is

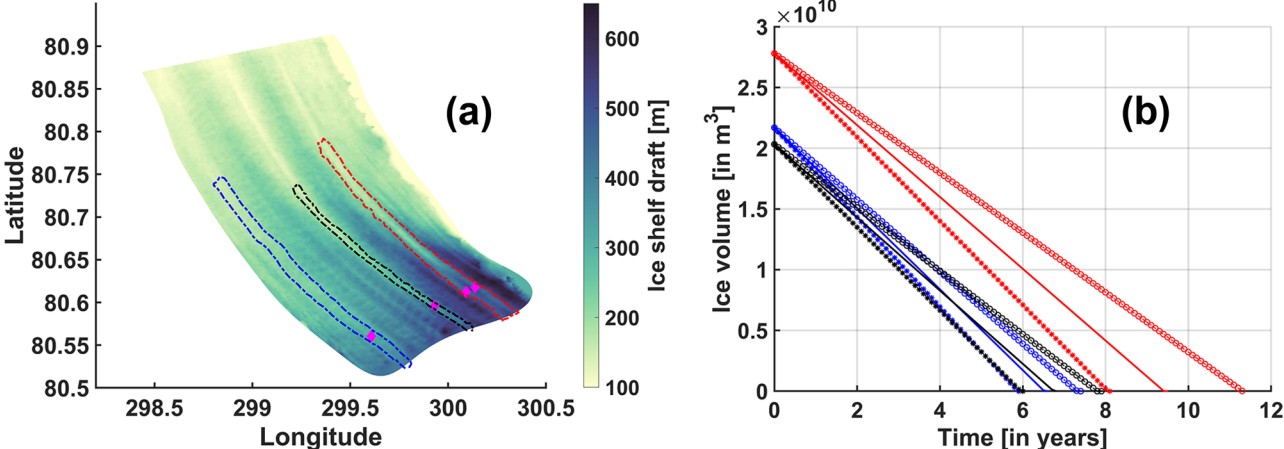

**Fig. 5 | Rapid convergence of meltwater toward the crests of basal channels and melt amplification in the RCP 8.5 scenario.** Cross-section profile of the Petermann Ice Shelf draft (**a**) at the transect ca. 5 km from the grounding line shown in Fig. 1b. Apex locations of four prominent longitudinal ice shelf basal channels at this section are highlighted using green stripes along which the Petermann Ice Shelf draft reduces from 300–450 m at the transect location to 150 m over a distance of 25–30 km from grounding line (Supplementary Fig. B1). Across the same section, the summer mean surface layer salinity and melt rate ($m_b$) (**b**), thermal driving ($\Delta T$) and melt rate (**c**), and friction velocity ($u^*$) and melt rate (**d**) are shown for the control experiment and the discharge ($Q_{sg}$) experiment $Q_{sg}$-RCP 8.5.

**Fig. 6 | Evolution of ice volume above basal channels under the RCP 8.5 scenario.** Map of the Petermann Ice Shelf draft overlaid with channel sections along the western ($S_w$; blue), central ($S_c$; black), and eastern ($S_e$; red) regions (**a**) which are used in calculating the changes in respective channel averaged annual mean ice volume above the channel over time in panel (**b**) for the RCP 8.5 discharge experiment, following Eqns. (5) and (6) (Table 2). Apex locations of four prominent longitudinal ice shelf basal channels at the across-fjord transect shown in Fig. 5 are represented using magenta diamonds. For each colour-coded channel section shown in (**a**), corresponding changes in mean ice volume above the channel section are shown in (**b**) for three different ice advection rates, namely, 0.5 km/yr (asterisk), 1 km/yr (solid line) and 1.5 km/yr (circle). Channel averaged annual mean basal melt rates for $S_w$, $S_c$, and $S_e$ are, respectively, 45 m/yr, 51 m/yr, and 47 m/yr.

**Table 2 | Dimensions associated with the Petermann Ice Shelf basal channel sections illustrated in Fig. 6(a)**

| Channel Name | Channel Length ($L_c$ [m]) | Mean Channel Depth ($H_c$ [m]) |
|---|---|---|
| $S_w$ | 30000 | 241 |
| $S_c$ | 25000 | 271 |
| $S_e$ | 28000 | 331 |

Channel sections along the western, central, and eastern regions are termed as $S_w$, $S_c$, and $S_e$, respectively.

imperative. To enhance the reliability of continent-scale projections, observational findings from sub-ice shelf environments - particularly near grounding zones - should be combined with insights from non-idealised, high-resolution numerical modelling studies informed by these datasets. Improved understanding of critical small-scale processes at the marine-ice margins should then inform the refinement of existing parameterisations and the development of new ones, which would facilitate large-scale models to better capture the influence of dynamics at the metre-scale. We note that in our experiments, discharge is distributed uniformly across the grounding line width, which may not be consistent with the ground truth. For an idealised tidewater glacier, modelling work has shown that distributed discharge generates more melting than channelised discharge[60]. With glacier hydrology exhibiting increased dynamism, which is expected to intensify in a warming climate, a concerted effort is warranted. This involves deriving insights into the relationships between moulin locations and density, meltwater routing, and eventually its magnitude and emergence location(s) at the grounding line from observational evidence and sophisticated subglacial hydrological models, which should then be incorporated into numerical simulations of ice shelf-ocean interactions. Furthermore, investigations of the interaction between subglacial discharge and tidally-modulated seawater intrusions in the grounding zone[14] will be essential for improving our understanding of glacier dynamics and stability.

## Methods

### The Finite Volume Community Ocean Model (FVCOM) setup

The unstructured grid, free-surface, 3-D primitive equation Finite-Volume Community Ocean Model (FVCOM)[61], augmented by modules for an ice shelf[62] and sea ice[22], is used. The model domain is centred on Petermann Fjord, covering the region between 75°N – 87°N and 29°W – 81°W (Fig. 1a). The grid comprises of unstructured triangle elements, featuring a total of 112709 nodes and 219836 cells, including 309 open boundary nodes in the Lincoln Sea and Baffin Bay. The topological flexibility of the unstructured grid allows accurate fitting of the complex irregular coastal geometry along the northern Greenland Ice Sheet margin. In particular, the grid is partitioned into polygons of varying horizontal resolution: A 200 m resolution is provided in the Petermann Fjord region to resolve the steep slopes characteristic of the fjord seafloor and ice shelf basal topography, which is relaxed to 2 km in the adjacent areas outside the fjord, and further to 4 km towards the open ocean boundary regions. For an extensive regional model setup, such a configuration enables us to strike a balance between computational efficiency and the need for accurate representation of topographical features and their variation over small spatial scales.

Petermann Ice Shelf draft derived from the BedMachine v3 dataset[8] (150 m resolution) represents a pre-2010 calving geometry[10], and is interpolated to the 200 m FVCOM model grid[22]. As such, the basal channels are resolved at 200 m scale in our model, which is sufficient to represent their 1-2 km width[11]. Bathymetry outside Petermann Fjord comes from the 1 km gridded IBCAO v3.0 dataset[63], whereas inside it, the BedMachine v3 dataset is used. In the vertical, the

domain between the irregular seafloor topography and the ice shelf draft (or the open ocean surface) is discretized into 23 terrain-following $\sigma$-layers. The ice shelf module dynamically adjusts horizontal pressure gradient forces to account for the weight of a freely floating ice shelf, with terrain-following $\sigma$-layers adapted to subduct beneath the prescribed ice shelf draft[62]. To alleviate errors in the discretization of horizontal pressure gradients along the sloping $\sigma$-layers, smoothing of the bathymetry is applied as a function of the mesh resolution before its implementation in the model[22]. Without seafloor depth measurements beneath the ice shelf, the BedMachine bathymetry is inferred by extending the known seafloor depth from seaward of the calving front landward towards the grounding line and aligning it with the depth of the ice shelf draft at the grounding line using natural neighbour interpolation. For Petermann Fjord, this leads to implausible abrupt changes in the water column thickness[22]. Specifically, the BedMachine bathymetry underestimates the water column thickness under the deeper ($> 200$ m) regions of the ice shelf, with sub-100 m values along the flanks and in the central zone (Supplementary Fig. E1a–c). More critically, this misrepresentation results in the formation of an artificial inner sill, which, if not accounted for, would block the inflow of the warm and dense (27.9 kg/m³) Atlantic Water at depth (Supplementary Fig. E1d) that contacts the grounding line[12,17]. Therefore, addressing these artefacts is imperative to ensuring that the model is able to reproduce the fundamental aspects of glacial fjord circulation[28]. Aerogravity data from Operation IceBridge was utilised to model the bathymetry beneath the ice shelf, revealing the presence of an inner sill, ca. 540–610 m deep, located about 25 km from the grounding line[23]. In this study, we implement a smoothed bathymetry product, wherein the BedMachine bathymetry has been modified to account for the unrealistic sub-ice shelf water column thickness and further integrated with the aerogravity data[22].

### Experiment design

The model is run from July 01, 2014 00:00:00 UTC – January 01, 2017 00:00:00 UTC with hourly oceanic (temperature, salinity, velocities, and sea surface elevation), 3-hourly atmospheric (heat and freshwater fluxes, and wind speeds), and daily averaged sea ice (sea ice concentration, thickness, and velocities, and bulk ice salinity) forcings obtained from respective high-resolution regional models over this period[17,22], and without discharge, which represents the Control experiment (Table 1). To allow investigations of the response of ice shelf basal melt to variations in subglacial discharge, our FVCOM setup has been augmented to include discharge across the ca. 20 km wide grounding line. Three discharge experiments are initialised from the stable model solution of the Control experiment from January 01, 2016 00:00:00 UTC and run until January 01, 2017 00:00:00 UTC (Table 1). Lacking information regarding the precise location(s) of emergence of subglacial meltwater across the grounding line, discharge is injected uniformly across the 90 cells (hereafter denoted by $\Delta_{Q_{sg}}$) representing the grounding line (Fig. 1b). At each $\Delta_{Q_{sg}}$, discharge is vertically uniform across the 23 $\sigma$-layers. The salinity at each $\Delta_{Q_{sg}}$ is set to 0, and the temperature is set to the (in situ) pressure-dependent freezing point of freshwater. The June-July-August (JJA)-mean discharge values presented in Table 1 represent the daily discharge values over the summer (JJA) months used in our model. Outside of this period, discharge is set to zero (similar to ref. 18). Results for the Control and discharge experiments are presented from the final year of the simulation (January 01, 2016 – January 01, 2017). We note that the pre-2010 ice shelf geometry provided by BedMachine v3 is not consistent with the forcing period(s). Change in ice shelf geometry (e.g., due to calving) has been shown to impact fjord circulation and basal melt rates[64]. However, the two prior large calving events in 2010 and 2012[10] removed the softer and thinner ($< 150$ m) outer sections of the ice shelf[7]. As such, we posit that the findings of our study - discharge-driven regime change in heat flux efficiency in the ice shelf cavity, and amplified channelised

melting under the deeper drafts - are robust, irrespective of the geometry used.

The present-day JJA-mean discharge ($Q_{sg}$-present) and its increase in a future warming climate under the RCP 8.5 scenario ($Q_{sg}$-RCP 8.5) for the Petermann catchment is obtained from simulations using a regional model[43] (Table 1). Here, a subset of the Coupled Model Intercomparison Project (CMIP5) Atmosphere and Ocean General Circulation Models (AOGCMs)[65] (Fig. 1c) is used to force the Modéle Atmosphérique Régional (MAR) 3.9.6[66] for the period 1950 – 2100. The projected MAR meltwater runoff over the Petermann catchment is then bias corrected to ensure that it is consistent with the best present-day (1995–2014) runoff estimate, and to also enable a smooth transition from present to RCP 8.5 forcing. It is then summed up over the Petermann catchment to give the discharge estimates. We note that the modelled runoff outputs were stored as annual means and were converted to the JJA-mean runoff by multiplying the output by a factor of 365/92, where 365 = number of days in a year, and 92 = number of days in JJA. This coarse conversion introduces a minor discrepancy between the converted and true modelled JJA-mean values; however, we maintain that this does not significantly impact the findings of our study. While inter-model differences can be large (Fig.1c), all five CMIP5 AOGCMs were used to prepare the multi-model mean discharge used in this study. Note that the median discharge magnitude ($Q_{sg}$-median) is not obtained from the model[43], but is constructed using the $Q_{sg}$-present and $Q_{sg}$-RCP 8.5 magnitudes (as the median of $Q_{sg}$-present and $Q_{sg}$-RCP 8.5). As such, it represents an intermediate future mean discharge magnitude, however, it does not correspond to any known projection scenario or a particular time period (Table 1).

### Diagnostics of basal melt rate and its drivers

For the investigation of ice shelf basal melt rate, its expression in the thermodynamic framework[62,67] of the primitive equation setting is recalled:

$$\rho_{fw} m_b L = -\rho_{sw} c_w \Gamma_T u^* \Delta T . \tag{1}$$

Here, $\rho_{fw}$ and $\rho_{sw}$ are the densities of freshwater and ocean water, respectively. $L$ is the latent heat of fusion of ice, $c_w$ is the specific heat capacity of ocean water, and $\Gamma_T$ is a non-dimensional heat-transfer coefficient. $m_b$ is basal melt rate, controlled primarily by friction velocity ($u^*$) and thermal driving ($\Delta T$), and where

$$u^* = \sqrt{C_D(u_w^2 + u_{res}^2)}, \qquad \Delta T = T_b - T_w , \tag{2}$$

$$T_b = \lambda_1 S_b + \lambda_2 + \lambda_3 P_b, \tag{3}$$

$$\rho_{fw} m_b S_b = -\rho_{sw} \Gamma_S u^*(S_b - S_w) . \tag{4}$$

Above, $C_D$ is the drag coefficient, $u_w$ is the velocity magnitude some distance away from the ice shelf-ocean boundary, and $u_{res}$ is the velocity magnitude of small sub-grid scale residual currents[68]. $T_b$, $S_b$, and $P_b$ are the potential temperature, salinity, and pressure at the ice shelf-ocean boundary, and $T_w$ and $S_w$ are the potential temperature and salinity some distance away from it. $\lambda_1$, $\lambda_2$, and $\lambda_3$, respectively, are the slope, intercept, and pressure coefficient of the liquidus, and $\Gamma_S$ is a non-dimensional salt-transfer coefficient. Note that $T_w$, $S_w$, and $u_w$ are obtained from the first (i.e., uppermost) $\sigma$-layer. Furthermore, $\Gamma_T$ and $\Gamma_S$ are application-specific, and are herein tuned according to the applied boundary conditions, Petermann Ice Shelf basal topography, and mixing schemes so as to be consistent with the contemporary observational estimates of the Petermann Ice Shelf basal melt (Supplementary Table A1). The drivers of basal melt rate ($m_b$), namely, friction velocity ($u^*$) and thermal driving ($\Delta T$) are used to investigate the impact of discharge on the summer mean basal melt rate.

### Changes in ice volume above basal channels over time

We calculate the changes in the annual mean channel-averaged ice volume above basal channels over time ($V_c(N)$) to determine the number of years ($N$) it would take for $V_c(N)$ to become 0 as

$$0 = V_i + (N \times V_a) - (N \times V_m), \tag{5}$$

$$\Rightarrow N = \frac{V_i}{V_m - V_a} \tag{6}$$

where $V_i$ is the initial (contemporary) ice volume above a channel expressed as $L_c$ (channel length [m]) $\times W_c$ (mean channel width [m]) $\times H_c$ (mean channel depth [m]). $V_a$ is the annual mean volume of ice advected downstream of the grounding line per year, expressed as $AR_c$ (mean advection rate [m/yr]) $\times W_c \times H_c$. $V_m$ is the annual mean volume of ice melted per year, expressed as $m_{bc}$ (annual mean channel-averaged basal melt rate [m/yr]) $\times L_c \times W_c$. Dimensions associated with each channel section are detailed in Table 2. Note that $W_c$ drops out of the calculation, and as such, is not detailed in Table 2.

## Data availability

Our study is based on numerical modelling, and we provide the model input files required to reproduce our simulations, which are publicly available on Zenodo (https://doi.org/10.5281/zenodo.12803093).

## Code availability

The open source code Finite Volume Community Ocean Model version 4.0 (FVCOM v4.0), augmented by both the ice shelf and sea ice modules, that has been used to conduct the numerical experiments is publicly available at: https://doi.org/10.5281/zenodo.15084570.

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

## Acknowledgements

This work was funded by the Swedish Research Council (VR) grant 2022-03718 awarded to N.K. Q.Z. was supported by the Research Council of Norway (RCN) projects 244319 and 295075. T.H. was supported by the RCN projects 314570 and 332635. We thank Donald Slater for providing the present-day and RCP 8.5 runoff estimates for the Petermann Glacier. The simulations were performed on resources provided by UNINETT Sigma2 - the National Infrastructure for High Performance Computing and Data Storage in Norway. The supercomputer Betzy was used under the project NN9824K.

## Author contributions

A.P., Q.Z., T.H. and N.K. contributed to the conceptualisation of the study. A.P. designed the experiments with technical input from Q.Z. and T.H. A.P. prepared and ran the numerical simulations. A.P. analysed the model output with scientific advice from Q.Z. and T.H. A.P. and N.K. wrote the manuscript, with contributions from Q.Z. and T.H.

## Funding

## Competing interests

The authors declare no competing interests.
