## [Transparent Peer Review file · Nature Communications]

Enhanced subglacial discharge amplifies Petermann Ice Shelf melting when ocean thermal forcing saturates

Corresponding Author: Dr Abhay Prakash

Version 0:

Reviewer comments:

Reviewer #1

(Remarks to the Author)

In this study the authors investigate the impact of changes in subglacial discharge on submarine melting of the Petermann Glacier ice shelf. Submarine melting is assumed to scale with thermal forcing and water velocity. Increases in subglacial discharge cause the thermal forcing to increase by drawing warm water into the sub-ice shelf cavity, but only to a point. At high values of subglacial discharge the thermal forcing saturates and melting is primarily affected by water velocity. This is an interesting result that has important implications for destabilization of the marine-terminating glaciers and ice sheets.

The oceanographic modeling and results are solid, but I am concerned about the implications for ice shelf stability. The ice shelf is not modeled (unless I am missing something), which means that the channels will only grow with time. Channels advect downstream and are eventually calved, and additionally I would expect changes in stress associated with channel enlargement to cause ice to flow toward the channels and at least partially offset melting. Neither of these things are included in the model, which causes me to question to what extent future runoff would destabilize the glacier.

I like this paper, although I do also think that it could benefit from some revisions to improve readability. I understand that Nature Communications wants/requires the methods to come at the end of the paper, but it felt to me that the methods was initially written as section 2 (before the results) and only later moved to the end of the paper, with the consequence that the results were difficult to interpret. For example, it took me awhile to understand what was meant by the various model scenarios. I think the introduction would benefit from a few sentences, or perhaps another paragraph, that would more fully describe the model set-up and simulations in layman's terms.

Finally, I found the use of acronyms and variables within text a bit cumbersome. I encourage the authors to eliminate them to the extent possible in order to make the paper more accessible. "basal melting", "subglacial discharge", "friction velocity", and "grounding line" are much easier to read than m_b , Q_{sg} , u^* , and GL. I don't see why the glacier and fjord names need to be abbreviated, and some acronyms are not used very often (such as GrIS and RG). There are many places where the text can be tightened up if space is an issue.

Specific comments:

Title: I feel that the title is a little misleading, although I agree with the sentiment, as this study is not actually modeling glacier flow or fracture propagation.

L20: How do "channels carve incisions"? Aren't the channels themselves products of melting?

L21: Are the thickness values stated for the areas immediately above the channels, or is this a statement about the average thickness of the ice shelf? Also, I assume that at least some of the thickness gradient is due to longitudinal extension, but this reads as though it is entirely due to submarine melting of channels.

L54: Please provide a definition of friction velocity here.

L113: What do you mean by "peaks"? Fig. 3 shows that the temperature continues to increase as you increase the subglacial discharge, just at a lower rate.

L117-L118: Similar to previous comment. Contribution to what? And do you just mean that the thermal forcing increases at a lower rate?

Fig. 3: What does “27.9” refer to?

L213-216: I'm not sure that I follow this. Are you essentially saying that the addition of cold freshwater will limit the thermal pump?

L370-373: How sensitive is the model to variability in subglacial discharge? Your simulated discharge follows a boxcar function, while we know that subglacial discharge follows more of a gaussian-like pattern during summer and is highly variable. I found myself wondering how important the variability is. If you have a few days or weeks of exceptionally high or low subglacial discharge, can that push the model into a different regime?

L419-430 (and Fig. 6): I found this description confusing. It's not clear to me what exactly you are calculating. Are you calculating the volume of ice above a channel? Maybe I am just hung up on the term “channel ice volume”, when to me the word channel implies a place where there is no ice.

Table 1: Is Q_{sg} -median just the average the present and RCP 8.5 scenarios? Or is that just coincidence?

(Remarks on code availability)

Reviewer #2

(Remarks to the Author)

The impact of subglacial water on ice shelf melt rates has been an understudied topic, but is now gaining prominence in the field with increasing recognition of its impact on ice sheet dynamics. This paper presents interesting insights into the mechanisms by which enhanced subglacial water flow might affect melt rates (i.e. shear-driven turbulence vs. thermal driving) and will therefore be of interest to the research community.

The study is well designed, using a very neat set of experiments to draw interesting conclusions, and the paper is well written: clear, and easy to follow. The figures are also well chosen, and very convincingly underpin the conclusions in the results and discussion sections, although they are not as polished and easy to read as I would like. I make some further suggestions for changes to the text below, but they are only minor. Overall, I think this is a very high quality contribution to the field.

Minor comments

Lines 3-4: “contemporary mass loss rates are underestimated”. Underestimated by whom? Observational studies, or ice sheet models?

Line 14: “ice volume of ca. 4m global mean sea level equivalent” I can't understand from this sentence which sectors of the ice sheet you are referring to. Please check the reference too – is Morlighem et al. (2017) not more appropriate here (BedMachine, see reference list below)?

Line 16: “Erstwhile” is not wrong here, but it's a rarely used word. You could consider replacing with “Formerly” or “Previously” to aid non-native speakers.

Line 21: “thinning it from ca. 600 m near the GL to ca. 100 m near the calving front.” Phrased this way, the sentence can be misinterpreted... do you mean “with ice thickness over the channels of 600 m near the GL to ca. 100 m near the calving front”?

Figure 1b: I suggest labelling the calving front with a date.

Line 34-36: “estimated area-averaged melt rate maximum of up to 170 m/yr” the phrase “area-average” initially implies that this was estimated over a large sector of the shelf. I suggest rephrasing slightly to make clear that this estimate was calculated only within the basal channel.

Line 38: “with warming expected to amplify” I suggest clarifying whether you mean atmospheric warming, ocean warming, or both.

Final paragraph of introduction (lines 43-59): It's unusual to see the results summarised so thoroughly at the end of the introduction. It might be more helpful to give a very brief summary of your experiment design instead e.g. from line 49 onwards something along the lines of “We performed a control run with zero Q_{sg} , and three test simulations designed to simulate present day subglacial conditions, increased Q_{sg} under high-end scenario climate warming, and an intermediate Q_{sg} scenario between the two.”

Paragraph beginning line 62: This paragraph jumps into discussion of modelled melt rates without describing the results clearly first, which means the reader has to work hard to catch up. I suggest an opening sentence or two describing the melt pattern in the control and preset day runs, before going on to compare them to observations.

Figure 2: You could make the model results larger and easier to see by removing unnecessarily duplicated color scales and axis labels (see also general comments on figures below).

Figure 3: The isopycnal label (27.9) is large and distracting (see also general comments on figures below).

Figure 5: The lines here can be a little hard to see – consider changing the color scheme to improve readability and increasing line thickness (see also general comments on figures below).

Line 151 & Figure 6: The two sets of location names are a bit confusing here (Sw, Sc, and Se vs. Lw, Lc, and Le). Are they both needed? Lw, Lc, and Le are barely mentioned in the text, and could perhaps be removed?

Figure 6: The pink dots in panel 6a are not described in the caption, and the symbols marking different lines in panel 6b are far too small to be readable (see also general comments on figures below)

Line 222: Should read "buoyant meltwater rises"

Discussion line 219-240 & Figure 6b: Given the very straight erosion trajectory in figure 6b, I initially thought you must have applied the summer value for Q_{sg} continuously over the whole year. Looking at the methods it seems that Q_{sg} does go to zero outside of the summer, so why don't we see a seasonal cycle in the channel volume? Presumably over winter the melt substantially decreases but the ice advection should change much less? I think the discussion section could do with a bit more context on seasonality, including how seasonally varying ocean temperatures affect erosion rates.

Line 266-267: There has been further recent work on this subject in Antarctica beyond your references, so I think the statement "insights into this process are limited to tacit reasoning" is a little unfair. I suggest reading and citing: Goldberg et al. 2023, Gwyther et al. 2023, and Pelle et al. 2024.

All figures: Please take a look at the text within your figures and make sure it meets minimum font size guidelines, but is not excessively large and distracting (as with some subpanel labels). Some axis labels and legends are very small indeed. In general, using consistent font across all figure will also improve perception of your figure quality. You could also improve clarity of figures 2, 3 & 4 by indicating on colour scales where a variable is absolute or relative (e.g. "Salinity" in panel 4a and "Change in salinity" in panel 4b, use a delta symbol if space is tight).

References

Goldberg, Daniel N., Andrew G. Twelves, Paul R. Holland, and Martin G. Wearing. "The non-local impacts of Antarctic subglacial runoff." *Journal of Geophysical Research: Oceans* 128, no. 10 (2023): e2023JC019823.

Gwyther, David E., Christine F. Dow, Stefan Jendersie, Noel Gourmelen, and Benjamin K. Galton-Fenzi. "Subglacial freshwater drainage increases simulated basal melt of the Totten Ice Shelf." *Geophysical Research Letters* 50, no. 12 (2023): e2023GL103765.

Morlighem, Mathieu, Chris N. Williams, Eric Rignot, Lu An, Jan Erik Arndt, Jonathan L. Bamber, Ginny Catania et al. "BedMachine v3: Complete bed topography and ocean bathymetry mapping of Greenland from multibeam echo sounding combined with mass conservation." *Geophysical research letters* 44, no. 21 (2017): 11-051.

Pelle, T., J_S Greenbaum, S. Ehrenfeucht, C_F Dow, and F_S McCormack. "Subglacial discharge accelerates dynamic retreat of aurora subglacial basin outlet glaciers, East Antarctica, over the 21st century." *Journal of Geophysical Research: Earth Surface* 129, no. 7 (2024): e2023JF007513.

(Remarks on code availability)

I have checked that the code is accessible, but having never used the FVCOM model I can't say whether it is sufficient to reproduce the results

Reviewer #3

(Remarks to the Author)

Petermann glacier in NW Greenland is rapidly losing mass and contributing strongly to modern global sea-level rise. Ice flow from Petermann is currently slowed by the presence of a ~30km floating ice tongue which buttresses flow. With ocean warming across the last ~two decades, that ice shelf has thinned, and future breakup of the ice shelf is possible. This paper uses numerical modeling to estimate the role that future atmospheric warming may have on ice shelf break-up. Where atmospheric temperatures warm, ice surface melt increases. This surface melt is able to reach the subglacial environment at moulins, and then flows into fjords, where it influences ocean circulation and melting. Using a coupled ice-shelf/ocean/sea ice model, this work finds that subglacial discharge causes enhanced subglacial melt, particularly within subglacial channels. Enhanced subglacial discharge acts to accelerate melting by

Generally speaking, this work is timely, impactful, and worthy of publication in *Nature Communications* with revisions. The majority of my suggested improvements below are easy to implement or require simple clarification. I have three more major concerns, one of which is easily addressed (point two below) by additional discussion or re-framing of the work. The other two points may require additional modeling work.

My primary issue with the work is that it frames itself as an improvement on previous modeling work because it uses more realistic domain geometry. However, the domain geometry used in these model runs is not sufficiently explained. Maybe I missed it, but I cannot find where the authors got ice shelf draft geometry from. I assume it is a radar flight line? From what year and what instrument? More importantly, the 'improved' seafloor geometry is a modified form of BedMachine, but the improved seafloor geometry is never shown. What is the cavity shape? Why were the modifications made to BedMachine? It's also worth noting that — to the best of my knowledge — BedMachine has no data for the ice shelf cavity. It simply interpolates/kreigs between radar flightlines (and the mass-conservation estimate between them) for grounded ice and the downstream seabed bathymetry from ships. So really very little is known about the actual shape of the seafloor for the ocean cavity that is modeled here. To what degree does uncertainty in this cavity geometry impact the modeling results presented here? I recognize that a full suite of modeling runs across a range of seafloor geometries is likely beyond the scope of this work, but even a single run that presents a modified geometry would be useful for determining the sensitivity of the model results to the unknown cavity geometry.

The ice geometry used in the model runs was a pre-2010 ice geometry (pre large calving event) but the atmospheric and ocean forcing was for later years (I believe). Given the large change at the glacier in 2010/12 with the calving, what uncertainty is baked in to modeling results as a product of not using the modern ice configuration? Why was this choice made? Some discussion is warranted here.

A third concern I have is a lack of discussion of the role of tidal pumping within the grounding zone environment and its potential impact on grounding zone melt rates. Recent work shows that the high levels of subglacial melt at the Peterman grounding zone might be due to seawater movement with the tides (Gadi et al (2023, GRL) and Ciraci et al (2023, PNAS)). The sub-ice shelf community has recently been very focused on the role of tidal seawater pumping and potential melting inland of the grounding zone by this seawater. There is a rich literature describing both the theory (Walker et al., 2013), modeling seawater intrusions (Robel et al., 2022), and observations of intrusions (Rignot et al., 2024). For a nice overview and intro to this subject, Parizek 2024 (GRL) summarizes the current state of thinking of tidal pumping and its role at Peterman. I think it is a significant omission that this work is not considered in this paper, as tidal pumping across the grounding zone may be a large source of melt (and water flow) there. Again, I recognize that a full suite of model runs that incorporate tidal pumping across the grounding zone are beyond the scope of the work presented here, but from my prospective, not even mentioning the role of tidal pumping across the grounding zone is a significant omission of this work: if nothing else, the potential role of this process should be discussed in the introduction and discussion.

Line-by-line and more minor comments

The abstract leaves me confused about geometry: a mechanism is described that exacerbates melt at the crest of channels (which I assume is the highest elevation point where the ice is the thinnest) but then the next sentence says that this melt will complete erode channels, which sounds like a flattening of channels. Does 'erode channels' here mean a complete incision of the shelf? If so, I would change language to 'We estimate that sustained intensified summer melting over a decade may undermining the stability of PGIS by allowing channels to cut entirely through the ice shelf'

I found the reliance on terms (e.g. Qsg for subglacial discharge) to be occasionally difficult — for the casual reader, this paper presents a significant number of terms that must be held in memory to derive meaning. I might encourage the authors to be more selective in what they make terms for (is it a word-count problem to just say 'subglacial discharge?'), but I recognize that this is a style choice.

8-11 the two clauses of this sentence appear to be jarringly unrelated, and generally I have a hard time following the logic structure of 8-14

13 Does the Petermann shelf do much in the way of buttressing? I thought that there was minimal grounded ice velocity response to the 2010 and 2012 calving events, which would suggest perhaps not. There may be modeling work that has been done that quantifies this. If so, worth citing and quickly mentioning here (given that the ice shelf community has recently been surprised by the lack of buttressing at other ice shelves like eastern Thwaites).

21 thinning the ice shelf from..

21 is the 600m/100m geometry presented for *within* the basal channels or regions outside of the channels?

35-36 slightly confused by geometry here, is this an integrated melt rate for the region from GL to 16km downstream? What time of year were those observations made (i.e. sentence before says that melt rates triple in summer).

My sense is that this is a controversial statement, so I would soften indicates to suggests

45-46 reword to improved sub-ice shelf bathymetry using BedMachine3 (Morlighem et al., 2017). As written it reads as if Morlighem is a prior ice-ocean model that these results will be improving upon.

does estuarine circulation start (i.e. at present time) NOT saturated and then becomes saturated at a point in time (suggesting regime change at that point?) Clarify here. I also don't have a great sense of what saturated circulation means.

58-59 at risk of repeating myself, I still wish for a different term than 'complete erosion of the channels' — this makes it sound like channels are going away. What about 'complete erosion of the ice shelf within the channels'?

62-64 The numbers for each of these values (your melt, remotely sensed Melt, in-situ measured melt) really need to be shown here... 'they agree with remote sensing/field data' is not sufficient - give a detailed comparison.

95 One more sentence of summary/overview would be nice here - a less technical 'this is what this means'

97-129 Missing from this section is a physical description of the mechanism responsible for the saturation of the thermal pump. Is it increased (cold) meltwater pooling at locations in the cavity?

141-142 This is not intuitive for me: isn't meltwater quite fresh and cold (ie minimal thermal capacity, minimal melt)? Why is there highest melt in areas of meltwater trapping, I would expect melt to be *low* there because of cold temps and increased stratification. Explain.

147-148 I would expect that meltwater would pool at the base of channels and melt would be slowest at the channel apex. Why is this not the case? I believe this is what ROV submersible Icefin observed under Thwaites (e.g. Schmidt et al., 2023 and the subsequent work of Peter Washam) - high melt rates on channel sides and slow melt at channel apex.

Given that old (pre-2010) ice draft geometry is used here with modern (2024) melt rates, are these numbers realistic? How would they change if modern ice draft geometry was used?
Reword: The modeling work of Hill
I would suggest adding one more sentence here about WHY this has a significant bearing on PG stability ('this is significant' feels like a bit of a punt without more context — however I frequently do it in my writing as well!).

189-196 THIS! Cool. I would consider adding these ideas to the abstract and introduction.

288-292 This is a key insight (no edits, just applauding)

299-302 I'd push back on this — I see how improving our knowledge of grounding zone geometries is being prioritized by funding agencies and research groups. Your study highlights the need for such work, but saying that this work isn't being undertaken is an overstatement. Many good groups are doing targeted geophysical work to characterized the grounding zone environment, there are just major logistical and financial barriers to doing this on a continent-wide scale. Maybe re-cast this argument as a push for even more targeted airborne and ground-based geophysical campaigns to describe the grounding zone environment where possible.

304 worth throwing in a citation for BedMap3 here too

309 The argument that you're making — that meter-scale features are essential to capture in content-wide datasets — is admirable, but we're just so far from that reality in this moment. More useful, I wonder, might be to suggest ways to parameterize the meter-scale effects to make our continent-scale datasets more usable at this time? Realistically we are not going to send Icefin, Autosub, or Ran under every ice shelf to map every feature.

312 Worth noting though that modeling work has shown that distributed Qsg produces more subglacial melt than channelized Qsg (Slater et al., 2015 GRL)

337 is this seafloor geometry of ice base geometry?

340 include a note as to whether or not this geometry is reasonable, with a citation... I.e. how wide are the channels at various points across the ice shelf— is a 200m grid sufficient to capture their geometry?

Figure 1

To improve comprehension without needing to read long legend, I would suggest adding text to 1d that signifies that dark regions are deep water column and lighter regions are integrated

Figure 5

What is the source of data for the ice draft?

(Remarks on code availability)

Link works and code is available, however I have not attempted to compile it.

Version 1:

Reviewer comments:

Reviewer #1

(Remarks to the Author)

The authors have sufficiently addressed all of my prior concerns. I find this to be an interesting paper that advances our understanding of glacier-ocean interactions. However, I have three related editorial comments regarding the paper that are in the spirit of trying to increase its impact.

1. To me, the real impact of this paper is showing how subglacial discharge (and changes in subglacial discharge) affects submarine melting of ice shelves and not simply that increases in subglacial discharge will affect the stability of Petermann Ice Shelf. Previous studies, such as the work by Slater and Straneo, as well as that of Motyka et al. and probably many others, have shown that changes in subglacial discharge can lead to large changes in melt rates. What I take away from the paper is that the effect of subglacial discharge (i) at low fluxes is to draw in warm water and breakdown the boundary layer, whereas (ii) at high fluxes the temperature effect becomes saturated and further increases in discharge essentially affect the boundary layer. Much of the paper focuses on this. The uncertainty in the model topography, location and number of discharge outlets, and lack of ice dynamics model causes me to question the assertion that future runoff may destabilize Petermann Ice Shelf. (Not that I disagree, I just don't think that's really what this paper shows.) Personally, I think the deeper understanding of how runoff affects submarine melting to be more interesting and robust than whether or not one particular glacier will destabilize, but maybe that is just my personal bias coming through.

2. The number of panels in figures 2 and 3, and the long figure captions, make them somewhat difficult to digest. I don't think the figures need to be changed all that much, but I do think it would be helpful if they were labeled somehow so as to make it clear which panels correspond to the control run and each of the runoff scenarios (as was done for figure 4)

3. I think the writing can be tightened up and made more succinct in many places. As just an example, the paragraphs in the discussion section are quite long (almost a page long in draft mode). This can make it difficult to determine (or remember!) the main point of each paragraph. There are quite a few clauses and phrases throughout the paper that are either unnecessary or that could be shortened to improve readability.

(Remarks on code availability)

Reviewer #2

(Remarks to the Author)

The authors have been very thorough in their response to reviewers comments, and have made substantial changes to the text and figures. I feel that all of my comments have been very well addressed.

(Remarks on code availability)

Reviewer #3

(Remarks to the Author)

The authors have sufficiently met my concerns in their revised manuscript and I have no further substantive suggestions. I

would commend the authors on their complete and thorough addressing of all (not just my) reviewer suggestion. The updated manuscript has improved readability and the key points are more clearly made.

The dropping of terms in favor of words (e.g. Qsg → subglacial discharge) improves readability. The added 'in sum' at the end of the results section is helpful. I appreciate the new concluding sentence of the manuscript and agree — the interplay between surface melt and seawater intrusions at the grounding zone are a real research frontier right now.

Minor notes for the authors:

2 First sentence scans somewhat awkwardly, suggest rephrasing to clarify

'studies'
sea-ice ice-shelf (I think?). Is there a coupled ice flow model here? In looking to your methods (438) it looks like there is a coupled ice shelf model, but I would love one more sentence here describing it. For those not familiar with the Zhao ice shelf model, what is it? What ice rheology is used? Is it a higher-order model?

88 grounding line are

135-148 the continued reliance on Qsg, mbpgis, U*300m and other variable-based terminology really does make reading this difficult for someone unfamiliar with the work. I know one round of this has already been done, but I would suggest one more round of 'decreasing variable terminology' to improve readability.

estimates (somewhat awkward flow here)
are currently being experienced

284-89 maybe I just need an afternoon coffee, but this sentence is far too long to be digestible without multiple readings.

305 Idle curiosity: does your model allow for migration of channels, as has been observed on Thwaites and elsewhere? Do you observe migration of channel location at all? E.g. Chartrand et al., 2020. I wonder if this could delay Petermann collapse.

Kiya Riverman (Reviewer #3)

(Remarks on code availability)

REVIEWER COMMENTS

Reviewer #1 (Remarks to the Author):

In this study the authors investigate the impact of changes in subglacial discharge on submarine melting of the Peterman Glacier ice shelf. Submarine melting is assumed to scale with thermal forcing and water velocity. Increases in subglacial discharge cause the thermal forcing to increase by drawing warm water into the sub-ice shelf cavity, but only to a point. At high values of subglacial discharge the thermal forcing saturates and melting is primarily affected by water velocity. This is an interesting result that has important implications for destabilization of the marine-terminating glaciers and ice sheets.

We appreciate your time and effort in reviewing our manuscript. We are glad to hear that you find our work to be important and of value to the scientific community. We also appreciate the constructive feedback and have revised our manuscript following your suggestions. Please note that line numbers provided below refer to the *track changes* document (appended to this report for convenience), where new text appears in blue and removed text is marked with red strikethrough.

Firstly, we would like to take this opportunity to inform you that during the revision process, we noticed a (minor) mistake in our calculation of the drivers of melt, namely, thermal driving and friction velocity. While the updated numbers are slightly different, we note that this does not impact any of the findings of our study, and the revised figures remain qualitatively identical. We humbly request to proceed with these corrected values, and sincerely apologize for the oversight and for any inconvenience this may cause. Please see the revised percentages in Lines 133 – 134, 167 – 170, 184, 195, and 227, and Figures 1(d), 2(e – l), and 5(c,d).

Please find our detailed (point-by-point) responses below.

The oceanographic modeling and results are solid, but I am concerned about the implications for ice shelf stability. The ice shelf is not modeled (unless I am missing something), which means that the channels will only grow with time. Channels advect downstream and are eventually calved, and additionally I would expect changes in stress associated with channel enlargement to cause ice to flow toward the channels and at least partially offset melting. Neither of these things are included in the model, which causes me to question to what extent future runoff would destabilize the glacier.

Thank you for raising this concern. You are right, we have a static ice shelf geometry (now clarified in the revised text). Indeed, ideally there would be downstream advection and calving,

and secondary ice-flow which would offset the channel growth. Mean advection rate of ca. 1 km/yr has been reported for Petermann with a seasonal modulation of ca. 0.1 km/yr (Muenchow et al., 2012). However, to our knowledge, its evolution in a future warming climate has not yet been investigated. Therefore, in our study, we envelop the contemporary observational estimates from Muenchow et al. (2012) with ± 0.5 km/yr. Ice-flow modelling study from Wearing et al. (2021) has shown that secondary flow is insufficient to significantly delay the breakthrough of basal channels in ice shelves that are thinner than 400 m. We acknowledge that the Petermann ice shelf draft is deeper than 400 m in certain sectors (e.g., southeast sector near the grounding line). However, with the mean-state analysis (mean advection and basal melt rates) that was feasible to implement here, and without a high-resolution (at the scale of the topographical data) ice-flow model to capture the nuanced dynamical variability, it is difficult to ascertain how localized small-scale channel closures/openings would impact the wider ice dynamics, and as such, the long-term stability of the glacier. Coupled ice-flow - ocean models (frameworks of which have been recently developed, e.g., Zhou et al. (2024)) or offline coupling of an ice-flow model with an ocean model which comprehensively resolves basal melt rates in response to ocean dynamics is indeed a subject of future work in this field. We have revised our discussion to elucidate the context and limitations of our calculations and measures that need to be taken to overcome these limitations in the future. Please see lines 366 – 380.

I like this paper, although I do also think that it could benefit from some revisions to improve readability. I understand that Nature Communications wants/requires the methods to come at the end of the paper, but it felt to me that the methods was initially written as section 2 (before the results) and only later moved to the end of the paper, with the consequence that the results were difficult to interpret. For example, it took me awhile to understand what was meant by the various model scenarios. I think the introduction would benefit from a few sentences, or perhaps another paragraph, that would more fully describe the model set-up and simulations in layman's terms.

Thank you for bringing this to our notice. We have addressed it by amending the Introduction chapter to include more information regarding both the model and experiment setup. Please see lines 59 – 100.

Finally, I found the use of acronyms and variables within text a bit cumbersome. I encourage the authors to eliminate them to the extent possible in order to make the paper more accessible. "basal melting", "subglacial discharge", "friction velocity", and "grounding line" are much easier to read than m_b , Q_{sg} , u^* , and GL. I don't see why the glacier and fjord names need to be abbreviated, and some acronyms are not used very often (such as GrIS and RG). There are many places where the text can be tightened up if space is an issue.

Thank you for the suggestion. To our knowledge, we are within the word limit guidelines provided by the journal. Therefore, we have removed the abbreviations for “basal melting”, “subglacial discharge”, “thermal driving”, “friction velocity”, “grounding line”, “Greenland Ice Sheet”, and “Ryder Glacier”, as well as all other glacier and fjord names from the text for clarity.

Specific comments:

Title: I feel that the title is a little misleading, although I agree with the sentiment, as this study is not actually modeling glacier flow or fracture propagation.

This is true, we do not model the glacier flow or fracture propagation. Hence, we preferred the use of “may destabilize”, as we posit based on our ice volume change calculations that the high basal rates modelled in the channels (ca. 200 m/yr) in the RCP 8.5 scenario could be unsustainable for the ice shelf to withstand for longer periods of time.

L20: How do “channels carve incisions”? Aren’t the channels themselves products of melting?

Thank you for identifying this mistake. We have fixed it (see below) together with the comment on L21.

L21: Are the thickness values stated for the areas immediately above the channels, or is this a statement about the average thickness of the ice shelf? Also, I assume that at least some of the thickness gradient is due to longitudinal extension, but this reads as though it is entirely due to submarine melting of channels.

The statement was regarding the (average) thickness of the ice shelf – it ranges from 400 – 600 m deep near the GL to 50-100 m near the calving front. You are right, by putting it together with the channel description, it appears to come across as being entirely driven by submarine melting. We have fixed this in the revised manuscript. Please see lines 22 – 28.

L54: Please provide a definition of friction velocity here.

We have added a definition of friction velocity. Please see lines 95 – 96.

L113: What do you mean by “peaks”? Fig. 3 shows that the temperature continues to increase as you increase the subglacial discharge, just at a lower rate.

You are correct, thank you for identifying this mistake. “Peaks” is not the right word. We have changed this in the revised text. Please see lines 177 – 179.

L117-L118: Similar to previous comment. Contribution to what? And do you just mean that the thermal forcing increases at a lower rate?

Indeed, this is a more appropriate formulation. We have adapted it in the revised text. Please see lines 181 – 183.

Fig. 3: What does “27.9” refer to?

Thank you for noticing this. We forgot to describe it in the caption. It has now been added.

L213-216: I’m not sure that I follow this. Are you essentially saying that the addition of cold freshwater will limit the thermal pump?

No, that is not what we intend to say, we apologize for the lack of clarity. We wanted to conclude the paragraph by presenting a broad discussion regarding the relative contributions of rising ocean temperatures (Atlantic Water reaching Petermann by the end of this century) vs increased subglacial discharge in driving increased melting in a warmer climate. We have made revisions to clarify our argument. Please see lines 310 – 327.

L370-373: How sensitive is the model to variability in subglacial discharge? Your simulated discharge follows a boxcar function, while we know that subglacial discharge follows more of a gaussian-like pattern during summer and is highly variable. I found myself wondering how important the variability is. If you have a few days or weeks of exceptionally high or low subglacial discharge, can that push the model into a different regime?

Thank you for raising this point, as we have given it due consideration as well. We are currently in the process of setting up the Glacier Drainage System Model (GlaDS) for the Petermann catchment to determine both the magnitude and location(s) of emergence of subglacial discharge. The gaussian-like profiles of subglacial discharge retrieved from GlaDS reveal considerable seasonal and inter-annual variability. However, this work is still in its nascent stages, and we are yet to analyze the corresponding ocean model simulations. It is difficult to quantify the sensitivity without running appropriate experiments using such a setup, however, we have now included a discussion around it, and caveated its speculative nature, suggesting the need for further studies. Please see lines 267 – 275.

L419-430 (and Fig. 6): I found this description confusing. It’s not clear to me what exactly you are calculating. Are you calculating the volume of ice above a channel? Maybe I am just hung

up on the term “channel ice volume”, when to me the word channel implies a place where there is no ice.

This is correct, we are calculating the volume of ice above a channel. Thank you for bringing this to our attention. We have replaced all occurrences of the phrase “channel ice volume” with “ice volume above a channel”. The section is renamed to “Changes in ice volume above basal channels over time”. Please see, for e.g., lines 611 – 624.

Table 1: Is Q_{sg} -median just the average the present and RCP 8.5 scenarios? Or is that just coincidence?

It is just the average of the two scenarios. Subglacial discharge from Slater et al. (2020) was only available for present and the RCP 8.5 scenario. We generated Q_{sg} -median manually by taking the median of the present and RCP 8.5 values. We have made adjustments to the text to reflect this better. Please see lines 580 – 585.

Future runoff increase may destabilize Petermann ice shelf

Abhay Prakash^{1,2,*}, Qin Zhou³, Tore Hattermann^{4,5}, and Nina Kirchner^{1,2}

¹Department of Physical Geography, Stockholm University, Stockholm 10691, Sweden

²Bolin Centre for Climate Research, Stockholm University, Stockholm 10691, Sweden

³Akvaplan-niva, Tromsø 9296, Norway

⁴Norwegian Polar Institute, Tromsø 9296, Norway

⁵Department of Physics and Technology, University of Tromsø, Tromsø 9019, Norway

*Corresponding author, abhay.prakash@natgeo.su.se

Abstract

Increased basal melting of the Petermann ~~Glacier-Ice Shelf (PGIS)-ice shelf~~ is typically attributed to rising ocean temperatures. Although subglacial discharge (Q_{sg}) has been shown to intensify melt, the mechanisms behind this increase and their evolution with increasing Q_{sg} discharge in a warmer climate remain unresolved. Using a 3-D numerical regional ice shelf-ocean model centered on the Petermann Fjord, we show that heightened Q_{sg} discharge under the RCP 8.5 scenario leads to more than threefold increase in summer mean melt when averaged over the deeper (>300 m) drafts compared to conditions without Q_{sg} discharge. Notably, we identify a regime change in heat flux efficiency within the ~~PGIS-cavity-when- Q_{sg} -ice shelf cavity when discharge~~ exceeds the current peak summer value. Here, thermal driving saturates, and ~~Q_{sg} -intensified discharge-intensified~~ currents increase melt by enhancing shear-driven turbulent mixing across the ~~PGIS-ocean-ice shelf-ocean~~ boundary layer. Increases in melt are most profound at the crests of the basal channels, where vigorous meltwater confluence exacerbates friction velocity. We estimate that sustained intensified summer melting over a decade may ~~completely erode the channels, undermining the stability of PGIS, undermine the stability of the ice shelf by allowing channels to cut entirely through it. Challenging conventional attributions of increased basal melting of the ice shelf to ocean warming alone, our results demonstrate how atmospheric warming exacerbates ocean-driven melt processes, and is likely to play a dominant role in amplifying future basal melt.~~ Considering the impact of the channelized basal morphology of ~~PGIS-the ice shelf~~ on the spatial heterogeneity of melt, and projected increases in Q_{sg} discharge, we posit that our results have wider implications for similar 'warm cavity' environments.

Introduction
The Greenland Ice Sheet (~~GrIS~~) is currently the largest contributor to global mean sea
level rise (Bamber et al., 2018; Fox-Kemper et al., 2021), and as ~~contemporary mass loss~~
~~rates observational estimates of recent mass loss~~ are underestimated (Greene et al., 2024),
they are hence likely lower bounds for rates to be observed in decades to come (Briner
et al., 2020; Goelzer et al., 2020). The northern sector of the ~~GrIS-Greenland Ice Sheet~~
hosts several of the last remaining marine terminating outlet glaciers featuring floating
ice tongues (also called ice shelves) that exert buttressing forces to the inland ice. During
2001 ~~—~~ 2021, dominating factors ~~for-of~~ mass loss from these ice shelves were (in de-
creasing order) basal melt, calving, and negative surface mass balance, ~~while between,~~
~~Between~~ 2003 and 2010, collapses were observed at Hagen Bræ, C.H. Ostenfeld glacier,
and Zachariæ Isstrøm (Millan et al., 2023). Dynamic thinning, retreat and even collapse
combined with perturbations of the outlet glacier's grounding ~~lines-(GL)-line~~ can lead to
a loss of backstress and accelerated mass loss from the ~~adjacent-GrIS-interior-sectors~~
~~northern Greenland Ice Sheet sectors~~ (Hill et al., 2018) holding an ice volume of ca.
~~4m-4 m~~ global mean sea level equivalent (~~Mouginot et al., 2019~~)(Morlighem et al., 2017
: Mougnot et al., 2019).
Along the Nares Strait coast of the northern ~~GrIS, Petermann Glacier Ice Shelf (PGIS)~~
~~terminates into Petermann Fjord (PF). Erstwhile Greenland Ice Sheet, Petermann ice~~
~~shelf terminates into the Petermann Fjord. Formerly~~ the longest ice shelf in the northern
hemisphere at ca. 80 km in length, two recent massive episodic calving events in 2010
and 2012 reduced its length by ca. 30 km (Münchow et al., 2014). ~~The ice shelf draft~~
~~thins from ca. 400 – 600 m near the grounding line to ca. 50 – 100 m near the calving~~
~~front. Evenly spaced across its 15-20-15 — 20 km width are 1-2-1 – 2 km wide basal~~
~~melt channels that run along its length (Rignot and Steffen, 2008). These channels carve~~
~~incisions that are incise~~ several-100 m deep into ~~its draft, thinning it from ca. 600 m~~
~~near the GL to ca. 100 m near the calving front. the draft~~ (Rignot and Steffen, 2008
28). Warming of Atlantic water in ~~PF by $\sim 0.3^\circ\text{C}$~~ ~~Petermann Fjord by ca. 0.3°C~~ has been
observed since 1960, with much (~~\sim ca. 0.2°C~~) of the warming occurring in the last
two decades (Millan et al., 2022). Average basal melt rates (~~m_b~~) ~~at the PGIS-GL, at \sim~~
~~at the Petermann ice shelf grounding line, at ca. 600 m below sea level, are currently~~
reported to exceed 17 m/yr, leading Millan et al. (2023) to suggest that increasing ice
shelf basal melt is following increasing ocean temperatures in ~~GrIS fjords. Greenland~~
~~Ice Sheet fjords. Numerical modelling study and satellite observations have revealed~~
~~that highest basal melt rates occur in the kilometer-size grounding zone of the glacier,~~
~~driven by tidally modulated intrusions of seawater~~ (Ciraci et al., 2023; Gadi et al., 2023
37). Seasonal and climate-warming driven long-term changes in regional sea ice cover are
38 also known to enhance ~~m_b basal melting~~. According to model results by Shroyer et al.
(2017) and Prakash et al. (2023), elevated temperatures and an energetic ocean circula-
tion under ~~PGIS the ice shelf~~ driven by variations in Nares Strait sea ice mobility and
thickness increased ~~m_b basal melting~~ by a factor ranging between 1.2 and 1.7. Also, ~~m_b~~
~~melting~~ can be enhanced by routing of glacier surface melt to the glacier bed, and fur-
ther to the ~~GL grounding line~~ where it enters the fjord as subglacial discharge (~~Q_{sg}~~). ~~A~~
~~higher Q_{sg} during summer has been suggested to triple m_b at PGIS. Modelling study~~
~~suggests that a higher subglacial discharge during summer (June – August) triples basal~~
~~melt at Petermann ice shelf compared to winter (Cai et al., 2017)(Cai et al., 2017). Fur-~~
~~thermore, within the central basal channel, for the region extending from the grounding~~
~~line to 16 km seaward of it, between August and December 2015, Washam et al. (2020)~~
estimated area-averaged melt rate maximum of up to 170 m/yr ~~, upstream of a location~~
~~16 km from the GL in the central basal channel,~~ which were consistent with episodes
of enhanced ~~Q_{sg} driven discharge driven~~ sub-ice shelf currents. This ~~indicates that Q_{sg}~~
~~suggests that subglacial discharge~~ may play a more important role in modulating ~~PGIS~~
~~Petermann ice shelf~~ basal melt than sea ice driven changes in ocean circulation. However,
with ~~atmospheric and oceanic~~ warming expected to amplify in the Arctic by the end of
the century (Muntjewerf et al., 2020; Shu et al., 2022), a salient question remains unan-
swered: How would ~~PGIS Petermann ice shelf~~ basal melt respond to projected increases
in ~~Q_{sg} under a future warming discharge under a warmer future~~ climate, and what would
be the mechanisms driving this response?
~~Here~~ ~~In this study~~, we extend a high-resolution unstructured grid 3-D ocean-sea ice-
ice shelf regional model setup (Prakash et al., 2022, 2023) centered over ~~PGIS and PF~~
~~to include Q_{sg} at the GL. Novel features include moreover an improved (with respect to~~
~~Morlighem et al., 2017) the Petermann ice shelf and Petermann Fjord to include subglacial~~
~~discharge at the grounding line (Fig. 1). The fjord bathymetry and the ice shelf draft~~
~~are derived from BedMachine v3 (Morlighem et al., 2017). Modifications have been~~
~~applied to the topographical datasets (discussed further below) to amend an inaccurately~~
~~represented~~ sub-ice shelf ~~bathymetry and a realistic PGIS basal topography. With these,~~

Figure 1. (a) Map of our regional **PGIS-PF-Petermann ice shelf-Petermann Fjord** model domain (in a wider context shown in the inset in the upper left corner) which follows the Greenland and Ellesmere Island coastlines, and extends into the Lincoln Sea and Baffin Bay (red lines). The mesh size in the model domain varies regionally, as exemplified by the three green-framed boxes. **PF-Petermann Fjord** is surrounded by a purple polygon, detailed in panel (b). (b) **PGIS-Petermann ice shelf** extent (white outlines), **from CE 2009**, overlain on water column thickness (WCT) in **PF-Petermann Fjord** and adjacent parts of Nares Strait. Cyan line marks **GL-the grounding line**, where discharge (detailed in panel (c)) is injected into the model. Red line marks the location of the transect over which the temperature, salinity, and flow along the fjord are investigated. The magenta star denotes the node ca. 10 km from **GL-the grounding line** at which the vertical profile of temperature and salinity are shown. Blue and yellow lines, respectively, represent the transects across which overturning circulation and diagnostics for meltwater **trappings-convergence** are shown. (c) Discharge (Q_{sg}) scenarios, 1950–2100 CE, obtained by forcing MAR with five CMIP5 AOGCMs (Barthel et al., 2020; Slater et al., 2020). Shaded vertical stripes with blue and red stippled outlines denote the periods over which Q_{sg} -present (CE 1995–2014) and Q_{sg} -RCP 8.5 (CE 2081–2100), respectively, are averaged over (cf. Table 1). Blue, magenta, and red horizontal stippled lines correspond to the Q_{sg} -present ($372 \text{ m}^3/\text{s}$), Q_{sg} -median ($1563 \text{ m}^3/\text{s}$), and Q_{sg} -RCP 8.5 ($2754 \text{ m}^3/\text{s}$) discharge magnitudes, respectively. (d) A synoptic graphical summary of the results showing basal melt rate (m_b), and its drivers, namely, thermal driving (ΔT) and friction velocity (u^*). **Common to all plots:** The **total-length-of-bars (dark colors) show corresponding variable-averaged spatially-over the deeper (>300 m) draft regions of PGIS, while light-colored counterparts represent spatial averages over the entire PGIS base.** The discharge type categories (X-axis) correspond to the experiment design, cf. Table 1.

~~we resolve the~~ water column thickness, and furthermore, to implement a 540 – 610 m
deep inner sill situated ca. 25 km from the grounding line, retrieved using aerogravity
data from Operation IceBridge (Fig. E1; Tinto et al., 2015). Open boundaries in the
Lincoln Sea and Baffin Bay are used to provide lateral ocean boundary conditions. These
include temperature, salinity, and sea water velocities derived from A4 ROMS (4 km
pan-Arctic Regional Ocean Modeling System grid; Hattermann et al., 2016), and merged
sea surface elevation derived using A4 ROMS and 5 km AOTIM (Arctic Ocean Tidal
Inverse Model; Padman and Erofeeva, 2004). Data from the 5.5 km polar (p) version
of RACMO2.3p2 (Regional Atmospheric Climate Model; Noël et al., 2019) is used to
compute the momentum, heat, and freshwater fluxes. Sea ice concentration and thickness,
bulk ice salinity, and sea ice velocities are obtained from A4 ROMS – CICE (Community
Ice Code; Hunke et al., 2015). Using this setup, we perform a control run with zero
subglacial discharge, and three test runs designed to simulate the response of the cav-
ity circulation and ~~spatial melt rate variability arising due to the complex PGIS basal~~
~~morphology (Rignot and Steffen, 2008) in unprecedented detail. Our findings evidence~~
~~that Q_{sg} ice shelf basal melt rates to present day subglacial discharge, increased discharge~~
~~under high-emissions ‘RCP8.5’ climate warming scenario, and an intermediate discharge~~
~~scenario between the two (Table 1). We show that discharge increases basal melting by~~
(a) accelerating a quasi-estuarine circulation (Sciascia et al., 2013) which increases the
advective heat transport into the cavity, and by (b) enhancing the shear-driven mixing
across the ~~PGIS-ocean ice shelf-ocean~~ boundary layer. The latter exerts a dominant control
over the increase in ~~m_b seen for larger Q_{sg}~~ basal melting seen for higher discharge
under future warming scenarios while the former (advective heat transport) saturates,
suggesting a shift towards a regime where melt increases are primarily driven by increases
in friction velocity (u^*). ~~Within the basal channels, we find that the buoyant meltwater~~
~~accelerates along the steepest slopes of the ice shelf adjacent to the crests of the channels~~
~~which increases u^* and m_b , both of which peak at the crest. We estimate that the high~~
~~mean melt rates driven by summer Q_{sg} , if allowed to persist for a decade, may lead to~~
~~the complete erosion of the channels i.e., velocity scale representing shear stress in the~~
~~boundary layer). Our findings challenge the prevailing focus on increased oceanic heat~~
~~forcing as the primary driver of basal melting at the Petermann ice shelf, revealing that~~
~~heightened subglacial discharge significantly enhances (channelized) basal melt. This~~
~~suggests that atmospheric warming-induced excitation of oceanic melt processes will~~
~~critically influence the stability of the ice shelf.~~
Results
Response of cavity circulation and basal melt to contemporary discharge
~~Our mean modelled melt rates and their spatial variability, and furthermore, their increase~~
~~due to the introduction of contemporary summer mean Q_{sg} (Q_{sg} -present) into the PGIS-PF~~
~~system (e.f. Fig. 1(d), Fig. In the control run, Petermann ice shelf basal melt rates~~
~~largely range from a few m/yr near the calving front to ca. 40 – 60 m/yr under the deeper~~
~~(400 – 600 m) regions of the ice shelf base near the grounding line (Fig. 2(a,b) are~~
~~in agreement with)). Melt rates decreasing with distance from the grounding line is~~
~~consistent with results obtained from steady state ice discharge divergence approach~~
~~(Rignot and Steffen, 2008) and satellite remote sensing (Wilson et al., 2017). For the~~
~~period from 2011 – 2015, remotely sensed estimates of annual mean melt (reveal melt~~
~~rates of up to ca. 80 m/yr near the grounding line, although they largely range between~~
~~40 – 50 m/yr decreasing to ca. 10 m/yr under the shallower (<200 m) portions of~~
~~the ice shelf (Wilson et al., 2017) and in-situ conductivity-temperature and radar derived~~
 ~~estimates of discharge enhanced seasonal melt (Washam et al., 2020), and are comparable~~
 ~~to our modelled estimates.~~ With the ~~inclusion of introduction of contemporary summer~~
 ~~mean subglacial discharge (Q_{sg} -present at the GL) into the ice shelf-fjord system,~~ the
 modelled mean melt strengthens nearly throughout the domain and appears to be concen-
 trated along the basal channels that run along the length of the ice shelf (Rignot and Steffen, 2008
 ~~;~~ Fig. 1(d), Fig. 2(a,b)). ~~Strongest increases are seen~~ Spatial variability of our
 ~~modelled melt on scales similar to the basal channel configuration is consistent with the~~
 ~~findings of Rignot and Steffen (2008) and Wilson et al. (2017).~~ Strongest increases of up
 ~~to ca. 100 m/yr are seen within these channels under the deeper regions of the PGIS base~~
 ~~near the GL, and which gradually decrease toward,~~ which gradually decreases towards
 the calving front. ~~This increase lies within the in-situ conductivity-temperature and radar~~
 ~~derived estimates of discharge enhanced seasonal melt rate maxima of 40 — 170 m/yr~~
 ~~(Washam et al., 2020).~~ The summer mean basal melt rate (m_b), friction velocity (u^*),
 and thermal driving (ΔT) are averaged over (a) the entire PGIS ice shelf base, and (b)
 over parts of the PGIS ice shelf where draft is deeper than 300 m, and are indicated, re-
 spectively, by subscripts "pgis" and "300m" appended to the variable in question. Com-
 pared to the summer mean modelled melt rates in the control experiment, the contempo-
 rary discharge-driven increase in m_{bpgis} and m_{b300m} , respectively, is 41.2% and 68.1%,
 driven by similar increases in u^*_{pgis} (40.747%) and u^*_{300m} (64.167.4%) and modest in-
 creases in ΔT_{pgis} (1.61.1%) and ΔT_{300m} (5.64%) (Fig. 1(d)).

Figure 2. Response of **PGIS-ice shelf** basal melt rates and its drivers to Q_{sg} -discharge distributed uniformly across the **GL-grounding line** (Fig. 1(b)), for the experiments defined in Table 1. Summer mean basal melt rate (m_b) (a-d), friction velocity (u^*) (e-h), and thermal driving (ΔT) (i-l) for the control (first row), Q_{sg} -present relative to control (δ ; second row), Q_{sg} -median relative to control (δ ; third row), and Q_{sg} -RCP 8.5 relative to control (δ ; last row) experiments. Contours of the ice shelf draft are plotted at 100 m intervals (450 m – 150 m; **panel(f)**) and overlaid as dotted black lines on each panel.

The presence of Q_{sg} at the **GL-subglacial discharge at the grounding line** enhances the
 fjord-scale overturning circulation (Fig. 3(i)). We observe a more robust Atlantic Water
 inflow at depth, extending into the inner fjord basin and up to the **GL-grounding line**

(Fig. 3(c,f)). Hereafter, we refer to this process as the "thermal pump", which promotes the entrainment of heat and salt deep into the **PGIS-ice shelf** cavity (columns 1 and 2 in Fig. 3). We note an increase in bottom temperature of 0.03°C at a location ca. 10 km from the **GL-grounding line**, and a temperature excess ($\Delta T_{\text{thermal driving}}$) of 0.1°C under the ice shelf base at 300 m depth (Fig. 2(j); Fig. 3(g)). A higher $\Delta T_{\text{near the GL}}$ **thermal driving near the grounding line** (Fig. 2(i,j)) drives more melting at depth. A more energetic outflow/strengthened meltwater plume dynamics, driven by the introduction of Q_{sg} -**discharge**, exacerbates the shear-driven turbulent mixing and heat transfer across the **PGIS-ocean-ice shelf-ocean** interface (Fig. 2(e,f)), thus driving more melt. The spatial variability of u^* -**friction velocity** anomalies is consistent with the corresponding melt rate anomalies (cf. panels b and f in Fig. 2). Regions exhibiting strong increase in u^* -**friction velocity** are concentrated along the longitudinal basal channels near the **GL-grounding line**, and gradually decrease downstream of it. However, a higher meltwater production (Fig. 2(b)) and stronger outflow (Fig. 3(f,i)) deposits larger volumes of meltwater from the deeper to the shallower regions of the **PGIS-ice shelf** base, considerably lowering the $\Delta T_{\text{thermal driving}}$ locally (Fig. 2(j); Fig. 3(d,e)). Thus, while the increase in u^*_{pgis} and u^*_{300m} is significant, the contribution of $\Delta T_{\text{thermal driving}}$ is largely restricted to the deeper regions of the **PGIS-ice shelf** draft (ΔT_{300m}) (Fig. 1(d)). **This implies that for subglacial discharge equivalent to present-day summer mean magnitudes, increase in basal melt under the deeper (>300 m) drafts is driven by increases in both thermal driving (increased thermal pumping) and friction velocity (increased shear-driven mixing).**

Heightened discharge and saturation of the thermal pump
When transitioning from the control experiment to RCP 8.5 (Q_{sg} -RCP 8.5), and an inter-
 mediate scenario between the two (Q_{sg} -median) (Table 1), a higher modelled Q_{sg} -**discharge**,
 as expected in a future warmer atmosphere (Aschwanden et al., 2019; Golledge et al.,
 2019), results in considerably higher increases in basal melt as compared to Q_{sg} -present
 (Fig. 1(d); Fig. 2(a–d)). In particular, an increase in Q_{sg} -**discharge** from control to RCP
 8.5 results in more than twofold (168.4%) and threefold (217.2%) increase in m_{bpgis}
 and m_{b300m} , respectively. These increases are primarily driven by increases in u^*_{pgis}
 (**165.5191.8%**) and u^*_{300m} (**205.7216.9%**), whereas ΔT_{pgis} and ΔT_{300m} show rela-
 tively nominal increases of **5.1% and 9.82.9% and 6.5%**, respectively. For the median
 experiment, **3.8% and 8.62.3% and 5.9%** increase in ΔT_{pgis} and ΔT_{300m} , and **114% and**
 **154.7132.5% and 163.4%** increase in u^*_{pgis} and u^*_{300m} generate 116.3% and 163.6% in-
 crease in m_{bpgis} and m_{b300m} , respectively. As noted for Q_{sg} -present, strongest increases
 occur under the deeper regions of the **PGIS-ice shelf** base which gradually decreases
 downstream (Fig. 2 (c,d)), however, channelized melting is seen to strengthen substan-
 tially (discussed further below).
Combining insights from all the experiments provides a novel perspective on how
 Q_{sg} -**discharge** dictates the interplay between **PGIS-Petermann ice shelf** basal melt and
 its drivers: Increase in the heat entrained in the **PGIS-ice shelf** cavity as a result of the
 thermal pump **peaks is strongest** at the inclusion of present-day summer mean estimate
 of Q_{sg} -**subglacial discharge**, with much of the heat delivered by the thermal pump con-
 centrated under the deeper **PGIS-ice shelf** base in the inner fjord basin (Fig. 1(d); Fig.
 3(d,g)). At a summer mean Q_{sg} -**discharge** value which is approximately fourfold higher
 than the present (i.e. Q_{sg} -median vs. Q_{sg} -present), **the contribution of the thermal pump**
 **is relatively weaker thermal driving increases at a lower rate** (Fig. 1(d); Fig. 3(g); cf.
 Fig. 3(d) and Fig. C1(a)). We note an increase of **2.91.9%** in the Q_{sg} -median ΔT_{300m}
 as compared to Q_{sg} -present (Fig. 1(d)). Moreover, a larger meltwater deposition further
 lowers the $\Delta T_{\text{thermal driving}}$ under the shallower drafts (Fig. 2 (b,c,j,k); Fig. 3(i);
Fig. C1(a,b,c)) and we do not observe a noticeable which results in a relatively weaker
 increase in ΔT_{pgis} (Fig. 1(d)). Lastly, in the RCP 8.5 scenario, Q_{sg} discharge is nearly
 doubled as compared to the Q_{sg} -median experiment, however, the efficiency of the thermal
 pump in increasing ~~m_b~~ basal melting saturates (Fig. 3(g); Fig. C1). Here, increase
 increases in ΔT_{pgis} ~~is~~ and ΔT_{300m} are indiscernible as compared to the Q_{sg} -median
 experiment, whereas ΔT_{300m} increases negligibly (Fig. 1(d)). Instead, we find that
 Q_{sg} under a discharge under the high-emissions RCP 8.5 scenario enhances melting by
 further accelerating the cavity circulation, wherein, 24.1% and 20.4% increase in m_{bpgis}
 and m_{b300m} are seen, driven predominantly by ~~24% and 20~~ 25.5% and 20.3% increase in
 u^*_{pgis} and u^*_{300m} , respectively.

Figure 3. Along-fjord summer mean temperature (a), salinity (b), and flow (c) for the control experiment for the section shown in Fig. 1(b). Mean isopycnals (solid black lines) are overlaid at equal intervals of 0.2 kg/m^3 . The 27.9 kg/m^3 isopycnal corresponds to the dense Atlantic Water that contacts the grounding line. For the same section, changes (δ) in summer mean temperature (d), salinity (e), and flow (f) for Q_{sg} -present relative to control are shown. Black stippled lines in panels d and e indicate the zero temperature and salinity difference contour, respectively. In panel f, solid black lines and stippled green lines depict the mean isopycnals for the control and Q_{sg} -present experiments, respectively, plotted at equal intervals of 0.2 kg/m^3 . In each panel, PGIS-GL-Petermann ice shelf grounding line (at \sim ca. 600 m depth) is on the right margin and open ocean is to the left. Vertical magenta line in panels a, b, d, and e show the location of the model node ca. 10 km from the GL-grounding line (Fig. 1(b)) at which the vertical profile of summer mean temperature (g) and salinity (h) are shown for the control, Q_{sg} -present, Q_{sg} -median, and Q_{sg} -RCP 8.5 experiments (cf. Table 1). (i) Laterally integrated summer mean vertical overturning profile [in Sv] for all the experiments (cf. Table 1) for the section shown in Fig. 1(b). Positive [Sv] values correspond to (laterally integrated) inflow and negative [Sv] values correspond to (laterally integrated) outflow across the section. Calculation of transport diagnostics follows from Prakash et al. (2023).

In sum, our experiments reveal a regime change in heat flux efficiency within the ice
shelf cavity. In the control experiment, the ice shelf-ocean boundary exchange consumes
 a considerable amount of heat from the ocean that is available for melting the ice. As
 such, the latent heat loss and meltwater release cools the ambient ocean inside the cavity.
 In such a scenario, an increased fjord-scale overturning circulation due to increased
 discharge (e.g., $< Q_{sg}$ -median experiment) entrains more heat in the cavity (thermal
 pump) which increases melting through enhanced thermal driving, in addition to the
 enhanced shear-driven turbulent heat exchange. This saturates at some point, when more
 discharge (e.g., $> Q_{sg}$ -median experiment, as seen for the RCP 8.5 scenario) accelerates
 the overturning circulation such that there is always enough heat coming into the cavity
 to maintain a high thermal driving (i.e., heat supply at an elevated level is maintained).
 Here, melt rates become entirely a function of the friction velocity.
**Trapping Convergence** of meltwater and channel stability

Figure 4. Maps of the summer mean surface layer salinity (a,b) and basal melt rate (c,d) for the control (column 1) and Q_{sg} -RCP 8.5 relative to control (δ ; column 2) experiments. Contours of the ice shelf draft are plotted at 100 m intervals (450 m – 150 m) and overlaid as dotted red and blue lines over the surface layer salinity and melt rate maps, respectively.

We use surface layer salinity as a proxy for tracking the spatial distribution of meltwater
 under the deeper regions of PGIS the ice shelf (Fig. 4; Fig. D1). Freshening occurs nearly
 throughout the domain in the Q_{sg} -driven discharge-driven experiments as compared to
 the control run. Furthermore, regions of substantial freshening align with four prominent
 ice shelf basal channels (Fig. 4 (a,b); 5 (a,b); Fig. B1) along which the draft thins from
 300 – 450 m ca. 5 km from the GL grounding line to 150 m ca. 25 – 30 km seaward
 of it, suggesting a topographically steered horizontal pathway for the meltwater outflow
 that is determined by the PGIS-ice shelf basal morphology and which in turn influences
 it (discussed further below). Changes appear to be most profound within these channels,
 particularly, at the erestcrests, where salinity decreases by up to 1.06 psu in the RCP 8.5
 experiment with respect to the control run (Fig. 4; Fig. 5(a,b)). These locations where
meltwater is trapped of meltwater convergence are concordant with locations of largest
 melt rate increases (discussed further below) of up to 181.5 m/yr (a sevenfold increase
 compared to the control run). ΔT is higher. Thermal driving is not higher at the channel
 crests where meltwater converges, but under the deeper drafts, which is consistent with
 the pressure dependent melting point of seawater. Although ΔT (Fig. 5(c)). Although
 thermal driving may not increase the most at the channel crests, we observe increases
 ranging from 0.06 to 0.19 0.02 to 0.12 °C in the RCP 8.5 experiment compared to the
 control run across the transverse cross-section near the GL grounding line (Fig. 5(c)).
 We find that u^* profiles the friction velocity profile across this section are is (nearly)
 congruent with the m_b profiles, and exhibit largest basal melt rate profile, and exhibits
 the highest increases at the channel crests of up to 0.03 m/s (Fig. 5(d)).

Figure 5. Maps of the summer mean surface layer salinity (a,b) and basal melt rate (c) Cross-section profile of the Petermann ice shelf draft, derived from BedMachine v3 (Morlighem et al., 2017), for the control transect ca. 5 km from the grounding line shown in Fig. 1 (column 1b) and Q_{sg} -RCP 8.5 relative to control (column 2) experiments. Contours Apex locations of four prominent longitudinal ice shelf basal channels at this section are highlighted using green stripes along which the Petermann ice shelf draft are plotted at 100 m intervals (reduces from 300 – 450 m – at the transect location to 150 m) and overlaid as dotted red and blue lines over a distance of 25 – 30 km from grounding line (Fig. B1). Across the same section, the summer mean surface layer salinity and melt rate maps (m_b) (b), respectively thermal driving (ΔT) and melt rate (c), and friction velocity (u^*) and melt rate (d) are shown for the control and Q_{sg} -RCP 8.5 experiments.

Three ice shelf basal channel sections are outlined (Fig. 6(a)), whose lengths are
 demarcated based on the apex of the 150 m ice shelf draft contour (e.g. Fig. 4(a); Table
 2). The channel averaged annual mean basal melt rates for the western (S_w), central
 (S_c), and eastern (S_e) sections for the RCP 8.5 experiment are calculated as 45 m/yr,
 51 m/yr, and 47 m/yr, respectively. The enhanced Enhanced melting at the crests will
 promote channel opening and we can calculate the changes in channel averaged annual
 mean ice volume above each channel over time to estimate the incision/erosion timescale
 for each that channel under the respective basal melt rates (Eqn. 5, 6). To bracket the
role of ice advection from upstream of the GL grounding line, three different rates (AR_c)
 are considered for each channel which correspond to the present contemporary mean (1
 242 km/yr; Münchow et al. (2014)), and 50% lower (0.5 km/yr) and higher (1.5 km/yr) than
 243 present contemporary mean values. While S_w exhibits the lowest mean basal melt rate, it
 also features the shallowest mean draft value. Notably, its erosion-incision timescales are
 comparable to those of S_c , which has the highest mean melt rate but a relatively deeper
 mean draft compared to S_w . The number of years (N) required for the mean channel-ice
 volume-ice volume above a channel to become zero is estimated to be 5.88, 6.52, and 7.31
 248 years for S_w , in contrast to 5.95, 6.75, and 7.8 years for S_c , at AR_c of 0.5 km/yr, 1 km/yr,
 and 1.5 km/yr, respectively (Fig. 6(b)). It has been shown that the viscous ice response
 to a basal channel is insufficient to significantly delay the breakthrough of basal channels
 in ice shelves that are thinner than 400 m (Wearing et al., 2021). With a lower mean melt
 rate than S_c and the deepest mean draft value, S_e demonstrates the greatest resilience. We
 estimate N to be 8.06, 9.41, and 11.31 years for AR_c of 0.5 km/yr, 1 km/yr, and 1.5 km/yr,
 respectively (Fig. 6(b)). These estimates suggest that ice advection from upstream of the
 PGIS-GL grounding line (alone) may not be sufficient to compensate for its thinning the
 thinning of the ice shelf due to intense channelized basal melting that may be expected
 under the RCP 8.5 scenario (discussed further below).

(a) Cross-section profile of the PGIS draft for the transect ca. 5 km from the GL shown in Fig. 1(b). Four prominent longitudinal (L) ice shelf basal channels are represented using green stripes as L_w (west), L_c (central), $L_e(1)$ (east-1) and $L_e(2)$ (east-2) along which the PGIS draft reduces from 300–450 m at the transect location to 150 m over a distance of 25–30 km from GL (see Fig. B1). For clarity, $L_e(1)$ and $L_e(2)$ are collectively classified as L_e (east) in panels b, c, and d. Across the same section, the summer mean surface layer salinity and melt rate (m_b) (b), thermal driving (ΔT) and m_b (c), and friction velocity (u^*) and m_b (d) are shown for the control and Q_{sg} -RCP 8.5 experiments.

Figure 6. (a) Map of the PGIS Petermann ice shelf draft overlaid with channel sections along the western (S_w ; blue), central (S_c ; black), and eastern (S_e ; red) regions which are used in calculating the changes in respective channel averaged annual mean ice volume above the channel over time in panel (b) for the RCP 8.5 experiment following Eqns. 5 and 6 (Table 2). Apex locations of four prominent longitudinal ice shelf basal channels at the across-fjord transect shown in Fig. 5 are represented using magenta diamonds. For each color-coded channel section shown in (a), channel-corresponding changes in mean ice volume changes above the channel section are shown in (b) are shown for three different ice advection rates, namely, 0.5 km/yr (asterisk), 1 km/yr (solid line) and 1.5 km/yr (circle). Channel averaged annual mean basal melt rates for S_w , S_c , and S_e are, respectively, 45 m/yr, 51 m/yr, and 47 m/yr.

**Discussion**
We note that our model is forced with contemporary summer mean Q_{sg} -estimates discharge
 estimate and expected increases of it under a future warming atmosphere. The present-
 261 day estimates of maximum Q_{sg} -discharge (e.g. Cai et al., 2017) are two to fourfold
 higher than the present-day summer mean estimate of Q_{sg} -discharge used in this study,
 and are even comparable to the Q_{sg} -median experiment. Therefore, while Q_{sg} -median
 represents an intermediate state between present and extreme future mean scenario in
 our study, it is plausible that such increases in Q_{sg} -discharge, and thus basal melt, are
 being experienced by **PGIS-the Petermann ice shelf** for brief periods during summer,
 with varying degrees of inter-annual variability. ~~Modelled study from~~ We posit that
 during the summer (JJA) months, if for e.g., exceptionally high discharge is sustained
 for a considerable (e.g., several weeks) period of time, it may temporarily nudge the ice
 shelf-fjord system into a regime where basal melting is largely driven by friction velocity.
 We note that as opposed to Gaussian-like patterns (Ehrenfeucht et al., 2023), our applied
 discharge forcing follows a boxcar function. Thus, studies investigating the sensitivity
 of the system to seasonal and inter-annual variability in subglacial discharge, derived for
 e.g., from a subglacial hydrological model, need to be conducted to ascertain and quantify
 any resulting transitions that may occur in the mechanisms driving basal melting. The
 modelling work of Hill et al. (2018) has demonstrated that the loss of thicker and rhe-
 ologically stiffer sections of **PGIS-the Petermann ice shelf** within 12 km from the **GL**
 **grounding line** (regions where draft ≥ 300 m) could disrupt **GL-grounding line** stresses
 sufficiently to greatly accelerate interior ice flow and increase dynamic ice discharge, sig-
 nificantly contributing to global mean sea level rise. We thus suggest that the Q_{sg} -driven
 discharge-driven intensified basal melting of the deeper (draft ≥ 300 m) regions near the
 **GL-grounding line** that are dynamically resilient (Fig. 1(d); Fig. 2) will have a significant
 bearing on the future stability of **PGthe Petermann Glacier**, as their loss could trigger an
 unopposed acceleration of the inland ice.
Thus far, basal melting of **PGIS-the Petermann ice shelf** has been largely investigated
 from the perspective of (increased) oceanic heat delivery to **PFPetermann Fjord**. How-
 ever, both observational and modelled estimates concur on the abundance of heat supply
 into the fjord, i.e. heat availability is in excess of what is needed to generate contempo-
 rary estimates of basal melt (Johnson et al., 2011; Washam et al., 2018; Prakash et al.,
 2023). Our findings show that increased basal melting of **PGIS-the Petermann ice shelf** in
 a warming climate is not limited to increase in oceanic heat forcing, and underscores the
 importance of atmospheric warming in controlling the basal melting of Northern **GrIS**
 **Greenland Ice Sheet** glaciers (Slater and Straneo, 2022). Uncovering how this unfolds
 from the **PGIS-ocean-an ice shelf-ocean** interaction perspective, we find that the effect
 of the thermal pump on melt is the strongest when Q_{sg} -discharge is introduced into the
 system (as Q_{sg} -present; Table 1) and saturates for Q_{sg} -discharge $> Q_{sg}$ -median (Fig.
 1(d); Fig. 3; Fig. C1), wherein, Q_{sg} -median is equivalent to present-day estimates of
 maximum Q_{sg} -discharge (c.f. Cai et al. (2017)). We see an increase in bottom temper-
 ature near the **GL-grounding line** (increased heat advection towards the interior of the
 **PGIS-ice shelf** cavity via the thermal pump) as Q_{sg} -discharge is increased from control
 to present to median, but less so for substantially higher discharge under the RCP 8.5
 scenario (Fig. 3(g)). Therefore, we posit that a regime shift of the driving mechanism
 behind m_b -Petermann ice shelf basal melting may occur when the catchment integrated
 meltwater runoff under (extreme) future warming scenarios exceeds the discharge thresh-
 old of present-day Q_{sg} -maximummaximum. In such a scenario, the significance of the
 thermal pump in driving increased melting ebbs off, and the higher discharge being re-
leased into the fjord accelerates the sub-ice shelf currents, which nudges the system to-
 wards a regime where increase in basal melt is predominantly dictated by substantially
 higher shear-driven turbulence which increases the heat fluxes at the ~~PGIS-ocean-ice~~
 ~~shelf-ocean~~ interface (Fig. 2). ~~We~~ Millan et al. (2023) show that Atlantic Water reaching
 the fjord in 2020 are 0.3°C warmer as compared to 1970, of which a warming of 0.2°C
 has occurred between 2000 and 2020. Further modelling work is required to determine
 the characteristics of Atlantic Water reaching the fjord by the end of this century under
 different climate warming scenarios. While we do not take into consideration the impact
 of rising ocean temperatures in our study, we acknowledge that realistically, in a future
 ~~warming climate, rising ocean temperatures warmer climate,~~ it would likely act in concert
 with the ~~enhanced Q_{sg} increased discharge~~ to further amplify ~~m_b basal melting~~. How-
 ever, with a much lower heat capacity as compared to the ocean, increase in surface melt
 over grounded ice in response to atmospheric warming (and thus increase in subglacial
 discharge) is highly likely to outpace the warming of the deep ocean water masses that
 (alone) would be required to bring about such substantial increases in ~~m_b basal melting~~
 in the upcoming centuries. ~~Thus, it is reasonable to posit that the atmospheric impact~~
 ~~on m_b is~~ (e.g., increases of up to ca. 200 m/yr in the RCP 8.5 scenario) as modelled
 here. Thus, while the impact of increase in ocean heat forcing and its interplay with
 increased discharge on basal melting in a future warming climate needs to be examined,
 discharge-driven basal melting is likely to pose a more immediate and pertinent threat to
 the glacier's long-term stability.
~~Across all Q_{sg}~~ Seasonal changes including, but not limited to, an increase in fjord
 temperature of ca. 0.1°C during summer as compared to winter (Prakash et al., 2023)
 drives up to ca. 47% increase in mean channelized (sections S_w , S_c , and S_e in Fig. 6(a);
 Table 2) basal melting. Without taking into consideration the compensating viscous ice
 response (discussed further below), and for contemporary mean ice advection rate from
 upstream of the grounding line of 1 km/yr, this increase results in up to ca. three-fold
 increase in erosion rates. The strongest seasonal influence on melt is imposed by the
 addition of subglacial discharge into the fjord. Compared to winter (no discharge), we
 calculate that contemporary summer mean discharge increases mean melting in the channels
 by up to ca. three-fold, yielding up to an order of magnitude increase in erosion rates.
 We note that although S_w and S_c may exhibit higher rates of erosion as compared to S_e ,
 highest increases in erosion rates are modelled for S_e .
~~Across all discharge~~ experiments (Table 1), we consistently find channelized strength-
 ening of basal melt (Fig. 2(a-d)), imposed by the complex ~~PGIS-ice shelf~~ basal morphol-
 ogy (Rignot and Steffen, 2008; discussed further below). Within these channels (e.g., ~~L_w ,~~
 ~~L_{S_w} , S_c , and L_{S_e}~~ in Fig. 5), 6(a), compared to the control experiment, larger volumes
 of buoyant meltwater rise along the steepest slopes of the ice base towards the channel's
 crests (Fig. 5), from where the stronger buoyancy-driven overturning circulation in the
 fjord carries it further downstream along the longitudinal channel axis (Fig. 3(i); Fig. 4;
 Fig. B1; Fig. D1). The steeper ice shelf basal slopes that characterize these channels
 support stronger entrainment of the ambient Atlantic Water into the buoyant meltwater
 plume (Wilson et al., 2017). Furthermore, the vigorous confluence of meltwater towards
 the ~~crest increases u^* which further amplifies melt. crests, and its downstream advection,~~
 ~~increases the friction velocity which predominantly drives the increased melting~~ (Fig. 2;
 Fig. 5(b,d)). The additional buoyancy forcing provided by ~~Q_{sg} under future warming~~
 ~~scenarios increased subglacial discharge in a warmer future climate,~~ in particular, in as
 modelled here for the RCP 8.5 experiment, ~~acts will act~~ to enhance these mechanisms
 (Fig. 5(b,d)) which may advance the vertical growth of the channel. To that end, our
 ~~channel erosion estimates~~ estimates of mean ice volume changes above the channels in-
dicates that sustained high mean melt rates due to enhanced summer Q_{sg} discharge, as seen
in the RCP 8.5 experiment, could undermine the structural integrity of ~~PGIS~~ the ice shelf
(Fig. 6). Persistence of such intensified melting over durations ranging from ca. 6 to
11 years may result in complete ~~erosion-incision~~ of the channels, leading to a possible
disintegration of the ice shelf. While the accuracy of this estimate depends on the un-
certainty of the applied melt rate parameterization, e.g., not accounting for the effects of
enhanced stratification due to the meltwater discharge (Rosevear et al., 2022; Davis et al.,
2023), the simulated acceleration of the cavity circulation is a robust feature of our sim-
ulations, which, together with observational evidence (Washam et al., 2020) indicates
significant impacts on the basal mass loss of ~~PGIS~~ the Petermann ice shelf. We also
note that our static ice shelf geometry does not account for the ice dynamical processes,
for e.g., downstream advection and calving, and viscous ice response which would act
to (partially) offset melting. While it has been shown that the secondary ice-flow is not
strong enough to substantially hinder the development of basal channels in ice shelves that
are less than 400 m thick (Wearing et al., 2021), we observe that the Petermann ice shelf
draft is deeper than 400 m in certain sectors (e.g., southeastern sector near the grounding
line). However, in the absence of a high-resolution (at the scale of the topographical data)
ice-flow model, it is difficult to ascertain how small-scale (sub-km) localized channel
closures/openings would impact the wider ice dynamics, and as such, the long-term
stability of the glacier. We suggest that future work with coupled ice-flow - ocean models
(frameworks of which have been recently developed, e.g., Zhou et al. (2024)) or offline
coupling of an ice-flow model with comprehensively resolved ice shelf basal melt rates
derived from sophisticated regional ocean models is needed to improve our understanding
of the nuanced dynamical variability that occur within these channels.
Our results could be extended to other 'warm cavity' environments where pronounced
increases in Q_{sg} discharge are expected, and the ice shelf is characterized by complex
basal morphology. We suggest that such conditions could be met for other Northern
~~GrIS~~ Greenland Ice Sheet glaciers that terminate as floating ice shelves, namely the Ry-
der Glacier (~~RG~~) and Nioghalvfjærdsbræ (79N) Glacier. The ice shelves of both ~~RG~~ Ryder
Glacier and 79N also feature channelized geometry and thus exhibit notable spatial melt
rate variability, as well as a dependency of basal melt on slope that is qualitatively similar
to ~~PGIS~~ the Petermann ice shelf (Wilson et al., 2017; Prakash et al., 2023). The lower
rates of ice advection from upstream ~~make RG of the grounding line~~ make Ryder Glacier
more prone to developing deep channels. At 79N, it has been suggested that the recent
expansion of its central basal channel is a consequence of heightened Q_{sg} discharge, at-
tributed to a significantly expanded surface melt area during summer, brought about by
atmospheric warming (Narkevic et al., 2023; Zeising et al., 2023). Moreover, the exten-
sive network of moulins observed along the ~~GrIS~~ Greenland Ice Sheet margins in summer
(Turton et al., 2021; Ehrenfeucht et al., 2023; Rawlins et al., 2023), coupled with expected
increases in surface melt ~~on the GrIS over the ice sheet~~ in a warmer climate (Aschwan-
den et al., 2019; Muntjewerf et al., 2020), are likely to yield end-of-century discharge
estimates at these sites that are comparable to those of Petermann (see projected runoff
for the northern, northeastern and northwestern ~~GrIS sectors~~ sectors of the ice sheet in
Slater et al. (2020)).
Analogous to Greenland, the intricate basal morphology of Antarctic ice shelves,
sculpted by their interactions with the bed and/or melting, also exhibits a diverse range
of features (e.g., channels, keels, crevasses, and terraces) that exert a significant in-
fluence on their melting and structural integrity, which is contingent upon the slope
(Drews et al., 2017; Schmidt et al., 2023). ~~While the~~ The role of an additional buoy-
ancy source that is injected basally into the fjord on enhancing basal melting and retreat
has been extensively investigated in Greenlandic geophysical settings (Sciascia et al.,
2013; Rignot et al., 2015; Cai et al., 2017; Cook et al., 2020), ~~in Antarctica, insights into~~
~~this process are limited to tacit reasoning~~ (Le Brocq et al., 2013; Lepp et al., 2022) ~~and~~
~~remain largely unaccounted for in modelling efforts that aim to constrain future sea-level~~
~~projections.~~ ~~Nonetheless, recent studies~~ and has garnered recent attention in Antarctica
(Goldberg et al., 2023; Gwyther et al., 2023; Pelle et al., 2024). Studies have shown that
Q_{sg} subglacial discharge has significantly influenced the mass balance of the Antarctic
Ice Sheet during the last deglaciation period and is expected to maintain its importance
in a warming climate (Li et al., 2023; Pelle et al., 2023). In addition, ice shelf collapses
due to hydrofracturing and flexurally-induced fractures resulting from supraglacial lake
drainage events have also been noted (Scambos et al., 2009; Banwell and MacAyeal,
2015; Dow et al., 2018). However, unlike Greenland (Forster et al., 2014; Willis et al.,
2015; Otto et al., 2022) there is no evidence of surface meltwater infiltrating to the bed.
Greenland's surface and subglacial hydrological systems that are connected by drainage
pathways offer a blueprint for understanding how Antarctica might evolve in a warmer
world, as its surface hydrology starts to resemble present conditions observed at ~~GIS~~ the
Greenland Ice Sheet (Bell et al., 2018). Projected increases in surface melt rates over
Antarctica (Noël et al., 2023) suggest a shift towards a more extensive and active surface
hydrology, one that could effectively link the presently isolated bed via seasonal influx
of surface meltwater. We note that such scenarios are dependent upon future emission
pathways and whether the ambitious targets proposed by the Paris Climate Agreement
are upheld (DeConto et al., 2021). However, if these targets are exceeded, as is expected
under the RCP 8.5 projection, we posit that a fundamental change in Antarctica's dy-
namics and the processes controlling it cannot be precluded. In particular, energetic
currents driven by intensified discharge in a cavity characterized by complex channelized
basal morphology, as modelled here, will accelerate the heat flow towards the ice shelf
base. The substantial shear-driven turbulence could independently drive increased melt-
ing, and may even be sufficient to abrade the stable boundary layer stratification that has
been observed in critical regions where ~~m_b~~ basal melting remains hitherto suppressed
due to latent ocean environments (Davis et al., 2023).
Ice shelf basal topography remains largely represented in numerical models in an
idealised tapered form, that is either completely flat or has some smooth curvature that
is imparted as a function of along-fjord slope. Moreover, the sub-ice shelf bathymetry
employed in these models is derived using rudimentary techniques that attempt to inter-
polate between known depths from ~~the continental shelf/fjord mouth to the GL~~ near the
fjord mouth and the grounding line. These critical shortcomings are addressed in our
model setup (Prakash et al., 2022), however, efforts to mitigate such topographic (both
seafloor and ice shelf basal ~~)~~ caveats topography caveats in numerical modelling efforts
that aim to investigate real-world locations, despite its well acknowledged impetus on
~~m_b~~ basal melting, are still not being widely undertaken. Indeed, an unknown sub-ice
shelf bathymetry ~~at PGIS and~~ for a vast majority of ice shelf-fjord systems is a limiting
factor. As such, wherever possible, more targeted airborne and ground-based geophysical
campaigns must be conducted to describe these environments. While our findings em-
phasize the need for implementing realistic ice shelf geometries, it is noteworthy to high-
light the limitations of the available resolutions (typically hundred to several-hundred
meters; e.g. Morlighem et al. (2017); Frémand et al., 2023) which remain relatively
coarse to capture the small-scale basal features, on the order of several (tens of) me-
ters as revealed by in-situ under-ice shelf observations (Davis et al., 2023; Schmidt et al.,
2023), that play a crucial role in dictating the spatial ~~variability in m_b~~ melt rate variability.
To comprehensively understand the intricate interplay between Q_{sg} subglacial discharge,
basal morphology, and melt processes, we emphasize that a more extensive dataset of
such measurements is imperative ~~which then needs to be integrated into~~. To enhance
the reliability of continent-scale projections, observational findings from sub-ice shelf
environments – including near the grounding zones – must be combined with insights
from non-idealized high-resolution ice shelf-ocean models numerical modelling studies
driven by these datasets. Improved understanding of critical small-scale processes at
the marine-ice margins should then inform the refinement of existing parameterizations
and the development of new ones, which would facilitate large-scale models to better
capture the influence of dynamics at the meter-scale. We note that in our experiments,
Q_{sg} discharge is distributed uniformly across the GL-grounding line width which may
not be consistent with ground truth. For an idealized tidewater glacier, modelling work
has shown that distributed discharge generates more melt than channelized discharge
(Slater et al., 2015). With glacier hydrology exhibiting increased dynamism, which is
likely to continue into the future, a concerted effort is warranted. This necessitates de-
riving insights into the interplay between moulin locations/density, meltwater routing,
and eventually its magnitude and emergence location(s) at the GL-grounding line from
observational evidence and sophisticated subglacial hydrological modelling tools. ~~These~~
~~insights, which~~ should then be incorporated into numerical models of ice shelf-ocean in-
teractions, ~~to better predict the evolving dynamics of glacier-ocean systems in a warming~~
~~climate.~~ Furthermore, investigations of the interaction between subglacial discharge and
tidally-modulated seawater intrusions in the grounding zone (Ciraci et al., 2023) will be
essential in improving our understanding of the dynamics and stability of the glacier.
**Methods**
**The Finite Volume Community Ocean Model (FVCOM) setup**
The unstructured grid, free-surface, 3-D primitive equation Finite-Volume Community
Ocean Model (FVCOM) (Chen et al., 2007), augmented by modules for an ice shelf
(Zhou and Hattermann, 2020) and sea ice (*Ice Nudge*) (Prakash et al., 2022), is used. The
model domain is centered on PF Petermann Fjord, covering the region between 75°N -
87°N and 29°W - 81°W (Fig. 1a). The grid comprises of unstructured triangle elements,
featuring a total of 112709 nodes and 219836 cells, including 309 open boundary nodes
in the Lincoln Sea and Baffin Bay. The topological flexibility of the unstructured grid
allows accurate fitting of the complex irregular coastal geometry along the northern GIS
Greenland Ice Sheet margin. In particular, the grid is partitioned into polygons of varying
horizontal resolution: A 200-m resolution is provided in the PF Petermann Fjord region
to resolve the steep slopes characteristic of the fjord seafloor and PGIS-ice shelf basal
topography, which is relaxed to 2 km in the adjacent areas outside the fjord, and further
to 4 km towards the open ocean boundary regions. For an extensive regional model setup,
such a configuration enables us to strike a balance between computational efficiency and
the need for accurate representation of topographical features and their variation over
small spatial scales.
PGIS-basal topography Petermann ice shelf draft derived from the BedMachine v3
(Morlighem et al., 2017) dataset (150-m resolution) represents a pre-2010 PGIS-calv-
ing geometry (Münchow et al., 2014), and is interpolated to the 200 m FVCOM model
grid (Prakash et al., 2022). As such, the basal channels (~~Rignot and Steffen, 2008~~) are
resolved at 200 m scale in our model, which is sufficient to represent their 1–2 km width
(Rignot and Steffen, 2008). Bathymetry outside PF Petermann Fjord comes from the 1-
503 km gridded IBCAO v3.0 dataset (Jakobsson et al., 2012), whereas inside it, the Bed-

Machine v3 dataset is used. In the vertical, the domain between the irregular seafloor topography and the ice shelf draft (or the open ocean surface) is discretized into 23 terrain-following σ -layers. To alleviate errors in the discretization of horizontal pressure gradients along the sloping σ -layers, smoothing of the bathymetry is applied as a function of the mesh resolution before its implementation in the model (Prakash et al., 2022). Further, the poorly constrained Without seafloor depth measurements beneath the ice shelf, the BedMachine bathymetry is inferred by extending the known seafloor depth from seaward of the calving front landward towards the grounding line and aligning it with the depth of the ice shelf draft at the grounding line using natural neighbor interpolation. For Petermann Fjord, Prakash et al. (2022) show that this leads to implausible abrupt changes in the water column thickness along either flanks and. Specifically, the BedMachine bathymetry underestimates the water column thickness under the deeper regions of PGIS is improved, for e.g., by implementing a ca. 600 m deepinner sill (Tinto et al., 2015) situated approximately (>200 m) regions of the ice shelf, with sub-100 m values along the flanks and in the central zone (Fig. E1(a-c)). More critically, this misrepresentation results in the formation of an artificial inner sill, which, if not accounted for, would block the inflow of the warm and dense (27.9 kg/m^3) Atlantic Water at depth (Fig. E1(d)) that contacts the grounding line (Washam et al., 2018; Prakash et al., 2023). Therefore, addressing these artefacts is imperative to ensuring that the model is able to reproduce the fundamental aspects of glacial fjord circulation (Sciascia et al., 2013). Tinto et al. (2015) used aerogravity data from Operation IceBridge to model the bathymetry underneath the ice shelf. Importantly, their findings revealed the presence of an inner sill, ca. 540 – 610 m deep, located about 25 km from the GL which is not included in the BedMachine v3 dataset (Prakash et al., 2022). Initial and lateral ocean boundary conditions, and atmospheric and sea ice surface boundary conditions are derived from a suite of high-resolution regional (ocean, tidal, atmospheric and sea ice) models (downscaling procedure is detailed in Prakash et al. (2022)) grounding line. In this study, we implement the smoothed bathymetry product generated by Prakash et al. (2022), wherein, the BedMachine bathymetry has been modified to account for the unrealistic sub-ice shelf water column thickness and further integrated with the aerogravity data from Tinto et al. (2015).

Experiment Design

Table 1. The ensemble averaged June-July-August (JJA) mean discharge (Q_{sg}) magnitude for the Petermann catchment [in m^3/s] that is distributed uniformly across the GL grounding line for present conditions (1995-2014), future conditions (2081-2100, under the RCP 8.5 scenario), a control scenario, and a median scenario.

Experiment	JJA-mean Q_{sg} [m^3/s]	JJA-mean time period
Control	N/A	N/A
Q_{sg} -present	372	1995–2014
Q_{sg} -median	1563	N/A
Q_{sg} -RCP 8.5	2754	2081-2100

The model is run from July 01, 2014 00:00:00 UTC – January 01, 2017 00:00:00 UTC with hourly oceanic (temperature, salinity, velocities, and sea surface elevation), 3-hourly atmospheric (heat and freshwater fluxes, and wind speeds), and daily averaged sea ice (sea ice concentration, thickness, and velocities, and bulk ice salinity) forcings obtained from respective high-resolution regional models over this period (Prakash et al., 2022, 2023), and without Q_{sg} discharge, which represents the *Control* experiment (Table 1). To allow

investigations of the response of **PGIS-ice shelf** basal melt to variations in Q_{sg} **subglacial**
 **discharge**, our FVCOM setup has been augmented to include Q_{sg} **discharge** across the
 ca. 20 km wide **PGIS-GL-Three- Q_{sg} -grounding line, Three discharge** experiments are
 initialised from the stable model solution of the *Control* experiment from January 01,
 2016 00:00:00 UTC and run until January 01, 2017 00:00:00 UTC (Table 1). Lacking
 information regarding the precise location(s) of emergence of subglacial meltwater across
 the **PGIS-GL- Q_{sg} -grounding line, discharge** is injected uniformly across the 90 cells
 (hereafter denoted by $\Delta_{Q_{sg}}$) representing the **GL-grounding line** (Fig. 1b). At each $\Delta_{Q_{sg}}$,
 discharge is vertically uniform across the 23 σ -layers. The salinity at each $\Delta_{Q_{sg}}$ is set to 0,
 and the temperature is set to the (in-situ) pressure dependent freezing point of freshwater.
 The June-July-August (JJA)-mean Q_{sg} **discharge** values presented in Table 1 represent
 the daily Q_{sg} **discharge** values over the summer (JJA) months used in our model. Outside
 of this period, Q_{sg} **discharge** is set to zero (similar to Cai et al. (2017)). Results for the
 *Control* and Q_{sg} **discharge** experiments are presented from the final year of the simulation
 (January 01, 2016 – January 01, 2017). We note that the pre-2010 ice shelf geometry
 provided by BedMachine v3 is not consistent with the forcing period(s). Change in ice
 shelf geometry (e.g., due to calving) has been shown to impact fjord circulation and
 basal melt rates (Poinelli et al., 2023). However, the two prior large calving events in
 2010 and 2012 (Münchow et al., 2014) removed the softer and thinner (<150 m) outer
 sections of the ice shelf (Hill et al., 2018). As such, we posit that the findings of our
 study – discharge driven regime change in heat flux efficiency in the ice shelf cavity, and
 amplified channelized melting under the deeper drafts – are robust, irrespective of the
 geometry used.
The present day JJA-mean Q_{sg} **discharge** (Q_{sg} -present) and its increase in a future
 warming climate under the RCP 8.5 scenario (Q_{sg} -RCP 8.5) for the Petermann catchment
 is obtained from Slater et al. (2020) (Table 1). Here, a subset of the Coupled Model
 Intercomparison Project (CMIP5) Atmosphere and Ocean general circulation models
 (AOGCMs) (Barthel et al., 2020; Fig. 1c) is used to force the Modèle Atmosphérique
 Régional (MAR) 3.9.6 (Fettweis et al., 2013) for the period 1950 – 2100. The projected
 MAR meltwater runoff over the Petermann catchment is then bias corrected to ensure
 that it is consistent with the best present-day (1995–2014) runoff estimate, and to also
 enable a smooth transition from present to RCP 8.5 forcing. It is then summed up over
 the Petermann catchment to give the Q_{sg} **discharge** estimates. We note that the modelled
 runoff outputs were stored as annual means and were converted to the JJA-mean runoff by
 multiplying the output by a factor of 365/92, where 365 = number of days in a year, and
 92 = number of days in JJA. This coarse conversion introduces a minor discrepancy be-
 tween the converted and true modelled JJA-mean values; however, we maintain that this
 does not significantly impact the findings of our study. While inter-model differences
 can be large (Fig.1 (c)), all five CMIP5 AOGCMs were used to prepare the multi-model
 mean Q_{sg} **discharge** used in this study. Note that the median Q_{sg} **discharge** magnitude
 (Q_{sg} -median) is not obtained from Slater et al. (2020), but is constructed using the Q_{sg} -
 present and Q_{sg} -RCP 8.5 magnitudes (as median ~~(of Q_{sg} -present and Q_{sg} -RCP 8.5)~~;
 ~~and as such,~~ As such, it represents an intermediate future mean Q_{sg} **discharge** mag-
 nitude, however, it does not correspond to any known projection scenario or a particular
 time period (Table 1).
**Diagnostics of basal melt rate and its drivers**
For the investigation of **PGIS-ice shelf** basal melt rate, its expression in the thermody-
 namic framework (Holland and Jenkins, 1999; Zhou and Hattermann, 2020) of the prim-
 itive equation setting is recalled:

$$\rho_{fw} m_b L = -\rho_{sw} c_w \Gamma_T u^* \Delta T. \quad (1)$$

Here, ρ_{fw} and ρ_{sw} are the densities of freshwater and ocean water, respectively. L is the latent heat of fusion of ice, c_w is the specific heat capacity of ocean water, and Γ_T is a non-dimensional heat-transfer coefficient. m_b is basal melt rate, controlled primarily by friction velocity (u^*) and thermal driving (ΔT), and where

$$u^* = \sqrt{C_D(u_w^2 + u_{res}^2)}, \quad \Delta T = T_b - T_w, \quad (2)$$

$$T_b = \lambda_1 S_b + \lambda_2 + \lambda_3 P_b, \quad (3)$$

$$\rho_{fw} m_b S_b = -\rho_{sw} \Gamma_S u^* (S_b - S_w). \quad (4)$$

Above, C_D is the drag coefficient, u_w is the velocity magnitude some distance away from the ice shelf-ocean boundary, and u_{res} is the velocity of small sub-grid scale residual currents (Asay-Davis et al., 2016). T_b , S_b , and P_b are the potential temperature, salinity, and pressure at the ice shelf-ocean boundary, and T_w and S_w are the potential temperature and salinity some distance away from it. λ_1 , λ_2 , and λ_3 , respectively, are the slope, intercept, and pressure coefficient of the liquidus, and Γ_S is a non-dimensional salt-transfer coefficient. Note that T_w , S_w , and u_w are obtained from the first (i.e. uppermost) σ -layer. Furthermore, Γ_T and Γ_S are application specific, and are herein tuned according to the applied boundary conditions, PGIS-Petermann ice shelf basal topography, and mixing schemes so as to be consistent with the contemporary observational estimates of PGIS the Petermann ice shelf basal melt (see Table A1). The drivers of basal melt rate (\$m_b\$ ), namely, friction velocity (\$u^*\$ and) and thermal driving (\$\Delta T\$ ) are used to investigate the impact of \$Q_{sg}\$ discharge on the summer mean \$m_b\$ basal melt rate.

Basal channel Changes in ice volume changes above basal channels over time
We calculate the changes in the annual mean channel averaged ice volume above basal
channels over time ($V_c(N)$) to determine the number of years (N) it would take for $V_c(N)$
to become 0 as

$$0 = V_i + (N \times V_a) - (N \times V_m), \quad (5)$$

$$\implies N = \frac{V_i}{V_m - V_a} \quad (6)$$

where, V_i is the initial (contemporary) channel ice volume ice volume above a channel
expressed as L_c (channel length [m]) \times W_c (mean channel width [m]) \times H_c (mean chan-
nel depth [m]). V_a is the annual mean volume of ice advected downstream of the GL
grounding line per year expressed as AR_c (annual-mean advection rate [m/yr]) \times W_c
621 \times H_c . V_m is the annual mean volume of ice melted per year expressed as m_{bc} (annual
mean channel averaged basal melt rate [m/yr]) \times $L_c \times W_c$. Dimensions associated with
each channel section are detailed in Table 2. Note that W_c drops out of the calculation,
and as such, is not detailed in Table 2.

Table 2. Dimensions associated with the PGIS-Petermann ice shelf basal channel sections illustrated in Fig. 6(a). Channel section along the western, central, and eastern region are termed as S_w , S_c , and S_e , respectively.

Channel Name	L_c [m]	H_c [m]
S_w	30000	241
S_c	25000	271
S_e	28000	331

Data availability
Our study is based on numerical modelling and we provide the model input files (<https://zenodo.org/doi/10.5281/zenodo.12803093>) and code (see Code availability below) that
627 //zenodo.org/doi/10.5281/zenodo.12803093) and code (see Code availability below) that
are required to reproduce our simulations.
Code availability
The open source code Finite Volume Community Ocean Model version 4.0 (FVCOM
v4.0), augmented by both the ice shelf and sea ice modules, that has been used to con-
duct the numerical experiments is publicly available at: [https://github.com/abhay26992/](https://github.com/abhay26992/FVCOM_Petermann_Code.git)
FVCOM_Petermann_Code.git
References
- Asay-Davis XS, Cornford SL, Durand G, Galton-Fenzi BK, Gladstone RM, Gudmundsson GH,
Hattermann T, Holland DM, Holland D, Holland PR, Martin DF, Mathiot P, Pattyn F, Seroussi
H. 2016. Experimental design for three interrelated marine ice sheet and ocean model in-
tercomparison projects: MISMIP v. 3 (MISMIP+), ISOMIP v. 2 (ISOMIP+) and MIS-
OMIP v. 1 (MISOMIP1). *Geoscientific Model Development* 9:2471–2497. doi:10.5194/
gmd-9-2471-2016.
- Aschwanden A, Fahnestock MA, Truffer M, Brinkerhoff DJ, Hock R, Khroulev C, Mottram R,
Khan SA. 2019. Contribution of the Greenland Ice Sheet to sea level over the next millennium.
*Science advances* 5:eaav9396.
- Bamber JL, Westaway RM, Marzeion B, Wouters B. 2018. The land ice contribution to sea level
during the satellite era. *Environmental Research Letters* 13. doi:10.1088/1748-9326/aac2f0.
- Banwell AF, MacAyeal DR. 2015. Ice-shelf fracture due to viscoelastic flexure stress induced by
fill/drain cycles of supraglacial lakes. *Antarctic Science* 27:587–597.
- Barthel A, Agosta C, Little CM, Hattermann T, Jourdain NC, Goelzer H, Nowicki S, Seroussi H,
Straneo F, Bracegirdle TJ. 2020. CMIP5 model selection for ISMIP6 ice sheet model forcing:
Greenland and Antarctica. *The Cryosphere* 14:855–879.
- Bell RE, Banwell AF, Trusel LD, Kingslake J. 2018. Antarctic surface hydrology and impacts on
ice-sheet mass balance. *Nature Climate Change* 8:1044–1052.
- Briner JP, Cuzzone JK, Badgeley JA, Young NE, Steig EJ, Morlighem M, Schlegel N, Hakim GJ,
Schaefer JM, Johnson JV, Lesnek AJ, Thomas EK, Allan E, Bennike O, Cluett AA, Csatho B,
de Vernal A, Downs J, Larour E, Nowicki S. 2020. Rate of mass loss from the Greenland Ice
Sheet will exceed Holocene values this century. *Nature* doi:10.1038/s41586-020-2742-6.
- Cai C, Rignot E, Menemenlis D, Nakayama Y. 2017. Observations and modeling of ocean-
induced melt beneath Petermann Glacier Ice Shelf in northwestern Greenland. *Geophysical*
*Research Letters* 44:8396–8403.
- Chen C, Huang H, Beardsley RC, Liu H, Xu Q, Cowles G. 2007. A finite volume numerical
approach for coastal ocean circulation studies: Comparisons with finite difference models.
*Journal of Geophysical Research: Oceans* 112.
- Ciraci E, Rignot E, Scheuchl B, Tolpekin V, Wollersheim M, An L, Milillo P, Bueso-Bello JL,
Rizzoli P, Dini L. 2023. Melt rates in the kilometer-size grounding zone of petermann glacier,
greenland, before and during a retreat. *Proceedings of the National Academy of Sciences*
120:e2220924120.
- Cook SJ, Christoffersen P, Todd J, Slater D, Chauché N. 2020. Coupled modelling of sub-
glacial hydrology and calving-front melting at Store Glacier, West Greenland. *The Cryosphere*
14:905–924.
- Davis PE, Nicholls KW, Holland DM, Schmidt BE, Washam P, Riverman KL, Arthern RJ,
Vaňková I, Eayrs C, Smith JA, et al. 2023. Suppressed basal melting in the eastern Thwaites
Glacier grounding zone. *Nature* 614:479–485.
- DeConto RM, Pollard D, Alley RB, Velicogna I, Gasson E, Gomez N, Sadai S, Condrón A,
Gilford DM, Ashe EL, et al. 2021. The Paris Climate Agreement and future sea-level rise
from Antarctica. *Nature* 593:83–89.
- Dow CF, Lee WS, Greenbaum JS, Greene CA, Blankenship DD, Poinar K, Forrest AL, Young
DA, Zappa CJ. 2018. Basal channels drive active surface hydrology and transverse ice shelf
fracture. *Science Advances* 4:eaa07212.
- Drews R, Pattyn F, Hewitt I, Ng F, Berger S, Matsuoka K, Helm V, Bergeot N, Favier L, Neckel
680 N. 2017. Actively evolving subglacial conduits and eskers initiate ice shelf channels at an
681 Antarctic grounding line. *Nature communications* 8:15228.
- Ehrenfeucht S, Morlighem M, Rignot E, Dow CF, Mouginot J. 2023. Seasonal acceleration of
Petermann Glacier, Greenland, from changes in subglacial hydrology. *Geophysical Research*
*Letters* 50:e2022GL098009.
- Fettweis X, Franco B, Tedesco M, Van Angelen J, Lenaerts JT, van den Broeke MR, Gallée H.
2013. Estimating the Greenland ice sheet surface mass balance contribution to future sea level
rise using the regional atmospheric climate model MAR. *The Cryosphere* 7:469–489.
- Forster RR, Box JE, Van Den Broeke MR, Miège C, Burgess EW, Van Angelen JH, Lenaerts JT,
Koenig LS, Paden J, Lewis C, et al. 2014. Extensive liquid meltwater storage in firn within the
Greenland ice sheet. *Nature Geoscience* 7:95–98.
- Fox-Kemper B, Hewitt HT, Xiao C, Aðalgeirsdóttir G, Drijfhout SS, Edwards TL, Golledge NR,
Hemer M, Kopp RE, Krinner G, Mix A, Notz D, Nowicki S, Nurhati IS, Ruiz L, Sallée JB,
BA SA, Y Y. 2021. Ocean, cryosphere and sea level change. In *Climate Change 2021: The*
*Physical Science Basis. Contribution of Working Group I to the Sixth Assessment Report of*
*the Intergovernmental Panel on Climate Change [Masson-Delmotte, V., P. Zhai, A. Pirani,*
*S.L. Connors, C. Péan, S. Berger, N. Caud, Y. Chen, L. Goldfarb, M.I. Gomis, M. Huang, K.*
*Leitzell, E. Lonnoy, J.B.R. Matthews, T.K. Maycock, T. Waterfield, O. Yelekçi, R. Yu, and B.*
*Zhou (eds.)]. Cambridge University Press, Cambridge, United Kingdom and New York, NY,*
*USA, pp. 1211–1362 doi:10.1017/9781009157896.011.*
- Frémand AC, Fretwell P, Bodart JA, Pritchard HD, Aitken A, Bamber JL, Bell R, Bianchi C,
Bingham RG, Blankenship DD, et al. 2023. Antarctic bedmap data: Findable, accessible,
interoperable, and reusable (fair) sharing of 60 years of ice bed, surface, and thickness data.
*Earth System Science Data* 15:2695–2710.
- Gadi R, Rignot E, Menemenlis D. 2023. Modeling ice melt rates from seawater intrusions
in the grounding zone of petermann gletscher, greenland. *Geophysical Research Letters*
50:e2023GL105869.
- Goelzer H, Nowicki S, Payne A, Larour E, Seroussi H, Lipscomb WH, Gregory J, Abe-Ouchi A,
Shepherd A, Simon E, Agosta C, Alexander P, Aschwanden A, Barthel A, Calov R, Chambers
C, Choi Y, Cuzzzone J, Dumas C, Edwards T, Felikson D, Fettweis X, Golledge NR, Greve R,
Humbert A, Huybrechts P, Le clec’h S, Lee V, Leguy G, Little C, Lowry DP, Morlighem M,
Nias I, Quiquet A, Rückamp M, Schlegel NJ, Slater DA, Smith RS, Straneo F, Tarasov L, van de
Wal R, van den Broeke M. 2020. The future sea-level contribution of the Greenland ice sheet:
a multi-model ensemble study of ISMIP6. *The Cryosphere*. doi:10.5194/tc-14-3071-2020.
- Goldberg DN, Twelves AG, Holland PR, Wearing MG. 2023. The non-local impacts of antarctic
subglacial runoff. *Journal of Geophysical Research: Oceans* 128:e2023JC019823.
- Golledge NR, Keller ED, Gomez N, Naughten KA, Bernales J, Trusel LD, Edwards TL. 2019.
Global environmental consequences of twenty-first-century ice-sheet melt. *Nature* 566:65–72.
- Greene CA, Gardner AS, Wood M, Cuzzzone JK. 2024. Ubiquitous acceleration in Greenland Ice
Sheet calving from 1985 to 2022. *Nature* 625. doi:doi.org/10.1038/s41586-023-06863-2.
Gwyther DE, Dow CF, Jendersie S, Gourmelen N, Galton-Fenzi BK. 2023. Subglacial freshwater
drainage increases simulated basal melt of the Totten Ice Shelf. *Geophysical Research Letters*
50:e2023GL103765.
Hattermann T, Isachsen PE, von Appen WJ, Albretsen J, Sundfjord A. 2016. Eddy-driven recir-
culation of atlantic water in fram strait. *Geophysical Research Letters* 43:3406–3414.
Hill EA, Gudmundsson GH, Carr JR, Stokes CR. 2018. Velocity response of Petermann Glacier,
northwest Greenland, to past and future calving events. *The Cryosphere* 12:3907–3921.
Holland DM, Jenkins A. 1999. Modeling thermodynamic ice–ocean interactions at the base of
an ice shelf. *Journal of physical oceanography* 29:1787–1800.
Hunke EC, Lipscomb WH, Turner AK, Jeffery N, Elliott S. 2015. Cice: The los alamos sea ice
model documentation and software user’s manual version 5.1 la-cc-06-012. T-3 Fluid Dynam-
ics Group, Los Alamos National Laboratory 675:15.
Jakobsson M, Mayer L, Coakley B, Dowdeswell JA, Forbes S, Fridman B, Hodnesdal H,
Noormets R, Pedersen R, Rebesco M, et al. 2012. The International Bathymetric Chart of
the Arctic Ocean (IBCAO) version 3.0. *Geophysical Research Letters* 39.
Johnson H, Münchow A, Falkner K, Melling H. 2011. Ocean circulation and properties in Peter-
manns Fjord, Greenland. *Journal of Geophysical Research: Oceans* 116.
Le Brocq AM, Ross N, Griggs JA, Bingham RG, Corr HF, Ferraccioli F, Jenkins A, Jordan TA,
Payne AJ, Rippin DM, et al. 2013. Evidence from ice shelves for channelized meltwater flow
beneath the Antarctic Ice Sheet. *Nature Geoscience* 6:945–948.
Lepp A, Simkins L, Anderson J, Clark R, Wellner J, Hillenbrand C, Smith J, Lehrmann A, Totten
R, Larter R, et al. 2022. Sedimentary signatures of persistent subglacial meltwater drainage
from Thwaites Glacier, Antarctica. *Frontiers in Earth Science* 10:863200.
Li T, Robinson LF, MacGilchrist GA, Chen T, Stewart JA, Burke A, Wang M, Li G, Chen J, Rae
JW. 2023. Enhanced subglacial discharge from Antarctica during meltwater pulse 1A. *Nature*
*Communications* 14:7327.
Millan R, Jager E, Mouginit J, Wood MH, Larsen SH, Mathiot P, Jourdain NC, Bjørk A. 2023.
Rapid disintegration and weakening of ice shelves in North Greenland. *Nature Communica-*
*tions* doi:10.1038/s41467-023-42198-2.
Millan R, Mouginit J, Derkacheva A, Rignot E, Milillo P, Ciraci E, Dini L, Bjørk A. 2022. On-
going grounding line retreat and fracturing initiated at the Petermann Glacier Ice Shelf, Green-
land, after 2016. *The Cryosphere* 16:3021–3031.
Morlighem M, Williams CN, Rignot E, An L, Arndt JE, Bamber JL, Catania G, Chauché N,
Dowdeswell JA, Dorschel B, et al. 2017. Bedmachine v3: Complete bed topography and
ocean bathymetry mapping of Greenland from multibeam echo sounding combined with mass
conservation. *Geophysical research letters* 44:11–051.
Mouginit J, Rignot E, Bjørk AA, Van den Broeke M, Millan R, Morlighem M, Noël B, Scheuchl
B, Wood M. 2019. Forty-six years of Greenland Ice Sheet mass balance from 1972 to 2018.
*Proceedings of the National Academy of Sciences* 116:9239–9244.
Münchow A, Padman L, Fricker HA. 2014. Interannual changes of the floating ice shelf of Pe-
termann Gletscher, North Greenland, from 2000 to 2012. *Journal of Glaciology* 60:489–499.
Muntjewerf L, Petrini M, Vizcaino M, Ernani da Silva C, Sellevold R, Scherrenberg MD, Thayer-
Calder K, Bradley SL, Lenaerts JT, Lipscomb WH, et al. 2020. Greenland Ice Sheet contribu-
tion to 21st century sea level rise as simulated by the coupled CESM2. 1-cism2. 1. *Geophysical*
*Research Letters* 47:e2019GL086836.
Narkevic A, Csatho B, Schenk T. 2023. Rapid basal channel growth beneath Greenland’s longest
floating ice shelf. *Geophysical Research Letters* 50:e2023GL103226.
Noël B, van de Berg WJ, Lhermitte S, van den Broeke MR. 2019. Rapid ablation zone expansion
amplifies north greenland mass loss. *Science advances* 5:eaaw0123.
Noël B, van Wessem JM, Wouters B, Trusel L, Lhermitte S, van den Broeke MR. 2023. Higher
Antarctic ice sheet accumulation and surface melt rates revealed at 2 km resolution. *Nature*
*communications* 14:7949.
Otto J, Holmes FA, Kirchner N. 2022. Supraglacial lake expansion, intensified lake drainage
frequency, and first observation of coupled lake drainage, during 1985–2020 at Ryder Glacier,
Northern Greenland. *Frontiers in Earth Science* 10:978137.
Padman L, Erofeeva S. 2004. A barotropic inverse tidal model for the arctic ocean. *Geophysical*
*Research Letters* 31.
- Pelle T, Greenbaum J, Ehrenfeucht S, Dow C, McCormack F. 2024. Subglacial discharge accel-
erates dynamic retreat of aurora subglacial basin outlet glaciers, east antarctica, over the 21st
century. *Journal of Geophysical Research: Earth Surface* 129:e2023JF007513.
- Pelle T, Greenbaum JS, Dow CF, Jenkins A, Morlighem M. 2023. Subglacial discharge accelerates
future retreat of Denman and Scott Glaciers, East Antarctica. *Science Advances* 9:ead9014.
- Poinelli M, Nakayama Y, Larour E, Vizcaino M, Riva R. 2023. Ice-front retreat controls
on ocean dynamics under larsen c ice shelf, antarctica. *Geophysical Research Letters*
50:e2023GL104588.
- Prakash A, Zhou Q, Hattermann T, Bao W, Graverson R, Kirchner N. 2022. A nested high-
resolution unstructured grid 3-D ocean-sea ice-ice shelf setup for numerical investigations of
the Petermann ice shelf and fjord. *MethodsX* 9:101668.
- Prakash A, Zhou Q, Hattermann T, Kirchner N. 2023. Impact of the Nares Strait sea ice arches
on the long-term stability of the Petermann Glacier ice shelf. *The Cryosphere* 17:5255–5281.
doi:10.5194/tc-17-5255-2023.
- Rawlins LD, Rippin DM, Sole AJ, Livingstone SJ, Yang K. 2023. Seasonal evolution of the
supraglacial drainage network at Humboldt Glacier, North Greenland, between 2016 and 2020.
*The Cryosphere Discussions* :1–32.
- Rignot E, Fenty I, Xu Y, Cai C, Kemp C. 2015. Undercutting of marine-terminating glaciers in
West Greenland. *Geophysical Research Letters* 42:5909–5917.
- Rignot E, Steffen K. 2008. Channelized bottom melting and stability of floating ice shelves.
*Geophysical Research Letters* 35.
- Rosevear MG, Gayen B, Galton-Fenzi BK. 2022. Regimes and transitions in the basal melting of
antarctic ice shelves. *Journal of Physical Oceanography* 52:2589–2608.
- Scambos T, Fricker HA, Liu CC, Bohlander J, Fastook J, Sargent A, Massom R, Wu AM. 2009.
Ice shelf disintegration by plate bending and hydro-fracture: Satellite observations and model
results of the 2008 Wilkins ice shelf break-ups. *Earth and Planetary Science Letters* 280:51–60.
- Schmidt BE, Washam P, Davis PE, Nicholls KW, Holland DM, Lawrence JD, Riverman KL,
Smith JA, Spears A, Dichek D, et al. 2023. Heterogeneous melting near the Thwaites Glacier
grounding line. *Nature* 614:471–478.
- Sciascia R, Straneo F, Cenedese C, Heimbach P. 2013. Seasonal variability of submarine melt
rate and circulation in an east Greenland fjord. *Journal of Geophysical Research: Oceans*
118:2492–2506.
- Shroyer EL, Padman L, Samelson R, Münchow A, Stearns LA. 2017. Seasonal control of Peter-
mann Gletscher ice-shelf melt by the ocean’s response to sea-ice cover in Nares Strait. *Journal*
*of Glaciology* 63:324–330.
- Shu Q, Wang Q, Årthun M, Wang S, Song Z, Zhang M, Qiao F. 2022. Arctic Ocean Amplification
in a warming climate in CMIP6 models. *Science Advances* 8:eabn9755.
- Slater D, Nienow P, Cowton T, Goldberg D, Sole A. 2015. Effect of near-terminus subglacial
hydrology on tidewater glacier submarine melt rates. *Geophysical Research Letters* 42:2861–
2868.
- Slater D, Straneo F. 2022. Submarine melting of glaciers in Greenland amplified by atmospheric
warming. *Nature Geoscience* 15:794–799.
- Slater DA, Felikson D, Straneo F, Goelzer H, Little CM, Morlighem M, Fettweis X, Nowicki S.
2020. Twenty-first century ocean forcing of the Greenland ice sheet for modelling of sea level
contribution. *The Cryosphere* 14:985–1008.
- Tinto KJ, Bell RE, Cochran JR, Münchow A. 2015. Bathymetry in Petermann fjord from Oper-
ation IceBridge aerogravity. *Earth and Planetary Science Letters* 422:58–66.
- Turton JV, Hochreuther P, Reimann N, Blau MT. 2021. The distribution and evolution of
supraglacial lakes on 79 N Glacier (north-eastern Greenland) and interannual climatic con-
trols. *The Cryosphere* 15:3877–3896.
- Washam P, Münchow A, Nicholls KW. 2018. A decade of ocean changes impacting the ice shelf
of Petermann Gletscher, Greenland. *Journal of Physical Oceanography* 48:2477–2493.
- Washam P, Nicholls KW, Münchow A, Padman L. 2020. Tidal modulation of buoyant flow and
basal melt beneath Petermann Gletscher ice Shelf, Greenland. *Journal of Geophysical Re-*
*search: Oceans* 125:e2020JC016427.
- Wearing M, Stevens L, Dutrieux P, Kingslake J. 2021. Ice-shelf basal melt channels stabilized by
secondary flow. *Geophysical Research Letters* 48:e2021GL094872.
- Willis MJ, Herried BG, Bevis MG, Bell RE. 2015. Recharge of a subglacial lake by surface
meltwater in northeast Greenland. *Nature* 518:223–227.
- Wilson N, Straneo F, Heimbach P. 2017. Satellite-derived submarine melt rates and mass balance
(2011–2015) for Greenland’s largest remaining ice tongues. *The Cryosphere* 11:2773–2782.
- Zeising O, Neckel N, Dörr N, Helm V, Steinhage D, Timmermann R, Humbert A. 2023. Extreme
melting at Greenland’s largest floating ice tongue. *EGUsphere* :1–35.
- Zhou Q, Hattermann T. 2020. Modeling ice shelf cavities in the unstructured-grid, finite volume
community ocean model: Implementation and effects of resolving small-scale topography.
*Ocean Modelling* 146:101536.
- Zhou Q, Zhao C, Gladstone R, Hattermann T, Gwyther D, Galton-Fenzi B. 2024. Evaluat-
ing an accelerated forcing approach for improving computational efficiency in coupled ice
sheet–ocean modelling. *Geoscientific Model Development* 17:8243–8265. doi:10.5194/
gmd-17-8243-2024.
**Acknowledgements**
This work was funded by the Swedish Research Council VR grant 2022-03718 to N.K..
Q.Z. was supported by the Research Council of Norway (RCN) project 244319 and
295075. T.H. was supported by the RCN project 314570. We thank Donald Slater for
providing the present-day and RCP 8.5 runoff estimates for the Petermann Glacier. The
simulations were performed on resources provided by UNINETT Sigma2 - the National
Infrastructure for High Performance Computing and Data Storage in Norway. The super-
computer Betzy was used under the project NN9824K.
**Contributions**
All authors contributed to the conceptualization of the study. A.P. designed the exper-
iments with technical input from Q.Z. and T.H.. A.P. prepared and ran the numerical
simulations. A.P. analyzed the model output with scientific advice from Q.Z. and T.H..
859 A.P. and N.K. wrote the manuscript, with contributions from Q.Z. and T.H..

**Competing interests**
The authors declare no competing interests.
**A Model tuning**

Table A1. PGIS Petermann ice shelf thermodynamical and ocean mixing parameters used in this study

Parameter	Value	Description
ρ_{fw}	1000 kg m^{-3}	Freshwater density
L	$3.34 \times 10^5 \text{ J kg}^{-1}$	Latent heat of fusion of ice
ρ_{sw}	1028 kg m^{-3}	Seawater density
c_w	$3974 \text{ J }^\circ\text{C}^{-1} \text{ kg}^{-1}$	Specific heat capacity of seawater
Γ_T	1.2×10^{-2}	Non-dimensional heat-transfer coefficient
C_D	2.5×10^{-3}	Drag coefficient
u_{res}	$1.0 \times 10^{-2} \text{ m s}^{-1}$	Residual velocity
λ_1	$-5.73 \times 10^{-2} \text{ }^\circ\text{C PSU}^{-1}$	Liquidus slope
λ_2	$8.32 \times 10^{-2} \text{ }^\circ\text{C}$	Liquidus intercept
λ_3	$-7.53 \times 10^{-8} \text{ }^\circ\text{C Pa}^{-1}$	Liquidus pressure coefficient
Γ_S	$\Gamma_T/35$	Non-dimensional salt-transfer coefficient
Z_o	1.0×10^{-3}	Roughness length scale
Ro_{min}	2.5×10^{-3}	Roughness minimum
K_m	1.0×10^{-5}	Vertical eddy viscosity
P_v	1.0	Vertical Prandtl Number
P_h	1.0×10^{-1}	Horizontal Prandtl Number
C_h	1.0×10^{-1}	Scaling constant

863 **B Longitudinal ice shelf draft profile**

Figure B1. (a) Map of the PGIS Petermann ice shelf draft with the across-fjord transect ca. 5 km from the GL grounding line shown in Fig. 1(b) overlaid. Locations Apex locations of the four prominent longitudinal ice shelf basal channels at this transect (highlighted in Fig. 5) at this transect are represented using magenta diamonds. From left to right, these are \$L_w\$ (west), \$L_c\$ (central), \$L_e(1)\$ (east-1) and \$L_e(2)\$ (east-2). (b) Along-fjord profile of the PGIS Petermann ice shelf draft over the channel sections (Fig. 6(a)) along the western (\$S_w\$; blue), central (\$S_c\$; black) and eastern (\$S_e\$; red) regions channel sections (Fig. 6(a)) as indicated in panel a.

C Saturation of the thermal pump

Figure C1. Along-fjord summer mean temperature (left column), salinity (middle column), and flow (right column) anomalies (δ) for Q_{sg} -median relative to control (first row) and Q_{sg} -RCP 8.5 relative to control (second row) experiments for the section shown in Fig. 1(b). Black stippled lines in panels a, b, d and e indicate the zero temperature and salinity difference contours. Solid black lines in panels c and f depict the mean isopycnals for the control experiment, whereas the stippled green lines correspond to the Q_{sg} -median (c) and Q_{sg} -RCP 8.5 (f) experiments. Isopycnals are plotted at equal intervals of 0.2 kg/m^3 . The 27.9 kg/m^3 isopycnal corresponds to the dense Atlantic Water that contacts the grounding line. In each panel, PGIS-GL the grounding line (at ca. 600 m depth) is on the right margin and open ocean is to the left. Vertical magenta line in panels a, b, d, and e show the location of the model node ca. 10 km from the GL-grounding line at which the vertical profile of summer mean temperature and salinity are shown for the control, Q_{sg} -present, Q_{sg} -median, and Q_{sg} -RCP 8.5 experiments in Fig. 3(g,h) (cf. Table 1).

865 **D** Trapping Convergence of meltwater

Figure D1. Anomaly maps (δ) of the summer mean surface layer salinity (a,b) and basal melt rate (c,d) for the Q_{sg} -present relative to control (column 1) and Q_{sg} -median relative to control (column 2) experiments. Contours of the ice shelf draft are plotted at 100 m intervals (450 m – 150 m) and overlaid as dotted red and blue lines over the surface layer salinity and melt rate maps, respectively.

E Modified sub-ice shelf water column thickness

Figure E1. (a) BedMachine v3 (Morlighem et al., 2017) water column thickness, and (b) modified BedMachine v3 water column thickness used in this study, obtained from Prakash et al. (2022). (c) Along-fjord profile of the water column thickness for the western (blue), central (black), and eastern (red) sections of the fjord as highlighted in panels (a) and (b). Stippled and solid lines correspond, respectively, to the BedMachine and modified BedMachine profiles. (d) Along-fjord summer mean temperature for the section shown in Fig. 1(b). Mean isopycnals (solid black lines) are overlaid at equal intervals of 0.2 kg/m^3 . The 27.9 kg/m^3 isopycnal surface represents the dense Atlantic Water that contacts the grounding line. Solid red line represents the (unmodified) BedMachine v3 bathymetry, wherein, a synthetic sill blocks the 27.9 kg/m^3 isopycnal surface from contacting the grounding line.

Reviewer #2 (Remarks to the Author):

The impact of subglacial water on ice shelf melt rates has been an understudied topic, but is now gaining prominence in the field with increasing recognition of its impact on ice sheet dynamics. This paper presents interesting insights into the mechanisms by which enhanced subglacial water flow might affect melt rates (i.e. shear-driven turbulence vs. thermal driving) and will therefore be of interest to the research community.

The study is well designed, using a very neat set of experiments to draw interesting conclusions, and the paper is well written: clear, and easy to follow. The figures are also well chosen, and very convincingly underpin the conclusions in the results and discussion sections, although they are not as polished and easy to read as I would like. I make some further suggestions for changes to the text below, but they are only minor. Overall, I think this is a very high quality contribution to the field.

Thank you for taking the time to read our manuscript. We are pleased to hear that you found it interesting and well-written. We agree with all of the suggestions that you have provided and have revised our manuscript accordingly. Please note that line numbers provided below refer to the *track changes* document (appended to this report for convenience), where new text appears in blue and removed text is marked with red strikethrough.

Firstly, we would like to take this opportunity to inform you that during the revision process, we noticed a (minor) mistake in our calculation of the drivers of melt, namely, thermal driving and friction velocity. While the updated numbers are slightly different, we note that this does not impact any of the findings of our study, and the revised figures remain qualitatively identical. We humbly request to proceed with these corrected values, and sincerely apologize for the oversight and for any inconvenience this may cause. Please see the revised percentages in Lines 133 – 134, 167 – 170, 184, 195, and 227, and Figures 1(d), 2(e – l), and 5(c,d).

Please find our detailed (point-by-point) responses below.

Minor comments

Lines 3-4: "contemporary mass loss rates are underestimated". Underestimated by whom? Observational studies, or ice sheet models?

Thank you for pointing this out. Greene et al., 2024 suggest that observational estimates of GrIS mass balance have underestimated recent (1992 – 2020) mass loss by ca. 20%. We have fixed this in the revised manuscript. Please see lines 3 – 4.

Line 14: “ice volume of ca. 4m global mean sea level equivalent” I can’t understand from this sentence which sectors of the ice sheet you are referring to. Please check the reference too – is Morlighem et al. (2017) not more appropriate here (BedMachine, see reference list below)?

We agree with the concerns raised above. In Line 5, we begin the thread about the northern sector of the ice sheet, however, the passage is quite long. Furthermore, the use of “adjacent” is confusing, as it implies the exclusion of the sector Petermann corresponds to. Upon reviewing, we do agree that Morlighem et al. (2017) is appropriate. However, we maintain that Mouginit et al. (2019) is also relevant, as they quantified the sectorial mass balance of the GrIS in unprecedented detail. Herein, the integrated contribution from the northern sector amounts to ca. 4 m. We have fixed both the phrasing and citation in the revised manuscript. Please see lines 14 – 17.

Line 16: “Erstwhile” is not wrong here, but it’s a rarely used word. You could consider replacing with “Formerly” or “Previously” to aid non-native speakers.

We agree, and have replaced it with Formerly. Please see line 20.

Line 21: “thinning it from ca. 600 m near the GL to ca. 100 m near the calving front.” Phrased this way, the sentence can be misinterpreted... do you mean “with ice thickness over the channels of 600 m near the GL to ca. 100 m near the calving front”?

Reviewer 1 has also raised concerns regarding the phrasing of this sentence, and we apologize for the confusion it has caused. In Lines 18-21, we aimed at detailing the average thickness of the ice shelf, extending from the grounding line (400 – 600 m) to the calving front (50 – 100 m). However, we also do not want to attribute the thinning completely to basal melting (Reviewer 1). We have rephrased this sentence in the revised manuscript, bearing in mind both the comments. We first document the average thickness gradient from the grounding line to the calving front (driven by all factors). Thereafter, we describe the basal melt channel geometries. Please see lines 22 – 28.

Figure 1b: I suggest labelling the calving front with a date.

Thank you for the suggestion. We have now included the date (2009) for improved clarity. Please see Figure 1(b).

Line 34-36: "estimated area-averaged melt rate maximum of up to 170 m/yr" the phrase "area-average" initially implies that this was estimated over a large sector of the shelf. I suggest rephrasing slightly to make clear that this estimate was calculated only within the basal channel.

This is true, thank you for noticing this. We have rephrased it in the revised text. Please see lines 44 – 51.

Line 38: "with warming expected to amplify" I suggest clarifying whether you mean atmospheric warming, ocean warming, or both.

Thank you for noticing this. We meant both, and have clarified this in this revised version. Please see line 54.

Final paragraph of introduction (lines 43-59): It's unusual to see the results summarised so thoroughly at the end of the introduction. It might be more helpful to give a very brief summary of your experiment design instead e.g. from line 49 onwards something along the lines of "We performed a control run with zero Q_{sg} , and three test simulations designed to simulate present day subglacial conditions, increased Q_{sg} under high-end scenario climate warming, and an intermediate Q_{sg} scenario between the two."

Thank you, the results are indeed described in detail here, and without proper context (i.e., experiment setup), its purpose is not really served. We have introduced a summary of our experiment design, and have condensed the description of the results in the revised manuscript. Please see lines 59 – 100, which also combines suggestions from both Reviewers 1 and 3.

Paragraph beginning line 62: This paragraph jumps into discussion of modelled melt rates without describing the results clearly first, which means the reader has to work hard to catch up. I suggest an opening sentence or two describing the melt pattern in the control and preset day runs, before going on to compare them to observations.

Thank you for providing this perspective. We agree with providing more context first, and to also provide a more detailed comparison between data and our modelled results (see Reviewer 3's comment). We have fixed this paragraph in the revised manuscript. Please see lines 103 – 127.

Figure 2: You could make the model results larger and easier to see by removing unnecessarily duplicated color scales and axis labels (see also general comments on figures below).

Thank you for the recommendations. We have removed the duplicated axis labels in favour of common X (Longitude) and Y (Latitude) axis labels. The duplicated color scales have also been replaced with one common color scale for each set of panels representing changes in melt rates, friction velocity, and thermal driving. A delta symbol has also been added to the c-axis labels to denote changes. Please see Figure 2.

Figure 3: The isopycnal label (27.9) is large and distracting (see also general comments on figures below).

Thank you, we have replaced it with a smaller font size that is consistent with the rest of the labels. Please see Figure 3.

Figure 5: The lines here can be a little hard to see – consider changing the color scheme to improve readability and increasing line thickness (see also general comments on figures below).

Thank you for the feedback. We have changed the color scheme and increased the line thickness. Please see Figure 5.

Line 151 & Figure 6: The two sets of location names are a bit confusing here (Sw, Sc, and Se vs. Lw, Lc, and Le). Are they both needed? Lw, Lc, and Le are barely mentioned in the text, and could perhaps be removed?

We agree, having two sets of location names is confusing and not required. We have removed all occurrences of Lw, Lc, and Le. Labels in panels a – d, Figure 5 have also been removed. We have kept the green stripes in Figure 5(a) to highlight the locations of the prominent basal channels.

Figure 6: The pink dots in panel 6a are not described in the caption, and the symbols marking different lines in panel 6b are far too small to be readable (see also general comments on figures below)

Thank you for noticing this, we have now added the description to the caption. We noticed that the cumbersome legend of Figure 6(b) can be conveniently added to the figure description, and as such has been removed from the panel.

Line 222: Should read “buoyant meltwater rises”

Thank you for noticing this. Following comments from Reviewer 3, we have rephrased this sentence. As such, this fix is no longer applicable.

Discussion line 219-240 & Figure 6b: Given the very straight erosion trajectory in figure 6b, I initially thought you must have applied the summer value for Q_{sg} continuously over the whole year. Looking at the methods it seems that Q_{sg} does go to zero outside of the summer, so why don't we see a seasonal cycle in the channel volume? Presumably over winter the melt substantially decreases but the ice advection should change much less? I think the discussion section could do with a bit more context on seasonality, including how seasonally varying ocean temperatures affect erosion rates.

Thank you for raising this concern. We do not see a seasonal cycle since we use mean annual basal melt rates and advection rates, and calculate the yearly change in ice volume above the channels until it reaches 0. The annual mean basal melt rates in these channels are still considerably high because of the discharge-driven strong summer signal. While it does not explicitly concern the issue raised in this comment, we note that we are also limited by the lack of systematic high (temporal) resolution data for present and future warming scenarios that model the seasonal variability of subglacial discharge (gaussian-like profiles) and ice advection rates.

Indeed, during winter, basal melt rates are lower, and ice advection rate does not vary significantly. Muenchow et al. (2012) have reported mean advection rate of ca. 1 km/yr with a modest seasonal modulation of ca. 0.1 km/yr. Our model is consistent with data from the fjord which suggests that seasonal ocean temperatures in Petermann Fjord vary by ca. 0.1 °C, which does not have as significant an impact on basal melt as subglacial discharge. We have now expanded our discussion to quantify the impact of seasonal changes in the fjord (ocean and subglacial discharge) on erosion rates in the revised text. Please see lines 328 – 339.

Line 266-267: There has been further recent work on this subject in Antarctica beyond your references, so I think the statement "insights into this process are limited to tacit reasoning" is a little unfair. I suggest reading and citing: Goldberg et al. 2023, Gwyther et al. 2023, and Pelle et al. 2024.

Thank you for providing us with this information. Upon reviewing the recent literature, we have revised the text and added these as references. Please see lines 405 – 415.

All figures: Please take a look at the text within your figures and make sure it meets minimum font size guidelines, but is not excessively large and distracting (as with some subpanel labels). Some axis labels and legends are very small indeed. In general, using consistent font across all figure will also improve perception of your figure quality. You could also improve clarity of

figures 2, 3 & 4 by indicating on colour scales where a variable is absolute or relative (e.g. "Salinity" in panel 4a and "Change in salinity" in panel 4b, use a delta symbol if space is tight).

Thank you for providing detailed suggestions. We have now implemented a consistent font size across all our figures which meets the font size guidelines. Absolute and relative values are now clearly differentiated by the use of a delta symbol.

References

Goldberg, Daniel N., Andrew G. Twelves, Paul R. Holland, and Martin G. Wearing. "The non-local impacts of Antarctic subglacial runoff." *Journal of Geophysical Research: Oceans* 128, no. 10 (2023): e2023JC019823.

Gwyther, David E., Christine F. Dow, Stefan Jendersie, Noel Gourmelen, and Benjamin K. Galton-Fenzi. "Subglacial freshwater drainage increases simulated basal melt of the Totten Ice Shelf." *Geophysical Research Letters* 50, no. 12 (2023): e2023GL103765.

Morlighem, Mathieu, Chris N. Williams, Eric Rignot, Lu An, Jan Erik Arndt, Jonathan L. Bamber, Ginny Catania et al. "BedMachine v3: Complete bed topography and ocean bathymetry mapping of Greenland from multibeam echo sounding combined with mass conservation." *Geophysical research letters* 44, no. 21 (2017): 11-051.

Pelle, T., J_S Greenbaum, S. Ehrenfeucht, C_F Dow, and F_S McCormack. "Subglacial discharge accelerates dynamic retreat of aurora subglacial basin outlet glaciers, East Antarctica, over the 21st century." *Journal of Geophysical Research: Earth Surface* 129, no. 7 (2024): e2023JF007513.

Reviewer #2 (Remarks on code availability):

I have checked that the code is accessible, but having never used the FVCOM model I can't say whether it is sufficient to reproduce the results

Thank you for raising this concern. We can assure that the model code and input files, together with the detailed instructions that accompany them, are sufficient to reproduce the results.

Future runoff increase may destabilize Petermann ice shelf

Abhay Prakash^{1,2,*}, Qin Zhou³, Tore Hattermann^{4,5}, and Nina Kirchner^{1,2}

¹Department of Physical Geography, Stockholm University, Stockholm 10691, Sweden

²Bolin Centre for Climate Research, Stockholm University, Stockholm 10691, Sweden

³Akvaplan-niva, Tromsø 9296, Norway

⁴Norwegian Polar Institute, Tromsø 9296, Norway

⁵Department of Physics and Technology, University of Tromsø, Tromsø 9019, Norway

*Corresponding author, abhay.prakash@natgeo.su.se

Abstract

Increased basal melting of the Petermann ~~Glacier-Ice Shelf (PGIS)-ice shelf~~ is typically attributed to rising ocean temperatures. Although subglacial discharge (Q_{sg}) has been shown to intensify melt, the mechanisms behind this increase and their evolution with increasing Q_{sg} discharge in a warmer climate remain unresolved. Using a 3-D numerical regional ice shelf-ocean model centered on the Petermann Fjord, we show that heightened Q_{sg} discharge under the RCP 8.5 scenario leads to more than threefold increase in summer mean melt when averaged over the deeper (>300 m) drafts compared to conditions without Q_{sg} discharge. Notably, we identify a regime change in heat flux efficiency within the ~~PGIS-cavity-when- Q_{sg} -ice shelf cavity when discharge~~ exceeds the current peak summer value. Here, thermal driving saturates, and ~~Q_{sg} -intensified discharge-intensified~~ currents increase melt by enhancing shear-driven turbulent mixing across the ~~PGIS-ocean-ice shelf-ocean~~ boundary layer. Increases in melt are most profound at the crests of the basal channels, where vigorous meltwater confluence exacerbates friction velocity. We estimate that sustained intensified summer melting over a decade may ~~completely erode the channels, undermining the stability of PGIS, undermine the stability of the ice shelf by allowing channels to cut entirely through it. Challenging conventional attributions of increased basal melting of the ice shelf to ocean warming alone, our results demonstrate how atmospheric warming exacerbates ocean-driven melt processes, and is likely to play a dominant role in amplifying future basal melt.~~ Considering the impact of the channelized basal morphology of ~~PGIS-the ice shelf~~ on the spatial heterogeneity of melt, and projected increases in Q_{sg} discharge, we posit that our results have wider implications for similar 'warm cavity' environments.

Introduction
The Greenland Ice Sheet (~~GrIS~~) is currently the largest contributor to global mean sea
level rise (Bamber et al., 2018; Fox-Kemper et al., 2021), and as ~~contemporary mass loss~~
~~rates observational estimates of recent mass loss~~ are underestimated (Greene et al., 2024),
they are hence likely lower bounds for rates to be observed in decades to come (Briner
et al., 2020; Goelzer et al., 2020). The northern sector of the ~~GrIS-Greenland Ice Sheet~~
hosts several of the last remaining marine terminating outlet glaciers featuring floating
ice tongues (also called ice shelves) that exert buttressing forces to the inland ice. During
2001 ~~—~~ 2021, dominating factors ~~for-of~~ mass loss from these ice shelves were (in de-
creasing order) basal melt, calving, and negative surface mass balance, ~~while between,~~
~~Between~~ 2003 and 2010, collapses were observed at Hagen Bræ, C.H. Ostenfeld glacier,
and Zachariæ Isstrøm (Millan et al., 2023). Dynamic thinning, retreat and even collapse
combined with perturbations of the outlet glacier's grounding ~~lines-(GL)-line~~ can lead to
a loss of backstress and accelerated mass loss from the ~~adjacent-GrIS-interior-sectors~~
~~northern Greenland Ice Sheet sectors~~ (Hill et al., 2018) holding an ice volume of ca.
~~4m-4 m~~ global mean sea level equivalent (~~Mouginot et al., 2019~~)(Morlighem et al., 2017
: Mougnot et al., 2019).
Along the Nares Strait coast of the northern ~~GrIS, Petermann Glacier Ice Shelf (PGIS)~~
~~terminates into Petermann Fjord (PF). Erstwhile Greenland Ice Sheet, Petermann ice~~
~~shelf terminates into the Petermann Fjord. Formerly~~ the longest ice shelf in the northern
hemisphere at ca. 80 km in length, two recent massive episodic calving events in 2010
and 2012 reduced its length by ca. 30 km (Münchow et al., 2014). ~~The ice shelf draft~~
~~thins from ca. 400 – 600 m near the grounding line to ca. 50 – 100 m near the calving~~
~~front. Evenly spaced across its 15-20-15 — 20 km width are 1-2-1 – 2 km wide basal~~
~~melt channels that run along its length (Rignot and Steffen, 2008). These channels carve~~
~~incisions that are incise~~ several-100 m deep into ~~its draft, thinning it from ca. 600 m~~
~~near the GL to ca. 100 m near the calving front. the draft~~ (Rignot and Steffen, 2008
28). Warming of Atlantic water in ~~PF by $\sim 0.3^\circ\text{C}$~~ ~~Petermann Fjord by ca. 0.3°C~~ has been
observed since 1960, with much (~~\sim ca. 0.2°C~~) of the warming occurring in the last
two decades (Millan et al., 2022). Average basal melt rates (~~m_b~~) ~~at the PGIS-GL, at \sim~~
~~at the Petermann ice shelf grounding line, at ca. 600 m below sea level, are currently~~
reported to exceed 17 m/yr, leading Millan et al. (2023) to suggest that increasing ice
shelf basal melt is following increasing ocean temperatures in ~~GrIS fjords. Greenland~~
~~Ice Sheet fjords. Numerical modelling study and satellite observations have revealed~~
~~that highest basal melt rates occur in the kilometer-size grounding zone of the glacier,~~
~~driven by tidally modulated intrusions of seawater~~ (Ciraci et al., 2023; Gadi et al., 2023
37). Seasonal and climate-warming driven long-term changes in regional sea ice cover are
38 also known to enhance ~~m_b basal melting~~. According to model results by Shroyer et al.
(2017) and Prakash et al. (2023), elevated temperatures and an energetic ocean circula-
tion under ~~PGIS the ice shelf~~ driven by variations in Nares Strait sea ice mobility and
thickness increased ~~m_b basal melting~~ by a factor ranging between 1.2 and 1.7. Also, ~~m_b~~
~~melting~~ can be enhanced by routing of glacier surface melt to the glacier bed, and fur-
ther to the ~~GL grounding line~~ where it enters the fjord as subglacial discharge (~~Q_{sg}~~). ~~A~~
~~higher Q_{sg} during summer has been suggested to triple m_b at PGIS. Modelling study~~
~~suggests that a higher subglacial discharge during summer (June – August) triples basal~~
~~melt at Petermann ice shelf compared to winter (Cai et al., 2017)(Cai et al., 2017). Fur-~~
~~thermore, within the central basal channel, for the region extending from the grounding~~
~~line to 16 km seaward of it, between August and December 2015, Washam et al. (2020)~~
~~estimated area-averaged melt rate maximum of up to 170 m/yr, upstream of a location~~
~~16 km from the GL in the central basal channel, which were consistent with episodes~~
~~of enhanced Q_{sg} driven discharge driven~~ sub-ice shelf currents. This ~~indicates that Q_{sg}~~
~~suggests that subglacial discharge~~ may play a more important role in modulating ~~PGIS~~
~~Petermann ice shelf~~ basal melt than sea ice driven changes in ocean circulation. However,
with ~~atmospheric and oceanic~~ warming expected to amplify in the Arctic by the end of
the century (Muntjewerf et al., 2020; Shu et al., 2022), a salient question remains unan-
swered: How would ~~PGIS Petermann ice shelf~~ basal melt respond to projected increases
in ~~Q_{sg} under a future warming discharge under a warmer future~~ climate, and what would
be the mechanisms driving this response?
~~Here~~In this study, we extend a high-resolution unstructured grid 3-D ocean-sea ice-
ice shelf regional model setup (Prakash et al., 2022, 2023) centered over ~~PGIS and PF~~
~~to include Q_{sg} at the GL. Novel features include moreover an improved (with respect to~~
~~Morlighem et al., 2017) the Petermann ice shelf and Petermann Fjord to include subglacial~~
~~discharge at the grounding line (Fig. 1). The fjord bathymetry and the ice shelf draft~~
~~are derived from BedMachine v3 (Morlighem et al., 2017). Modifications have been~~
~~applied to the topographical datasets (discussed further below) to amend an inaccurately~~
~~represented~~ sub-ice shelf ~~bathymetry and a realistic PGIS basal topography. With these,~~

Figure 1. (a) Map of our regional **PGIS-PF-Petermann ice shelf-Petermann Fjord** model domain (in a wider context shown in the inset in the upper left corner) which follows the Greenland and Ellesmere Island coastlines, and extends into the Lincoln Sea and Baffin Bay (red lines). The mesh size in the model domain varies regionally, as exemplified by the three green-framed boxes. **PF-Petermann Fjord** is surrounded by a purple polygon, detailed in panel (b). (b) **PGIS-Petermann ice shelf** extent (white outlines), **from CE 2009**, overlain on water column thickness (WCT) in **PF-Petermann Fjord** and adjacent parts of Nares Strait. Cyan line marks **GL-the grounding line**, where discharge (detailed in panel (c)) is injected into the model. Red line marks the location of the transect over which the temperature, salinity, and flow along the fjord are investigated. The magenta star denotes the node ca. 10 km from **GL-the grounding line** at which the vertical profile of temperature and salinity are shown. Blue and yellow lines, respectively, represent the transects across which overturning circulation and diagnostics for meltwater **trappings-convergence** are shown. (c) Discharge (Q_{sg}) scenarios, 1950–2100 CE, obtained by forcing MAR with five CMIP5 AOGCMs (Barthel et al., 2020; Slater et al., 2020). Shaded vertical stripes with blue and red stippled outlines denote the periods over which Q_{sg} -present (CE 1995–2014) and Q_{sg} -RCP 8.5 (CE 2081–2100), respectively, are averaged over (cf. Table 1). Blue, magenta, and red horizontal stippled lines correspond to the Q_{sg} -present (372 m³/s), Q_{sg} -median (1563 m³/s), and Q_{sg} -RCP 8.5 (2754 m³/s) discharge magnitudes, respectively. (d) A synoptic graphical summary of the results showing basal melt rate (m_b), and its drivers, namely, thermal driving (ΔT) and friction velocity (u^*). **Common to all plots:** The **total-length-of-bars (dark colors) show corresponding variable-averaged spatially-over the deeper (>300 m) draft regions of PGIS, while light-colored counterparts represent spatial averages over the entire PGIS base.** The discharge type categories (X-axis) correspond to the experiment design, cf. Table 1.

~~we resolve the~~ water column thickness, and furthermore, to implement a 540 – 610 m
deep inner sill situated ca. 25 km from the grounding line, retrieved using aerogravity
data from Operation IceBridge (Fig. E1; Tinto et al., 2015). Open boundaries in the
Lincoln Sea and Baffin Bay are used to provide lateral ocean boundary conditions. These
include temperature, salinity, and sea water velocities derived from A4 ROMS (4 km
pan-Arctic Regional Ocean Modeling System grid; Hattermann et al., 2016), and merged
sea surface elevation derived using A4 ROMS and 5 km AOTIM (Arctic Ocean Tidal
Inverse Model; Padman and Erofeeva, 2004). Data from the 5.5 km polar (p) version
of RACMO2.3p2 (Regional Atmospheric Climate Model; Noël et al., 2019) is used to
compute the momentum, heat, and freshwater fluxes. Sea ice concentration and thickness,
bulk ice salinity, and sea ice velocities are obtained from A4 ROMS – CICE (Community
Ice Code; Hunke et al., 2015). Using this setup, we perform a control run with zero
subglacial discharge, and three test runs designed to simulate the response of the cav-
ity circulation and ~~spatial melt rate variability arising due to the complex PGIS basal~~
~~morphology (Rignot and Steffen, 2008) in unprecedented detail. Our findings evidence~~
~~that Q_{sg} ice shelf basal melt rates to present day subglacial discharge, increased discharge~~
~~under high-emissions ‘RCP8.5’ climate warming scenario, and an intermediate discharge~~
~~scenario between the two (Table 1). We show that discharge increases basal melting by~~
(a) accelerating a quasi-estuarine circulation (Sciascia et al., 2013) which increases the
advective heat transport into the cavity, and by (b) enhancing the shear-driven mixing
across the ~~PGIS-ocean ice shelf-ocean~~ boundary layer. The latter exerts a dominant control
over the increase in ~~m_b seen for larger Q_{sg}~~ basal melting seen for higher discharge
under future warming scenarios while the former (advective heat transport) saturates,
suggesting a shift towards a regime where melt increases are primarily driven by increases
in friction velocity (u^*). ~~Within the basal channels, we find that the buoyant meltwater~~
~~accelerates along the steepest slopes of the ice shelf adjacent to the crests of the channels~~
~~which increases u^* and m_b , both of which peak at the crest. We estimate that the high~~
~~mean melt rates driven by summer Q_{sg} , if allowed to persist for a decade, may lead to~~
~~the complete erosion of the channels i.e., velocity scale representing shear stress in the~~
~~boundary layer). Our findings challenge the prevailing focus on increased oceanic heat~~
~~forcing as the primary driver of basal melting at the Petermann ice shelf, revealing that~~
~~heightened subglacial discharge significantly enhances (channelized) basal melt. This~~
~~suggests that atmospheric warming-induced excitation of oceanic melt processes will~~
~~critically influence the stability of the ice shelf.~~
Results
Response of cavity circulation and basal melt to contemporary discharge
~~Our mean modelled melt rates and their spatial variability, and furthermore, their increase~~
~~due to the introduction of contemporary summer mean Q_{sg} (Q_{sg} -present) into the PGIS-PF~~
~~system (e.f. Fig. 1(d), Fig. In the control run, Petermann ice shelf basal melt rates~~
~~largely range from a few m/yr near the calving front to ca. 40 – 60 m/yr under the deeper~~
~~(400 – 600 m) regions of the ice shelf base near the grounding line (Fig. 2(a,b) are~~
~~in agreement with)). Melt rates decreasing with distance from the grounding line is~~
~~consistent with results obtained from steady state ice discharge divergence approach~~
~~(Rignot and Steffen, 2008) and satellite remote sensing (Wilson et al., 2017). For the~~
~~period from 2011 – 2015, remotely sensed estimates of annual mean melt (reveal melt~~
~~rates of up to ca. 80 m/yr near the grounding line, although they largely range between~~
~~40 – 50 m/yr decreasing to ca. 10 m/yr under the shallower (<200 m) portions of~~
~~the ice shelf (Wilson et al., 2017) and in-situ conductivity-temperature and radar derived~~
 ~~estimates of discharge enhanced seasonal melt (Washam et al., 2020), and are comparable~~
 ~~to our modelled estimates.~~ With the ~~inclusion of introduction of contemporary summer~~
 ~~mean subglacial discharge (Q_{sg} -present at the GL) into the ice shelf-fjord system,~~ the
 modelled mean melt strengthens nearly throughout the domain and appears to be concen-
 trated along the basal channels that run along the length of the ice shelf (Rignot and Steffen, 2008
 ~~;~~ Fig. 1(d), Fig. 2(a,b)). ~~Strongest increases are seen~~ Spatial variability of our
 modelled melt on scales similar to the basal channel configuration is consistent with the
 findings of Rignot and Steffen (2008) and Wilson et al. (2017). Strongest increases of up
 to ca. 100 m/yr are seen within these channels under the deeper regions of the PGIS base
 ~~near the GL, and which gradually decrease toward,~~ which gradually decreases towards
 the calving front. This increase lies within the in-situ conductivity-temperature and radar
 derived estimates of discharge enhanced seasonal melt rate maxima of 40 — 170 m/yr
 (Washam et al., 2020). The summer mean basal melt rate (m_b), friction velocity (u^*),
 and thermal driving (ΔT) are averaged over (a) the entire PGIS ice shelf base, and (b)
 over parts of the PGIS ice shelf where draft is deeper than 300 m, and are indicated, re-
 spectively, by subscripts "pgis" and "300m" appended to the variable in question. Com-
 pared to the summer mean modelled melt rates in the control experiment, the contempo-
 rary discharge-driven increase in m_{bpgis} and m_{b300m} , respectively, is 41.2% and 68.1%,
 driven by similar increases in u^*_{pgis} (40.747%) and u^*_{300m} (64.167.4%) and modest in-
 creases in ΔT_{pgis} (1.61.1%) and ΔT_{300m} (5.64%) (Fig. 1(d)).

Figure 2. Response of **PGIS-ice shelf** basal melt rates and its drivers to Q_{sg} -discharge distributed uniformly across the **GL-grounding line** (Fig. 1(b)), for the experiments defined in Table 1. Summer mean basal melt rate (m_b) (a-d), friction velocity (u^*) (e-h), and thermal driving (ΔT) (i-l) for the control (first row), Q_{sg} -present relative to control (δ ; second row), Q_{sg} -median relative to control (δ ; third row), and Q_{sg} -RCP 8.5 relative to control (δ ; last row) experiments. Contours of the ice shelf draft are plotted at 100 m intervals (450 m – 150 m; panel(f)) and overlaid as dotted black lines on each panel.

The presence of Q_{sg} at the **GL-subglacial discharge at the grounding line** enhances the
 fjord-scale overturning circulation (Fig. 3(i)). We observe a more robust Atlantic Water
 inflow at depth, extending into the inner fjord basin and up to the **GL-grounding line**

(Fig. 3(c,f)). Hereafter, we refer to this process as the "thermal pump", which promotes the entrainment of heat and salt deep into the **PGIS-ice shelf** cavity (columns 1 and 2 in Fig. 3). We note an increase in bottom temperature of 0.03°C at a location ca. 10 km from the **GL-grounding line**, and a temperature excess ($\Delta T_{\text{thermal driving}}$) of 0.1°C under the ice shelf base at 300 m depth (Fig. 2(j); Fig. 3(g)). A higher $\Delta T_{\text{near the GL}}$ **thermal driving near the grounding line** (Fig. 2(i,j)) drives more melting at depth. A more energetic outflow/strengthened meltwater plume dynamics, driven by the introduction of Q_{sg} -**discharge**, exacerbates the shear-driven turbulent mixing and heat transfer across the **PGIS-ocean-ice shelf-ocean** interface (Fig. 2(e,f)), thus driving more melt. The spatial variability of u^* -**friction velocity** anomalies is consistent with the corresponding melt rate anomalies (cf. panels b and f in Fig. 2). Regions exhibiting strong increase in u^* -**friction velocity** are concentrated along the longitudinal basal channels near the **GL-grounding line**, and gradually decrease downstream of it. However, a higher meltwater production (Fig. 2(b)) and stronger outflow (Fig. 3(f,i)) deposits larger volumes of meltwater from the deeper to the shallower regions of the **PGIS-ice shelf** base, considerably lowering the $\Delta T_{\text{thermal driving}}$ locally (Fig. 2(j); Fig. 3(d,e)). Thus, while the increase in u^*_{pgis} and u^*_{300m} is significant, the contribution of $\Delta T_{\text{thermal driving}}$ is largely restricted to the deeper regions of the **PGIS-ice shelf** draft (ΔT_{300m}) (Fig. 1(d)). **This implies that for subglacial discharge equivalent to present-day summer mean magnitudes, increase in basal melt under the deeper (>300 m) drafts is driven by increases in both thermal driving (increased thermal pumping) and friction velocity (increased shear-driven mixing).**

Heightened discharge and saturation of the thermal pump
When transitioning from the control experiment to RCP 8.5 (Q_{sg} -RCP 8.5), and an inter-
 mediate scenario between the two (Q_{sg} -median) (Table 1), a higher modelled Q_{sg} -**discharge**,
 as expected in a future warmer atmosphere (Aschwanden et al., 2019; Golledge et al.,
 2019), results in considerably higher increases in basal melt as compared to Q_{sg} -present
 (Fig. 1(d); Fig. 2(a–d)). In particular, an increase in Q_{sg} -**discharge** from control to RCP
 8.5 results in more than twofold (168.4%) and threefold (217.2%) increase in m_{bpgis}
 and m_{b300m} , respectively. These increases are primarily driven by increases in u^*_{pgis}
 (**165.5191.8%**) and u^*_{300m} (**205.7216.9%**), whereas ΔT_{pgis} and ΔT_{300m} show rela-
 tively nominal increases of **5.1% and 9.82.9% and 6.5%**, respectively. For the median
 experiment, **3.8% and 8.62.3% and 5.9%** increase in ΔT_{pgis} and ΔT_{300m} , and **114% and**
 **154.7132.5% and 163.4%** increase in u^*_{pgis} and u^*_{300m} generate 116.3% and 163.6% in-
 crease in m_{bpgis} and m_{b300m} , respectively. As noted for Q_{sg} -present, strongest increases
 occur under the deeper regions of the **PGIS-ice shelf** base which gradually decreases
 downstream (Fig. 2 (c,d)), however, channelized melting is seen to strengthen substan-
 tially (discussed further below).
Combining insights from all the experiments provides a novel perspective on how
 Q_{sg} -**discharge** dictates the interplay between **PGIS-Petermann ice shelf** basal melt and
 its drivers: Increase in the heat entrained in the **PGIS-ice shelf** cavity as a result of the
 thermal pump **peaks is strongest** at the inclusion of present-day summer mean estimate
 of Q_{sg} -**subglacial discharge**, with much of the heat delivered by the thermal pump con-
 centrated under the deeper **PGIS-ice shelf** base in the inner fjord basin (Fig. 1(d); Fig.
 3(d,g)). At a summer mean Q_{sg} -**discharge** value which is approximately fourfold higher
 than the present (i.e. Q_{sg} -median vs. Q_{sg} -present), **the contribution of the thermal pump**
 **is relatively weaker thermal driving increases at a lower rate** (Fig. 1(d); Fig. 3(g); cf.
 Fig. 3(d) and Fig. C1(a)). We note an increase of **2.91.9%** in the Q_{sg} -median ΔT_{300m}
 as compared to Q_{sg} -present (Fig. 1(d)). Moreover, a larger meltwater deposition further
 lowers the $\Delta T_{\text{thermal driving}}$ under the shallower drafts (Fig. 2 (b,c,j,k); Fig. 3(i);
Fig. C1(a,b,c)) and we do not observe a noticeable which results in a relatively weaker
 increase in ΔT_{pgis} (Fig. 1(d)). Lastly, in the RCP 8.5 scenario, Q_{sg} discharge is nearly
 doubled as compared to the Q_{sg} -median experiment, however, the efficiency of the thermal
 pump in increasing ~~m_b~~ basal melting saturates (Fig. 3(g); Fig. C1). Here, increase
 increases in ΔT_{pgis} ~~is~~ and ΔT_{300m} are indiscernible as compared to the Q_{sg} -median
 experiment, whereas ΔT_{300m} increases negligibly (Fig. 1(d)). Instead, we find that
 Q_{sg} under a discharge under the high-emissions RCP 8.5 scenario enhances melting by
 further accelerating the cavity circulation, wherein, 24.1% and 20.4% increase in m_{bpgis}
 and m_{b300m} are seen, driven predominantly by ~~24% and 20~~ 25.5% and 20.3% increase in
 u^*_{pgis} and u^*_{300m} , respectively.

Figure 3. Along-fjord summer mean temperature (a), salinity (b), and flow (c) for the control experiment for the section shown in Fig. 1(b). Mean isopycnals (solid black lines) are overlaid at equal intervals of 0.2 kg/m^3 . The 27.9 kg/m^3 isopycnal corresponds to the dense Atlantic Water that contacts the grounding line. For the same section, changes (δ) in summer mean temperature (d), salinity (e), and flow (f) for Q_{sg} -present relative to control are shown. Black stippled lines in panels d and e indicate the zero temperature and salinity difference contour, respectively. In panel f, solid black lines and stippled green lines depict the mean isopycnals for the control and Q_{sg} -present experiments, respectively, plotted at equal intervals of 0.2 kg/m^3 . In each panel, PGIS-GL-Petermann ice shelf grounding line (at \sim ca. 600 m depth) is on the right margin and open ocean is to the left. Vertical magenta line in panels a, b, d, and e show the location of the model node ca. 10 km from the GL-grounding line (Fig. 1(b)) at which the vertical profile of summer mean temperature (g) and salinity (h) are shown for the control, Q_{sg} -present, Q_{sg} -median, and Q_{sg} -RCP 8.5 experiments (cf. Table 1). (i) Laterally integrated summer mean vertical overturning profile [in Sv] for all the experiments (cf. Table 1) for the section shown in Fig. 1(b). Positive [Sv] values correspond to (laterally integrated) inflow and negative [Sv] values correspond to (laterally integrated) outflow across the section. Calculation of transport diagnostics follows from Prakash et al. (2023).

In sum, our experiments reveal a regime change in heat flux efficiency within the ice
shelf cavity. In the control experiment, the ice shelf-ocean boundary exchange consumes
 a considerable amount of heat from the ocean that is available for melting the ice. As
 such, the latent heat loss and meltwater release cools the ambient ocean inside the cavity.
 In such a scenario, an increased fjord-scale overturning circulation due to increased
 discharge (e.g., $< Q_{sg}$ -median experiment) entrains more heat in the cavity (thermal
 pump) which increases melting through enhanced thermal driving, in addition to the
 enhanced shear-driven turbulent heat exchange. This saturates at some point, when more
 discharge (e.g., $> Q_{sg}$ -median experiment, as seen for the RCP 8.5 scenario) accelerates
 the overturning circulation such that there is always enough heat coming into the cavity
 to maintain a high thermal driving (i.e., heat supply at an elevated level is maintained).
 Here, melt rates become entirely a function of the friction velocity.
**Trapping Convergence** of meltwater and channel stability

Figure 4. Maps of the summer mean surface layer salinity (a,b) and basal melt rate (c,d) for the control (column 1) and Q_{sg} -RCP 8.5 relative to control (δ ; column 2) experiments. Contours of the ice shelf draft are plotted at 100 m intervals (450 m – 150 m) and overlaid as dotted red and blue lines over the surface layer salinity and melt rate maps, respectively.

We use surface layer salinity as a proxy for tracking the spatial distribution of meltwater
 under the deeper regions of PGIS the ice shelf (Fig. 4; Fig. D1). Freshening occurs nearly
 throughout the domain in the Q_{sg} -driven discharge-driven experiments as compared to
 the control run. Furthermore, regions of substantial freshening align with four prominent
 ice shelf basal channels (Fig. 4 (a,b); 5 (a,b); Fig. B1) along which the draft thins from
 300 – 450 m ca. 5 km from the GL grounding line to 150 m ca. 25 – 30 km seaward
 of it, suggesting a topographically steered horizontal pathway for the meltwater outflow
 that is determined by the PGIS-ice shelf basal morphology and which in turn influences
 it (discussed further below). Changes appear to be most profound within these channels,
 particularly, at the erestcrests, where salinity decreases by up to 1.06 psu in the RCP 8.5
 experiment with respect to the control run (Fig. 4; Fig. 5(a,b)). These locations where
meltwater is trapped of meltwater convergence are concordant with locations of largest
 melt rate increases (discussed further below) of up to 181.5 m/yr (a sevenfold increase
 compared to the control run). ΔT is higher. Thermal driving is not higher at the channel
 crests where meltwater converges, but under the deeper drafts, which is consistent with
 the pressure dependent melting point of seawater. Although ΔT (Fig. 5(c)). Although
 thermal driving may not increase the most at the channel crests, we observe increases
 ranging from 0.06 to 0.19 0.02 to 0.12 °C in the RCP 8.5 experiment compared to the
 control run across the transverse cross-section near the GL grounding line (Fig. 5(c)).
 We find that u^* profiles the friction velocity profile across this section are is (nearly)
 congruent with the m_b profiles, and exhibit largest basal melt rate profile, and exhibits
 the highest increases at the channel crests of up to 0.03 m/s (Fig. 5(d)).

Figure 5. Maps of the summer mean surface layer salinity (a,b) and basal melt rate (c) Cross-section profile
of the Petermann ice shelf draft, derived from BedMachine v3 (Morlighem et al., 2017), for the control
transect ca. 5 km from the grounding line shown in Fig. 1 (column 1b) and Q_{sg} -RCP 8.5 relative to control
(column 2) experiments. Contours Apex locations of four prominent longitudinal ice shelf basal channels
at this section are highlighted using green stripes along which the Petermann ice shelf draft are plotted at
100-m intervals (reduces from 300 – 450 m – at the transect location to 150 m) and overlaid as dotted red
and blue lines over a distance of 25 – 30 km from grounding line (Fig. B1). Across the same section, the
summer mean surface layer salinity and melt rate maps (m_b) (b), respectively thermal driving (ΔT) and melt
rate (c), and friction velocity (u^*) and melt rate (d) are shown for the control and Q_{sg} -RCP 8.5 experiments.

Three ice shelf basal channel sections are outlined (Fig. 6(a)), whose lengths are
 demarcated based on the apex of the 150 m ice shelf draft contour (e.g. Fig. 4(a); Table
 2). The channel averaged annual mean basal melt rates for the western (S_w), central
 (S_c), and eastern (S_e) sections for the RCP 8.5 experiment are calculated as 45 m/yr,
 51 m/yr, and 47 m/yr, respectively. The enhanced Enhanced
 melting at the crests will promote channel opening and we can calculate the changes in channel averaged annual
 mean ice volume above each channel over time to estimate the incision/erosion timescale
 for each that channel under the respective basal melt rates (Eqn. 5, 6). To bracket the
role of ice advection from upstream of the GL grounding line, three different rates (AR_c)
 are considered for each channel which correspond to the present contemporary mean (1
 242 km/yr; Münchow et al. (2014)), and 50% lower (0.5 km/yr) and higher (1.5 km/yr) than
 243 present contemporary mean values. While S_w exhibits the lowest mean basal melt rate, it
 also features the shallowest mean draft value. Notably, its erosion-incision timescales are
 comparable to those of S_c , which has the highest mean melt rate but a relatively deeper
 mean draft compared to S_w . The number of years (N) required for the mean channel-ice
 volume-ice volume above a channel to become zero is estimated to be 5.88, 6.52, and 7.31
 248 years for S_w , in contrast to 5.95, 6.75, and 7.8 years for S_c , at AR_c of 0.5 km/yr, 1 km/yr,
 and 1.5 km/yr, respectively (Fig. 6(b)). It has been shown that the viscous ice response
 to a basal channel is insufficient to significantly delay the breakthrough of basal channels
 in ice shelves that are thinner than 400 m (Wearing et al., 2021). With a lower mean melt
 rate than S_c and the deepest mean draft value, S_e demonstrates the greatest resilience. We
 estimate N to be 8.06, 9.41, and 11.31 years for AR_c of 0.5 km/yr, 1 km/yr, and 1.5 km/yr,
 respectively (Fig. 6(b)). These estimates suggest that ice advection from upstream of the
 PGIS-GL grounding line (alone) may not be sufficient to compensate for its thinning the
 thinning of the ice shelf due to intense channelized basal melting that may be expected
 under the RCP 8.5 scenario (discussed further below).

(a) Cross-section profile of the PGIS draft for the transect ca. 5 km from the GL shown in Fig. 1(b). Four prominent longitudinal (L) ice shelf basal channels are represented using green stripes as L_w (west), L_c (central), $L_e(1)$ (east-1) and $L_e(2)$ (east-2) along which the PGIS draft reduces from 300–450 m at the transect location to 150 m over a distance of 25–30 km from GL (see Fig. B1). For clarity, $L_e(1)$ and $L_e(2)$ are collectively classified as L_e (east) in panels b, c, and d. Across the same section, the summer mean surface layer salinity and melt rate (m_b) (b), thermal driving (ΔT) and m_b (c), and friction velocity (u^*) and m_b (d) are shown for the control and Q_{sg} -RCP 8.5 experiments.

Figure 6. (a) Map of the PGIS Petermann ice shelf draft overlaid with channel sections along the western (S_w ; blue), central (S_c ; black), and eastern (S_e ; red) regions which are used in calculating the changes in respective channel averaged annual mean ice volume above the channel over time in panel (b) for the RCP 8.5 experiment following Eqns. 5 and 6 (Table 2). Apex locations of four prominent longitudinal ice shelf basal channels at the across-fjord transect shown in Fig. 5 are represented using magenta diamonds. For each color-coded channel section shown in (a), channel-corresponding changes in mean ice volume changes above the channel section are shown in (b) are shown for three different ice advection rates, namely, 0.5 km/yr (asterisk), 1 km/yr (solid line) and 1.5 km/yr (circle). Channel averaged annual mean basal melt rates for S_w , S_c , and S_e are, respectively, 45 m/yr, 51 m/yr, and 47 m/yr.

**Discussion**
We note that our model is forced with contemporary summer mean Q_{sg} -estimates discharge
 estimate and expected increases of it under a future warming atmosphere. The present-
 261 day estimates of maximum Q_{sg} -discharge (e.g. Cai et al., 2017) are two to fourfold
 higher than the present-day summer mean estimate of Q_{sg} -discharge used in this study,
 and are even comparable to the Q_{sg} -median experiment. Therefore, while Q_{sg} -median
 represents an intermediate state between present and extreme future mean scenario in
 our study, it is plausible that such increases in Q_{sg} -discharge, and thus basal melt, are
 being experienced by **PGIS-the Petermann ice shelf** for brief periods during summer,
 with varying degrees of inter-annual variability. ~~Modelled study from~~ We posit that
 during the summer (JJA) months, if for e.g., exceptionally high discharge is sustained
 for a considerable (e.g., several weeks) period of time, it may temporarily nudge the ice
 shelf-fjord system into a regime where basal melting is largely driven by friction velocity.
 We note that as opposed to Gaussian-like patterns (Ehrenfeucht et al., 2023), our applied
 discharge forcing follows a boxcar function. Thus, studies investigating the sensitivity
 of the system to seasonal and inter-annual variability in subglacial discharge, derived for
 e.g., from a subglacial hydrological model, need to be conducted to ascertain and quantify
 any resulting transitions that may occur in the mechanisms driving basal melting. The
 modelling work of Hill et al. (2018) has demonstrated that the loss of thicker and rhe-
 ologically stiffer sections of **PGIS-the Petermann ice shelf** within 12 km from the **GL**
 **grounding line** (regions where draft ≥ 300 m) could disrupt **GL-grounding line** stresses
 sufficiently to greatly accelerate interior ice flow and increase dynamic ice discharge, sig-
 nificantly contributing to global mean sea level rise. We thus suggest that the Q_{sg} -driven
 discharge-driven intensified basal melting of the deeper (draft ≥ 300 m) regions near the
 **GL-grounding line** that are dynamically resilient (Fig. 1(d); Fig. 2) will have a significant
 bearing on the future stability of **PGthe Petermann Glacier**, as their loss could trigger an
 unopposed acceleration of the inland ice.
Thus far, basal melting of **PGIS-the Petermann ice shelf** has been largely investigated
 from the perspective of (increased) oceanic heat delivery to **PFPetermann Fjord**. How-
 ever, both observational and modelled estimates concur on the abundance of heat supply
 into the fjord, i.e. heat availability is in excess of what is needed to generate contempo-
 rary estimates of basal melt (Johnson et al., 2011; Washam et al., 2018; Prakash et al.,
 2023). Our findings show that increased basal melting of **PGIS-the Petermann ice shelf** in
 a warming climate is not limited to increase in oceanic heat forcing, and underscores the
 importance of atmospheric warming in controlling the basal melting of Northern **GrIS**
 **Greenland Ice Sheet** glaciers (Slater and Straneo, 2022). Uncovering how this unfolds
 from the **PGIS-ocean-an ice shelf-ocean** interaction perspective, we find that the effect
 of the thermal pump on melt is the strongest when Q_{sg} -discharge is introduced into the
 system (as Q_{sg} -present; Table 1) and saturates for Q_{sg} -discharge $> Q_{sg}$ -median (Fig.
 1(d); Fig. 3; Fig. C1), wherein, Q_{sg} -median is equivalent to present-day estimates of
 maximum Q_{sg} -discharge (c.f. Cai et al. (2017)). We see an increase in bottom temper-
 ature near the **GL-grounding line** (increased heat advection towards the interior of the
 **PGIS-ice shelf** cavity via the thermal pump) as Q_{sg} -discharge is increased from control
 to present to median, but less so for substantially higher discharge under the RCP 8.5
 scenario (Fig. 3(g)). Therefore, we posit that a regime shift of the driving mechanism
 behind m_b -Petermann ice shelf basal melting may occur when the catchment integrated
 meltwater runoff under (extreme) future warming scenarios exceeds the discharge thresh-
 old of present-day Q_{sg} -maximummaximum. In such a scenario, the significance of the
 thermal pump in driving increased melting ebbs off, and the higher discharge being re-
leased into the fjord accelerates the sub-ice shelf currents, which nudges the system to-
 wards a regime where increase in basal melt is predominantly dictated by substantially
 higher shear-driven turbulence which increases the heat fluxes at the ~~PGIS-ocean-ice~~
 ~~shelf-ocean~~ interface (Fig. 2). ~~We Millan et al. (2023) show that Atlantic Water reaching~~
 ~~the fjord in 2020 are 0.3°C warmer as compared to 1970, of which a warming of 0.2°C~~
 ~~has occurred between 2000 and 2020. Further modelling work is required to determine~~
 ~~the characteristics of Atlantic Water reaching the fjord by the end of this century under~~
 ~~different climate warming scenarios. While we do not take into consideration the impact~~
 ~~of rising ocean temperatures in our study, we acknowledge that realistically, in a future~~
 ~~warming climate, rising ocean temperatures warmer climate, it would likely act in concert~~
 ~~with the enhanced Q_{sg} increased discharge to further amplify m_b basal melting.~~ How-
 ever, with a much lower heat capacity as compared to the ocean, increase in surface melt
 over grounded ice in response to atmospheric warming (and thus increase in subglacial
 discharge) is highly likely to outpace the warming of the deep ocean water masses that
 (alone) would be required to bring about such substantial increases in m_b basal melting
 in the upcoming centuries. ~~Thus, it is reasonable to posit that the atmospheric impact~~
 ~~on m_b is~~ (e.g., increases of up to ca. 200 m/yr in the RCP 8.5 scenario) as modelled
 here. Thus, while the impact of increase in ocean heat forcing and its interplay with
 increased discharge on basal melting in a future warming climate needs to be examined,
 discharge-driven basal melting is likely to pose a more immediate and pertinent threat to
 the glacier's long-term stability.
~~Across all Q_{sg}~~ Seasonal changes including, but not limited to, an increase in fjord
 temperature of ca. 0.1°C during summer as compared to winter (Prakash et al., 2023)
 drives up to ca. 47% increase in mean channelized (sections S_w , S_c , and S_e in Fig. 6(a);
 Table 2) basal melting. Without taking into consideration the compensating viscous ice
 response (discussed further below), and for contemporary mean ice advection rate from
 upstream of the grounding line of 1 km/yr, this increase results in up to ca. three-fold
 increase in erosion rates. The strongest seasonal influence on melt is imposed by the
 addition of subglacial discharge into the fjord. Compared to winter (no discharge), we
 calculate that contemporary summer mean discharge increases mean melting in the channels
 by up to ca. three-fold, yielding up to an order of magnitude increase in erosion rates.
 We note that although S_w and S_c may exhibit higher rates of erosion as compared to S_e ,
 highest increases in erosion rates are modelled for S_e .
~~Across all discharge~~ experiments (Table 1), we consistently find channelized strength-
 ening of basal melt (Fig. 2(a-d)), imposed by the complex ~~PGIS-ice shelf~~ basal morphol-
 ogy (Rignot and Steffen, 2008; discussed further below). Within these channels (e.g., ~~L_w ,~~
 ~~L_{S_w} , S_c , and L_{S_e}~~ in Fig. 5), 6(a)), compared to the control experiment, larger volumes
 of buoyant meltwater rise along the steepest slopes of the ice base towards the channel's
 crests (Fig. 5), from where the stronger buoyancy-driven overturning circulation in the
 fjord carries it further downstream along the longitudinal channel axis (Fig. 3(i); Fig. 4;
 Fig. B1; Fig. D1). The steeper ice shelf basal slopes that characterize these channels
 support stronger entrainment of the ambient Atlantic Water into the buoyant meltwater
 plume (Wilson et al., 2017). Furthermore, the vigorous confluence of meltwater towards
 the ~~crest increases u^* which further amplifies melt. crests, and its downstream advection,~~
 ~~increases the friction velocity which predominantly drives the increased melting~~ (Fig. 2;
 Fig. 5(b,d)). The additional buoyancy forcing provided by ~~Q_{sg} under future warming~~
 ~~scenarios increased subglacial discharge in a warmer future climate, in particular, in as~~
 ~~modelled here for~~ the RCP 8.5 experiment, ~~acts will act~~ to enhance these mechanisms
 (Fig. 5(b,d)) which may advance the vertical growth of the channel. To that end, our
 ~~channel erosion estimates~~ estimates of mean ice volume changes above the channels in-
dicates that sustained high mean melt rates due to enhanced summer Q_{sg} discharge, as seen
in the RCP 8.5 experiment, could undermine the structural integrity of ~~PGIS~~ the ice shelf
(Fig. 6). Persistence of such intensified melting over durations ranging from ca. 6 to
11 years may result in complete ~~erosion-incision~~ of the channels, leading to a possible
disintegration of the ice shelf. While the accuracy of this estimate depends on the un-
certainty of the applied melt rate parameterization, e.g., not accounting for the effects of
enhanced stratification due to the meltwater discharge (Rosevear et al., 2022; Davis et al.,
2023), the simulated acceleration of the cavity circulation is a robust feature of our sim-
ulations, which, together with observational evidence (Washam et al., 2020) indicates
significant impacts on the basal mass loss of ~~PGIS~~ the Petermann ice shelf. We also
note that our static ice shelf geometry does not account for the ice dynamical processes,
for e.g., downstream advection and calving, and viscous ice response which would act
to (partially) offset melting. While it has been shown that the secondary ice-flow is not
strong enough to substantially hinder the development of basal channels in ice shelves that
are less than 400 m thick (Wearing et al., 2021), we observe that the Petermann ice shelf
draft is deeper than 400 m in certain sectors (e.g., southeastern sector near the grounding
line). However, in the absence of a high-resolution (at the scale of the topographical data)
ice-flow model, it is difficult to ascertain how small-scale (sub-km) localized channel
closures/openings would impact the wider ice dynamics, and as such, the long-term
stability of the glacier. We suggest that future work with coupled ice-flow - ocean models
(frameworks of which have been recently developed, e.g., Zhou et al. (2024)) or offline
coupling of an ice-flow model with comprehensively resolved ice shelf basal melt rates
derived from sophisticated regional ocean models is needed to improve our understanding
of the nuanced dynamical variability that occur within these channels.
Our results could be extended to other 'warm cavity' environments where pronounced
increases in Q_{sg} discharge are expected, and the ice shelf is characterized by complex
basal morphology. We suggest that such conditions could be met for other Northern
~~GrIS~~ Greenland Ice Sheet glaciers that terminate as floating ice shelves, namely the Ry-
der Glacier (~~RG~~) and Nioghalvfjærdsbræ (79N) Glacier. The ice shelves of both ~~RG~~ Ryder
Glacier and 79N also feature channelized geometry and thus exhibit notable spatial melt
rate variability, as well as a dependency of basal melt on slope that is qualitatively similar
to ~~PGIS~~ the Petermann ice shelf (Wilson et al., 2017; Prakash et al., 2023). The lower
rates of ice advection from upstream ~~make RG of the grounding line~~ make Ryder Glacier
more prone to developing deep channels. At 79N, it has been suggested that the recent
expansion of its central basal channel is a consequence of heightened Q_{sg} discharge, at-
tributed to a significantly expanded surface melt area during summer, brought about by
atmospheric warming (Narkevic et al., 2023; Zeising et al., 2023). Moreover, the exten-
sive network of moulins observed along the ~~GrIS~~ Greenland Ice Sheet margins in summer
(Turton et al., 2021; Ehrenfeucht et al., 2023; Rawlins et al., 2023), coupled with expected
increases in surface melt ~~on the GrIS over the ice sheet~~ in a warmer climate (Aschwan-
den et al., 2019; Muntjewerf et al., 2020), are likely to yield end-of-century discharge
estimates at these sites that are comparable to those of Petermann (see projected runoff
for the northern, northeastern and northwestern ~~GrIS sectors~~ sectors of the ice sheet in
Slater et al. (2020)).
Analogous to Greenland, the intricate basal morphology of Antarctic ice shelves,
sculpted by their interactions with the bed and/or melting, also exhibits a diverse range
of features (e.g., channels, keels, crevasses, and terraces) that exert a significant in-
fluence on their melting and structural integrity, which is contingent upon the slope
(Drews et al., 2017; Schmidt et al., 2023). ~~While the~~ The role of an additional buoy-
ancy source that is injected basally into the fjord on enhancing basal melting and retreat
has been extensively investigated in Greenlandic geophysical settings (Sciascia et al.,
2013; Rignot et al., 2015; Cai et al., 2017; Cook et al., 2020), ~~in Antarctica, insights into~~
~~this process are limited to tacit reasoning~~ (Le Brocq et al., 2013; Lepp et al., 2022) ~~and~~
~~remain largely unaccounted for in modelling efforts that aim to constrain future sea-level~~
~~projections.~~ ~~Nonetheless, recent studies~~ and has garnered recent attention in Antarctica
(Goldberg et al., 2023; Gwyther et al., 2023; Pelle et al., 2024). Studies have shown that
Q_{sg} subglacial discharge has significantly influenced the mass balance of the Antarctic
Ice Sheet during the last deglaciation period and is expected to maintain its importance
in a warming climate (Li et al., 2023; Pelle et al., 2023). In addition, ice shelf collapses
due to hydrofracturing and flexurally-induced fractures resulting from supraglacial lake
drainage events have also been noted (Scambos et al., 2009; Banwell and MacAyeal,
2015; Dow et al., 2018). However, unlike Greenland (Forster et al., 2014; Willis et al.,
2015; Otto et al., 2022) there is no evidence of surface meltwater infiltrating to the bed.
Greenland's surface and subglacial hydrological systems that are connected by drainage
pathways offer a blueprint for understanding how Antarctica might evolve in a warmer
world, as its surface hydrology starts to resemble present conditions observed at ~~GIS~~ the
Greenland Ice Sheet (Bell et al., 2018). Projected increases in surface melt rates over
Antarctica (Noël et al., 2023) suggest a shift towards a more extensive and active surface
hydrology, one that could effectively link the presently isolated bed via seasonal influx
of surface meltwater. We note that such scenarios are dependent upon future emission
pathways and whether the ambitious targets proposed by the Paris Climate Agreement
are upheld (DeConto et al., 2021). However, if these targets are exceeded, as is expected
under the RCP 8.5 projection, we posit that a fundamental change in Antarctica's dy-
namics and the processes controlling it cannot be precluded. In particular, energetic
currents driven by intensified discharge in a cavity characterized by complex channelized
basal morphology, as modelled here, will accelerate the heat flow towards the ice shelf
base. The substantial shear-driven turbulence could independently drive increased melt-
ing, and may even be sufficient to abrade the stable boundary layer stratification that has
been observed in critical regions where ~~m_b~~ basal melting remains hitherto suppressed
due to latent ocean environments (Davis et al., 2023).
Ice shelf basal topography remains largely represented in numerical models in an
idealised tapered form, that is either completely flat or has some smooth curvature that
is imparted as a function of along-fjord slope. Moreover, the sub-ice shelf bathymetry
employed in these models is derived using rudimentary techniques that attempt to inter-
polate between known depths from ~~the continental shelf/fjord mouth to the GL~~ near the
fjord mouth and the grounding line. These critical shortcomings are addressed in our
model setup (Prakash et al., 2022), however, efforts to mitigate such topographic (both
seafloor and ice shelf basal ~~)~~ caveats topography caveats in numerical modelling efforts
that aim to investigate real-world locations, despite its well acknowledged impetus on
~~m_b~~ basal melting, are still not being widely undertaken. Indeed, an unknown sub-ice
shelf bathymetry ~~at PGIS and~~ for a vast majority of ice shelf-fjord systems is a limiting
factor. As such, wherever possible, more targeted airborne and ground-based geophysical
campaigns must be conducted to describe these environments. While our findings em-
phasize the need for implementing realistic ice shelf geometries, it is noteworthy to high-
light the limitations of the available resolutions (typically hundred to several-hundred
meters; e.g. Morlighem et al. (2017); Frémand et al., 2023) which remain relatively
coarse to capture the small-scale basal features, on the order of several (tens of) me-
ters as revealed by in-situ under-ice shelf observations (Davis et al., 2023; Schmidt et al.,
2023), that play a crucial role in dictating the spatial ~~variability in m_b~~ melt rate variability.
To comprehensively understand the intricate interplay between Q_{sg} subglacial discharge,
basal morphology, and melt processes, we emphasize that a more extensive dataset of
such measurements is imperative ~~which then needs to be integrated into~~. To enhance
the reliability of continent-scale projections, observational findings from sub-ice shelf
environments – including near the grounding zones – must be combined with insights
from non-idealized high-resolution ice shelf-ocean models numerical modelling studies
driven by these datasets. Improved understanding of critical small-scale processes at
the marine-ice margins should then inform the refinement of existing parameterizations
and the development of new ones, which would facilitate large-scale models to better
capture the influence of dynamics at the meter-scale. We note that in our experiments,
Q_{sg} discharge is distributed uniformly across the GL-grounding line width which may
not be consistent with ground truth. For an idealized tidewater glacier, modelling work
has shown that distributed discharge generates more melt than channelized discharge
(Slater et al., 2015). With glacier hydrology exhibiting increased dynamism, which is
likely to continue into the future, a concerted effort is warranted. This necessitates de-
riving insights into the interplay between moulin locations/density, meltwater routing,
and eventually its magnitude and emergence location(s) at the GL-grounding line from
observational evidence and sophisticated subglacial hydrological modelling tools. ~~These~~
~~insights, which~~ should then be incorporated into numerical models of ice shelf-ocean in-
teractions, ~~to better predict the evolving dynamics of glacier-ocean systems in a warming~~
~~climate.~~ Furthermore, investigations of the interaction between subglacial discharge and
tidally-modulated seawater intrusions in the grounding zone (Ciraci et al., 2023) will be
essential in improving our understanding of the dynamics and stability of the glacier.
**Methods**
**The Finite Volume Community Ocean Model (FVCOM) setup**
The unstructured grid, free-surface, 3-D primitive equation Finite-Volume Community
Ocean Model (FVCOM) (Chen et al., 2007), augmented by modules for an ice shelf
(Zhou and Hattermann, 2020) and sea ice (*Ice Nudge*) (Prakash et al., 2022), is used. The
model domain is centered on PF Petermann Fjord, covering the region between 75°N -
87°N and 29°W - 81°W (Fig. 1a). The grid comprises of unstructured triangle elements,
featuring a total of 112709 nodes and 219836 cells, including 309 open boundary nodes
in the Lincoln Sea and Baffin Bay. The topological flexibility of the unstructured grid
allows accurate fitting of the complex irregular coastal geometry along the northern GIS
Greenland Ice Sheet margin. In particular, the grid is partitioned into polygons of varying
horizontal resolution: A 200-m resolution is provided in the PF Petermann Fjord region
to resolve the steep slopes characteristic of the fjord seafloor and PGIS-ice shelf basal
topography, which is relaxed to 2 km in the adjacent areas outside the fjord, and further
to 4 km towards the open ocean boundary regions. For an extensive regional model setup,
such a configuration enables us to strike a balance between computational efficiency and
the need for accurate representation of topographical features and their variation over
small spatial scales.
PGIS-basal topography Petermann ice shelf draft derived from the BedMachine v3
(Morlighem et al., 2017) dataset (150-m resolution) represents a pre-2010 PGIS-calv-
ing geometry (Münchow et al., 2014), and is interpolated to the 200 m FVCOM model
grid (Prakash et al., 2022). As such, the basal channels (~~Rignot and Steffen, 2008~~) are
resolved at 200 m scale in our model, which is sufficient to represent their 1–2 km width
(Rignot and Steffen, 2008). Bathymetry outside PF Petermann Fjord comes from the 1-
503 km gridded IBCAO v3.0 dataset (Jakobsson et al., 2012), whereas inside it, the Bed-

Machine v3 dataset is used. In the vertical, the domain between the irregular seafloor topography and the ice shelf draft (or the open ocean surface) is discretized into 23 terrain-following σ -layers. To alleviate errors in the discretization of horizontal pressure gradients along the sloping σ -layers, smoothing of the bathymetry is applied as a function of the mesh resolution before its implementation in the model (Prakash et al., 2022). Further, the poorly constrained Without seafloor depth measurements beneath the ice shelf, the BedMachine bathymetry is inferred by extending the known seafloor depth from seaward of the calving front landward towards the grounding line and aligning it with the depth of the ice shelf draft at the grounding line using natural neighbor interpolation. For Petermann Fjord, Prakash et al. (2022) show that this leads to implausible abrupt changes in the water column thickness along either flanks and. Specifically, the BedMachine bathymetry underestimates the water column thickness under the deeper regions of PGIS is improved, for e.g., by implementing a ca. 600 m deepinner sill (Tinto et al., 2015) situated approximately (>200 m) regions of the ice shelf, with sub-100 m values along the flanks and in the central zone (Fig. E1(a-c)). More critically, this misrepresentation results in the formation of an artificial inner sill, which, if not accounted for, would block the inflow of the warm and dense (27.9 kg/m^3) Atlantic Water at depth (Fig. E1(d)) that contacts the grounding line (Washam et al., 2018; Prakash et al., 2023). Therefore, addressing these artefacts is imperative to ensuring that the model is able to reproduce the fundamental aspects of glacial fjord circulation (Sciascia et al., 2013). Tinto et al. (2015) used aerogravity data from Operation IceBridge to model the bathymetry underneath the ice shelf. Importantly, their findings revealed the presence of an inner sill, ca. 540 – 610 m deep, located about 25 km from the GL which is not included in the BedMachine v3 dataset (Prakash et al., 2022). Initial and lateral ocean boundary conditions, and atmospheric and sea ice surface boundary conditions are derived from a suite of high-resolution regional (ocean, tidal, atmospheric and sea ice) models (downscaling procedure is detailed in Prakash et al. (2022)) grounding line. In this study, we implement the smoothed bathymetry product generated by Prakash et al. (2022), wherein, the BedMachine bathymetry has been modified to account for the unrealistic sub-ice shelf water column thickness and further integrated with the aerogravity data from Tinto et al. (2015).

Experiment Design

Table 1. The ensemble averaged June-July-August (JJA) mean discharge (Q_{sg}) magnitude for the Petermann catchment [in m^3/s] that is distributed uniformly across the GL grounding line for present conditions (1995-2014), future conditions (2081-2100, under the RCP 8.5 scenario), a control scenario, and a median scenario.

Experiment	JJA-mean Q_{sg} [m^3/s]	JJA-mean time period
Control	N/A	N/A
Q_{sg} -present	372	1995–2014
Q_{sg} -median	1563	N/A
Q_{sg} -RCP 8.5	2754	2081-2100

The model is run from July 01, 2014 00:00:00 UTC – January 01, 2017 00:00:00 UTC with hourly oceanic (temperature, salinity, velocities, and sea surface elevation), 3-hourly atmospheric (heat and freshwater fluxes, and wind speeds), and daily averaged sea ice (sea ice concentration, thickness, and velocities, and bulk ice salinity) forcings obtained from respective high-resolution regional models over this period (Prakash et al., 2022, 2023), and without Q_{sg} discharge, which represents the *Control* experiment (Table 1). To allow

investigations of the response of **PGIS-ice shelf** basal melt to variations in Q_{sg} **subglacial**
 **discharge**, our FVCOM setup has been augmented to include Q_{sg} **discharge** across the
 ca. 20 km wide **PGIS-GL-Three- Q_{sg} -grounding line, Three discharge** experiments are
 initialised from the stable model solution of the *Control* experiment from January 01,
 2016 00:00:00 UTC and run until January 01, 2017 00:00:00 UTC (Table 1). Lacking
 information regarding the precise location(s) of emergence of subglacial meltwater across
 the **PGIS-GL- Q_{sg} -grounding line, discharge** is injected uniformly across the 90 cells
 (hereafter denoted by $\Delta_{Q_{sg}}$) representing the **GL-grounding line** (Fig. 1b). At each $\Delta_{Q_{sg}}$,
 discharge is vertically uniform across the 23 σ -layers. The salinity at each $\Delta_{Q_{sg}}$ is set to 0,
 and the temperature is set to the (in-situ) pressure dependent freezing point of freshwater.
 The June-July-August (JJA)-mean Q_{sg} **discharge** values presented in Table 1 represent
 the daily Q_{sg} **discharge** values over the summer (JJA) months used in our model. Outside
 of this period, Q_{sg} **discharge** is set to zero (similar to Cai et al. (2017)). Results for the
 *Control* and Q_{sg} **discharge** experiments are presented from the final year of the simulation
 (January 01, 2016 – January 01, 2017). We note that the pre-2010 ice shelf geometry
 provided by BedMachine v3 is not consistent with the forcing period(s). Change in ice
 shelf geometry (e.g., due to calving) has been shown to impact fjord circulation and
 basal melt rates (Poinelli et al., 2023). However, the two prior large calving events in
 2010 and 2012 (Münchow et al., 2014) removed the softer and thinner (<150 m) outer
 sections of the ice shelf (Hill et al., 2018). As such, we posit that the findings of our
 study – discharge driven regime change in heat flux efficiency in the ice shelf cavity, and
 amplified channelized melting under the deeper drafts – are robust, irrespective of the
 geometry used.
The present day JJA-mean Q_{sg} **discharge** (Q_{sg} -present) and its increase in a future
 warming climate under the RCP 8.5 scenario (Q_{sg} -RCP 8.5) for the Petermann catchment
 is obtained from Slater et al. (2020) (Table 1). Here, a subset of the Coupled Model
 Intercomparison Project (CMIP5) Atmosphere and Ocean general circulation models
 (AOGCMs) (Barthel et al., 2020; Fig. 1c) is used to force the Modèle Atmosphérique
 Régional (MAR) 3.9.6 (Fettweis et al., 2013) for the period 1950 – 2100. The projected
 MAR meltwater runoff over the Petermann catchment is then bias corrected to ensure
 that it is consistent with the best present-day (1995–2014) runoff estimate, and to also
 enable a smooth transition from present to RCP 8.5 forcing. It is then summed up over
 the Petermann catchment to give the Q_{sg} **discharge** estimates. We note that the modelled
 runoff outputs were stored as annual means and were converted to the JJA-mean runoff by
 multiplying the output by a factor of 365/92, where 365 = number of days in a year, and
 92 = number of days in JJA. This coarse conversion introduces a minor discrepancy be-
 tween the converted and true modelled JJA-mean values; however, we maintain that this
 does not significantly impact the findings of our study. While inter-model differences
 can be large (Fig.1 (c)), all five CMIP5 AOGCMs were used to prepare the multi-model
 mean Q_{sg} **discharge** used in this study. Note that the median Q_{sg} **discharge** magnitude
 (Q_{sg} -median) is not obtained from Slater et al. (2020), but is constructed using the Q_{sg} -
 present and Q_{sg} -RCP 8.5 magnitudes (as median ~~(of Q_{sg} -present and Q_{sg} -RCP 8.5)~~;
 ~~and as such,~~ As such, it represents an intermediate future mean Q_{sg} **discharge** mag-
 nitude, however, it does not correspond to any known projection scenario or a particular
 time period (Table 1).
**Diagnostics of basal melt rate and its drivers**
For the investigation of **PGIS-ice shelf** basal melt rate, its expression in the thermody-
 namic framework (Holland and Jenkins, 1999; Zhou and Hattermann, 2020) of the prim-
 itive equation setting is recalled:

$$\rho_{fw} m_b L = -\rho_{sw} c_w \Gamma_T u^* \Delta T. \quad (1)$$

Here, ρ_{fw} and ρ_{sw} are the densities of freshwater and ocean water, respectively. L is the latent heat of fusion of ice, c_w is the specific heat capacity of ocean water, and Γ_T is a non-dimensional heat-transfer coefficient. m_b is basal melt rate, controlled primarily by friction velocity (u^*) and thermal driving (ΔT), and where

$$u^* = \sqrt{C_D(u_w^2 + u_{res}^2)}, \quad \Delta T = T_b - T_w, \quad (2)$$

$$T_b = \lambda_1 S_b + \lambda_2 + \lambda_3 P_b, \quad (3)$$

$$\rho_{fw} m_b S_b = -\rho_{sw} \Gamma_S u^* (S_b - S_w). \quad (4)$$

Above, C_D is the drag coefficient, u_w is the velocity magnitude some distance away from the ice shelf-ocean boundary, and u_{res} is the velocity of small sub-grid scale residual currents (Asay-Davis et al., 2016). T_b , S_b , and P_b are the potential temperature, salinity, and pressure at the ice shelf-ocean boundary, and T_w and S_w are the potential temperature and salinity some distance away from it. λ_1 , λ_2 , and λ_3 , respectively, are the slope, intercept, and pressure coefficient of the liquidus, and Γ_S is a non-dimensional salt-transfer coefficient. Note that T_w , S_w , and u_w are obtained from the first (i.e. uppermost) σ -layer. Furthermore, Γ_T and Γ_S are application specific, and are herein tuned according to the applied boundary conditions, PGIS-Petermann ice shelf basal topography, and mixing schemes so as to be consistent with the contemporary observational estimates of PGIS the Petermann ice shelf basal melt (see Table A1). The drivers of basal melt rate (\$m_b\$ ), namely, friction velocity (\$u^*\$ and) and thermal driving (\$\Delta T\$ ) are used to investigate the impact of \$Q_{sg}\$ discharge on the summer mean \$m_b\$ basal melt rate.

Basal channel Changes in ice volume changes above basal channels over time
We calculate the changes in the annual mean channel averaged ice volume above basal
channels over time ($V_c(N)$) to determine the number of years (N) it would take for $V_c(N)$
to become 0 as

$$0 = V_i + (N \times V_a) - (N \times V_m), \quad (5)$$

$$\implies N = \frac{V_i}{V_m - V_a} \quad (6)$$

where, V_i is the initial (contemporary) channel ice volume ice volume above a channel
expressed as L_c (channel length [m]) \times W_c (mean channel width [m]) \times H_c (mean chan-
nel depth [m]). V_a is the annual mean volume of ice advected downstream of the GL
grounding line per year expressed as AR_c (annual-mean advection rate [m/yr]) \times W_c
621 \times H_c . V_m is the annual mean volume of ice melted per year expressed as m_{bc} (annual
mean channel averaged basal melt rate [m/yr]) \times $L_c \times W_c$. Dimensions associated with
each channel section are detailed in Table 2. Note that W_c drops out of the calculation,
and as such, is not detailed in Table 2.

Table 2. Dimensions associated with the PGIS-Petermann ice shelf basal channel sections illustrated in Fig. 6(a). Channel section along the western, central, and eastern region are termed as S_w , S_c , and S_e , respectively.

Channel Name	L_c [m]	H_c [m]
S_w	30000	241
S_c	25000	271
S_e	28000	331

Data availability
Our study is based on numerical modelling and we provide the model input files (<https://zenodo.org/doi/10.5281/zenodo.12803093>) and code (see Code availability below) that
627 //zenodo.org/doi/10.5281/zenodo.12803093) and code (see Code availability below) that
are required to reproduce our simulations.
Code availability
The open source code Finite Volume Community Ocean Model version 4.0 (FVCOM
v4.0), augmented by both the ice shelf and sea ice modules, that has been used to con-
duct the numerical experiments is publicly available at: [https://github.com/abhay26992/](https://github.com/abhay26992/FVCOM_Petermann_Code.git)
FVCOM_Petermann_Code.git
References
- Asay-Davis XS, Cornford SL, Durand G, Galton-Fenzi BK, Gladstone RM, Gudmundsson GH,
Hattermann T, Holland DM, Holland D, Holland PR, Martin DF, Mathiot P, Pattyn F, Seroussi
H. 2016. Experimental design for three interrelated marine ice sheet and ocean model in-
tercomparison projects: MISMIP v. 3 (MISMIP+), ISOMIP v. 2 (ISOMIP+) and MIS-
OMIP v. 1 (MISOMIP1). *Geoscientific Model Development* 9:2471–2497. doi:10.5194/
gmd-9-2471-2016.
- Aschwanden A, Fahnestock MA, Truffer M, Brinkerhoff DJ, Hock R, Khroulev C, Mottram R,
Khan SA. 2019. Contribution of the Greenland Ice Sheet to sea level over the next millennium.
*Science advances* 5:eaav9396.
- Bamber JL, Westaway RM, Marzeion B, Wouters B. 2018. The land ice contribution to sea level
during the satellite era. *Environmental Research Letters* 13. doi:10.1088/1748-9326/aac2f0.
- Banwell AF, MacAyeal DR. 2015. Ice-shelf fracture due to viscoelastic flexure stress induced by
fill/drain cycles of supraglacial lakes. *Antarctic Science* 27:587–597.
- Barthel A, Agosta C, Little CM, Hattermann T, Jourdain NC, Goelzer H, Nowicki S, Seroussi H,
Straneo F, Bracegirdle TJ. 2020. CMIP5 model selection for ISMIP6 ice sheet model forcing:
Greenland and Antarctica. *The Cryosphere* 14:855–879.
- Bell RE, Banwell AF, Trusel LD, Kingslake J. 2018. Antarctic surface hydrology and impacts on
ice-sheet mass balance. *Nature Climate Change* 8:1044–1052.
- Briner JP, Cuzzone JK, Badgeley JA, Young NE, Steig EJ, Morlighem M, Schlegel N, Hakim GJ,
Schaefer JM, Johnson JV, Lesnek AJ, Thomas EK, Allan E, Bennike O, Cluett AA, Csatho B,
de Vernal A, Downs J, Larour E, Nowicki S. 2020. Rate of mass loss from the Greenland Ice
Sheet will exceed Holocene values this century. *Nature* doi:10.1038/s41586-020-2742-6.
- Cai C, Rignot E, Menemenlis D, Nakayama Y. 2017. Observations and modeling of ocean-
induced melt beneath Petermann Glacier Ice Shelf in northwestern Greenland. *Geophysical*
*Research Letters* 44:8396–8403.
- Chen C, Huang H, Beardsley RC, Liu H, Xu Q, Cowles G. 2007. A finite volume numerical
approach for coastal ocean circulation studies: Comparisons with finite difference models.
*Journal of Geophysical Research: Oceans* 112.
- Ciraci E, Rignot E, Scheuchl B, Tolpekin V, Wollersheim M, An L, Milillo P, Bueso-Bello JL,
Rizzoli P, Dini L. 2023. Melt rates in the kilometer-size grounding zone of petermann glacier,
greenland, before and during a retreat. *Proceedings of the National Academy of Sciences*
120:e2220924120.
- Cook SJ, Christoffersen P, Todd J, Slater D, Chauché N. 2020. Coupled modelling of sub-
glacial hydrology and calving-front melting at Store Glacier, West Greenland. *The Cryosphere*
14:905–924.
- Davis PE, Nicholls KW, Holland DM, Schmidt BE, Washam P, Riverman KL, Arthern RJ,
Vaňková I, Eayrs C, Smith JA, et al. 2023. Suppressed basal melting in the eastern Thwaites
Glacier grounding zone. *Nature* 614:479–485.
- DeConto RM, Pollard D, Alley RB, Velicogna I, Gasson E, Gomez N, Sadai S, Condrón A,
Gilford DM, Ashe EL, et al. 2021. The Paris Climate Agreement and future sea-level rise
from Antarctica. *Nature* 593:83–89.
- Dow CF, Lee WS, Greenbaum JS, Greene CA, Blankenship DD, Poinar K, Forrest AL, Young
DA, Zappa CJ. 2018. Basal channels drive active surface hydrology and transverse ice shelf
fracture. *Science Advances* 4:eaa07212.
- Drews R, Pattyn F, Hewitt I, Ng F, Berger S, Matsuoka K, Helm V, Bergeot N, Favier L, Neckel
680 N. 2017. Actively evolving subglacial conduits and eskers initiate ice shelf channels at an
681 Antarctic grounding line. *Nature communications* 8:15228.
- Ehrenfeucht S, Morlighem M, Rignot E, Dow CF, Mouginot J. 2023. Seasonal acceleration of
Petermann Glacier, Greenland, from changes in subglacial hydrology. *Geophysical Research*
*Letters* 50:e2022GL098009.
- Fettweis X, Franco B, Tedesco M, Van Angelen J, Lenaerts JT, van den Broeke MR, Gallée H.
2013. Estimating the Greenland ice sheet surface mass balance contribution to future sea level
rise using the regional atmospheric climate model MAR. *The Cryosphere* 7:469–489.
- Forster RR, Box JE, Van Den Broeke MR, Miège C, Burgess EW, Van Angelen JH, Lenaerts JT,
Koenig LS, Paden J, Lewis C, et al. 2014. Extensive liquid meltwater storage in firn within the
Greenland ice sheet. *Nature Geoscience* 7:95–98.
- Fox-Kemper B, Hewitt HT, Xiao C, Aðalgeirsdóttir G, Drijfhout SS, Edwards TL, Golledge NR,
Hemer M, Kopp RE, Krinner G, Mix A, Notz D, Nowicki S, Nurhati IS, Ruiz L, Sallée JB,
BA SA, Y Y. 2021. Ocean, cryosphere and sea level change. In *Climate Change 2021: The*
*Physical Science Basis. Contribution of Working Group I to the Sixth Assessment Report of*
*the Intergovernmental Panel on Climate Change [Masson-Delmotte, V., P. Zhai, A. Pirani,*
*S.L. Connors, C. Péan, S. Berger, N. Caud, Y. Chen, L. Goldfarb, M.I. Gomis, M. Huang, K.*
*Leitzell, E. Lonnoy, J.B.R. Matthews, T.K. Maycock, T. Waterfield, O. Yelekçi, R. Yu, and B.*
*Zhou (eds.)]. Cambridge University Press, Cambridge, United Kingdom and New York, NY,*
*USA, pp. 1211–1362 doi:10.1017/9781009157896.011.*
- Frémand AC, Fretwell P, Bodart JA, Pritchard HD, Aitken A, Bamber JL, Bell R, Bianchi C,
Bingham RG, Blankenship DD, et al. 2023. Antarctic bedmap data: Findable, accessible,
interoperable, and reusable (fair) sharing of 60 years of ice bed, surface, and thickness data.
*Earth System Science Data* 15:2695–2710.
- Gadi R, Rignot E, Menemenlis D. 2023. Modeling ice melt rates from seawater intrusions
in the grounding zone of petermann gletscher, greenland. *Geophysical Research Letters*
50:e2023GL105869.
- Goelzer H, Nowicki S, Payne A, Larour E, Seroussi H, Lipscomb WH, Gregory J, Abe-Ouchi A,
Shepherd A, Simon E, Agosta C, Alexander P, Aschwanden A, Barthel A, Calov R, Chambers
C, Choi Y, Cuzzzone J, Dumas C, Edwards T, Felikson D, Fettweis X, Golledge NR, Greve R,
Humbert A, Huybrechts P, Le clec’h S, Lee V, Leguy G, Little C, Lowry DP, Morlighem M,
Nias I, Quiquet A, Rückamp M, Schlegel NJ, Slater DA, Smith RS, Straneo F, Tarasov L, van de
Wal R, van den Broeke M. 2020. The future sea-level contribution of the Greenland ice sheet:
a multi-model ensemble study of ISMIP6. *The Cryosphere*. doi:10.5194/tc-14-3071-2020.
- Goldberg DN, Twelves AG, Holland PR, Wearing MG. 2023. The non-local impacts of antarctic
subglacial runoff. *Journal of Geophysical Research: Oceans* 128:e2023JC019823.
- Golledge NR, Keller ED, Gomez N, Naughten KA, Bernales J, Trusel LD, Edwards TL. 2019.
Global environmental consequences of twenty-first-century ice-sheet melt. *Nature* 566:65–72.
- Greene CA, Gardner AS, Wood M, Cuzzzone JK. 2024. Ubiquitous acceleration in Greenland Ice
Sheet calving from 1985 to 2022. *Nature* 625. doi:doi.org/10.1038/s41586-023-06863-2.
- Gwyther DE, Dow CF, Jendersie S, Gourmelen N, Galton-Fenzi BK. 2023. Subglacial freshwater
drainage increases simulated basal melt of the Totten Ice Shelf. *Geophysical Research Letters*
50:e2023GL103765.
- Hattermann T, Isachsen PE, von Appen WJ, Albretsen J, Sundfjord A. 2016. Eddy-driven recir-
culation of atlantic water in fram strait. *Geophysical Research Letters* 43:3406–3414.
- Hill EA, Gudmundsson GH, Carr JR, Stokes CR. 2018. Velocity response of Petermann Glacier,
northwest Greenland, to past and future calving events. *The Cryosphere* 12:3907–3921.
- Holland DM, Jenkins A. 1999. Modeling thermodynamic ice–ocean interactions at the base of
an ice shelf. *Journal of physical oceanography* 29:1787–1800.
- Hunke EC, Lipscomb WH, Turner AK, Jeffery N, Elliott S. 2015. Cice: The los alamos sea ice
model documentation and software user’s manual version 5.1 la-cc-06-012. T-3 Fluid Dynam-
ics Group, Los Alamos National Laboratory 675:15.
- Jakobsson M, Mayer L, Coakley B, Dowdeswell JA, Forbes S, Fridman B, Hodnesdal H,
Noormets R, Pedersen R, Rebesco M, et al. 2012. The International Bathymetric Chart of
the Arctic Ocean (IBCAO) version 3.0. *Geophysical Research Letters* 39.
- Johnson H, Münchow A, Falkner K, Melling H. 2011. Ocean circulation and properties in Peter-
manns Fjord, Greenland. *Journal of Geophysical Research: Oceans* 116.
- Le Brocq AM, Ross N, Griggs JA, Bingham RG, Corr HF, Ferraccioli F, Jenkins A, Jordan TA,
Payne AJ, Rippin DM, et al. 2013. Evidence from ice shelves for channelized meltwater flow
beneath the Antarctic Ice Sheet. *Nature Geoscience* 6:945–948.
- Lepp A, Simkins L, Anderson J, Clark R, Wellner J, Hillenbrand C, Smith J, Lehrmann A, Totten
R, Larter R, et al. 2022. Sedimentary signatures of persistent subglacial meltwater drainage
from Thwaites Glacier, Antarctica. *Frontiers in Earth Science* 10:863200.
- Li T, Robinson LF, MacGilchrist GA, Chen T, Stewart JA, Burke A, Wang M, Li G, Chen J, Rae
JW. 2023. Enhanced subglacial discharge from Antarctica during meltwater pulse 1A. *Nature*
*Communications* 14:7327.
- Millan R, Jager E, Mouginit J, Wood MH, Larsen SH, Mathiot P, Jourdain NC, Bjørk A. 2023.
Rapid disintegration and weakening of ice shelves in North Greenland. *Nature Communica-*
*tions* doi:10.1038/s41467-023-42198-2.
- Millan R, Mouginit J, Derkacheva A, Rignot E, Milillo P, Ciraci E, Dini L, Bjørk A. 2022. On-
going grounding line retreat and fracturing initiated at the Petermann Glacier Ice Shelf, Green-
land, after 2016. *The Cryosphere* 16:3021–3031.
- Morlighem M, Williams CN, Rignot E, An L, Arndt JE, Bamber JL, Catania G, Chauché N,
Dowdeswell JA, Dorschel B, et al. 2017. Bedmachine v3: Complete bed topography and
ocean bathymetry mapping of Greenland from multibeam echo sounding combined with mass
conservation. *Geophysical research letters* 44:11–051.
- Mouginit J, Rignot E, Bjørk AA, Van den Broeke M, Millan R, Morlighem M, Noël B, Scheuchl
B, Wood M. 2019. Forty-six years of Greenland Ice Sheet mass balance from 1972 to 2018.
*Proceedings of the National Academy of Sciences* 116:9239–9244.
- Münchow A, Padman L, Fricker HA. 2014. Interannual changes of the floating ice shelf of Pe-
termann Gletscher, North Greenland, from 2000 to 2012. *Journal of Glaciology* 60:489–499.
- Muntjewerf L, Petrini M, Vizcaino M, Ernani da Silva C, Sellevold R, Scherrenberg MD, Thayer-
Calder K, Bradley SL, Lenaerts JT, Lipscomb WH, et al. 2020. Greenland Ice Sheet contribu-
tion to 21st century sea level rise as simulated by the coupled CESM2. 1-cism2. 1. *Geophysical*
*Research Letters* 47:e2019GL086836.
- Narkevic A, Csatho B, Schenk T. 2023. Rapid basal channel growth beneath Greenland’s longest
floating ice shelf. *Geophysical Research Letters* 50:e2023GL103226.
- Noël B, van de Berg WJ, Lhermitte S, van den Broeke MR. 2019. Rapid ablation zone expansion
amplifies north greenland mass loss. *Science advances* 5:eaaw0123.
- Noël B, van Wessem JM, Wouters B, Trusel L, Lhermitte S, van den Broeke MR. 2023. Higher
Antarctic ice sheet accumulation and surface melt rates revealed at 2 km resolution. *Nature*
*communications* 14:7949.
- Otto J, Holmes FA, Kirchner N. 2022. Supraglacial lake expansion, intensified lake drainage
frequency, and first observation of coupled lake drainage, during 1985–2020 at Ryder Glacier,
Northern Greenland. *Frontiers in Earth Science* 10:978137.
- Padman L, Erofeeva S. 2004. A barotropic inverse tidal model for the arctic ocean. *Geophysical*
*Research Letters* 31.
- Pelle T, Greenbaum J, Ehrenfeucht S, Dow C, McCormack F. 2024. Subglacial discharge accel-
erates dynamic retreat of aurora subglacial basin outlet glaciers, east antarctica, over the 21st
century. *Journal of Geophysical Research: Earth Surface* 129:e2023JF007513.
- Pelle T, Greenbaum JS, Dow CF, Jenkins A, Morlighem M. 2023. Subglacial discharge accelerates
future retreat of Denman and Scott Glaciers, East Antarctica. *Science Advances* 9:ead9014.
- Poinelli M, Nakayama Y, Larour E, Vizcaino M, Riva R. 2023. Ice-front retreat controls
on ocean dynamics under larsen c ice shelf, antarctica. *Geophysical Research Letters*
50:e2023GL104588.
- Prakash A, Zhou Q, Hattermann T, Bao W, Graverson R, Kirchner N. 2022. A nested high-
resolution unstructured grid 3-D ocean-sea ice-ice shelf setup for numerical investigations of
the Petermann ice shelf and fjord. *MethodsX* 9:101668.
- Prakash A, Zhou Q, Hattermann T, Kirchner N. 2023. Impact of the Nares Strait sea ice arches
on the long-term stability of the Petermann Glacier ice shelf. *The Cryosphere* 17:5255–5281.
doi:10.5194/tc-17-5255-2023.
- Rawlins LD, Rippin DM, Sole AJ, Livingstone SJ, Yang K. 2023. Seasonal evolution of the
supraglacial drainage network at Humboldt Glacier, North Greenland, between 2016 and 2020.
*The Cryosphere Discussions* :1–32.
- Rignot E, Fenty I, Xu Y, Cai C, Kemp C. 2015. Undercutting of marine-terminating glaciers in
West Greenland. *Geophysical Research Letters* 42:5909–5917.
- Rignot E, Steffen K. 2008. Channelized bottom melting and stability of floating ice shelves.
*Geophysical Research Letters* 35.
- Rosevear MG, Gayen B, Galton-Fenzi BK. 2022. Regimes and transitions in the basal melting of
antarctic ice shelves. *Journal of Physical Oceanography* 52:2589–2608.
- Scambos T, Fricker HA, Liu CC, Bohlander J, Fastook J, Sargent A, Massom R, Wu AM. 2009.
Ice shelf disintegration by plate bending and hydro-fracture: Satellite observations and model
results of the 2008 Wilkins ice shelf break-ups. *Earth and Planetary Science Letters* 280:51–60.
- Schmidt BE, Washam P, Davis PE, Nicholls KW, Holland DM, Lawrence JD, Riverman KL,
Smith JA, Spears A, Dichek D, et al. 2023. Heterogeneous melting near the Thwaites Glacier
grounding line. *Nature* 614:471–478.
- Sciascia R, Straneo F, Cenedese C, Heimbach P. 2013. Seasonal variability of submarine melt
rate and circulation in an east Greenland fjord. *Journal of Geophysical Research: Oceans*
118:2492–2506.
- Shroyer EL, Padman L, Samelson R, Münchow A, Stearns LA. 2017. Seasonal control of Peter-
mann Gletscher ice-shelf melt by the ocean’s response to sea-ice cover in Nares Strait. *Journal*
*of Glaciology* 63:324–330.
- Shu Q, Wang Q, Årthun M, Wang S, Song Z, Zhang M, Qiao F. 2022. Arctic Ocean Amplification
in a warming climate in CMIP6 models. *Science Advances* 8:eabn9755.
- Slater D, Nienow P, Cowton T, Goldberg D, Sole A. 2015. Effect of near-terminus subglacial
hydrology on tidewater glacier submarine melt rates. *Geophysical Research Letters* 42:2861–
2868.
- Slater D, Straneo F. 2022. Submarine melting of glaciers in Greenland amplified by atmospheric
warming. *Nature Geoscience* 15:794–799.
- Slater DA, Felikson D, Straneo F, Goelzer H, Little CM, Morlighem M, Fettweis X, Nowicki S.
2020. Twenty-first century ocean forcing of the Greenland ice sheet for modelling of sea level
contribution. *The Cryosphere* 14:985–1008.
- Tinto KJ, Bell RE, Cochran JR, Münchow A. 2015. Bathymetry in Petermann fjord from Oper-
ation IceBridge aerogravity. *Earth and Planetary Science Letters* 422:58–66.
- Turton JV, Hochreuther P, Reimann N, Blau MT. 2021. The distribution and evolution of
supraglacial lakes on 79 N Glacier (north-eastern Greenland) and interannual climatic con-
trols. *The Cryosphere* 15:3877–3896.
- Washam P, Münchow A, Nicholls KW. 2018. A decade of ocean changes impacting the ice shelf
of Petermann Gletscher, Greenland. *Journal of Physical Oceanography* 48:2477–2493.
- Washam P, Nicholls KW, Münchow A, Padman L. 2020. Tidal modulation of buoyant flow and
basal melt beneath Petermann Gletscher ice Shelf, Greenland. *Journal of Geophysical Re-*
*search: Oceans* 125:e2020JC016427.
- Wearing M, Stevens L, Dutrieux P, Kingslake J. 2021. Ice-shelf basal melt channels stabilized by
secondary flow. *Geophysical Research Letters* 48:e2021GL094872.
- Willis MJ, Herried BG, Bevis MG, Bell RE. 2015. Recharge of a subglacial lake by surface
meltwater in northeast Greenland. *Nature* 518:223–227.
- Wilson N, Straneo F, Heimbach P. 2017. Satellite-derived submarine melt rates and mass balance
(2011–2015) for Greenland’s largest remaining ice tongues. *The Cryosphere* 11:2773–2782.
- Zeising O, Neckel N, Dörr N, Helm V, Steinhage D, Timmermann R, Humbert A. 2023. Extreme
melting at Greenland’s largest floating ice tongue. *EGUsphere* :1–35.
- Zhou Q, Hattermann T. 2020. Modeling ice shelf cavities in the unstructured-grid, finite volume
community ocean model: Implementation and effects of resolving small-scale topography.
*Ocean Modelling* 146:101536.
- Zhou Q, Zhao C, Gladstone R, Hattermann T, Gwyther D, Galton-Fenzi B. 2024. Evaluat-
ing an accelerated forcing approach for improving computational efficiency in coupled ice
sheet–ocean modelling. *Geoscientific Model Development* 17:8243–8265. doi:10.5194/
gmd-17-8243-2024.
**Acknowledgements**
This work was funded by the Swedish Research Council VR grant 2022-03718 to N.K..
Q.Z. was supported by the Research Council of Norway (RCN) project 244319 and
295075. T.H. was supported by the RCN project 314570. We thank Donald Slater for
providing the present-day and RCP 8.5 runoff estimates for the Petermann Glacier. The
simulations were performed on resources provided by UNINETT Sigma2 - the National
Infrastructure for High Performance Computing and Data Storage in Norway. The super-
computer Betzy was used under the project NN9824K.
**Contributions**
All authors contributed to the conceptualization of the study. A.P. designed the exper-
iments with technical input from Q.Z. and T.H.. A.P. prepared and ran the numerical
simulations. A.P. analyzed the model output with scientific advice from Q.Z. and T.H..
859 A.P. and N.K. wrote the manuscript, with contributions from Q.Z. and T.H..

**Competing interests**
The authors declare no competing interests.
**A Model tuning**

Table A1. PGIS Petermann ice shelf thermodynamical and ocean mixing parameters used in this study

Parameter	Value	Description
ρ_{fw}	1000 kg m^{-3}	Freshwater density
L	$3.34 \times 10^5 \text{ J kg}^{-1}$	Latent heat of fusion of ice
ρ_{sw}	1028 kg m^{-3}	Seawater density
c_w	$3974 \text{ J }^\circ\text{C}^{-1} \text{ kg}^{-1}$	Specific heat capacity of seawater
Γ_T	1.2×10^{-2}	Non-dimensional heat-transfer coefficient
C_D	2.5×10^{-3}	Drag coefficient
u_{res}	$1.0 \times 10^{-2} \text{ m s}^{-1}$	Residual velocity
λ_1	$-5.73 \times 10^{-2} \text{ }^\circ\text{C PSU}^{-1}$	Liquidus slope
λ_2	$8.32 \times 10^{-2} \text{ }^\circ\text{C}$	Liquidus intercept
λ_3	$-7.53 \times 10^{-8} \text{ }^\circ\text{C Pa}^{-1}$	Liquidus pressure coefficient
Γ_S	$\Gamma_T/35$	Non-dimensional salt-transfer coefficient
Z_o	1.0×10^{-3}	Roughness length scale
Ro_{min}	2.5×10^{-3}	Roughness minimum
K_m	1.0×10^{-5}	Vertical eddy viscosity
P_v	1.0	Vertical Prandtl Number
P_h	1.0×10^{-1}	Horizontal Prandtl Number
C_h	1.0×10^{-1}	Scaling constant

863 **B Longitudinal ice shelf draft profile**

Figure B1. (a) Map of the PGIS Petermann ice shelf draft with the across-fjord transect ca. 5 km from the GL grounding line shown in Fig. 1(b) overlaid. Locations Apex locations of the four prominent longitudinal ice shelf basal channels at this transect (highlighted in Fig. 5) at this transect are represented using magenta diamonds. From left to right, these are \$L_w\$ (west), \$L_c\$ (central), \$L_e(1)\$ (east-1) and \$L_e(2)\$ (east-2). (b) Along-fjord profile of the PGIS Petermann ice shelf draft over the channel sections (Fig. 6(a)) along the western (\$S_w\$; blue), central (\$S_c\$; black) and eastern (\$S_e\$; red) regions channel sections (Fig. 6(a)) as indicated in panel a.

C Saturation of the thermal pump

Figure C1. Along-fjord summer mean temperature (left column), salinity (middle column), and flow (right column) anomalies (δ) for Q_{sg} -median relative to control (first row) and Q_{sg} -RCP 8.5 relative to control (second row) experiments for the section shown in Fig. 1(b). Black stippled lines in panels a, b, d and e indicate the zero temperature and salinity difference contours. Solid black lines in panels c and f depict the mean isopycnals for the control experiment, whereas the stippled green lines correspond to the Q_{sg} -median (c) and Q_{sg} -RCP 8.5 (f) experiments. Isopycnals are plotted at equal intervals of 0.2 kg/m^3 . The 27.9 kg/m^3 isopycnal corresponds to the dense Atlantic Water that contacts the grounding line. In each panel, PGIS-GL the grounding line (at ca. 600 m depth) is on the right margin and open ocean is to the left. Vertical magenta line in panels a, b, d, and e show the location of the model node ca. 10 km from the GL-grounding line at which the vertical profile of summer mean temperature and salinity are shown for the control, Q_{sg} -present, Q_{sg} -median, and Q_{sg} -RCP 8.5 experiments in Fig. 3(g,h) (cf. Table 1).

865 **D** Trapping Convergence of meltwater

Figure D1. Anomaly maps (δ) of the summer mean surface layer salinity (a,b) and basal melt rate (c,d) for the Q_{sg} -present relative to control (column 1) and Q_{sg} -median relative to control (column 2) experiments. Contours of the ice shelf draft are plotted at 100 m intervals (450 m – 150 m) and overlaid as dotted red and blue lines over the surface layer salinity and melt rate maps, respectively.

E Modified sub-ice shelf water column thickness

Figure E1. (a) BedMachine v3 (Morlighem et al., 2017) water column thickness, and (b) modified BedMachine v3 water column thickness used in this study, obtained from Prakash et al. (2022). (c) Along-fjord profile of the water column thickness for the western (blue), central (black), and eastern (red) sections of the fjord as highlighted in panels (a) and (b). Stippled and solid lines correspond, respectively, to the BedMachine and modified BedMachine profiles. (d) Along-fjord summer mean temperature for the section shown in Fig. 1(b). Mean isopycnals (solid black lines) are overlaid at equal intervals of 0.2 kg/m^3 . The 27.9 kg/m^3 isopycnal surface represents the dense Atlantic Water that contacts the grounding line. Solid red line represents the (unmodified) BedMachine v3 bathymetry, wherein, a synthetic sill blocks the 27.9 kg/m^3 isopycnal surface from contacting the grounding line.

Reviewer #3 (Remarks to the Author):

Petermann glacier in NW Greenland is rapidly losing mass and contributing strongly to modern global sea-level rise. Ice flow from Petermann is currently slowed by the presence of a ~30km floating ice tongue which buttresses flow. With ocean warming across the last ~two decades, that ice shelf has thinned, and future breakup of the ice shelf is possible. This paper uses numerical modeling to estimate the role that future atmospheric warming may have on ice shelf break-up. Where atmospheric temperatures warm, ice surface melt increases. This surface melt is able to reach the subglacial environment at moulins, and then flows into fjords, where it influences ocean circulation and melting. Using a coupled ice-shelf/ocean/sea ice model, this work finds that subglacial discharge causes enhanced subglacial melt, particularly within subglacial channels.

Generally speaking, this work is timely, impactful, and worthy of publication in Nature Communications with revisions. The majority of my suggested improvements below are easy to implement or require simple clarification. I have three more major concerns, one of which is easily addressed (point two below) by additional discussion or re-framing of the work. The other two points may require additional modeling work.

We appreciate your time and effort in reviewing our manuscript and are pleased to hear of the decision. We have provided clarifications wherever necessary, and revised our manuscript following your comments. Please note that unless stated otherwise, line numbers provided below refer to the *track changes* document (appended to this report for convenience), where new text appears in blue and removed text is marked with red strikethrough.

Firstly, we would like to take this opportunity to inform you that during the revision process, we noticed a (minor) mistake in our calculation of the drivers of melt, namely, thermal driving and friction velocity. While the updated numbers are slightly different, we note that this does not impact any of the findings of our study, and the revised figures remain qualitatively identical. We humbly request to proceed with these corrected values, and sincerely apologize for the oversight and for any inconvenience this may cause. Please see the revised percentages in Lines 133 – 134, 167 – 170, 184, 195, and 227, and Figures 1(d), 2(e – l), and 5(c,d).

Please find our detailed (point-by-point) responses below.

My primary issue with the work is that it frames itself as an improvement on previous modeling work because it uses more realistic domain geometry. However, the domain geometry used in these model runs is not sufficiently explained. Maybe I missed it, but I cannot find where the authors got ice shelf draft geometry from. I assume it is a radar flight

line? From what year and what instrument? More importantly, the 'improved' seafloor geometry is a modified form of BedMachine, but the improved seafloor geometry is never shown. What is the cavity shape? Why were the modifications made to BedMachine? It's also worth noting that — to the best of my knowledge — BedMachine has no data for the ice shelf cavity. It simply interpolates/kreigs between radar flightlines (and the mass-conservation estimate between them) for grounded ice and the downstream seabed bathymetry from ships. So really very little is known about the actual shape of the seafloor for the ocean cavity that is modeled here. To what degree does uncertainty in this cavity geometry impact the modeling results presented here? I recognize that a full suite of modeling runs across a range of seafloor geometries is likely beyond the scope of this work, but even a single run that presents a modified geometry would be useful for determining the sensitivity of the model results to the unknown cavity geometry.

Thank you for bringing this shortcoming to our notice. The ice shelf draft comes from BedMachine (line 337, original manuscript). You are correct regarding the unavailability of data in the ice shelf cavity and the interpolation that is adopted to fill this gap. For Petermann, this technique results in the formation of an artificial inner sill that blocks the inflow of the warm and dense (27.9 kg/m^3) Atlantic Water at depth that contacts the grounding line at ca. 600 m depth (Prakash et al., 2022, Prakash et al., 2023). It is imperative to account for this artifact, otherwise, the model will fail to reproduce the fundamental aspects of Arctic fjord circulation and clearly disagree with the findings of Washam et al. (2018) who show that the 27.9 kg/m^3 isopycnal surface overflows the sills and contacts the grounding line. Fortunately, there exists IceBridge data for Petermann from Tinto et al. (2015) (lines 347 – 351, original manuscript), which provides an improved understanding of the sub-ice shelf bathymetry. It revealed the presence of a 540 – 610 m deep inner sill ca. 25 km from the grounding line. A study conducted by Shroyer et al. (2017), who directly implemented the BedMachine bathymetry, presented a novel mechanism that was responsible for delivering more heat from the Nares Strait into the fjord, however, they suggested that its effect was limited to the shallower (<200 m) regions of the ice shelf. This did not agree with the modelled and observed findings of Cai et al. (2017) (who set up their 2-D model along the well-mapped along-fjord IceBridge flight track) and Washam et al. (2020), respectively. Thus, sufficient evidence was available to us which suggested that these topographic adjustments (integrating modified BedMachine data with IceBridge data) would be an integral component of our model setup (Prakash et al., 2022), and simulations performed without it would not be meaningful. Indeed, this was confirmed and discussed further in Prakash et al. (2023).

The technical details involving the generation of our smoothed topography datasets are too cumbersome to include in this study, however, they have been documented and justified in detail in Prakash et al. (2022), and further discussed in Prakash et al. (2023), and referred to in this study. However, we acknowledge the shortcoming of the methods chapter where these

modifications should be detailed further. We have revised the text to clarify the topographic datasets that have been used, and modifications that have been made together with appropriate justifications (as described above) for the same. We now also provide a supplementary figure (Figure E1) which shows the old (BedMachine) vs new (modified-BedMachine + IceBridge) water column thickness maps and cross-section profiles. The latter clearly illustrates the aforementioned artificial sill and blocking of Atlantic Water in the non-modified geometry.

Please see lines 59 – 69, lines 497 – 533, and Figure E1 in the *track changes* document.

The ice geometry used in the model runs was a pre-2010 ice geometry (pre large calving event) but the atmospheric and ocean forcing was for later years (I believe). Given the large change at the glacier in 2010/12 with the calving, what uncertainty is baked in to modeling results as a product of not using the modern ice configuration? Why was this choice made? Some discussion is warranted here.

Thank you for raising this concern. Indeed, the ice geometry used is a pre-2010 geometry which is not consistent with the period of the oceanic and surface forcing. This is because, to the best of our knowledge, only this geometry of the Petermann ice shelf is provided in the BedMachine Greenland dataset that is publicly available. However, we are happy to inform that following two successful fieldwork campaigns to Petermann in 2019 and 2024, we have managed to geolocate the ongoing rift and derive the updated draft from airborne radar surveys. By combining them, we have assembled together the updated geometry, and simulations investigating changes driven by change in geometry (pre- vs. post-calving geometry) are currently being analyzed. While we expect to note changes in magnitude and spatial variability of melt imparted by (a) change in ice shelf geometry, and (b) combined effect of geometry change + addition (and increase) of subglacial discharge, it is difficult to make speculations at this stage regarding both (a) and (b), and any mechanisms that may get uncovered that drive these changes. We would also like to note that the 2010 and 2012 calving events (and the likely future calving event) removed (will remove) the soft and thin outer sections of the ice shelf (drafts ca. <150 m). As such, the results presented here – the saturation of the thermal pump and amplified channelized melting under the deeper drafts – are robust, irrespective of the geometry used. We have revised the text in the methods chapter to clarify the reason behind our choice, and discuss any potential implications on our results. Please see lines 555 – 563 in the *track changes* document.

A third concern I have is a lack of discussion of the role of tidal pumping within the grounding zone environment and its potential impact on grounding zone melt rates. Recent work shows that the high levels of subglacial melt at the Peterman grounding zone might be due to seawater movement with the tides (Gadi et al (2023, GRL) and Ciraci et al (2023, PNAS)). The sub-ice shelf community has recently been very focused on the role of tidal seawater pumping

and potential melting inland of the grounding zone by this seawater. There is a rich literature describing both the theory (Walker et al., 2013), modeling seawater intrusions (Robel et al., 2022), and observations of intrusions (Rignot et al., 2024). For a nice overview and intro to this subject, Parizek 2024 (GRL) summarizes the current state of thinking of tidal pumping and its role at Peterman. I think it is a significant omission that this work is not considered in this paper, as tidal pumping across the grounding zone may be a large source of melt (and water flow) there. Again, I recognize that a full suite of model runs that incorporate tidal pumping across the grounding zone are beyond the scope of the work presented here, but from my prospective, not even mentioning the role of tidal pumping across the grounding zone is a significant omission of this work: if nothing else, the potential role of this process should be discussed in the introduction and discussion.

Thank you for this insightful comment, and for the valuable literature recommendations. We acknowledge that tidally-driven seawater intrusions are critical to the stability of the Petermann Glacier, however, we also agree with your remark that conducting such simulations falls beyond the scope of this work, both from a technical and scientific perspective. From a technical standpoint, substantial amount of work would have to be carried out to extend our 3-D regional model setup to incorporate the requisite (e.g., cavity opening) features, conduct tests (validation), and perform simulations. From a scientific perspective, our study is intended to highlight the mechanisms that come-forth when climate warming driven heightened subglacial discharge enters the fjord. To uncover these mechanisms, it is imperative to not convolute our experiments by investigating other mechanisms that interact (discussion paragraph 3 in Gadi et al. (2024)) with the ones we aim to highlight here. While our findings remain numerically robust, we acknowledge that studies investigating the interaction between discharge and tidally-modulated seawater intrusions will be essential in improving our understanding of the dynamics and stability of the glacier. We have made amendments to the introduction and discussions chapter to reflect this. Please see lines 34 – 37, and lines 476 – 478 in the *track changes* document.

Line-by-line and more minor comments

The abstract leaves me confused about geometry: a mechanism is described that exacerbates melt at the crest of channels (which I assume is the highest elevation point where the ice is the thinnest) but then the next sentence says that this melt will complete erode channels, which sounds like a flattening of channels. Does 'erode channels' here mean a complete incision of the shelf? If so, I would change language to 'We estimate that sustained intensified summer melting over a decade may undermining the stability of PGIS by allowing channels to cut entirely through the ice shelf'

Indeed, we mean to say a complete incision of the ice shelf. Thank you for the helpful reformulation. We have now changed this sentence in the revised manuscript. Please see lines 12 – 15 of the abstract in the *track changes* document.

I found the reliance on terms (e.g. Qsg for subglacial discharge) to be occasionally difficult — for the casual reader, this paper presents a significant number of terms that must be held in memory to derive meaning. I might encourage the authors to be more selective in what they make terms for (is it a word-count problem to just say ‘subglacial discharge?’), but I recognize that this is a style choice.

Thank you for bringing this to our notice. We have received similar concerns from Reviewer 1. Since, to our knowledge, we do not have a word count problem, we have decided to drop the abbreviations for “basal melting”, “subglacial discharge”, “thermal driving”, “friction velocity”, “grounding line”, “Greenland Ice Sheet”, and “Ryder Glacier”, as well as all other glacier and fjord names from the text for clarity.

8-11 the two clauses of this sentence appear to be jarringly unrelated, and generally I have a hard time following the logic structure of 8-14

In the original manuscript, we realize that the use of “while” in line 9 is perhaps the cause of confusion. In lines 8 – 11, we intended to report on the factors (basal melting, calving, and negative SMB) that drive mass loss. The recent collapses of ice shelves of Northern Greenland, while important to address, are not excluded from these factors. Hence, the use of while is incorrect. Thereafter, in lines 11 – 14, we mention how dynamic thinning and retreat (driven by these factors) could result in reduced backstress and increase the sector’s contribution to sea level rise. Please see the reformulated version (*track changes*) in lines 9 – 12.

13 Does the Petermann shelf do much in the way of buttressing? I thought that there was minimal grounded ice velocity response to the 2010 and 2012 calving events, which would suggest perhaps not. There may be modeling work that has been done that quantifies this. If so, worth citing and quickly mentioning here (given that the ice shelf community has recently been surprised by the lack of buttressing at other ice shelves like eastern Thwaites).

Indeed, grounded ice velocity response to the previous two calving events has been minimal. However, as we have mentioned in the discussion chapter (lines 181 – 188, original manuscript), modelling work of Hill et al. (2018) has shown that loss of the stiffer and deeper drafts (as opposed to the softer and thinner sections that were lost in 2010 and 2012) within 12 km from the grounding line could perturb grounding line stresses enough to significantly increase inland flow speeds, grounded ice discharge, and the glacier’s contribution to sea level rise. We have now included a citation to their work here as well. Please see lines 12 – 17 in the *track changes* document.

21 thinning the ice shelf from..

21 is the 600m/100m geometry presented for *within* the basal channels or regions outside of the channels?

We have received similar comments from Reviewers 1 and 2 regarding this sentence, and we apologize for the confusion it has caused. We aimed at detailing the average thickness gradient of the ice shelf – it ranges from 400 – 600 m deep near the grounding line to 50-100 m near the calving front. We also acknowledge that this gradient is not entirely driven by basal melting (as lines 18 – 21 are currently phrased in the original manuscript), and that at least some of it is due to longitudinal extension (as pointed out by Reviewer 1). We have revised lines 18 – 21 by first documenting the average thickness gradient extending from the grounding line to the calving front (driven by all possible factors). Thereafter, we describe the basal melt channel geometries. Please see lines 22 – 28 in the *track changes* document.

35-36 slightly confused by geometry here, is this an integrated melt rate for the region from GL to 16km downstream? What time of year were those observations made (i.e. sentence before says that melt rates triple in summer).

Yes, the geometry is as you have specified (grounding line to 16 km downstream of it), however, confined to the central basal channel. We have clarified this in the revised text, together with the time of the year (August – December 2015) when the observations were made. The sentence prior is describing the modelling work from Cai et al. (2017). Here, summer is June-July-August, and winter refers to months outside JJA. We have now revised our text to make this clear. Please see lines 44 – 51 in the *track changes* document.

37 My sense is that this is a controversial statement, so I would soften indicates to suggests

We agree with your opinion and have reformulated it in the revised text. Please see lines 51 – 52 in the *track changes* document.

45-46 reword to improved sub-ice shelf bathymetry using BedMachine3 (Morlighem et al., 2017). As written it reads as if Morlighem is a prior ice-ocean model that these results will be improving upon.

Thank you for bringing this to our attention. We agree with the suggested reformulation, however, we were advised by Reviewers 1 and 2 to revise the last paragraph of the Introduction chapter to include more detailed information regarding the model and experiment setup. As such, based on their suggestions, we have revised lines 43 – 59 (original manuscript), which we believe also fixes the problem addressed in this comment. Please see lines 59 – 100 in the *track changes* document.

53 does estuarine circulation start (i.e. at present time) NOT saturated and then becomes saturated at a point in time (suggesting regime change at that point?) Clarify here. I also don't have a great sense of what saturated circulation means.

Yes, that is correct. By saying "the former saturates", we intended to say that the advective heat transport (which increases the thermal driving) into the cavity saturates. We realize now that the phrasing is misleading. We have now explicitly stated what we mean by "the former" (i.e. heat transport/thermal driving and not circulation) in the revised text. Please see line 89 in the *track changes* document.

58-59 at risk of repeating myself, I still wish for a different term than 'complete erosion of the channels' — this makes it sound like channels are going away. What about 'complete erosion of the ice shelf within the channels'?

Thank you for raising this concern, and for the help with reformulation. We would have adapted it in our revised formulation, however, following remarks from Reviewer 2 (unusually long description of results at the end of the Introduction chapter), we have condensed the description of the results. As such, this sentence has been removed from the revised version.

62-64 The numbers for each of these values (your melt, remotely sensed Melt, in-situ measured melt) really need to be shown here... 'they agree with remote sensing/field data' is not sufficient - give a detailed comparison.

Thank you for pointing this out. We agree, and have now provided a more detailed comparison between data and our modelled estimates of basal melt. Please see lines 103 – 127 in the *track changes* document.

95 One more sentence of summary/overview would be nice here - a less technical 'this is what this means'

Thank you for this helpful insight, we agree and have added an overview to the revised text. Please see lines 155 – 158 in the *track changes* document.

97-129 Missing from this section is a physical description of the mechanism responsible for the saturation of the thermal pump. Is it increased (cold) meltwater pooling at locations in the cavity?

Thank you for noticing this. Pooling of meltwater under the shallower (<200 m) drafts is a factor when thermal driving is averaged over the entire ice shelf base, as we have described in lines 119-122 in the original manuscript. We have now added an elaborate description of the regime change in heat flux efficiency within the ice shelf cavity which explains the saturation of the thermal pump. Please see lines 197 – 208 in the *track changes* document.

141-142 This is not intuitive for me: isn't meltwater quite fresh and cold (ie minimal thermal capacity, minimal melt)? Why is there highest melt in areas of meltwater trapping, I would expect melt to be *low* there because of cold temps and increased stratification. Explain.

Yes, meltwater is fresh and cold, as can be inferred from Fig. 5(b,c), where we note that thermal driving is not higher at the channel crests where meltwater converges. It is, however, this vigorous convergence of meltwater towards the crests of the channels which results in substantial increases in friction velocity, which predominantly drives the increased melting (Fig. 5(d)). We discuss this in lines 219 – 230 in the original manuscript: As increased discharge drives more channelized melt, larger volumes of buoyant meltwater rise along the steep slopes, vigorously converging towards the crest which increases the friction velocity (Fig. 5). At the crest, this meltwater doesn't stagnate (pool up), but is advected downstream along the longitudinal channel axis by the stronger overturning circulation in the fjord (discharge-enhanced outflow) (Fig. 3 (i), Fig. 4, Fig. B1), which further exacerbates the friction velocity (Fig. 2). Thus, the growth of the channel in the vertical direction is primarily driven by the rapid convergence of meltwater towards the crests, which increases the friction velocity and melt. We have made the following refinements to the text to better convey our message:

i) We believe that convergence (instead of trapping) of meltwater is more appropriate and less confusing. As such, all occurrences of words such as trapping in this context have been replaced with convergence or similar.

ii) We add a note saying that this unintuitive process (higher melt at the crests/locations of meltwater convergence) is discussed further. We also explicitly state that thermal driving is not higher at the crests where meltwater converges. Please see lines 220 – 225 in the *track changes* document.

iii) We have made slight refinements to the text in the discussion chapter to improve readability. Please see lines 340 – 355 in the *track changes* document.

147-148 I would expect that meltwater would pool at the base of channels and melt would be slowest at the channel apex. Why is this not the case? I believe this is what ROV submersible Icefin observed under Thwaites (e.g. Schmidt et al., 2023 and the subsequent work of Peter Washam) - high melt rates on channel sides and slow melt at channel apex.

Please see our reply to the comment above. In the discussion paragraph centered around the likely implications of our findings for Antarctica (lines 259 – 292, original manuscript), we put forward our hypothesis in lines 274 – 292 (original manuscript).

164 Given that old (pre-2010) ice draft geometry is used here with modern (2024) melt rates, are these numbers realistic? How would they change if modern ice draft geometry was used?

Please see our reply to major comment #2.

181 Reword: The modeling work of Hill

Thank you for noticing this, we have fixed it in the revised text. Please see lines 275 – 276 in the *track changes* document.

188 I would suggest adding one more sentence here about WHY this has a significant bearing on PG stability ('this is significant' feels like a bit of a punt without more context — however I frequently do it in my writing as well!).

This is true, we have now added an explanation to the revised text. Please see lines 282 – 284 in the *track changes* document.

189-196 THIS! Cool. I would consider adding these ideas to the abstract and introduction.

Thank you for the suggestion. We have included some of these ideas to the abstract and the introduction chapter in the revised text. Please see lines 15 – 17 in the abstract and lines 96 – 100 in the introduction in the *track changes* document.

288-292 This is a key insight (no edits, just applauding)

AP: Thank you!

299-302 I'd push back on this — I see how improving our knowledge of grounding zone geometries is being prioritized by funding agencies and research groups. Your study highlights the need for such work, but saying that this work isn't being undertaken is an overstatement. Many good groups are doing targeted geophysical work to characterize the grounding zone environment, there are just major logistical and financial barriers to doing this on a continent-wide scale. Maybe re-cast this argument as a push for even more targeted airborne and ground-based geophysical campaigns to describe the grounding zone environment where possible.

Thank you for raising this concern. We are cognizant of, and also greatly appreciate the fieldwork campaigns that have significantly improved our understanding of these largely inaccessible environments. There might not be data available for every glacier-fjord system due to financial and logistical challenges as you have rightly pointed out, but to our knowledge, there exists a lot of valuable data for the well-known regions. This is true for Petermann, where BedMachine and IceBridge aerogravity data can be combined, and carefully extrapolated, to generate a meaningful picture of the fjord (including the sub-ice shelf region) bathymetry. Through these sentences, we wanted to encourage future modelling efforts to make use of the rich resources that are already available, if possible. It is important to clarify here that idealized studies are extremely vital, but if presenting a real-world case, real-world geometries should be preferred if possible. Indeed, it takes a fair amount of time to set-up and tune the model, but our results highlight why this is important. We have

addressed this by fixing line 299 (original manuscript). Furthermore, your suggestion of highlighting the need for more targeted campaigns is meaningful, and we have added it to line 302 (original manuscript). Please see lines 444 – 445 and lines 448 – 449 in the *track changes* document.

worth throwing in a citation for BedMap3 here too

Indeed, thank you, we have added this to the revised text. Please see line 452 in the *track changes* document.

The argument that you're making — that meter-scale features are essential to capture in continent-wide datasets — is admirable, but we're just so far from that reality in this moment. More useful, I wonder, might be to suggest ways to parameterize the meter-scale effects to make our continent-scale datasets more usable at this time? Realistically we are not going to send Icefin, Autosub, or Ran under every ice shelf to map every feature.

This is correct. We acknowledge that it is important to present feasible recommendations here, and have revised the text to reflect this. Please see lines 458 – 465 in the *track changes* document.

Worth noting though that modeling work has shown that distributed Qsg produces more subglacial melt than channelized Qsg (Slater et al., 2015 GRL)

Thank you for bringing this to our attention. We have added this to the revised text. Please see lines 467 – 469 in the *track changes* document.

is this seafloor geometry of ice base geometry?

This is the ice base geometry. We have replaced the word "basal topography" with "draft" to avoid confusion with "seafloor topography". Please see line 497 in the *track changes* document.

include a note as to whether or not this geometry is reasonable, with a citation... I.e. how wide are the channels at various points across the ice shelf— is a 200m grid sufficient to capture their geometry?

Thank you for noticing this, we have added a note to the revised text. Please see lines 500 – 502 in the *track changes* document.

Figure 1

To improve comprehension without needing to read long legend, I would suggest adding text to 1d that signifies that dark regions are deep water column and lighter regions are integrated

Thank you for the suggestion. We have added this information as text to Figure 1(d).

Figure 5

What is the source of data for the ice draft?

The ice draft comes from BedMachinev3 (Morlighem et al., 2017), and we have now clarified this in the figure caption.

Reviewer #3 (Remarks on code availability):

Link works and code is available, however I have not attempted to compile it.

AP: Thank you for raising this concern. We can assure that the model code and input files, together with the detailed instructions that accompany them, are sufficient to reproduce the results.

Future runoff increase may destabilize Petermann ice shelf

Abhay Prakash^{1,2,*}, Qin Zhou³, Tore Hattermann^{4,5}, and Nina Kirchner^{1,2}

¹Department of Physical Geography, Stockholm University, Stockholm 10691, Sweden

²Bolin Centre for Climate Research, Stockholm University, Stockholm 10691, Sweden

³Akvaplan-niva, Tromsø 9296, Norway

⁴Norwegian Polar Institute, Tromsø 9296, Norway

⁵Department of Physics and Technology, University of Tromsø, Tromsø 9019, Norway

*Corresponding author, abhay.prakash@natgeo.su.se

Abstract

Increased basal melting of the Petermann ~~Glacier-Ice Shelf (PGIS)-ice shelf~~ is typically attributed to rising ocean temperatures. Although subglacial discharge (Q_{sg}) has been shown to intensify melt, the mechanisms behind this increase and their evolution with increasing Q_{sg} discharge in a warmer climate remain unresolved. Using a 3-D numerical regional ice shelf-ocean model centered on the Petermann Fjord, we show that heightened Q_{sg} discharge under the RCP 8.5 scenario leads to more than threefold increase in summer mean melt when averaged over the deeper (>300 m) drafts compared to conditions without Q_{sg} discharge. Notably, we identify a regime change in heat flux efficiency within the ~~PGIS-cavity-when- Q_{sg} -ice shelf cavity when discharge~~ exceeds the current peak summer value. Here, thermal driving saturates, and ~~Q_{sg} -intensified discharge-intensified~~ currents increase melt by enhancing shear-driven turbulent mixing across the ~~PGIS-ocean-ice shelf-ocean~~ boundary layer. Increases in melt are most profound at the crests of the basal channels, where vigorous meltwater confluence exacerbates friction velocity. We estimate that sustained intensified summer melting over a decade may ~~completely erode the channels, undermining the stability of PGIS, undermine the stability of the ice shelf by allowing channels to cut entirely through it. Challenging conventional attributions of increased basal melting of the ice shelf to ocean warming alone, our results demonstrate how atmospheric warming exacerbates ocean-driven melt processes, and is likely to play a dominant role in amplifying future basal melt.~~ Considering the impact of the channelized basal morphology of ~~PGIS-the ice shelf~~ on the spatial heterogeneity of melt, and projected increases in Q_{sg} discharge, we posit that our results have wider implications for similar 'warm cavity' environments.

Introduction
The Greenland Ice Sheet (~~GrIS~~) is currently the largest contributor to global mean sea
level rise (Bamber et al., 2018; Fox-Kemper et al., 2021), and as ~~contemporary mass loss~~
~~rates~~ observational estimates of recent mass loss are underestimated (Greene et al., 2024),
they are hence likely lower bounds for rates to be observed in decades to come (Briner
et al., 2020; Goelzer et al., 2020). The northern sector of the ~~GrIS~~ Greenland Ice Sheet
hosts several of the last remaining marine terminating outlet glaciers featuring floating
ice tongues (also called ice shelves) that exert buttressing forces to the inland ice. During
2001 ~~—~~ 2021, dominating factors ~~for~~ of mass loss from these ice shelves were (in de-
creasing order) basal melt, calving, and negative surface mass balance, ~~while between,~~
Between 2003 and 2010, collapses were observed at Hagen Bræ, C.H. Ostenfeld glacier,
and Zachariæ Isstrøm (Millan et al., 2023). Dynamic thinning, retreat and even collapse
combined with perturbations of the outlet glacier's grounding ~~lines (GL)-line~~
a loss of backstress and accelerated mass loss from the ~~adjacent-GrIS-interior-sectors~~
northern Greenland Ice Sheet sectors (Hill et al., 2018) holding an ice volume of ca.
~~4m~~ 4 m global mean sea level equivalent (~~Mouginot et al., 2019~~) (Morlighem et al., 2017
: Mougnot et al., 2019).
Along the Nares Strait coast of the northern ~~GrIS, Petermann Glacier Ice Shelf (PGIS)~~
~~terminates into Petermann Fjord (PF). Erstwhile Greenland Ice Sheet, Petermann ice~~
~~shelf terminates into the Petermann Fjord. Formerly~~ the longest ice shelf in the northern
hemisphere at ca. 80 km in length, two recent massive episodic calving events in 2010
and 2012 reduced its length by ca. 30 km (Münchow et al., 2014). ~~The ice shelf draft~~
~~thins from ca. 400 – 600 m near the grounding line to ca. 50 – 100 m near the calving~~
~~front. Evenly spaced across its 15-20-15 — 20 km width are 1-2-1 – 2 km wide basal~~
~~melt channels that run along its length (Rignot and Steffen, 2008). These channels carve~~
~~incisions that are incise~~ several-100 m deep into ~~its draft, thinning it from ca. 600 m~~
~~near the GL to ca. 100 m near the calving front. the draft~~ (Rignot and Steffen, 2008
28). Warming of Atlantic water in ~~PF by $\sim 0.3^\circ\text{C}$~~ ~~Petermann Fjord by ca. 0.3°C~~ has been
observed since 1960, with much (~~\sim ca. 0.2°C~~) of the warming occurring in the last
two decades (Millan et al., 2022). Average basal melt rates (~~m_b~~) ~~at the PGIS-GL, at \sim~~
~~at the Petermann ice shelf grounding line, at ca. 600 m below sea level, are currently~~
reported to exceed 17 m/yr, leading Millan et al. (2023) to suggest that increasing ice
shelf basal melt is following increasing ocean temperatures in ~~GrIS fjords. Greenland~~
~~Ice Sheet fjords. Numerical modelling study and satellite observations have revealed~~
~~that highest basal melt rates occur in the kilometer-size grounding zone of the glacier,~~
~~driven by tidally modulated intrusions of seawater~~ (Ciraci et al., 2023; Gadi et al., 2023
37). Seasonal and climate-warming driven long-term changes in regional sea ice cover are
38 also known to enhance ~~m_b basal melting~~. According to model results by Shroyer et al.
(2017) and Prakash et al. (2023), elevated temperatures and an energetic ocean circula-
tion under ~~PGIS the ice shelf~~ driven by variations in Nares Strait sea ice mobility and
thickness increased ~~m_b basal melting~~ by a factor ranging between 1.2 and 1.7. Also, ~~m_b~~
~~melting~~ can be enhanced by routing of glacier surface melt to the glacier bed, and fur-
ther to the ~~GL grounding line~~ where it enters the fjord as subglacial discharge (~~Q_{sg}~~). ~~A~~
~~higher Q_{sg} during summer has been suggested to triple m_b at PGIS. Modelling study~~
~~suggests that a higher subglacial discharge during summer (June – August) triples basal~~
~~melt at Petermann ice shelf compared to winter (Cai et al., 2017)(Cai et al., 2017). Fur-~~
~~thermore, within the central basal channel, for the region extending from the grounding~~
~~line to 16 km seaward of it, between August and December 2015, Washam et al. (2020)~~
estimated area-averaged melt rate maximum of up to 170 m/yr, ~~upstream of a location~~
~~16 km from the GL in the central basal channel,~~ which were consistent with episodes
of enhanced ~~Q_{sg} driven discharge driven~~ sub-ice shelf currents. This ~~indicates that Q_{sg}~~
~~suggests that subglacial discharge~~ may play a more important role in modulating ~~PGIS~~
~~Petermann ice shelf~~ basal melt than sea ice driven changes in ocean circulation. However,
with ~~atmospheric and oceanic~~ warming expected to amplify in the Arctic by the end of
the century (Muntjewerf et al., 2020; Shu et al., 2022), a salient question remains unan-
swered: How would ~~PGIS Petermann ice shelf~~ basal melt respond to projected increases
in ~~Q_{sg} under a future warming discharge under a warmer future~~ climate, and what would
be the mechanisms driving this response?
~~Here~~In this study, we extend a high-resolution unstructured grid 3-D ocean-sea ice-
ice shelf regional model setup (Prakash et al., 2022, 2023) centered over ~~PGIS and PF~~
~~to include Q_{sg} at the GL. Novel features include moreover an improved (with respect to~~
~~Morlighem et al., 2017)the Petermann ice shelf and Petermann Fjord to include subglacial~~
~~discharge at the grounding line (Fig. 1). The fjord bathymetry and the ice shelf draft~~
~~are derived from BedMachine v3 (Morlighem et al., 2017). Modifications have been~~
~~applied to the topographical datasets (discussed further below) to amend an inaccurately~~
~~represented~~ sub-ice shelf ~~bathymetry and a realistic PGIS basal topography. With these,~~

Figure 1. (a) Map of our regional **PGIS-PF-Petermann ice shelf-Petermann Fjord** model domain (in a wider context shown in the inset in the upper left corner) which follows the Greenland and Ellesmere Island coastlines, and extends into the Lincoln Sea and Baffin Bay (red lines). The mesh size in the model domain varies regionally, as exemplified by the three green-framed boxes. **PF-Petermann Fjord** is surrounded by a purple polygon, detailed in panel (b). (b) **PGIS-Petermann ice shelf** extent (white outlines), from CE 2009, overlain on water column thickness (WCT) in **PF-Petermann Fjord** and adjacent parts of Nares Strait. Cyan line marks **GL-the grounding line**, where discharge (detailed in panel (c)) is injected into the model. Red line marks the location of the transect over which the temperature, salinity, and flow along the fjord are investigated. The magenta star denotes the node ca. 10 km from **GL-the grounding line** at which the vertical profile of temperature and salinity are shown. Blue and yellow lines, respectively, represent the transects across which overturning circulation and diagnostics for meltwater **trappings-convergence** are shown. (c) Discharge (Q_{sg}) scenarios, 1950–2100 CE, obtained by forcing MAR with five CMIP5 AOGCMs (Barthel et al., 2020; Slater et al., 2020). Shaded vertical stripes with blue and red stippled outlines denote the periods over which Q_{sg} -present (CE 1995–2014) and Q_{sg} -RCP 8.5 (CE 2081–2100), respectively, are averaged over (cf. Table 1). Blue, magenta, and red horizontal stippled lines correspond to the Q_{sg} -present ($372 \text{ m}^3/\text{s}$), Q_{sg} -median ($1563 \text{ m}^3/\text{s}$), and Q_{sg} -RCP 8.5 ($2754 \text{ m}^3/\text{s}$) discharge magnitudes, respectively. (d) A synoptic graphical summary of the results showing basal melt rate (m_b), and its drivers, namely, thermal driving (ΔT) and friction velocity (u^*). **Common to all plots:** The **total-length-of-bars (dark colors) show corresponding variable-averaged spatially-over the deeper (>300 m) draft regions of PGIS, while light-colored counterparts represent spatial averages over the entire PGIS base.** The discharge type categories (X-axis) correspond to the experiment design, cf. Table 1.

~~we resolve the~~ water column thickness, and furthermore, to implement a 540 – 610 m
deep inner sill situated ca. 25 km from the grounding line, retrieved using aerogravity
data from Operation IceBridge (Fig. E1; Tinto et al., 2015). Open boundaries in the
Lincoln Sea and Baffin Bay are used to provide lateral ocean boundary conditions. These
include temperature, salinity, and sea water velocities derived from A4 ROMS (4 km
pan-Arctic Regional Ocean Modeling System grid; Hattermann et al., 2016), and merged
sea surface elevation derived using A4 ROMS and 5 km AOTIM (Arctic Ocean Tidal
Inverse Model; Padman and Erofeeva, 2004). Data from the 5.5 km polar (p) version
of RACMO2.3p2 (Regional Atmospheric Climate Model; Noël et al., 2019) is used to
compute the momentum, heat, and freshwater fluxes. Sea ice concentration and thickness,
bulk ice salinity, and sea ice velocities are obtained from A4 ROMS – CICE (Community
Ice Code; Hunke et al., 2015). Using this setup, we perform a control run with zero
subglacial discharge, and three test runs designed to simulate the response of the cav-
ity circulation and ~~spatial melt rate variability arising due to the complex PGIS basal~~
~~morphology (Rignot and Steffen, 2008) in unprecedented detail. Our findings evidence~~
~~that Q_{sg} ice shelf basal melt rates to present day subglacial discharge, increased discharge~~
~~under high-emissions ‘RCP8.5’ climate warming scenario, and an intermediate discharge~~
~~scenario between the two (Table 1). We show that discharge increases basal melting by~~
(a) accelerating a quasi-estuarine circulation (Sciascia et al., 2013) which increases the
advective heat transport into the cavity, and by (b) enhancing the shear-driven mixing
across the ~~PGIS-ocean ice shelf-ocean~~ boundary layer. The latter exerts a dominant control
over the increase in ~~m_b seen for larger Q_{sg}~~ basal melting seen for higher discharge
under future warming scenarios while the former (advective heat transport) saturates,
suggesting a shift towards a regime where melt increases are primarily driven by increases
in friction velocity (u^*). ~~Within the basal channels, we find that the buoyant meltwater~~
~~accelerates along the steepest slopes of the ice shelf adjacent to the crests of the channels~~
~~which increases u^* and m_b , both of which peak at the crest. We estimate that the high~~
~~mean melt rates driven by summer Q_{sg} , if allowed to persist for a decade, may lead to~~
~~the complete erosion of the channels i.e., velocity scale representing shear stress in the~~
~~boundary layer). Our findings challenge the prevailing focus on increased oceanic heat~~
~~forcing as the primary driver of basal melting at the Petermann ice shelf, revealing that~~
~~heightened subglacial discharge significantly enhances (channelized) basal melt. This~~
~~suggests that atmospheric warming-induced excitation of oceanic melt processes will~~
~~critically influence the stability of the ice shelf.~~
Results
Response of cavity circulation and basal melt to contemporary discharge
~~Our mean modelled melt rates and their spatial variability, and furthermore, their increase~~
~~due to the introduction of contemporary summer mean Q_{sg} (Q_{sg} -present) into the PGIS-PF~~
~~system (e.f. Fig. 1(d), Fig. In the control run, Petermann ice shelf basal melt rates~~
~~largely range from a few m/yr near the calving front to ca. 40 – 60 m/yr under the deeper~~
~~(400 – 600 m) regions of the ice shelf base near the grounding line (Fig. 2(a,b) are~~
~~in agreement with)). Melt rates decreasing with distance from the grounding line is~~
~~consistent with results obtained from steady state ice discharge divergence approach~~
~~(Rignot and Steffen, 2008) and satellite remote sensing (Wilson et al., 2017). For the~~
~~period from 2011 – 2015, remotely sensed estimates of annual mean melt (reveal melt~~
~~rates of up to ca. 80 m/yr near the grounding line, although they largely range between~~
~~40 – 50 m/yr decreasing to ca. 10 m/yr under the shallower (<200 m) portions of~~
~~the ice shelf (Wilson et al., 2017) and in-situ conductivity-temperature and radar derived~~
 ~~estimates of discharge enhanced seasonal melt (Washam et al., 2020), and are comparable~~
 ~~to our modelled estimates.~~ With the ~~inclusion of introduction of contemporary summer~~
 ~~mean subglacial discharge (Q_{sg} -present at the GL) into the ice shelf-fjord system,~~ the
 modelled mean melt strengthens nearly throughout the domain and appears to be concen-
 trated along the basal channels that run along the length of the ice shelf (Rignot and Steffen, 2008
 ~~;~~ Fig. 1(d), Fig. 2(a,b)). ~~Strongest increases are seen~~ Spatial variability of our
 modelled melt on scales similar to the basal channel configuration is consistent with the
 findings of Rignot and Steffen (2008) and Wilson et al. (2017). Strongest increases of up
 to ca. 100 m/yr are seen within these channels under the deeper regions of the PGIS base
 ~~near the GL, and which gradually decrease toward,~~ which gradually decreases towards
 the calving front. This increase lies within the in-situ conductivity-temperature and radar
 derived estimates of discharge enhanced seasonal melt rate maxima of 40 — 170 m/yr
 (Washam et al., 2020). The summer mean basal melt rate (m_b), friction velocity (u^*),
 and thermal driving (ΔT) are averaged over (a) the entire PGIS ice shelf base, and (b)
 over parts of the PGIS ice shelf where draft is deeper than 300 m, and are indicated, re-
 spectively, by subscripts "pgis" and "300m" appended to the variable in question. Com-
 pared to the summer mean modelled melt rates in the control experiment, the contempo-
 rary discharge-driven increase in m_{bpgis} and m_{b300m} , respectively, is 41.2% and 68.1%,
 driven by similar increases in u^*_{pgis} (40.747%) and u^*_{300m} (64.167.4%) and modest in-
 creases in ΔT_{pgis} (1.61.1%) and ΔT_{300m} (5.64%) (Fig. 1(d)).

Figure 2. Response of **PGIS-ice shelf** basal melt rates and its drivers to Q_{sg} -discharge distributed uniformly across the **GL-grounding line** (Fig. 1(b)), for the experiments defined in Table 1. Summer mean basal melt rate (m_b) (a-d), friction velocity (u^*) (e-h), and thermal driving (ΔT) (i-l) for the control (first row), Q_{sg} -present relative to control (δ ; second row), Q_{sg} -median relative to control (δ ; third row), and Q_{sg} -RCP 8.5 relative to control (δ ; last row) experiments. Contours of the ice shelf draft are plotted at 100 m intervals (450 m – 150 m; panel(f)) and overlaid as dotted black lines on each panel.

The presence of Q_{sg} at the **GL-subglacial discharge at the grounding line** enhances the
 fjord-scale overturning circulation (Fig. 3(i)). We observe a more robust Atlantic Water
 inflow at depth, extending into the inner fjord basin and up to the **GL-grounding line**

(Fig. 3(c,f)). Hereafter, we refer to this process as the "thermal pump", which promotes the entrainment of heat and salt deep into the **PGIS-ice shelf** cavity (columns 1 and 2 in Fig. 3). We note an increase in bottom temperature of 0.03°C at a location ca. 10 km from the **GL-grounding line**, and a temperature excess ($\Delta T_{\text{thermal driving}}$) of 0.1°C under the ice shelf base at 300 m depth (Fig. 2(j); Fig. 3(g)). A higher $\Delta T_{\text{near the GL}}$ **thermal driving near the grounding line** (Fig. 2(i,j)) drives more melting at depth. A more energetic outflow/strengthened meltwater plume dynamics, driven by the introduction of Q_{sg} -**discharge**, exacerbates the shear-driven turbulent mixing and heat transfer across the **PGIS-ocean-ice shelf-ocean** interface (Fig. 2(e,f)), thus driving more melt. The spatial variability of u^* -**friction velocity** anomalies is consistent with the corresponding melt rate anomalies (cf. panels b and f in Fig. 2). Regions exhibiting strong increase in u^* -**friction velocity** are concentrated along the longitudinal basal channels near the **GL-grounding line**, and gradually decrease downstream of it. However, a higher meltwater production (Fig. 2(b)) and stronger outflow (Fig. 3(f,i)) deposits larger volumes of meltwater from the deeper to the shallower regions of the **PGIS-ice shelf** base, considerably lowering the $\Delta T_{\text{thermal driving}}$ locally (Fig. 2(j); Fig. 3(d,e)). Thus, while the increase in u^*_{pgis} and u^*_{300m} is significant, the contribution of $\Delta T_{\text{thermal driving}}$ is largely restricted to the deeper regions of the **PGIS-ice shelf** draft (ΔT_{300m}) (Fig. 1(d)). **This implies that for subglacial discharge equivalent to present-day summer mean magnitudes, increase in basal melt under the deeper (>300 m) drafts is driven by increases in both thermal driving (increased thermal pumping) and friction velocity (increased shear-driven mixing).**

Heightened discharge and saturation of the thermal pump
When transitioning from the control experiment to RCP 8.5 (Q_{sg} -RCP 8.5), and an inter-
 mediate scenario between the two (Q_{sg} -median) (Table 1), a higher modelled Q_{sg} -**discharge**,
 as expected in a future warmer atmosphere (Aschwanden et al., 2019; Golledge et al.,
 2019), results in considerably higher increases in basal melt as compared to Q_{sg} -present
 (Fig. 1(d); Fig. 2(a–d)). In particular, an increase in Q_{sg} -**discharge** from control to RCP
 8.5 results in more than twofold (168.4%) and threefold (217.2%) increase in m_{bpgis}
 and m_{b300m} , respectively. These increases are primarily driven by increases in u^*_{pgis}
 (**165.5191.8%**) and u^*_{300m} (**205.7216.9%**), whereas ΔT_{pgis} and ΔT_{300m} show rela-
 tively nominal increases of **5.1% and 9.82.9% and 6.5%**, respectively. For the median
 experiment, **3.8% and 8.62.3% and 5.9%** increase in ΔT_{pgis} and ΔT_{300m} , and **114% and**
 **154.7132.5% and 163.4%** increase in u^*_{pgis} and u^*_{300m} generate 116.3% and 163.6% in-
 crease in m_{bpgis} and m_{b300m} , respectively. As noted for Q_{sg} -present, strongest increases
 occur under the deeper regions of the **PGIS-ice shelf** base which gradually decreases
 downstream (Fig. 2 (c,d)), however, channelized melting is seen to strengthen substan-
 tially (discussed further below).
Combining insights from all the experiments provides a novel perspective on how
 Q_{sg} -**discharge** dictates the interplay between **PGIS-Petermann ice shelf** basal melt and
 its drivers: Increase in the heat entrained in the **PGIS-ice shelf** cavity as a result of the
 thermal pump **peaks is strongest** at the inclusion of present-day summer mean estimate
 of Q_{sg} -**subglacial discharge**, with much of the heat delivered by the thermal pump con-
 centrated under the deeper **PGIS-ice shelf** base in the inner fjord basin (Fig. 1(d); Fig.
 3(d,g)). At a summer mean Q_{sg} -**discharge** value which is approximately fourfold higher
 than the present (i.e. Q_{sg} -median vs. Q_{sg} -present), **the contribution of the thermal pump**
 **is relatively weaker thermal driving increases at a lower rate** (Fig. 1(d); Fig. 3(g); cf.
 Fig. 3(d) and Fig. C1(a)). We note an increase of **2.91.9%** in the Q_{sg} -median ΔT_{300m}
 as compared to Q_{sg} -present (Fig. 1(d)). Moreover, a larger meltwater deposition further
 lowers the $\Delta T_{\text{thermal driving}}$ under the shallower drafts (Fig. 2 (b,c,j,k); Fig. 3(i);
Fig. C1(a,b,c)) and we do not observe a noticeable which results in a relatively weaker
 increase in ΔT_{pgis} (Fig. 1(d)). Lastly, in the RCP 8.5 scenario, Q_{sg} discharge is nearly
 doubled as compared to the Q_{sg} -median experiment, however, the efficiency of the thermal
 pump in increasing ~~m_b~~ basal melting saturates (Fig. 3(g); Fig. C1). Here, increase
 increases in ΔT_{pgis} ~~is and~~ ΔT_{300m} are indiscernible as compared to the Q_{sg} -median
 experiment, ~~whereas ΔT_{300m} increases negligibly~~ (Fig. 1(d)). Instead, we find that
 Q_{sg} ~~under a discharge under the~~ high-emissions RCP 8.5 scenario enhances melting by
 further accelerating the cavity circulation, wherein, 24.1% and 20.4% increase in m_{bpgis}
 and m_{b300m} are seen, driven predominantly by ~~24% and 20~~ 25.5% and 20.3% increase in
 u^*_{pgis} and u^*_{300m} , respectively.

Figure 3. Along-fjord summer mean temperature (a), salinity (b), and flow (c) for the control experiment for the section shown in Fig. 1(b). Mean isopycnals (solid black lines) are overlaid at equal intervals of 0.2 kg/m^3 . The 27.9 kg/m^3 isopycnal corresponds to the dense Atlantic Water that contacts the grounding line. For the same section, changes (δ) in summer mean temperature (d), salinity (e), and flow (f) for Q_{sg} -present relative to control are shown. Black stippled lines in panels d and e indicate the zero temperature and salinity difference contour, respectively. In panel f, solid black lines and stippled green lines depict the mean isopycnals for the control and Q_{sg} -present experiments, respectively, plotted at equal intervals of 0.2 kg/m^3 . In each panel, PGIS-GL Petermann ice shelf grounding line (at \sim ca. 600 m depth) is on the right margin and open ocean is to the left. Vertical magenta line in panels a, b, d, and e show the location of the model node ca. 10 km from the GL grounding line (Fig. 1(b)) at which the vertical profile of summer mean temperature (g) and salinity (h) are shown for the control, Q_{sg} -present, Q_{sg} -median, and Q_{sg} -RCP 8.5 experiments (cf. Table 1). (i) Laterally integrated summer mean vertical overturning profile [in Sv] for all the experiments (cf. Table 1) for the section shown in Fig. 1(b). Positive [Sv] values correspond to (laterally integrated) inflow and negative [Sv] values correspond to (laterally integrated) outflow across the section. Calculation of transport diagnostics follows from Prakash et al. (2023).

In sum, our experiments reveal a regime change in heat flux efficiency within the ice
shelf cavity. In the control experiment, the ice shelf-ocean boundary exchange consumes
 a considerable amount of heat from the ocean that is available for melting the ice. As
 such, the latent heat loss and meltwater release cools the ambient ocean inside the cavity.
 In such a scenario, an increased fjord-scale overturning circulation due to increased
 discharge (e.g., $< Q_{sg}$ -median experiment) entrains more heat in the cavity (thermal
 pump) which increases melting through enhanced thermal driving, in addition to the
 enhanced shear-driven turbulent heat exchange. This saturates at some point, when more
 discharge (e.g., $> Q_{sg}$ -median experiment, as seen for the RCP 8.5 scenario) accelerates
 the overturning circulation such that there is always enough heat coming into the cavity
 to maintain a high thermal driving (i.e., heat supply at an elevated level is maintained).
 Here, melt rates become entirely a function of the friction velocity.
**Trapping Convergence** of meltwater and channel stability

Figure 4. Maps of the summer mean surface layer salinity (a,b) and basal melt rate (c,d) for the control (column 1) and Q_{sg} -RCP 8.5 relative to control (δ ; column 2) experiments. Contours of the ice shelf draft are plotted at 100 m intervals (450 m – 150 m) and overlaid as dotted red and blue lines over the surface layer salinity and melt rate maps, respectively.

We use surface layer salinity as a proxy for tracking the spatial distribution of meltwater
 under the deeper regions of PGIS the ice shelf (Fig. 4; Fig. D1). Freshening occurs nearly
 throughout the domain in the Q_{sg} -driven discharge-driven experiments as compared to
 the control run. Furthermore, regions of substantial freshening align with four prominent
 ice shelf basal channels (Fig. 4 (a,b); 5 (a,b); Fig. B1) along which the draft thins from
 300 – 450 m ca. 5 km from the GL grounding line to 150 m ca. 25 – 30 km seaward
 of it, suggesting a topographically steered horizontal pathway for the meltwater outflow
 that is determined by the PGIS-ice shelf basal morphology and which in turn influences
 it (discussed further below). Changes appear to be most profound within these channels,
 particularly, at the erestcrests, where salinity decreases by up to 1.06 psu in the RCP 8.5
 experiment with respect to the control run (Fig. 4; Fig. 5(a,b)). These locations where
meltwater is trapped of meltwater convergence are concordant with locations of largest
 melt rate increases (discussed further below) of up to 181.5 m/yr (a sevenfold increase
 compared to the control run). ΔT is higher. Thermal driving is not higher at the channel
 crests where meltwater converges, but under the deeper drafts, which is consistent with
 the pressure dependent melting point of seawater. Although ΔT (Fig. 5(c)). Although
 thermal driving may not increase the most at the channel crests, we observe increases
 ranging from 0.06 to 0.19 0.02 to 0.12 °C in the RCP 8.5 experiment compared to the
 control run across the transverse cross-section near the GL grounding line (Fig. 5(c)).
 We find that u^* profiles the friction velocity profile across this section are is (nearly)
 congruent with the m_b profiles, and exhibit largest basal melt rate profile, and exhibits
 the highest increases at the channel crests of up to 0.03 m/s (Fig. 5(d)).

Figure 5. Maps of the summer mean surface layer salinity (a,b) and basal melt rate (c) Cross-section profile
of the Petermann ice shelf draft, derived from BedMachine v3 (Morlighem et al., 2017), for the control
transect ca. 5 km from the grounding line shown in Fig. 1 (column 1b) and Q_{sg} -RCP 8.5 relative to control
(column 2) experiments. Contours Apex locations of four prominent longitudinal ice shelf basal channels
at this section are highlighted using green stripes along which the Petermann ice shelf draft are plotted at
100-m intervals (reduces from 300 – 450 m – at the transect location to 150 m) and overlaid as dotted red
and blue lines over a distance of 25 – 30 km from grounding line (Fig. B1). Across the same section, the
summer mean surface layer salinity and melt rate maps (m_b) (b), respectively thermal driving (ΔT) and melt
rate (c), and friction velocity (u^*) and melt rate (d) are shown for the control and Q_{sg} -RCP 8.5 experiments.

Three ice shelf basal channel sections are outlined (Fig. 6(a)), whose lengths are
 demarcated based on the apex of the 150 m ice shelf draft contour (e.g. Fig. 4(a); Table
 2). The channel averaged annual mean basal melt rates for the western (S_w), central
 (S_c), and eastern (S_e) sections for the RCP 8.5 experiment are calculated as 45 m/yr,
 51 m/yr, and 47 m/yr, respectively. The enhanced Enhanced
 melting at the crests will promote channel opening and we can calculate the changes in channel averaged annual
 mean ice volume above each channel over time to estimate the incision/erosion timescale
 for each that channel under the respective basal melt rates (Eqn. 5, 6). To bracket the
role of ice advection from upstream of the GL grounding line, three different rates (AR_c)
 are considered for each channel which correspond to the present contemporary mean (1
 242 km/yr; Münchow et al. (2014)), and 50% lower (0.5 km/yr) and higher (1.5 km/yr) than
 243 present contemporary mean values. While S_w exhibits the lowest mean basal melt rate, it
 also features the shallowest mean draft value. Notably, its erosion-incision timescales are
 comparable to those of S_c , which has the highest mean melt rate but a relatively deeper
 mean draft compared to S_w . The number of years (N) required for the mean channel-ice
 volume-ice volume above a channel to become zero is estimated to be 5.88, 6.52, and 7.31
 248 years for S_w , in contrast to 5.95, 6.75, and 7.8 years for S_c , at AR_c of 0.5 km/yr, 1 km/yr,
 and 1.5 km/yr, respectively (Fig. 6(b)). It has been shown that the viscous ice response
 to a basal channel is insufficient to significantly delay the breakthrough of basal channels
 in ice shelves that are thinner than 400 m (Wearing et al., 2021). With a lower mean melt
 rate than S_c and the deepest mean draft value, S_e demonstrates the greatest resilience. We
 estimate N to be 8.06, 9.41, and 11.31 years for AR_c of 0.5 km/yr, 1 km/yr, and 1.5 km/yr,
 respectively (Fig. 6(b)). These estimates suggest that ice advection from upstream of the
 PGIS-GL grounding line (alone) may not be sufficient to compensate for its thinning the
 thinning of the ice shelf due to intense channelized basal melting that may be expected
 under the RCP 8.5 scenario (discussed further below).

(a) Cross-section profile of the PGIS draft for the transect ca. 5 km from the GL shown in Fig. 1(b). Four prominent longitudinal (L) ice shelf basal channels are represented using green stripes as L_w (west), L_c (central), $L_e(1)$ (east-1) and $L_e(2)$ (east-2) along which the PGIS draft reduces from 300–450 m at the transect location to 150 m over a distance of 25–30 km from GL (see Fig. B1). For clarity, $L_e(1)$ and $L_e(2)$ are collectively classified as L_e (east) in panels b, c, and d. Across the same section, the summer mean surface layer salinity and melt rate (m_b) (b), thermal driving (ΔT) and m_b (c), and friction velocity (u^*) and m_b (d) are shown for the control and Q_{sg} -RCP 8.5 experiments.

Figure 6. (a) Map of the PGIS Petermann ice shelf draft overlaid with channel sections along the western (S_w ; blue), central (S_c ; black), and eastern (S_e ; red) regions which are used in calculating the changes in respective channel averaged annual mean ice volume above the channel over time in panel (b) for the RCP 8.5 experiment following Eqns. 5 and 6 (Table 2). Apex locations of four prominent longitudinal ice shelf basal channels at the across-fjord transect shown in Fig. 5 are represented using magenta diamonds. For each color-coded channel section shown in (a), channel-corresponding changes in mean ice volume changes above the channel section are shown in (b) are shown for three different ice advection rates, namely, 0.5 km/yr (asterisk), 1 km/yr (solid line) and 1.5 km/yr (circle). Channel averaged annual mean basal melt rates for S_w , S_c , and S_e are, respectively, 45 m/yr, 51 m/yr, and 47 m/yr.

**Discussion**
We note that our model is forced with contemporary summer mean Q_{sg} -estimates discharge
estimate and expected increases of it under a future warming atmosphere. The present-
261 day estimates of maximum Q_{sg} -discharge (e.g. Cai et al., 2017) are two to fourfold
higher than the present-day summer mean estimate of Q_{sg} -discharge used in this study,
and are even comparable to the Q_{sg} -median experiment. Therefore, while Q_{sg} -median
represents an intermediate state between present and extreme future mean scenario in
our study, it is plausible that such increases in Q_{sg} -discharge, and thus basal melt, are
being experienced by **PGIS-the Petermann ice shelf** for brief periods during summer,
with varying degrees of inter-annual variability. ~~Modelled study from~~ We posit that
during the summer (JJA) months, if for e.g., exceptionally high discharge is sustained
for a considerable (e.g., several weeks) period of time, it may temporarily nudge the ice
shelf-fjord system into a regime where basal melting is largely driven by friction velocity.
We note that as opposed to Gaussian-like patterns (Ehrenfeucht et al., 2023), our applied
discharge forcing follows a boxcar function. Thus, studies investigating the sensitivity
of the system to seasonal and inter-annual variability in subglacial discharge, derived for
e.g., from a subglacial hydrological model, need to be conducted to ascertain and quantify
any resulting transitions that may occur in the mechanisms driving basal melting. The
modelling work of Hill et al. (2018) has demonstrated that the loss of thicker and rhe-
ologically stiffer sections of **PGIS-the Petermann ice shelf** within 12 km from the **GL**
**grounding line** (regions where draft ≥ 300 m) could disrupt **GL-grounding line** stresses
sufficiently to greatly accelerate interior ice flow and increase dynamic ice discharge, sig-
nificantly contributing to global mean sea level rise. We thus suggest that the Q_{sg} -driven
discharge-driven intensified basal melting of the deeper (draft ≥ 300 m) regions near the
**GL-grounding line** that are dynamically resilient (Fig. 1(d); Fig. 2) will have a significant
bearing on the future stability of **PGthe Petermann Glacier**, as their loss could trigger an
unopposed acceleration of the inland ice.
Thus far, basal melting of **PGIS-the Petermann ice shelf** has been largely investigated
from the perspective of (increased) oceanic heat delivery to **PFPetermann Fjord**. How-
ever, both observational and modelled estimates concur on the abundance of heat supply
into the fjord, i.e. heat availability is in excess of what is needed to generate contempo-
rary estimates of basal melt (Johnson et al., 2011; Washam et al., 2018; Prakash et al.,
2023). Our findings show that increased basal melting of **PGIS-the Petermann ice shelf** in
a warming climate is not limited to increase in oceanic heat forcing, and underscores the
importance of atmospheric warming in controlling the basal melting of Northern **GrIS**
**Greenland Ice Sheet** glaciers (Slater and Straneo, 2022). Uncovering how this unfolds
from the **PGIS-ocean-an ice shelf-ocean** interaction perspective, we find that the effect
of the thermal pump on melt is the strongest when Q_{sg} -discharge is introduced into the
system (as Q_{sg} -present; Table 1) and saturates for Q_{sg} -discharge $> Q_{sg}$ -median (Fig.
1(d); Fig. 3; Fig. C1), wherein, Q_{sg} -median is equivalent to present-day estimates of
maximum Q_{sg} -discharge (c.f. Cai et al. (2017)). We see an increase in bottom temper-
ature near the **GL-grounding line** (increased heat advection towards the interior of the
**PGIS-ice shelf** cavity via the thermal pump) as Q_{sg} -discharge is increased from control
to present to median, but less so for substantially higher discharge under the RCP 8.5
scenario (Fig. 3(g)). Therefore, we posit that a regime shift of the driving mechanism
behind m_b -**Petermann ice shelf basal melting** may occur when the catchment integrated
meltwater runoff under (extreme) future warming scenarios exceeds the discharge thresh-
old of present-day Q_{sg} -maximummaximum. In such a scenario, the significance of the
thermal pump in driving increased melting ebbs off, and the higher discharge being re-
leased into the fjord accelerates the sub-ice shelf currents, which nudges the system to-
wards a regime where increase in basal melt is predominantly dictated by substantially
higher shear-driven turbulence which increases the heat fluxes at the ~~PGIS-ocean-ice~~
~~shelf-ocean~~ interface (Fig. 2). ~~We Millan et al. (2023) show that Atlantic Water reaching~~
~~the fjord in 2020 are 0.3°C warmer as compared to 1970, of which a warming of 0.2°C~~
~~has occurred between 2000 and 2020. Further modelling work is required to determine~~
~~the characteristics of Atlantic Water reaching the fjord by the end of this century under~~
~~different climate warming scenarios. While we do not take into consideration the impact~~
~~of rising ocean temperatures in our study, we acknowledge that realistically, in a future~~
~~warming climate, rising ocean temperatures warmer climate, it would likely act in concert~~
~~with the enhanced Q_{sg} increased discharge to further amplify m_b basal melting.~~ How-
ever, with a much lower heat capacity as compared to the ocean, increase in surface melt
over grounded ice in response to atmospheric warming (and thus increase in subglacial
discharge) is highly likely to outpace the warming of the deep ocean water masses that
(alone) would be required to bring about such substantial increases in m_b basal melting
in the upcoming centuries. ~~Thus, it is reasonable to posit that the atmospheric impact~~
~~on m_b is~~ (e.g., increases of up to ca. 200 m/yr in the RCP 8.5 scenario) as modelled
here. Thus, while the impact of increase in ocean heat forcing and its interplay with
increased discharge on basal melting in a future warming climate needs to be examined,
discharge-driven basal melting is likely to pose a more immediate and pertinent threat to
the glacier's long-term stability.
~~Across all Q_{sg}~~ Seasonal changes including, but not limited to, an increase in fjord
temperature of ca. 0.1°C during summer as compared to winter (Prakash et al., 2023)
drives up to ca. 47% increase in mean channelized (sections S_w , S_c , and S_e in Fig. 6(a);
Table 2) basal melting. Without taking into consideration the compensating viscous ice
response (discussed further below), and for contemporary mean ice advection rate from
upstream of the grounding line of 1 km/yr, this increase results in up to ca. three-fold
increase in erosion rates. The strongest seasonal influence on melt is imposed by the
addition of subglacial discharge into the fjord. Compared to winter (no discharge), we
calculate that contemporary summer mean discharge increases mean melting in the channels
by up to ca. three-fold, yielding up to an order of magnitude increase in erosion rates.
We note that although S_w and S_c may exhibit higher rates of erosion as compared to S_e ,
highest increases in erosion rates are modelled for S_e .
~~Across all discharge~~ experiments (Table 1), we consistently find channelized strength-
ening of basal melt (Fig. 2(a-d)), imposed by the complex ~~PGIS-ice shelf~~ basal morphol-
ogy (Rignot and Steffen, 2008; discussed further below). Within these channels (e.g., ~~L_w ,~~
~~L_{S_w} , S_c , and L_{S_e}~~ in Fig. 5), 6(a)), compared to the control experiment, larger volumes
of buoyant meltwater rise along the steepest slopes of the ice base towards the channel's
crests (Fig. 5), from where the stronger buoyancy-driven overturning circulation in the
fjord carries it further downstream along the longitudinal channel axis (Fig. 3(i); Fig. 4;
Fig. B1; Fig. D1). The steeper ice shelf basal slopes that characterize these channels
support stronger entrainment of the ambient Atlantic Water into the buoyant meltwater
plume (Wilson et al., 2017). Furthermore, the vigorous confluence of meltwater towards
the ~~crest increases u^* which further amplifies melt. crests, and its downstream advection,~~
~~increases the friction velocity which predominantly drives the increased melting~~ (Fig. 2;
Fig. 5(b,d)). The additional buoyancy forcing provided by ~~Q_{sg} under future warming~~
~~scenarios increased subglacial discharge in a warmer future climate, in particular, in as~~
~~modelled here for~~ the RCP 8.5 experiment, ~~acts will act~~ to enhance these mechanisms
(Fig. 5(b,d)) which may advance the vertical growth of the channel. To that end, our
~~channel erosion estimates~~ estimates of mean ice volume changes above the channels in-
dicates that sustained high mean melt rates due to enhanced summer Q_{sg} discharge, as seen
in the RCP 8.5 experiment, could undermine the structural integrity of ~~PGIS~~ the ice shelf
(Fig. 6). Persistence of such intensified melting over durations ranging from ca. 6 to
11 years may result in complete ~~erosion-incision~~ of the channels, leading to a possible
disintegration of the ice shelf. While the accuracy of this estimate depends on the un-
certainty of the applied melt rate parameterization, e.g., not accounting for the effects of
enhanced stratification due to the meltwater discharge (Rosevear et al., 2022; Davis et al.,
2023), the simulated acceleration of the cavity circulation is a robust feature of our sim-
ulations, which, together with observational evidence (Washam et al., 2020) indicates
significant impacts on the basal mass loss of ~~PGIS~~ the Petermann ice shelf. We also
note that our static ice shelf geometry does not account for the ice dynamical processes,
for e.g., downstream advection and calving, and viscous ice response which would act
to (partially) offset melting. While it has been shown that the secondary ice-flow is not
strong enough to substantially hinder the development of basal channels in ice shelves that
are less than 400 m thick (Wearing et al., 2021), we observe that the Petermann ice shelf
draft is deeper than 400 m in certain sectors (e.g., southeastern sector near the grounding
line). However, in the absence of a high-resolution (at the scale of the topographical data)
ice-flow model, it is difficult to ascertain how small-scale (sub-km) localized channel
closures/openings would impact the wider ice dynamics, and as such, the long-term
stability of the glacier. We suggest that future work with coupled ice-flow - ocean models
(frameworks of which have been recently developed, e.g., Zhou et al. (2024)) or offline
coupling of an ice-flow model with comprehensively resolved ice shelf basal melt rates
derived from sophisticated regional ocean models is needed to improve our understanding
of the nuanced dynamical variability that occur within these channels.
Our results could be extended to other 'warm cavity' environments where pronounced
increases in Q_{sg} discharge are expected, and the ice shelf is characterized by complex
basal morphology. We suggest that such conditions could be met for other Northern
~~GrIS~~ Greenland Ice Sheet glaciers that terminate as floating ice shelves, namely the Ry-
der Glacier (~~RG~~) and Nioghalvfjærdsbræ (79N) Glacier. The ice shelves of both ~~RG~~ Ryder
Glacier and 79N also feature channelized geometry and thus exhibit notable spatial melt
rate variability, as well as a dependency of basal melt on slope that is qualitatively similar
to ~~PGIS~~ the Petermann ice shelf (Wilson et al., 2017; Prakash et al., 2023). The lower
rates of ice advection from upstream ~~make RG of the grounding line~~ make Ryder Glacier
more prone to developing deep channels. At 79N, it has been suggested that the recent
expansion of its central basal channel is a consequence of heightened Q_{sg} discharge, at-
tributed to a significantly expanded surface melt area during summer, brought about by
atmospheric warming (Narkevic et al., 2023; Zeising et al., 2023). Moreover, the exten-
sive network of moulins observed along the ~~GrIS~~ Greenland Ice Sheet margins in summer
(Turton et al., 2021; Ehrenfeucht et al., 2023; Rawlins et al., 2023), coupled with expected
increases in surface melt ~~on the GrIS over the ice sheet~~ in a warmer climate (Aschwan-
den et al., 2019; Muntjewerf et al., 2020), are likely to yield end-of-century discharge
estimates at these sites that are comparable to those of Petermann (see projected runoff
for the northern, northeastern and northwestern ~~GrIS sectors~~ sectors of the ice sheet in
Slater et al. (2020)).
Analogous to Greenland, the intricate basal morphology of Antarctic ice shelves,
sculpted by their interactions with the bed and/or melting, also exhibits a diverse range
of features (e.g., channels, keels, crevasses, and terraces) that exert a significant in-
fluence on their melting and structural integrity, which is contingent upon the slope
(Drews et al., 2017; Schmidt et al., 2023). ~~While the~~ The role of an additional buoy-
ancy source that is injected basally into the fjord on enhancing basal melting and retreat
has been extensively investigated in Greenlandic geophysical settings (Sciascia et al.,
2013; Rignot et al., 2015; Cai et al., 2017; Cook et al., 2020), ~~in Antarctica, insights into~~
~~this process are limited to tacit reasoning~~ (Le Brocq et al., 2013; Lepp et al., 2022) ~~and~~
~~remain largely unaccounted for in modelling efforts that aim to constrain future sea-level~~
~~projections.~~ ~~Nonetheless, recent studies~~ and has garnered recent attention in Antarctica
(Goldberg et al., 2023; Gwyther et al., 2023; Pelle et al., 2024). Studies have shown that
Q_{sg} subglacial discharge has significantly influenced the mass balance of the Antarctic
Ice Sheet during the last deglaciation period and is expected to maintain its importance
in a warming climate (Li et al., 2023; Pelle et al., 2023). In addition, ice shelf collapses
due to hydrofracturing and flexurally-induced fractures resulting from supraglacial lake
drainage events have also been noted (Scambos et al., 2009; Banwell and MacAyeal,
2015; Dow et al., 2018). However, unlike Greenland (Forster et al., 2014; Willis et al.,
2015; Otto et al., 2022) there is no evidence of surface meltwater infiltrating to the bed.
Greenland's surface and subglacial hydrological systems that are connected by drainage
pathways offer a blueprint for understanding how Antarctica might evolve in a warmer
world, as its surface hydrology starts to resemble present conditions observed at ~~GIS~~ the
Greenland Ice Sheet (Bell et al., 2018). Projected increases in surface melt rates over
Antarctica (Noël et al., 2023) suggest a shift towards a more extensive and active surface
hydrology, one that could effectively link the presently isolated bed via seasonal influx
of surface meltwater. We note that such scenarios are dependent upon future emission
pathways and whether the ambitious targets proposed by the Paris Climate Agreement
are upheld (DeConto et al., 2021). However, if these targets are exceeded, as is expected
under the RCP 8.5 projection, we posit that a fundamental change in Antarctica's dy-
namics and the processes controlling it cannot be precluded. In particular, energetic
currents driven by intensified discharge in a cavity characterized by complex channelized
basal morphology, as modelled here, will accelerate the heat flow towards the ice shelf
base. The substantial shear-driven turbulence could independently drive increased melt-
ing, and may even be sufficient to abrade the stable boundary layer stratification that has
been observed in critical regions where ~~m_b~~ basal melting remains hitherto suppressed
due to latent ocean environments (Davis et al., 2023).
Ice shelf basal topography remains largely represented in numerical models in an
idealised tapered form, that is either completely flat or has some smooth curvature that
is imparted as a function of along-fjord slope. Moreover, the sub-ice shelf bathymetry
employed in these models is derived using rudimentary techniques that attempt to inter-
polate between known depths from ~~the continental shelf/fjord mouth to the GL~~ near the
fjord mouth and the grounding line. These critical shortcomings are addressed in our
model setup (Prakash et al., 2022), however, efforts to mitigate such topographic (both
seafloor and ice shelf basal ~~)~~ caveats topography caveats in numerical modelling efforts
that aim to investigate real-world locations, despite its well acknowledged impetus on
~~m_b~~ basal melting, are still not being widely undertaken. Indeed, an unknown sub-ice
shelf bathymetry ~~at PGIS and~~ for a vast majority of ice shelf-fjord systems is a limiting
factor. As such, wherever possible, more targeted airborne and ground-based geophysical
campaigns must be conducted to describe these environments. While our findings em-
phasize the need for implementing realistic ice shelf geometries, it is noteworthy to high-
light the limitations of the available resolutions (typically hundred to several-hundred
meters; e.g. Morlighem et al. (2017); Frémand et al., 2023) which remain relatively
coarse to capture the small-scale basal features, on the order of several (tens of) me-
ters as revealed by in-situ under-ice shelf observations (Davis et al., 2023; Schmidt et al.,
2023), that play a crucial role in dictating the spatial ~~variability in m_b~~ melt rate variability.
To comprehensively understand the intricate interplay between Q_{sg} subglacial discharge,
basal morphology, and melt processes, we emphasize that a more extensive dataset of
such measurements is imperative ~~which then needs to be integrated into~~. To enhance
the reliability of continent-scale projections, observational findings from sub-ice shelf
environments – including near the grounding zones – must be combined with insights
from non-idealized high-resolution ice shelf-ocean models numerical modelling studies
driven by these datasets. Improved understanding of critical small-scale processes at
the marine-ice margins should then inform the refinement of existing parameterizations
and the development of new ones, which would facilitate large-scale models to better
capture the influence of dynamics at the meter-scale. We note that in our experiments,
Q_{sg} discharge is distributed uniformly across the GL-grounding line width which may
not be consistent with ground truth. For an idealized tidewater glacier, modelling work
has shown that distributed discharge generates more melt than channelized discharge
(Slater et al., 2015). With glacier hydrology exhibiting increased dynamism, which is
likely to continue into the future, a concerted effort is warranted. This necessitates de-
riving insights into the interplay between moulin locations/density, meltwater routing,
and eventually its magnitude and emergence location(s) at the GL-grounding line from
observational evidence and sophisticated subglacial hydrological modelling tools. ~~These~~
~~insights, which~~ should then be incorporated into numerical models of ice shelf-ocean in-
teractions, ~~to better predict the evolving dynamics of glacier-ocean systems in a warming~~
~~climate.~~ Furthermore, investigations of the interaction between subglacial discharge and
tidally-modulated seawater intrusions in the grounding zone (Ciraci et al., 2023) will be
essential in improving our understanding of the dynamics and stability of the glacier.
**Methods**
**The Finite Volume Community Ocean Model (FVCOM) setup**
The unstructured grid, free-surface, 3-D primitive equation Finite-Volume Community
Ocean Model (FVCOM) (Chen et al., 2007), augmented by modules for an ice shelf
(Zhou and Hattermann, 2020) and sea ice (*Ice Nudge*) (Prakash et al., 2022), is used. The
model domain is centered on PF Petermann Fjord, covering the region between 75°N -
87°N and 29°W - 81°W (Fig. 1a). The grid comprises of unstructured triangle elements,
featuring a total of 112709 nodes and 219836 cells, including 309 open boundary nodes
in the Lincoln Sea and Baffin Bay. The topological flexibility of the unstructured grid
allows accurate fitting of the complex irregular coastal geometry along the northern GIS
Greenland Ice Sheet margin. In particular, the grid is partitioned into polygons of varying
horizontal resolution: A 200-m resolution is provided in the PF Petermann Fjord region
to resolve the steep slopes characteristic of the fjord seafloor and PGIS-ice shelf basal
topography, which is relaxed to 2 km in the adjacent areas outside the fjord, and further
to 4 km towards the open ocean boundary regions. For an extensive regional model setup,
such a configuration enables us to strike a balance between computational efficiency and
the need for accurate representation of topographical features and their variation over
small spatial scales.
PGIS-basal topography Petermann ice shelf draft derived from the BedMachine v3
(Morlighem et al., 2017) dataset (150-m resolution) represents a pre-2010 PGIS-calv-
ing geometry (Münchow et al., 2014), and is interpolated to the 200 m FVCOM model
grid (Prakash et al., 2022). As such, the basal channels (Rignot and Steffen, 2008) are
resolved at 200 m scale in our model, which is sufficient to represent their 1–2 km width
(Rignot and Steffen, 2008). Bathymetry outside PF Petermann Fjord comes from the 1-
503 km gridded IBCAO v3.0 dataset (Jakobsson et al., 2012), whereas inside it, the Bed-

Machine v3 dataset is used. In the vertical, the domain between the irregular seafloor topography and the ice shelf draft (or the open ocean surface) is discretized into 23 terrain-following σ -layers. To alleviate errors in the discretization of horizontal pressure gradients along the sloping σ -layers, smoothing of the bathymetry is applied as a function of the mesh resolution before its implementation in the model (Prakash et al., 2022). Further, the poorly constrained Without seafloor depth measurements beneath the ice shelf, the BedMachine bathymetry is inferred by extending the known seafloor depth from seaward of the calving front landward towards the grounding line and aligning it with the depth of the ice shelf draft at the grounding line using natural neighbor interpolation. For Petermann Fjord, Prakash et al. (2022) show that this leads to implausible abrupt changes in the water column thickness along either flanks and. Specifically, the BedMachine bathymetry underestimates the water column thickness under the deeper regions of PGIS is improved, for e.g., by implementing a ca. 600 m deepinner sill (Tinto et al., 2015) situated approximately (>200 m) regions of the ice shelf, with sub-100 m values along the flanks and in the central zone (Fig. E1(a-c)). More critically, this misrepresentation results in the formation of an artificial inner sill, which, if not accounted for, would block the inflow of the warm and dense (27.9 kg/m^3) Atlantic Water at depth (Fig. E1(d)) that contacts the grounding line (Washam et al., 2018; Prakash et al., 2023). Therefore, addressing these artefacts is imperative to ensuring that the model is able to reproduce the fundamental aspects of glacial fjord circulation (Sciascia et al., 2013). Tinto et al. (2015) used aerogravity data from Operation IceBridge to model the bathymetry underneath the ice shelf. Importantly, their findings revealed the presence of an inner sill, ca. 540 – 610 m deep, located about 25 km from the GL which is not included in the BedMachine v3 dataset (Prakash et al., 2022). Initial and lateral ocean boundary conditions, and atmospheric and sea ice surface boundary conditions are derived from a suite of high-resolution regional (ocean, tidal, atmospheric and sea ice) models (downscaling procedure is detailed in Prakash et al. (2022)) grounding line. In this study, we implement the smoothed bathymetry product generated by Prakash et al. (2022), wherein, the BedMachine bathymetry has been modified to account for the unrealistic sub-ice shelf water column thickness and further integrated with the aerogravity data from Tinto et al. (2015).

Experiment Design

Table 1. The ensemble averaged June-July-August (JJA) mean discharge (Q_{sg}) magnitude for the Petermann catchment [in m^3/s] that is distributed uniformly across the GL grounding line for present conditions (1995-2014), future conditions (2081-2100, under the RCP 8.5 scenario), a control scenario, and a median scenario.

Experiment	JJA-mean Q_{sg} [m^3/s]	JJA-mean time period
Control	N/A	N/A
Q_{sg} -present	372	1995–2014
Q_{sg} -median	1563	N/A
Q_{sg} -RCP 8.5	2754	2081-2100

The model is run from July 01, 2014 00:00:00 UTC – January 01, 2017 00:00:00 UTC with hourly oceanic (temperature, salinity, velocities, and sea surface elevation), 3-hourly atmospheric (heat and freshwater fluxes, and wind speeds), and daily averaged sea ice (sea ice concentration, thickness, and velocities, and bulk ice salinity) forcings obtained from respective high-resolution regional models over this period (Prakash et al., 2022, 2023), and without Q_{sg} discharge, which represents the *Control* experiment (Table 1). To allow

investigations of the response of **PGIS-ice shelf** basal melt to variations in Q_{sg} **subglacial**
 **discharge**, our FVCOM setup has been augmented to include Q_{sg} **discharge** across the
 ca. 20 km wide **PGIS-GL-Three- Q_{sg} -grounding line. Three discharge** experiments are
 initialised from the stable model solution of the *Control* experiment from January 01,
 2016 00:00:00 UTC and run until January 01, 2017 00:00:00 UTC (Table 1). Lacking
 information regarding the precise location(s) of emergence of subglacial meltwater across
 the **PGIS-GL- Q_{sg} -grounding line. discharge** is injected uniformly across the 90 cells
 (hereafter denoted by $\Delta_{Q_{sg}}$) representing the **GL-grounding line** (Fig. 1b). At each $\Delta_{Q_{sg}}$,
 discharge is vertically uniform across the 23 σ -layers. The salinity at each $\Delta_{Q_{sg}}$ is set to 0,
 and the temperature is set to the (in-situ) pressure dependent freezing point of freshwater.
 The June-July-August (JJA)-mean Q_{sg} **discharge** values presented in Table 1 represent
 the daily Q_{sg} **discharge** values over the summer (JJA) months used in our model. Outside
 of this period, Q_{sg} **discharge** is set to zero (similar to Cai et al. (2017)). Results for the
 *Control* and Q_{sg} **discharge** experiments are presented from the final year of the simulation
 (January 01, 2016 – January 01, 2017). We note that the pre-2010 ice shelf geometry
 provided by BedMachine v3 is not consistent with the forcing period(s). Change in ice
 shelf geometry (e.g., due to calving) has been shown to impact fjord circulation and
 basal melt rates (Poinelli et al., 2023). However, the two prior large calving events in
 2010 and 2012 (Münchow et al., 2014) removed the softer and thinner (<150 m) outer
 sections of the ice shelf (Hill et al., 2018). As such, we posit that the findings of our
 study – discharge driven regime change in heat flux efficiency in the ice shelf cavity, and
 amplified channelized melting under the deeper drafts – are robust, irrespective of the
 geometry used.
The present day JJA-mean Q_{sg} **discharge** (Q_{sg} -present) and its increase in a future
 warming climate under the RCP 8.5 scenario (Q_{sg} -RCP 8.5) for the Petermann catchment
 is obtained from Slater et al. (2020) (Table 1). Here, a subset of the Coupled Model
 Intercomparison Project (CMIP5) Atmosphere and Ocean general circulation models
 (AOGCMs) (Barthel et al., 2020; Fig. 1c) is used to force the Modèle Atmosphérique
 Régional (MAR) 3.9.6 (Fettweis et al., 2013) for the period 1950 – 2100. The projected
 MAR meltwater runoff over the Petermann catchment is then bias corrected to ensure
 that it is consistent with the best present-day (1995–2014) runoff estimate, and to also
 enable a smooth transition from present to RCP 8.5 forcing. It is then summed up over
 the Petermann catchment to give the Q_{sg} **discharge** estimates. We note that the modelled
 runoff outputs were stored as annual means and were converted to the JJA-mean runoff by
 multiplying the output by a factor of 365/92, where 365 = number of days in a year, and
 92 = number of days in JJA. This coarse conversion introduces a minor discrepancy be-
 tween the converted and true modelled JJA-mean values; however, we maintain that this
 does not significantly impact the findings of our study. While inter-model differences
 can be large (Fig.1 (c)), all five CMIP5 AOGCMs were used to prepare the multi-model
 mean Q_{sg} **discharge** used in this study. Note that the median Q_{sg} **discharge** magnitude
 (Q_{sg} -median) is not obtained from Slater et al. (2020), but is constructed using the Q_{sg} -
 present and Q_{sg} -RCP 8.5 magnitudes (as median ~~(of Q_{sg} -present and Q_{sg} -RCP 8.5)~~;
 ~~and as such,~~ As such, it represents an intermediate future mean Q_{sg} **discharge** mag-
 nitude, however, it does not correspond to any known projection scenario or a particular
 time period (Table 1).
**Diagnostics of basal melt rate and its drivers**
For the investigation of **PGIS-ice shelf** basal melt rate, its expression in the thermody-
 namic framework (Holland and Jenkins, 1999; Zhou and Hattermann, 2020) of the prim-
 itive equation setting is recalled:

$$590 \quad \rho_{fw} m_b L = -\rho_{sw} c_w \Gamma_T u^* \Delta T. \quad (1)$$

Here, ρ_{fw} and ρ_{sw} are the densities of freshwater and ocean water, respectively. L
 is the latent heat of fusion of ice, c_w is the specific heat capacity of ocean water, and Γ_T
 is a non-dimensional heat-transfer coefficient. m_b is basal melt rate, controlled primarily
 by friction velocity (u^*) and thermal driving (ΔT), and where

$$595 \quad u^* = \sqrt{C_D(u_w^2 + u_{res}^2)}, \quad \Delta T = T_b - T_w, \quad (2)$$

$$596 \quad T_b = \lambda_1 S_b + \lambda_2 + \lambda_3 P_b, \quad (3)$$

$$597 \quad \rho_{fw} m_b S_b = -\rho_{sw} \Gamma_S u^* (S_b - S_w). \quad (4)$$

Above, C_D is the drag coefficient, u_w is the velocity magnitude some distance away
 from the ice shelf-ocean boundary, and u_{res} is the velocity of small sub-grid scale residual
 currents (Asay-Davis et al., 2016). T_b , S_b , and P_b are the potential temperature, salinity,
 and pressure at the ice shelf-ocean boundary, and T_w and S_w are the potential temperature
 and salinity some distance away from it. λ_1 , λ_2 , and λ_3 , respectively, are the slope, inter-
 cept, and pressure coefficient of the liquidus, and Γ_S is a non-dimensional salt-transfer
 coefficient. Note that T_w , S_w , and u_w are obtained from the first (i.e. uppermost) σ -layer.
 Furthermore, Γ_T and Γ_S are application specific, and are herein tuned according to the
 applied boundary conditions, PGIS-Petermann ice shelf basal topography, and mixing
 schemes so as to be consistent with the contemporary observational estimates of PGIS
 the Petermann ice shelf basal melt (see Table A1). The drivers of basal melt rate (m_b),
 namely, friction velocity (u^* ~~and~~) and thermal driving (ΔT) are used to investigate the
 impact of \$Q_{sg}\$ discharge on the summer mean \$m_b\$ basal melt rate.
Basal channel Changes in ice volume changes above basal channels over time
We calculate the changes in the annual mean channel averaged ice volume above basal
 channels over time ($V_c(N)$) to determine the number of years (N) it would take for $V_c(N)$
 to become 0 as

$$615 \quad 0 = V_i + (N \times V_a) - (N \times V_m), \quad (5)$$

$$616 \quad \implies N = \frac{V_i}{V_m - V_a} \quad (6)$$

where, V_i is the initial (contemporary) channel ice volume ice volume above a channel
 expressed as L_c (channel length [m]) \times W_c (mean channel width [m]) \times H_c (mean chan-
 nel depth [m]). V_a is the annual mean volume of ice advected downstream of the GL
 grounding line per year expressed as AR_c (~~annual~~-mean advection rate [m/yr]) \times W_c
 621 \times H_c . V_m is the annual mean volume of ice melted per year expressed as m_{bc} (annual
 mean channel averaged basal melt rate [m/yr]) \times $L_c \times W_c$. Dimensions associated with
 each channel section are detailed in Table 2. Note that W_c drops out of the calculation,
 and as such, is not detailed in Table 2.

Table 2. Dimensions associated with the PGIS-Petermann ice shelf basal channel sections illustrated in Fig. 6(a). Channel section along the western, central, and eastern region are termed as S_w , S_c , and S_e , respectively.

Channel Name	L_c [m]	H_c [m]
S_w	30000	241
S_c	25000	271
S_e	28000	331

Data availability
Our study is based on numerical modelling and we provide the model input files (<https://zenodo.org/doi/10.5281/zenodo.12803093>) and code (see Code availability below) that
627 //zenodo.org/doi/10.5281/zenodo.12803093) and code (see Code availability below) that
are required to reproduce our simulations.
Code availability
The open source code Finite Volume Community Ocean Model version 4.0 (FVCOM
v4.0), augmented by both the ice shelf and sea ice modules, that has been used to con-
duct the numerical experiments is publicly available at: [https://github.com/abhay26992/](https://github.com/abhay26992/FVCOM_Petermann_Code.git)
FVCOM_Petermann_Code.git
References
- Asay-Davis XS, Cornford SL, Durand G, Galton-Fenzi BK, Gladstone RM, Gudmundsson GH,
Hattermann T, Holland DM, Holland D, Holland PR, Martin DF, Mathiot P, Pattyn F, Seroussi
H. 2016. Experimental design for three interrelated marine ice sheet and ocean model in-
tercomparison projects: MISMIP v. 3 (MISMIP+), ISOMIP v. 2 (ISOMIP+) and MIS-
OMIP v. 1 (MISOMIP1). *Geoscientific Model Development* 9:2471–2497. doi:10.5194/
gmd-9-2471-2016.
- Aschwanden A, Fahnestock MA, Truffer M, Brinkerhoff DJ, Hock R, Khroulev C, Mottram R,
Khan SA. 2019. Contribution of the Greenland Ice Sheet to sea level over the next millennium.
*Science advances* 5:eaav9396.
- Bamber JL, Westaway RM, Marzeion B, Wouters B. 2018. The land ice contribution to sea level
during the satellite era. *Environmental Research Letters* 13. doi:10.1088/1748-9326/aac2f0.
- Banwell AF, MacAyeal DR. 2015. Ice-shelf fracture due to viscoelastic flexure stress induced by
fill/drain cycles of supraglacial lakes. *Antarctic Science* 27:587–597.
- Barthel A, Agosta C, Little CM, Hattermann T, Jourdain NC, Goelzer H, Nowicki S, Seroussi H,
Straneo F, Bracegirdle TJ. 2020. CMIP5 model selection for ISMIP6 ice sheet model forcing:
Greenland and Antarctica. *The Cryosphere* 14:855–879.
- Bell RE, Banwell AF, Trusel LD, Kingslake J. 2018. Antarctic surface hydrology and impacts on
ice-sheet mass balance. *Nature Climate Change* 8:1044–1052.
- Briner JP, Cuzzone JK, Badgeley JA, Young NE, Steig EJ, Morlighem M, Schlegel N, Hakim GJ,
Schaefer JM, Johnson JV, Lesnek AJ, Thomas EK, Allan E, Bennike O, Cluett AA, Csatho B,
de Vernal A, Downs J, Larour E, Nowicki S. 2020. Rate of mass loss from the Greenland Ice
Sheet will exceed Holocene values this century. *Nature* doi:10.1038/s41586-020-2742-6.
- Cai C, Rignot E, Menemenlis D, Nakayama Y. 2017. Observations and modeling of ocean-
induced melt beneath Petermann Glacier Ice Shelf in northwestern Greenland. *Geophysical*
*Research Letters* 44:8396–8403.
- Chen C, Huang H, Beardsley RC, Liu H, Xu Q, Cowles G. 2007. A finite volume numerical
approach for coastal ocean circulation studies: Comparisons with finite difference models.
*Journal of Geophysical Research: Oceans* 112.
- Ciraci E, Rignot E, Scheuchl B, Tolpekin V, Wollersheim M, An L, Milillo P, Bueso-Bello JL,
Rizzoli P, Dini L. 2023. Melt rates in the kilometer-size grounding zone of petermann glacier,
greenland, before and during a retreat. *Proceedings of the National Academy of Sciences*
120:e2220924120.
- Cook SJ, Christoffersen P, Todd J, Slater D, Chauché N. 2020. Coupled modelling of sub-
glacial hydrology and calving-front melting at Store Glacier, West Greenland. *The Cryosphere*
14:905–924.
- Davis PE, Nicholls KW, Holland DM, Schmidt BE, Washam P, Riverman KL, Arthern RJ,
Vaňková I, Eayrs C, Smith JA, et al. 2023. Suppressed basal melting in the eastern Thwaites
Glacier grounding zone. *Nature* 614:479–485.
- DeConto RM, Pollard D, Alley RB, Velicogna I, Gasson E, Gomez N, Sadai S, Condrón A,
Gilford DM, Ashe EL, et al. 2021. The Paris Climate Agreement and future sea-level rise
from Antarctica. *Nature* 593:83–89.
- Dow CF, Lee WS, Greenbaum JS, Greene CA, Blankenship DD, Poinar K, Forrest AL, Young
DA, Zappa CJ. 2018. Basal channels drive active surface hydrology and transverse ice shelf
fracture. *Science Advances* 4:eaa07212.
- Drews R, Pattyn F, Hewitt I, Ng F, Berger S, Matsuoka K, Helm V, Bergeot N, Favier L, Neckel
680 N. 2017. Actively evolving subglacial conduits and eskers initiate ice shelf channels at an
681 Antarctic grounding line. *Nature communications* 8:15228.
- Ehrenfeucht S, Morlighem M, Rignot E, Dow CF, Mouginot J. 2023. Seasonal acceleration of
Petermann Glacier, Greenland, from changes in subglacial hydrology. *Geophysical Research*
*Letters* 50:e2022GL098009.
- Fettweis X, Franco B, Tedesco M, Van Angelen J, Lenaerts JT, van den Broeke MR, Gallée H.
2013. Estimating the Greenland ice sheet surface mass balance contribution to future sea level
rise using the regional atmospheric climate model MAR. *The Cryosphere* 7:469–489.
- Forster RR, Box JE, Van Den Broeke MR, Miège C, Burgess EW, Van Angelen JH, Lenaerts JT,
Koenig LS, Paden J, Lewis C, et al. 2014. Extensive liquid meltwater storage in firn within the
Greenland ice sheet. *Nature Geoscience* 7:95–98.
- Fox-Kemper B, Hewitt HT, Xiao C, Aðalgeirsdóttir G, Drijfhout SS, Edwards TL, Golledge NR,
Hemer M, Kopp RE, Krinner G, Mix A, Notz D, Nowicki S, Nurhati IS, Ruiz L, Sallée JB,
BA SA, Y Y. 2021. Ocean, cryosphere and sea level change. In *Climate Change 2021: The*
*Physical Science Basis. Contribution of Working Group I to the Sixth Assessment Report of*
*the Intergovernmental Panel on Climate Change* [Masson-Delmotte, V., P. Zhai, A. Pirani,
S.L. Connors, C. Péan, S. Berger, N. Caud, Y. Chen, L. Goldfarb, M.I. Gomis, M. Huang, K.
Leitzell, E. Lonnoy, J.B.R. Matthews, T.K. Maycock, T. Waterfield, O. Yelekçi, R. Yu, and B.
Zhou (eds.)]. Cambridge University Press, Cambridge, United Kingdom and New York, NY,
USA, pp. 1211–1362 doi:10.1017/9781009157896.011.
- Frémand AC, Fretwell P, Bodart JA, Pritchard HD, Aitken A, Bamber JL, Bell R, Bianchi C,
Bingham RG, Blankenship DD, et al. 2023. Antarctic bedmap data: Findable, accessible,
interoperable, and reusable (fair) sharing of 60 years of ice bed, surface, and thickness data.
*Earth System Science Data* 15:2695–2710.
- Gadi R, Rignot E, Menemenlis D. 2023. Modeling ice melt rates from seawater intrusions
in the grounding zone of petermann gletscher, greenland. *Geophysical Research Letters*
50:e2023GL105869.
- Goelzer H, Nowicki S, Payne A, Larour E, Seroussi H, Lipscomb WH, Gregory J, Abe-Ouchi A,
Shepherd A, Simon E, Agosta C, Alexander P, Aschwanden A, Barthel A, Calov R, Chambers
C, Choi Y, Cuzzzone J, Dumas C, Edwards T, Felikson D, Fettweis X, Golledge NR, Greve R,
Humbert A, Huybrechts P, Le clec’h S, Lee V, Leguy G, Little C, Lowry DP, Morlighem M,
Nias I, Quiquet A, Rückamp M, Schlegel NJ, Slater DA, Smith RS, Straneo F, Tarasov L, van de
Wal R, van den Broeke M. 2020. The future sea-level contribution of the Greenland ice sheet:
a multi-model ensemble study of ISMIP6. *The Cryosphere*. doi:10.5194/tc-14-3071-2020.
- Goldberg DN, Twelves AG, Holland PR, Wearing MG. 2023. The non-local impacts of antarctic
subglacial runoff. *Journal of Geophysical Research: Oceans* 128:e2023JC019823.
- Golledge NR, Keller ED, Gomez N, Naughten KA, Bernales J, Trusel LD, Edwards TL. 2019.
Global environmental consequences of twenty-first-century ice-sheet melt. *Nature* 566:65–72.
- Greene CA, Gardner AS, Wood M, Cuzzzone JK. 2024. Ubiquitous acceleration in Greenland Ice
Sheet calving from 1985 to 2022. *Nature* 625. doi:doi.org/10.1038/s41586-023-06863-2.
Gwyther DE, Dow CF, Jendersie S, Gourmelen N, Galton-Fenzi BK. 2023. Subglacial freshwater
drainage increases simulated basal melt of the Totten Ice Shelf. *Geophysical Research Letters*
50:e2023GL103765.
Hattermann T, Isachsen PE, von Appen WJ, Albretsen J, Sundfjord A. 2016. Eddy-driven recir-
culation of atlantic water in fram strait. *Geophysical Research Letters* 43:3406–3414.
Hill EA, Gudmundsson GH, Carr JR, Stokes CR. 2018. Velocity response of Petermann Glacier,
northwest Greenland, to past and future calving events. *The Cryosphere* 12:3907–3921.
Holland DM, Jenkins A. 1999. Modeling thermodynamic ice–ocean interactions at the base of
an ice shelf. *Journal of physical oceanography* 29:1787–1800.
Hunke EC, Lipscomb WH, Turner AK, Jeffery N, Elliott S. 2015. Cice: The los alamos sea ice
model documentation and software user’s manual version 5.1 la-cc-06-012. T-3 Fluid Dynam-
ics Group, Los Alamos National Laboratory 675:15.
Jakobsson M, Mayer L, Coakley B, Dowdeswell JA, Forbes S, Fridman B, Hodnesdal H,
Noormets R, Pedersen R, Rebesco M, et al. 2012. The International Bathymetric Chart of
the Arctic Ocean (IBCAO) version 3.0. *Geophysical Research Letters* 39.
Johnson H, Münchow A, Falkner K, Melling H. 2011. Ocean circulation and properties in Peter-
manns Fjord, Greenland. *Journal of Geophysical Research: Oceans* 116.
Le Brocq AM, Ross N, Griggs JA, Bingham RG, Corr HF, Ferraccioli F, Jenkins A, Jordan TA,
Payne AJ, Rippin DM, et al. 2013. Evidence from ice shelves for channelized meltwater flow
beneath the Antarctic Ice Sheet. *Nature Geoscience* 6:945–948.
Lepp A, Simkins L, Anderson J, Clark R, Wellner J, Hillenbrand C, Smith J, Lehrmann A, Totten
R, Larter R, et al. 2022. Sedimentary signatures of persistent subglacial meltwater drainage
from Thwaites Glacier, Antarctica. *Frontiers in Earth Science* 10:863200.
Li T, Robinson LF, MacGilchrist GA, Chen T, Stewart JA, Burke A, Wang M, Li G, Chen J, Rae
JW. 2023. Enhanced subglacial discharge from Antarctica during meltwater pulse 1A. *Nature*
*Communications* 14:7327.
Millan R, Jager E, Mouginit J, Wood MH, Larsen SH, Mathiot P, Jourdain NC, Bjørk A. 2023.
Rapid disintegration and weakening of ice shelves in North Greenland. *Nature Communica-*
*tions* doi:10.1038/s41467-023-42198-2.
Millan R, Mouginit J, Derkacheva A, Rignot E, Milillo P, Ciraci E, Dini L, Bjørk A. 2022. On-
going grounding line retreat and fracturing initiated at the Petermann Glacier Ice Shelf, Green-
land, after 2016. *The Cryosphere* 16:3021–3031.
Morlighem M, Williams CN, Rignot E, An L, Arndt JE, Bamber JL, Catania G, Chauché N,
Dowdeswell JA, Dorschel B, et al. 2017. Bedmachine v3: Complete bed topography and
ocean bathymetry mapping of Greenland from multibeam echo sounding combined with mass
conservation. *Geophysical research letters* 44:11–051.
Mouginit J, Rignot E, Bjørk AA, Van den Broeke M, Millan R, Morlighem M, Noël B, Scheuchl
B, Wood M. 2019. Forty-six years of Greenland Ice Sheet mass balance from 1972 to 2018.
*Proceedings of the National Academy of Sciences* 116:9239–9244.
Münchow A, Padman L, Fricker HA. 2014. Interannual changes of the floating ice shelf of Pe-
termann Gletscher, North Greenland, from 2000 to 2012. *Journal of Glaciology* 60:489–499.
Muntjewerf L, Petrini M, Vizcaino M, Ernani da Silva C, Sellevold R, Scherrenberg MD, Thayer-
Calder K, Bradley SL, Lenaerts JT, Lipscomb WH, et al. 2020. Greenland Ice Sheet contribu-
tion to 21st century sea level rise as simulated by the coupled CESM2. 1-cism2. 1. *Geophysical*
*Research Letters* 47:e2019GL086836.
Narkevic A, Csatho B, Schenk T. 2023. Rapid basal channel growth beneath Greenland’s longest
floating ice shelf. *Geophysical Research Letters* 50:e2023GL103226.
Noël B, van de Berg WJ, Lhermitte S, van den Broeke MR. 2019. Rapid ablation zone expansion
amplifies north greenland mass loss. *Science advances* 5:eaaw0123.
Noël B, van Wessem JM, Wouters B, Trusel L, Lhermitte S, van den Broeke MR. 2023. Higher
Antarctic ice sheet accumulation and surface melt rates revealed at 2 km resolution. *Nature*
*communications* 14:7949.
Otto J, Holmes FA, Kirchner N. 2022. Supraglacial lake expansion, intensified lake drainage
frequency, and first observation of coupled lake drainage, during 1985–2020 at Ryder Glacier,
Northern Greenland. *Frontiers in Earth Science* 10:978137.
Padman L, Erofeeva S. 2004. A barotropic inverse tidal model for the arctic ocean. *Geophysical*
*Research Letters* 31.
- Pelle T, Greenbaum J, Ehrenfeucht S, Dow C, McCormack F. 2024. Subglacial discharge accel-
erates dynamic retreat of aurora subglacial basin outlet glaciers, east antarctica, over the 21st
century. *Journal of Geophysical Research: Earth Surface* 129:e2023JF007513.
- Pelle T, Greenbaum JS, Dow CF, Jenkins A, Morlighem M. 2023. Subglacial discharge accelerates
future retreat of Denman and Scott Glaciers, East Antarctica. *Science Advances* 9:ead9014.
- Poinelli M, Nakayama Y, Larour E, Vizcaino M, Riva R. 2023. Ice-front retreat controls
on ocean dynamics under larsen c ice shelf, antarctica. *Geophysical Research Letters*
50:e2023GL104588.
- Prakash A, Zhou Q, Hattermann T, Bao W, Graverson R, Kirchner N. 2022. A nested high-
resolution unstructured grid 3-D ocean-sea ice-ice shelf setup for numerical investigations of
the Petermann ice shelf and fjord. *MethodsX* 9:101668.
- Prakash A, Zhou Q, Hattermann T, Kirchner N. 2023. Impact of the Nares Strait sea ice arches
on the long-term stability of the Petermann Glacier ice shelf. *The Cryosphere* 17:5255–5281.
doi:10.5194/tc-17-5255-2023.
- Rawlins LD, Rippin DM, Sole AJ, Livingstone SJ, Yang K. 2023. Seasonal evolution of the
supraglacial drainage network at Humboldt Glacier, North Greenland, between 2016 and 2020.
*The Cryosphere Discussions* :1–32.
- Rignot E, Fenty I, Xu Y, Cai C, Kemp C. 2015. Undercutting of marine-terminating glaciers in
West Greenland. *Geophysical Research Letters* 42:5909–5917.
- Rignot E, Steffen K. 2008. Channelized bottom melting and stability of floating ice shelves.
*Geophysical Research Letters* 35.
- Rosevear MG, Gayen B, Galton-Fenzi BK. 2022. Regimes and transitions in the basal melting of
antarctic ice shelves. *Journal of Physical Oceanography* 52:2589–2608.
- Scambos T, Fricker HA, Liu CC, Bohlander J, Fastook J, Sargent A, Massom R, Wu AM. 2009.
Ice shelf disintegration by plate bending and hydro-fracture: Satellite observations and model
results of the 2008 Wilkins ice shelf break-ups. *Earth and Planetary Science Letters* 280:51–60.
- Schmidt BE, Washam P, Davis PE, Nicholls KW, Holland DM, Lawrence JD, Riverman KL,
Smith JA, Spears A, Dichek D, et al. 2023. Heterogeneous melting near the Thwaites Glacier
grounding line. *Nature* 614:471–478.
- Sciascia R, Straneo F, Cenedese C, Heimbach P. 2013. Seasonal variability of submarine melt
rate and circulation in an east Greenland fjord. *Journal of Geophysical Research: Oceans*
118:2492–2506.
- Shroyer EL, Padman L, Samelson R, Münchow A, Stearns LA. 2017. Seasonal control of Peter-
mann Gletscher ice-shelf melt by the ocean’s response to sea-ice cover in Nares Strait. *Journal*
*of Glaciology* 63:324–330.
- Shu Q, Wang Q, Årthun M, Wang S, Song Z, Zhang M, Qiao F. 2022. Arctic Ocean Amplification
in a warming climate in CMIP6 models. *Science Advances* 8:eabn9755.
- Slater D, Nienow P, Cowton T, Goldberg D, Sole A. 2015. Effect of near-terminus subglacial
hydrology on tidewater glacier submarine melt rates. *Geophysical Research Letters* 42:2861–
2868.
- Slater D, Straneo F. 2022. Submarine melting of glaciers in Greenland amplified by atmospheric
warming. *Nature Geoscience* 15:794–799.
- Slater DA, Felikson D, Straneo F, Goelzer H, Little CM, Morlighem M, Fettweis X, Nowicki S.
2020. Twenty-first century ocean forcing of the Greenland ice sheet for modelling of sea level
contribution. *The Cryosphere* 14:985–1008.
- Tinto KJ, Bell RE, Cochran JR, Münchow A. 2015. Bathymetry in Petermann fjord from Oper-
ation IceBridge aerogravity. *Earth and Planetary Science Letters* 422:58–66.
- Turton JV, Hochreuther P, Reimann N, Blau MT. 2021. The distribution and evolution of
supraglacial lakes on 79 N Glacier (north-eastern Greenland) and interannual climatic con-
trols. *The Cryosphere* 15:3877–3896.
- Washam P, Münchow A, Nicholls KW. 2018. A decade of ocean changes impacting the ice shelf
of Petermann Gletscher, Greenland. *Journal of Physical Oceanography* 48:2477–2493.
- Washam P, Nicholls KW, Münchow A, Padman L. 2020. Tidal modulation of buoyant flow and
basal melt beneath Petermann Gletscher ice Shelf, Greenland. *Journal of Geophysical Re-*
*search: Oceans* 125:e2020JC016427.
- Wearing M, Stevens L, Dutrieux P, Kingslake J. 2021. Ice-shelf basal melt channels stabilized by
secondary flow. *Geophysical Research Letters* 48:e2021GL094872.
- Willis MJ, Herried BG, Bevis MG, Bell RE. 2015. Recharge of a subglacial lake by surface
meltwater in northeast Greenland. *Nature* 518:223–227.
- Wilson N, Straneo F, Heimbach P. 2017. Satellite-derived submarine melt rates and mass balance
(2011–2015) for Greenland’s largest remaining ice tongues. *The Cryosphere* 11:2773–2782.
- Zeising O, Neckel N, Dörr N, Helm V, Steinhage D, Timmermann R, Humbert A. 2023. Extreme
melting at Greenland’s largest floating ice tongue. *EGUsphere* :1–35.
- Zhou Q, Hattermann T. 2020. Modeling ice shelf cavities in the unstructured-grid, finite volume
community ocean model: Implementation and effects of resolving small-scale topography.
*Ocean Modelling* 146:101536.
- Zhou Q, Zhao C, Gladstone R, Hattermann T, Gwyther D, Galton-Fenzi B. 2024. Evaluat-
ing an accelerated forcing approach for improving computational efficiency in coupled ice
sheet–ocean modelling. *Geoscientific Model Development* 17:8243–8265. doi:10.5194/
gmd-17-8243-2024.
**Acknowledgements**
This work was funded by the Swedish Research Council VR grant 2022-03718 to N.K..
Q.Z. was supported by the Research Council of Norway (RCN) project 244319 and
295075. T.H. was supported by the RCN project 314570. We thank Donald Slater for
providing the present-day and RCP 8.5 runoff estimates for the Petermann Glacier. The
simulations were performed on resources provided by UNINETT Sigma2 - the National
Infrastructure for High Performance Computing and Data Storage in Norway. The super-
computer Betzy was used under the project NN9824K.
**Contributions**
All authors contributed to the conceptualization of the study. A.P. designed the exper-
iments with technical input from Q.Z. and T.H.. A.P. prepared and ran the numerical
simulations. A.P. analyzed the model output with scientific advice from Q.Z. and T.H..
859 A.P. and N.K. wrote the manuscript, with contributions from Q.Z. and T.H..

**Competing interests**
The authors declare no competing interests.
**A Model tuning**

Table A1. PGIS Petermann ice shelf thermodynamical and ocean mixing parameters used in this study

Parameter	Value	Description
ρ_{fw}	1000 kg m^{-3}	Freshwater density
L	$3.34 \times 10^5 \text{ J kg}^{-1}$	Latent heat of fusion of ice
ρ_{sw}	1028 kg m^{-3}	Seawater density
c_w	$3974 \text{ J }^\circ\text{C}^{-1} \text{ kg}^{-1}$	Specific heat capacity of seawater
Γ_T	1.2×10^{-2}	Non-dimensional heat-transfer coefficient
C_D	2.5×10^{-3}	Drag coefficient
u_{res}	$1.0 \times 10^{-2} \text{ m s}^{-1}$	Residual velocity
λ_1	$-5.73 \times 10^{-2} \text{ }^\circ\text{C PSU}^{-1}$	Liquidus slope
λ_2	$8.32 \times 10^{-2} \text{ }^\circ\text{C}$	Liquidus intercept
λ_3	$-7.53 \times 10^{-8} \text{ }^\circ\text{C Pa}^{-1}$	Liquidus pressure coefficient
Γ_S	$\Gamma_T/35$	Non-dimensional salt-transfer coefficient
Z_o	1.0×10^{-3}	Roughness length scale
Ro_{min}	2.5×10^{-3}	Roughness minimum
K_m	1.0×10^{-5}	Vertical eddy viscosity
P_v	1.0	Vertical Prandtl Number
P_h	1.0×10^{-1}	Horizontal Prandtl Number
C_h	1.0×10^{-1}	Scaling constant

863 **B Longitudinal ice shelf draft profile**

Figure B1. (a) Map of the PGIS Petermann ice shelf draft with the across-fjord transect ca. 5 km from the GL grounding line shown in Fig. 1(b) overlaid. Locations Apex locations of the four prominent longitudinal ice shelf basal channels at this transect (highlighted in Fig. 5) at this transect are represented using magenta diamonds. From left to right, these are \$L_w\$ (west), \$L_c\$ (central), \$L_e(1)\$ (east-1) and \$L_e(2)\$ (east-2). (b) Along-fjord profile of the PGIS Petermann ice shelf draft over the channel sections (Fig. 6(a)) along the western (\$S_w\$; blue), central (\$S_c\$; black) and eastern (\$S_e\$; red) regions channel sections (Fig. 6(a)) as indicated in panel a.

C Saturation of the thermal pump

Figure C1. Along-fjord summer mean temperature (left column), salinity (middle column), and flow (right column) anomalies (δ) for Q_{sg} -median relative to control (first row) and Q_{sg} -RCP 8.5 relative to control (second row) experiments for the section shown in Fig. 1(b). Black stippled lines in panels a, b, d and e indicate the zero temperature and salinity difference contours. Solid black lines in panels c and f depict the mean isopycnals for the control experiment, whereas the stippled green lines correspond to the Q_{sg} -median (c) and Q_{sg} -RCP 8.5 (f) experiments. Isopycnals are plotted at equal intervals of 0.2 kg/m^3 . The 27.9 kg/m^3 isopycnal corresponds to the dense Atlantic Water that contacts the grounding line. In each panel, PGIS-GL the grounding line (at ca. 600 m depth) is on the right margin and open ocean is to the left. Vertical magenta line in panels a, b, d, and e show the location of the model node ca. 10 km from the GL-grounding line at which the vertical profile of summer mean temperature and salinity are shown for the control, Q_{sg} -present, Q_{sg} -median, and Q_{sg} -RCP 8.5 experiments in Fig. 3(g,h) (cf. Table 1).

865 **D** Trapping Convergence of meltwater

Figure D1. Anomaly maps (δ) of the summer mean surface layer salinity (a,b) and basal melt rate (c,d) for the Q_{sg} -present relative to control (column 1) and Q_{sg} -median relative to control (column 2) experiments. Contours of the ice shelf draft are plotted at 100 m intervals (450 m – 150 m) and overlaid as dotted red and blue lines over the surface layer salinity and melt rate maps, respectively.

E Modified sub-ice shelf water column thickness

Figure E1. (a) BedMachine v3 (Morlighem et al., 2017) water column thickness, and (b) modified BedMachine v3 water column thickness used in this study, obtained from Prakash et al. (2022). (c) Along-fjord profile of the water column thickness for the western (blue), central (black), and eastern (red) sections of the fjord as highlighted in panels (a) and (b). Stippled and solid lines correspond, respectively, to the BedMachine and modified BedMachine profiles. (d) Along-fjord summer mean temperature for the section shown in Fig. 1(b). Mean isopycnals (solid black lines) are overlaid at equal intervals of 0.2 kg/m^3 . The 27.9 kg/m^3 isopycnal surface represents the dense Atlantic Water that contacts the grounding line. Solid red line represents the (unmodified) BedMachine v3 bathymetry, wherein, a synthetic sill blocks the 27.9 kg/m^3 isopycnal surface from contacting the grounding line.

Reviewer #1 (Remarks to the Author):

The authors have sufficiently addressed all of my prior concerns. I find this to be an interesting paper that advances our understanding of glacier-ocean interactions. However, I have three related editorial comments regarding the paper that are in the spirit of trying to increase its impact.

We sincerely appreciate your time and constructive feedback, which has strengthened our manuscript. We are pleased to hear your decision and grateful for your insightful suggestions. Please see our replies to your comments below.

Please note that the *track changes* document also includes revisions made to address editorial requests. Below are some of the most notable changes:

- The abstract has been shortened to less than 150 words.
- Citations are now formatted as superscripted numbers and referenced in the order they appear in the paper. In some cases, minor adjustments were made to the surrounding text to accommodate this citation style.
- Figures now include a brief title, added in bold at the beginning of each caption, summarizing the figure's content. Due to size limitations, high-resolution figures have not been included in the *track changes* document but can be found in the *revised manuscript* file and the *figure files*.

To me, the real impact of this paper is showing how subglacial discharge (and changes in subglacial discharge) affects submarine melting of ice shelves and not simply that increases in subglacial discharge will affect the stability of Petermann Ice Shelf. Previous studies, such as the work by Slater and Straneo, as well as that of Motyka et al. and probably many others, have shown that changes in subglacial discharge can lead to large changes in melt rates. What I take away from the paper is that the effect of subglacial discharge (i) at low fluxes is to draw in warm water and breakdown the boundary layer, whereas (ii) at high fluxes the temperature effect becomes saturated and further increases in discharge essentially affect the boundary layer. Much of the paper focuses on this. The uncertainty in the model topography, location and number of discharge outlets, and lack of ice dynamics model causes me to question the assertion that future runoff may destabilize Petermann Ice Shelf. (Not that I disagree, I just don't think that's really what this paper shows.) Personally, I think the deeper understanding of how runoff affects submarine melting to be more interesting and robust than whether or not one particular glacier will destabilize, but maybe that is just my personal bias coming through.

The arguments presented here are compelling and have led us to reconsider the title of our paper. We apologize for not addressing this concern in the first round of review and sincerely appreciate you bringing it to our attention again. Indeed, influence of subglacial discharge on basal melting has been well documented. Our primary research objective was to investigate the underlying mechanisms – *how* discharge drives melt, and *how* these mechanisms may evolve in a warming climate (abstract: lines 2 – 5, and Introduction: lines 47 – 49; *track changes* document). As you noted, much of the paper is dedicated to it. We also recognize and agree that the title should be grounded in the strengths of our paper, which primarily lie in the ice shelf-ocean model results. It should also highlight the novelty/key takeaway: the enhanced discharge-driven transition from a thermally regulated to a thermally saturated, fully velocity-regulated melt regime, and (importantly) the increased melting despite this saturation. Therefore, we have revised the title of the paper to “*Enhanced subglacial discharge amplifies Petermann Ice Shelf melting when ocean thermal forcing saturates*”.

The number of panels in figures 2 and 3, and the long figure captions, make them somewhat difficult to digest. I don't think the figures need to be changed all that much, but I do think it would be helpful if they were labeled somehow so as to make it clear which panels correspond to the control run and each of the runoff scenarios (as was done for figure 4)

Thank you for the feedback. We have revised Figure 3 by adding panel labels which indicate the corresponding experiments. To avoid cluttering, we have refrained from doing so in Figure 2. Please see the updated figure in the *revised manuscript* file or in the corresponding *figure file*.

I think the writing can be tightened up and made more succinct in many places. As just an example, the paragraphs in the discussion section are quite long (almost a page long in draft mode). This can make it difficult to determine (or remember!) the main point of each paragraph. There are quite a few clauses and phrases throughout the paper that are either unnecessary or that could be shortened to improve readability.

Thank you for the suggestion. We have revised the discussion chapter by streamlining the language, removing redundant clauses, and simplifying phrasing wherever applicable to address these concerns.

Reviewer #2 (Remarks to the Author):

The authors have been very thorough in their response to reviewers comments, and have made substantial changes to the text and figures. I feel that all of my comments have been very well addressed.

We sincerely appreciate your time and thoughtful review. Your constructive feedback has been valuable in refining our manuscript, and we are pleased that the revisions have effectively addressed your concerns.

Reviewer #3 (Remarks to the Author):

The authors have sufficiently met my concerns in their revised manuscript and I have no further substantive suggestions. I would commend the authors on their complete and thorough addressing of all (not just my) reviewer suggestion. The updated manuscript has improved readability and the key points are more clearly made.

The dropping of terms in favor of words (e.g. Qsg → subglacial discharge) improves readability. The added 'in sum' at the end of the results section is helpful. I appreciate the new concluding sentence of the manuscript and agree — the interplay between surface melt and seawater intrusions at the grounding zone are a real research frontier right now.

We are pleased to hear of your decision, and sincerely appreciate the time and effort you have put into reviewing our manuscript. Your detailed and constructive feedback has been invaluable in strengthening our work. Please see our response to the minor notes below.

Please note that the *track changes* document also includes revisions made to address editorial requests. Below are some of the most notable changes:

- The abstract has been shortened to less than 150 words.
- Citations are now formatted as superscripted numbers and referenced in the order they appear in the paper. In some cases, minor adjustments were made to the surrounding text to accommodate this citation style.
- Figures now include a brief title, added in bold at the beginning of each caption, summarizing the figure's content. Due to size limitations, high-resolution figures have not been included in the *track changes* document but can be found in the *revised manuscript* file and the *figure files*.

Minor notes for the authors:

2 First sentence scans somewhat awkwardly, suggest rephrasing to clarify

Thank you for informing us about it. We also noticed a typo – the Greenland Ice Sheet is currently the largest contributor to *barystatic (mass addition)* sea-level rise. Please see the revised formulation in lines 2 – 7.

37 'studies'

Here, study refers to Cai et al. (2017), who show tripling of melt in summer driven by subglacial discharge compared to winter (no discharge).

50 sea-ice ice-shelf (I think?). Is there a coupled ice flow model here? In looking to your methods (438) it looks like there is a coupled ice shelf model, but I would love one more sentence here describing it. For those not familiar with the Zhao ice shelf model, what is it? What ice rheology is used? Is it a higher-order model?

Thank you for bringing this to our attention. The correct phrasing would be ocean–sea ice–ice shelf, using en dashes instead of hyphens to separate distinct model components. We have fixed this in the revised manuscript. Please see lines 50 – 51.

Thank you for your question. There is no coupled ice flow model in this study. The ice shelf module referenced in our methods refers exclusively to modifications within the ocean model (FVCOM)

to account for the static effects of a prescribed ice shelf geometry on ocean dynamics and thermodynamics. As such, ice rheology, stress regimes, and ice sheet model order are not applicable here. The ice shelf module adjusts FVCOM's horizontal pressure gradient forces (dynamics) to account for the ice shelf's weight, with terrain-following sigma-layers adapting to the prescribed ice draft. We have included a brief description in lines 485 – 487. The three-equation melt parameterization (thermodynamics) is used to simulate melt at the ice shelf-ocean interface (lines 558 – 581).

88 grounding line are

We use the singular verb “is” to refer to the subject of the sentence “melt rates decreasing with distance from the grounding line”, which is treated as a single idea or concept.

135-148 the continued reliance on Qsg, mbpgis, U*300m and other variable-based terminology really does make reading this difficult for someone unfamiliar with the work. I know one round of this has already been done, but I would suggest one more round of ‘decreasing variable terminology’ to improve readability.

We acknowledge the concern raised here. We have thoroughly reviewed and eliminated all abbreviations for discharge, and melt and its drivers from the main text. However, in sentences in the results chapter where discharge experiments are being compared (as in this paragraph), it is necessary to describe changes in melt and its drivers averaged spatially over two domains. In such cases, the use of abbreviations allows us to avoid repetitive and excessively cumbersome phrasing.

228 estimates (somewhat awkward flow here)

Thank you for informing us about it. Following feedback from Reviewer 1, we have distilled the text and refined the writing in the discussion chapter wherever applicable to improve readability. Here, we use estimate to refer to the ensemble averaged time-mean value (372 m³/s) that was generated to force our model.

234 are currently being experienced

Thank you for spotting this. We have fixed it in the revised manuscript. Please see lines 239 – 240.

284-89 maybe I just need an afternoon coffee, but this sentence is far too long to be digestible without multiple readings.

Thank you for informing us. We have made adjustments in the revised version to improve readability. Please see lines 291 – 297.

305 Idle curiosity: does your model allow for migration of channels, as has been observed on Thwaites and elsewhere? Do you observe migration of channel location at all? E.g. Chartrand et al., 2020. I wonder if this could delay Petermann collapse.

Thank you for raising this point. In our current model setup, channel migration is not simulated, as the ice shelf cavity geometry (including basal channels) is prescribed and static. In other words, while our model resolves channelized melting, it does not dynamically adjust the ice shelf basal topography in response to melt-induced thinning or ice flow adjustments. However, recent advances in coupled ice sheet-ocean modelling (Gladstone et al., 2020; Zhou et al., 2024) now

enable time-evolving cavities with dynamic channel migration. These tools will help resolve whether channel mobility stabilizes ice shelves.